# Classification of the Genus *Harpalus* (Coleoptera, Carabidae) of the World Based on Imaginal Morphology †

Boris M. Kataev

Zoological Institute, Russian Academy of Sciences, Universitetskaya Nab. 1, St. Petersburg 199034, Russia; harpal@zin.ru
† urn:lsid:zoobank.org:pub:B3EEE405-FE8A-4BB4-9DC1-F6A549FD09B0.

**Abstract:** The genus *Harpalus*, Latreille, 1802, is the largest ground beetle genus of the tribe Harpalini and one of the most diverse genera of Carabidae (more than 400 described species). The definition and taxonomic boundaries of the genus, as well as previously proposed classifications, all based on regional faunas, are briefly reviewed in this paper from a historical perspective. The classification of the genus *Harpalus* proposed herein is based, for the first time, on a comparative morphological study of the world fauna and covers all the described species. The genus is divided into seventy subgenera combined into nineteen subgroups and ten groups; thirty-six subgenera are newly described: *Afroharpalus* **subg. n.** (type species *Harpalus fulvicornis* Thunberg, 1806), *Hyloharpalus* **subg. n.** (type species *Harpalus laevipes* Zetterstedt, 1828), *Sinoharpalus* **subg. n.** (type species *Harpalus puetzi* Kataev et Wrase, 1997), *Macroharpalus* **subg. n.** (type species *Erpeinus major* Motschulsky, 1850), *Meroharpalus* **subg. n.** (type species *Harpalus fulvilabris* Mannerheim, 1853), *Ameroharpalus* **subg. n.** (type species *Harpalus spadiceus* Dejean, 1829), *Drymoharpalus* **subg. n.** (type species *Harpalus atratus* Latreille, 1804), *Caucasoharpalus* **subg. n.** (type species *Omaseus aeneipennis* Faldermann, 1836), *Calathoderus* **subg. n.** (type species *Harpalus potanini* Tschitschérine, 1906), *Isoharpalus* **subg. n.** (type species *Carabus serripes* Quensel, 1806), *Psammoharpalus* **subg. n.** (type species *Harpalus kozlovi* Kataev), 1993, *Platyharpalus* **subg. n.** (type species *Harpalus ventralis* LeConte, 1848), *Asioharpalus* **subg. n.** (type species *Harpalus nigrans* Morawitz, 1862), *Anamblystus* **subg. n.** (type species *Carabus latus* Linnaeus, 1758), *Homaloharpalus* **subg. n.** (type species *Carabus tardus* Panzer, 1796), *Bactroharpalus* **subg. n.** (type species *Harpalus cautus* Dejean, 1829), *Diaharpalus* **subg. n.** (type species *Harpalus vittatus* Gebler, 1833), *Mesoharpalus* **subg. n.** (type species *Harpalus gisellae* Csiki, 1932), *Eremoharpalus* **subg. n.** (type species *Harpalus remboides* Solsky, 1874) *Oreoharpalus* **subg. n.** (type species *Harpalus famelicus* Tschitschérine, 1898), *Hypsoharpalus* **subg. n.** (type species *Harpalus arnoldii* Kataev, 1988), *Anophonus* **subg. n.** (type species *Ophonus cyanopterus* Tschitschérine, 1897), *Haloharpalus* **subg. n.** (type species *Harpalus salinulus* Reitter, 1900), *Megaharpalus* **subg. n.** (type species *Harpalus stoetznerianus* Schauberger, 1932), *Aristoharpalus* **subg. n.** (type species *Harpalus ingenuus* Tschitschérine, 1898), *Cycloharpalus* **subg. n.** (type species *Harpalus pulvinatus* Ménétries, 1848), *Euryharpalus* **subg. n.** (type species *Harpalus cisteloides* Motschulsky, 1844), *Brachyharpalus* **subg. n.** (type species *Carabus autumnalis* Duftschmid, 1812), *Hemipangus* **subg. n.** (type species *Harpalus klapperichi* Jedlička, 1955), *Mauriharpalus* **subg. n.** (type species *Harpalus cardoni* Antoine, 1922), *Caloharpalus* **subg. n.** (type species *Harpalus cupreus* Dejean, 1829), *Baryharpalus* **subg. n.** (type species *Carabus dimidiatus* Rossi, 1790), *Heteroharpalus* **subg. n.** (type species *Harpalus metallinus* Ménétries, 1839), *Idioharpalus* **subg. n.** (type species *Harpalus numidicus* Bedel, 1893), *Pachyharpalus* **subg. n.** (type species *Harpalus crates* Bates, 1883) and *Paraharpalus* **subg. n.** (type species *Harpalus oblitus* Dejean, 1829). A detailed morphological description of the genus, diagnoses, composition and distribution of all supraspecific taxa with brief biotopic notes are given.

**Keywords:** ground beetles; Harpalini; *Harpalus*; morphology; taxonomy; new subgenera

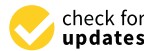



## 1. Introduction

"One of the largest genera of ground beetles is Harpalus. I would gladly keep silent about them, if many of the ground beetles commonly encountered by us did not belong to them. They should be considered boring insects when it comes to describing their differences from others. In fact, in their form, in the dull coloration of all species, there are so few features... so much in common that there is nothing to dwell on...".

Alfred Edmund Brehm. Illustrated Life of Animals (1876).

The genus *Harpalus* Latreille, 1802 is the largest genus of the tribe Harpalini (subfamily Harpalinae, according to Lorenz [1,2]) and one of the most species-rich genera of Carabidae, with over 400 species. This genus is also one of the taxonomically most difficult groups of ground beetles. These difficulties are due to a fairly uniform appearance with a large number of species, very high individual and geographical variability, very wide distributional ranges of many species, and also the fact that most taxa were described in the 19th or at the very beginning of the 20th century based on very limited material and usually without analysis of their geographical variability. The result is a very large number of synonyms for the genus, which includes more than 1500 names. The lack of a substantiated classification of *Harpalus* to date significantly complicates further taxonomic studies and other works on this genus, especially phylogenetic, faunistic and zoogeographical studies.

### 1.1. Definition and Taxonomic Boundaries of the Genus Harpalus

It is quite obvious that the development of a classification of any taxon is possible only when this taxon is distinctly defined and clearly distinguished by its diagnostic features from closely related taxa and is considered in full on the basis of the world fauna. In the case of *Harpalus* Latreille, 1802, this is especially true, since delimitation of this genus is an old and very difficult taxonomic problem.

*Harpalus* is the type genus of the tribe Harpalini. This tribe is one of the largest and most diverse in the family Carabidae. It includes more than 240 genera and subgenera and about 2000 species distributed in all biogeographic regions of the world. About 6300 names have been proposed so far by researchers in an attempt to describe the entire diversity of the numerous forms included in this group. As Ball [3] rightly noted, on the one hand, representatives of the tribe are quite diverse in their morphology, suggesting that they represent a large number of discrete evolutionary lineages; on the other hand, the abundance of parallelisms and the extremely high variability of many morphological structures greatly complicate the task of the definition of these lineages and consequently make it extremely difficult to work out the natural system of the tribe. As in many other tribes of ground beetles, most of the more or less diverse supraspecific taxa in Harpalini are defined polythetically in the understanding of Simpson [4] and Mayr [5] based on a complex of features, when each individual character may be absent in some representatives of this taxon; however, it is often present in a different combination in representatives of other taxa. Respectively, there are very conflicting opinions in the literature about the composition of many genera of Harpalini and their association into natural groups. Another reason for such disagreements is that almost all of the proposed classifications were developed on the basis of the regional faunas and usually on a limited number of analyzed characters. As a consequence, characters diagnosing groups in one region are often completely inappropriate for the same groups in other regions with a different species composition. With an uncritical approach, the wider application of such classifications developed on the regional faunas has always led either to the unification of heterogeneous elements, or, conversely, to the unjustified separation of many natural groups.

According to the modern data, the genus *Harpalus* belongs to the Harpali genus group of the subtribe Harpalina [6–8]. This group is defined mainly by setose paraglossae and the absence of many apomorphies characteristic of other generic groups (Acinopi, Trichotichni, Selenophori, Ophoni, Bradybaeni and some other). The diagnoses and, accordingly, the composition of most of these groups are still debatable.

The diagnosis, taxonomic boundaries, and, consequently, the composition of the genus *Harpalus* have also been the subject of much debate since its description up to the present time. The first ten species of *Harpalus* whose names are now considered valid were described in the genus *Carabus* before the description of the genus *Harpalus* itself, and the first species, *H. latus*, was described by Linnaeus as early as 1758 [9]. When describing the genus *Harpalus*, Latreille [10] included twenty-five species in it, of which only five are now assigned to the tribe Harpalini, and only two of these five species have retained their position within the genus *Harpalus*: *Carabus ruficornis* Fabricius, 1775 (=*C. rufipes* DeGeer, 1774), and *Carabus proteus* Paykull, 1790 (=*C. affinis* Schrank, 1781). Latreille [11] designated *C. ruficornis* as the type species of the genus; however, as Andrewes [12] noted, this species is now assigned to *Pseudoophonus*, and if Latreille's designation is accepted, significant nomenclatural changes will be required in the long-established and widely names used. Therefore, Andrewes [12] suggested taking *Carabus proteus* (=*C. affinis*) as the type species of the genus *Harpalus*. Almost all subsequent authors [6,13–16] and others accepted Andrewes' proposal and considered *C. proteus* to be the type species of the genus *Harpalus*. A recent attempt [17] to revise the nomenclature of the genus based on the recognition of *Carabus ruficornis* as its type species was not supported by subsequent authors (in this case, the name *Harpalus* would be assigned to *Pseudoophonus*, and most other species of the genus would have to be assigned to *Actephilus*). In this work, *Carabus proteus* (=*Harpalus affinis*) is also tentatively accepted as the type species of the genus *Harpalus*.

Since the name *Harpalus* is the oldest in the tribe Harpalini, almost all representatives of the tribe were originally described in the genus *Harpalus*. Subsequently, with the accumulation of knowledge, the most isolated forms were distinguished as separate genera, but due to the lack of clear diagnoses and the abundance of taxa with intermediate characters, a generally accepted point of view on the composition and boundaries of the genus has not yet been developed.

Most of the earlier authors who have worked with Harpalini ([6,15,16,18–32], etc.) treated the genus *Harpalus* as extremely broad, including almost all taxa Harpali and Ophoni. In such a broad interpretation, *Harpalus* is a very large and heterogeneous complex, in which, on one hand, undoubtedly heterogeneous elements are united together, and on the other, many natural groups turn out to be divided into separate units. An example of such an interpretation, covering the entire world fauna, is the system (more precisely, the worldwide synopsis of the supraspecific taxa) of the genus *Harpalus* published by Noonan [6], in which the genus unites 37 equivalent subgenera without indicating their diagnostic characters; the diagnosis of the genus was also absent. A similar system of the genus *Harpalus*, again without any diagnoses or comments, was given in the first edition of the systematic list of ground beetles of the world published by Lorenz [32]. Most of the other authors, who interpreted the genus *Harpalus* quite broadly, worked mainly with regional faunas, and the diagnoses they gave in most cases corresponded (and not always) only to these local faunas.

The heterogeneity of this complex (*Harpalus* sensu lato) has long been noted by researchers, and therefore a number of authors, especially those who studied mainly regional faunas ([17,33–44], etc.), accepted a narrower interpretation of the genus, dividing *Harpalus* into a number of separate genera, primarily distinguishing from *Harpalus* such easily recognizable and widespread groups as *Ophonus* and *Pseudoophonus*. Despite the fact that these authors limited their studies to only one local fauna, as a result of which the number of identified genera and their composition in different systems sometimes differ significantly, this approach undoubtedly contributed to a better understanding of the taxonomic structure of *Harpalus* and related taxa. Unfortunately, in most of these studies, the researchers were not always able to sufficiently substantiate their conclusions and, more importantly, to make reliable diagnoses for all groups, so none of the systems they proposed were universally accepted. In this connection, special mention should be made of the work of Brandmayr et al. [41], who managed to sufficiently justify the significant difference in the structure and biology of larvae in *Ophonus* and *Harpalus* (their work refers to the harpaloid

and ophonoid phyletic lineages within *Harpalus* sensu lato). For many groups, however, the larvae are not yet known, and therefore it is not yet possible to use their characters fully to clarify the relationships of *Harpalus* with other genera. At the same time, as expected, with an increase in the number of known larvae, it becomes more and more obvious that the variability in the larval characters of *Harpalus* and related taxa is much wider than previously thought [45].

Since most Harpalini taxa demonstrate a very wide variability in diagnostic characters with many parallelisms, the use of rigorous cladistic analysis results in the creation of many unnatural groups. My many years of experience with this group convince me that most of the structural characters of Harpalini, including punctation and pubescence of integuments, setigerous pores on elytra, adaptive modifications of protibia, sexual modifications of some sclerites and structures, the shape of the median lobe of aedeagus and the armament of its internal sac, etc., have arisen and disappeared among its various representatives repeatedly and independently in the course of its evolution (see, for example, [8,46]). The same character may be constant in members of one group and very variable in members of other groups. Even within one group of closely related species, the same character may be invariable in most of its species and individually changeable in one of its representatives, for example, the fusion of mentum and submentum in *Cryptophonus* Brandmayr et Zetto Brandmayr, 1982 [47]. In this case, recognition of monophyletic groups is possible only at a comprehensive analysis of these groups in full composition and on the basis of similarity of a complex of independent distinctive characters covering the most various structures and organs. A comparative morphological study of most representatives of all supraspecific taxa of *Harpalus* and related genera of the world fauna, which I carried out, made it possible to identify several of the most important morphological characters for this group and, on their basis, to revise the diagnosis and the taxonomic composition of the genus *Harpalus*.

According to recent data [8], the genus *Harpalus*, like most other genera of Harpalini, is defined polythetically and includes species with the following combination of characters: body glabrous or pubescent, eyes hairless; paraglossae setaceous at margins; frontal foveae small, often punctiform, fronto-ocular furrows absent; epilobes of mentum more or less narrow with inner margin more or less straight, not or at most indistinctly angular; and protarsomeres 1–4 and mesotarsomeres 1–4 or 2–4 in males are dilated and with two rows of adhesive scales ventrally. In addition, in most species, the metacoxa are without a posteromedial setigerous pore, metatarsomere 1 is shorter than metatarsomeres 2 and 3 combined, the basal labial palpomere is not carinate ventrally, and the aedeagus is asymmetrical, with the apical orifice of the median lobe in most species shifted to the left, and with or without an apical capitulum. Taken together, these distinctive features define the genus quite uniquely, although some of its members do not have a complete set of them. The presence of setae on the paraglossae is one of the most characteristic features not only of *Harpalus* but apparently of all Harpali.

The most important result of the comparative revision of *Harpalus* and all putatively related taxa is the justification of a generic status of *Ophonus* based on sufficient differences in the morphology of adults between this taxon and the genus *Harpalus* [46,48], which confirmed a similar conclusion made earlier [41] based on the study of larvae of both groups. The molecular studies of the tribe Harpalini based on the analysis of mitochondrial cytochrome oxidase [44] also confirmed the independence of the genera *Ophonus* and *Harpalus*.

The monotypic taxon *Nipponoharpalus* Habu, 1973, described as a subgenus of *Harpalus*, is also excluded from *Harpalus*. It is treated as a separate genus related to *Trichotichnus* Morawitz, 1863 [49–51].

The North American *Harpalus nitidulus* Chaudoir, 1843 (=*H. fulgens* Csiki, 1932), originally described within *Harpalus* and later considered a member of the monobasic *fulgens* group of this genus [15,16], is transferred to the subgenus *Iridessus* Bates, 1883, of *Trichotichnus* [7,52].

The monotypical *Microharpalus* Tschitschérine, 1901, which was described as a subgenus of *Harpalus* sensu lato and then considered a synonym of *Harpalus* sensu stricto without any evidence [6,23,53], was included into the genus *Microderes* Faldermann, 1936 [54].

The monotypical taxon *Harpalobrachys* Tschitschérine, 1899, which is often included in the genus *Harpalus* (for example [7,16]), is also considered a separate genus within Harpali. In contrast to all known *Harpalus*, in the males of its only species, the pro- and mesotarsi are not widened and do not bear adhesive scales ventrally. In other characters, including setose paraglossae, it is similar to *Harpalus*, but is characterized by a very peculiar appearance (somewhat similar to members of the genus *Chydaeus* Chadoir, 1854, of the subtribe Anisodactylina) and male genitalia and cannot be related to any of the *Harpalus* taxa. Since the features of adhesive vestiture on the widened tarsomeres of the male in Harpalini are the basis for the division of this tribe into its main phyletic lineages, a more accurate position of this taxon requires the use of additional characters, for example, the larval stage. Unfortunately, the larvae of this taxon are not yet known. Molecular studies have also not been carried out.

The taxon *Anisochirus* Jeannel, 1946, described as a subgenus of *Harpalus*, is also treated as a separate genus, to which most of the Madagascan and Mascarene species previously included in *Harpalus* are assigned [55]. The members of this genus differ from those of *Harpalus* in having glabrous paraglossae.

At the same time, *Cephalophonus* Ganglbauer, 1892, *Pseudoophonus* Motschulsky, 1844, *Semiophonus* Schauberger, 1933 and *Cryptophonus* Brandmayr et Zetto Brandmayr, 1982, which are sometimes given generic status or included in the genus *Ophonus*, and also *Harpalellus* Lindroth, 1968, are treated as members of *Harpalus* because they do not have clear morphological features that distinguish them from this genus.

Such a definition of the genus *Harpalus* was accepted in the most recent general works on Carabidae (e.g., [1,7,50,51,56,57]).

Although the diagnosis presented clearly identifies the genus *Harpalus*, due to the lack of common apomorphies for all of its members, the genus in such an interpretation is still likely a paraphyletic rather than a monophyletic group. Its relationships with some Afrotropical genera of Harplali and some Palaearctic ones, such as *Microderes* Faldermann, 1936, and even some Acinopi [58] require further special study.

*1.2. Classification of the Genus Harpalus*

1.2.1. Brief Review of Previous Classifications

The first attempt to divide the genus *Harpalus* (as it is defined here) into a number of subdivisions belongs to Motschulsky [34,59–61]. However, the taxa he recognized (*Pseudoophonus* Motschulsky, 1844, *Platus* Motschulsky,1844, *Erpeinus* Motschulsky, 1844, *Conicus* Motschulsky, 1844, *Pheuginus* Motschulsky, 1844, *Bioderus* Motschulsky, 1848, *Amblystus* Motschulsky, 1864 and *Ooistus* Motschulsky, 1864) were very poorly characterized, identified by single characters and had an eclectic composition. Not unexpectedly, the Motschulsky system was not adopted by subsequent authors. For example, Ganglbauer [36], the author of the most thoroughly developed classification of ground beetles for his time, divided *Harpalus* only into three subgenera (*Harpalus* s. str., *Artabas* Gozis, 1882 and *Actephilus* Stephens, 1833) and considered almost all the subgeneric names proposed by Motschulsky to be synonyms of *Harpalus* s. str. Note that Ganglbauer [36] considered *Pseudoophonus* to be a subgenus of the genus *Ophonus*.

In 1900, Reitter [38], based on the study of a significant part of the Palaearctic species known to him at that time, divided *Harpalus* into twelve subgenera: *Artabas* Gozis, 1882, *Loxophonus* Reitter, 1894, *Microderes* Faldermann, 1836, *Epiharpalus* Reitter, 1900, *Lasioharpalus* Reitter, 1900, *Harpaloxys* Reitter, 1900, *Asmerinx* Tschitschérine., 1898, *Amblystus* Motschulsky, 1864, *Harpaloderus* Reitter, 1900, *Harpalobius* Reitter, 1900, *Actephilus* Stephens, 1833 and *Pheuginus* Motschulsky, 1844. Unlike Ganglbauer (1892), Reitter (1900) considered *Ophonus* and *Pseudoophonus* to be separate genera, and *Harpalophonus* Ganglbauer, 1892, described in the genus *Ophonus*, a subgenus of *Pseudoophonus*. In the sub-

genus *Asmerinx*, Reitter (1900) combined one species of *Trichotichnus* Morawitz, 1863 in the modern sense and four species of *Harpalus*. Later he [39], continuing to consider *Trichotichnus* (=*Asmerinx*) as a subgenus of the genus *Harpalus*, established for these four *Harpalus* species one more subgenus, *Acardystus* Reitter, 1908. Reitter's [38,39] classification undoubtedly introduced a certain order into the variety in the Palaearctic *Harpalus*. Nevertheless, this system served almost exclusively for utilitarian tasks of identification, and the subgenera were characterized by an arbitrary combination of a few features of the external morphology, convenient for use in keys (punctation of the elytra and base of the pronotum, pubescence of the abdominal sternites, shape of the pronotum, etc.), and with a few exceptions, the majority of these subgenera were not natural groups, uniting phylogenetically different species. Reitter, it seems, did not set himself any other task. The artificial nature of his classification can be judged at least by the fact that he assigned only the nominative form of *H. tjanschanicus* Semenov, 1889, which is characterized by impunctate and glabrous elytra, to the subgenus *Lasioharpalus*. A variation of this species, which was described as var. *cyclopius* Reitter, 1900, characterized by the presence of punctures and setae on the lateral elytral intervals, was assigned by him to the subgenus *Epiharpalus*. Surprisingly, Reitter did not made any comments on this treatment. Despite the fact that the system proposed by Reitter was criticized by many authors ([21,22,62], etc.), it gained wide popularity. In particular, it was accepted by Jakobson [63] with only minor changes.

Tschitschérine [20], who proposed the most carefully developed classification of Palaearctic Harpalini for his time, adhered to a broad concept of the genus and placed most of the *Harpalus* species in the subgenus *Harpalus* s. str. He refrained from dividing this vast complex into smaller groups because of the extreme complexity of the issue, but noted the heterogeneity of such a taxon.

Lutshnik [21] adhered to the similar broad concept of the genus but considered it absolutely necessary in the future to split this temporarily accepted group (*Harpalus* sensu stricto) into a significant number of subgenera.

Schauberger ([25,64–68], etc.) made a great contribution to the development of the classification of the Palaearctic *Harpalus*; in particular, he was able to outline quite well a number of natural groups within this genus (for example, the "*fuscipalpis*", "*rufitarsis*", "*dimidiatus*", "*picipennis*" groups and some other). In addition, this author was the first to draw attention to the various structures of the protibia in different groups of Harpalini, including species of *Harpalus*, and based on their features provided a new diagnosis for the subgenus *Acardystus* and established two new subgenera: *Haploharpalus* Schauberger, 1926 and *Loboharpalus* Schauberger, 1932. However, Schauberger overestimated the taxonomic significance of some adaptive changes in the structure of the protibia, considering convergently similar groups of *Acardystus*, *Haploharpalus* and *Phygas* Motschulsky, 1848 (=*Neophygas* Noonan, 1976) as subgenera of the separate genus *Acardystus*, which he contrasted with the genus *Harpalus* (including *Ophonus* and *Pseudoophonus*). It should be noted that the structure of the protibia was used by Schauberger [69] as the basis of the classification of the world fauna of the tribe Harpalini, but, as in the case of the genus *Acardystus*, many of its groupings turned out to be artificial. Although the complete classification of Harpalini was not published by him, it is known [3] that Csiki consulted with Schauberger, and he used the Schauberger's system in his catalog of the world ground beetle fauna [23]. Unfortunately, the catalogue lacks comments or diagnoses.

The Nearctic species of *Harpalus* were fist revised by LeConte [70] and later by Casey [71,72]. The latter author not only described many new species but also established several supraspecific taxa. Unfortunately, most of the new species described by Casey were based on individual variation, with the result that his revision introduced more chaos into the system than order, making it very difficult to use.

A new stage in the development of *Harpalus* taxonomy began after the use of the structural features of the male genitalia and, above all, the features of the armament of the internal sac of the aedeagus, usually well developed in *Harpalus*, since the shape of the aedeagus itself, as it turned out, varies little within the Harpali genus group. Due to the

study of the internal sac, it was possible to clearly distinguish species, the identification of which had previously been very difficult. For the first time, the features of the internal sac of aedeagus for distinguishing closely related species were used by Lindroth [73]. Antoine [40] widely used the internal structure of aedeagus in the revision of Moroccan Harpalini, including representatives of the genus *Harpalus*. According to Antoine [40], the sclerotized structures in the internal sac of the aedeagus are an excellent specific feature, but they do not allow for understanding the relationships between species, since the development and the differentiation of these structures occurred in a similar way in different lines of Harpalini and often led to the same external result. As evidence, Antoine pointed to similar structures in the internal sac of the aedeagus in representatives of unrelated groups, for example, *Oedesis villosulus* (Reiche, 1859) and *Harpalus cupreus* Dejean, 1929, *Harpalus lateralis* Dejean, 1829 and *Ophonus berberus* Antoine, 1825, etc. Therefore, Antoine [40] divided the *Harpalus* species of the Moroccan fauna into groups mainly according to the characteristics of their external structure, with a wide use of secondary sexual characteristics. Lindroth [15] came to the opposite conclusion when studying North American species of *Harpalus*. This author convincingly showed that the number and location of sclerotized structures in the inner sac of the aedeagus is of great importance not only for species identification but also for the association of species into natural groups. According to Lindroth [15], the armament of the internal sac in most cases reflects the relationship between species much better than the characters of the external morphology that were previously proposed for dividing *Harpalus* into subgenera. Based mainly on the features of the internal sac, he grouped most of the North American species. Some of these groups partly corresponded to the taxa *Plectralidus* Casey, 1914, *Pharalus* Casey, 1914, *Harpalomerus* Casey, 1914, *Euharpalops* Casey, 1924 and *Cordoharpalus* Hatch, 1949, previously distinguished in the fauna of North America by Casey [71,72] and Hatch [74]. Lindroth's work, therefore, contributed new character evidence for these groups. Two groups ("*herbivagus*" and "*opacipennis*") recognized by Lindroth [15], in his opinion, corresponded to the Palaearctic subgenera of *Amblystus* and *Pheuginus*, and some remained under the informal name of species groups. Lindroth believed that a rational division of *Harpalus* into subgenera would be possible only when all the species of the genus and, first of all, the numerous species of the Palaearctic fauna were studied in a similar way.

Some Palaearctic species were revised, taking into account the structure of the internal sac of aedeagus, by Mlynář [53,75,76], who, like Lindroth, considered the armament of the internal sac to be the main feature that makes it possible to combine species into natural groups and define relationships between them. Based on the features of the armament of the internal sac, Mlynář characterized some groups of *Harpalus*, but he did not propose a general classification of the genus. Of particular interest is the fact of taxonomic vicariance established by this author, probably for the first time, between several species of *Harpalus* of the Palaearctic and Nearctic regions previously studied separately: *H. fuscipalpis* Sturm, 1818 and *H. basilaris* Kirby, 1837; *H. obtusus* Gebler, 1833 and *H. amputatus* Say, 1830; *H. quadripunctatus* Dejean, 1829 (=*H. laevipes* Zetterstedt, 1828) and *H. egregius* Casey, 1914; and *H. obesus* Morawitz, 1862 (=*H. major* Motschulsky, 1850) and *H. viduus* LeConte, 1865. Each of these pairs, except for the last one, is currently considered within the boundaries of one species.

An important step in the study of the North American *Harpalus* was the publication of two works by Noonan [16,77], in which he presented his results of a taxonomic revision and cladistic analysis of all species of this genus known north of Mexico, except for those that had already been revised by previous authors [78–80]. Thus, not only a few species from Mexico but also the subgenera *Pseudoophonus* Motschulsky, 1844 and *Glanodes* Casey, 1914 were not included in the analysis and, accordingly, were absent from the classification. Also not mentioned [16] was the Holarctic *Harpalus fuscipalpis* Sturm, 1818 (=*H. basilaris* Kirby, 1837), which he, following Lindroth [15], probably considered to be a member of the separate genus *Harpalellus* Lindroth, 1968 of the Selenophori genus group [6], although this species is also absent in his later revision of selenophorines [81]. No attempt was

made by this author to compare the faunas of the Palaearctic and Nearctic, traditionally studied independently of one another. The North American *Harpalus* species studied by Noonan were divided by him into twenty-two groups (sometimes also into subgroups), some of which (thirteen groups) were combined into two phyletic stocks ("*fraternus*" and "*caliginosus*"), and within the *fraternus* stock, a significant part of the species groups were united in the *fraternus* lineage. Nine groups were not included by Noonan in the cladograms, since, in his opinion, there are no synapomorphies that would allow for establishing any cladistic relationships both between these groups and between them and other *Harpalus*. Noonan [16] based his system mainly on the features of the structure and shape of the internal sac of the aedeagus, which he studied by turning the sac inside out but not inflating it. As a result of this limited approach and due to the fact that this author did not notice many important morphological features, some of the groupings he proposed turned out to be artificial, and, for example, the *rewolinskii* group was based on individuals with an aberrant armament of the internal sac [7,46]. In addition, having studied the types of all North American *Harpalus*, Noonan established many new synonyms; however, at least some of the species names fell into synonyms unjustifiably. Some of these errors were later corrected by Ball and Bousquet [7].

Ball and Bousquet [7], in the first volume of the American Beetles, followed mainly the classification of Noonan [16]. Based on the same data, they formally treated some of his species groups or their combinations as subgenera: *Euharpalops* Casey, 1914, *Opadius* Casey, 1914, *Pharalus* Casey, 1914, *Megapangus* Casey, 1914, *Plectralidus* Casey, 1914, *Harpalomerus* Casey, 1914 and *Harpalobius* Reitter, 1900. More recently, Bousquet [82] also largely followed the classification of Noonan [16] and Ball and Bousquet [7] but treated some subgenera more widely, making some additional combinations, although also without special taxonomic studies.

The results of my research on the taxonomy of *Harpalus*, mainly Palaearctic taxa, are presented in a series of articles that have been published since 1984 [8,46–48,54,55,83–119]. In these publications, many subgeneric taxa have been revised, the taxonomic status and distribution of numerous insufficiently described taxa have been clarified, many new taxa have been described, and many new synonyms have been established. In addition, the species were arranged into species groups and phyletic lineages, some of which were considered as subgenera [8,51,120]. However, most of the species groups remained within the highly diverse subgenus *Harpalus* s. str. At the first stage of research, before the completion of the revision of the genus as a whole, the allocation of informal groupings, in particular species groups, and the elucidation of their relationships seemed most appropriate as the basis of the future formal subgeneric system. Such an informal hierarchical classification, although it reflects well enough the relationships between taxa within *Harpalus*, is inconvenient from a practical point of view, since only subgenera can be formally recognized within the genus, and the division into informal groups is ignored in most catalogues and faunistic lists.

Therefore, in this paper, it is proposed to consider the majority of the recognized natural species groups of *Harpalus* as subgenera, which, in turn, are united within the genus into informal groups and subgroups, thus preserving the hierarchical nature of the subgenus system. Such a classification turns out to be rather fractional, with a large number of new subgenera, but it seems to be the most advisable and correct, since in this case the monophyly of subgenera uniting most closely related species seems to be the most obvious in comparison with their informal associations into groups and subgroups, for most of which monophyly has not been proven. In any case, the rank of monophyletic groups is always subjective and determined by the position of these groups in the system and by practical considerations.

### 1.2.2. Present Classification of *Harpalus*

The classification of the genus *Harpalus* proposed here covers all the described species of the world fauna. This work began almost 40 years ago. During this period, most of the described species and subspecies were studied, including almost all valid taxa of the

genus, and their morphological characters, both external and male and female genitalia, were analyzed. Since this is the first classification of the genus covering all the described species, it is certainly not without flaws and will require further refinement, including on the basis of larval morphology and molecular data, but I hope that it will serve as a basis for this kind of research. Some taxa of the Nearctic and Afrotropical regions as well as some areas of the Palaearctic still require revision at the species level, and, undoubtedly, serious clarifications will be made in the future regarding the composition of some subgenera and their relationships.

All supraspecific taxa of *Harpalus* recognized here, including subgenera, are provided with diagnoses; however, since these taxa are defined polythetically, I have only been able to give a dichotomous key for the identification of subgenus groups.

## 2. Materials and Methods

This new classification is based on the study of the morphology of most of the described species and subspecies of *Harpalus* and related taxa of the world fauna, including the type material for almost a thousand nominal taxa.

The collection of the Zoological Institute, Russian Academy of Sciences (St. Petersburg, Russia), was the main basis for this research. In addition, rich material including many species was studied from the Zoological Museum (Moscow, Russia); Moscow Pedagogical University (Moscow, Russia); Institute of Biology and Soil Science, Far East Branch, Russian Academy of Sciences (Vladivostok, Russia); Institute of Systematics and Ecology of Animals, Siberian Branch, Russian Academy of Sciences (Novosibirsk, Russia); Institute of Biological Problems of the North, Far East Branch, Russian Academy of Sciences (Magadan, Russia); Institute of plant and animal Ecology (Yekaterinburg, Russia); Muséum National d'Histoire Naturelle (Paris, France); Naturhistorisches Museum (Wien, Austria); Ober-Österreichisches Landesmuseum (Linz, Austria); Museum für Naturkunde (Berlin, Germany); Deutsches Entomologisches Institut (Müncheberg, Germany); Zoologische Staatssammlung (München, Germany); Staatliches Museum für Naturkunde (Stuttgart, Germany); Naturkundemuseum (Erfurt, Germany); Staatliches Museum fur Tierkunde (Dresden, Germany); Zoologisches Forschungsinstitut und Museum Alexander Koenig (Bonn, Germany); Národni Muzeum v Praze (National Museum) (Prague, Czech Republic); Moravian Museum (Brno, Czech Republic); Natural History Museum (London, Great Britain); Hope Entomological Collection, University of Oxford (Great Britain); Natural History Museum (Basel, Switzerland); Természettudományi Múzeum (Hungarian Natural History Museum) (Budapest, Hungary); Zoological Museum of the University of Helsinki (Finland); Naturhistoriska Riksmuseet (Swedish Museum of Natural History) (Stockholm, Sweden); Museum of Zoology, Lund University (Lund, Sweden); Uppsala University (Uppsala, Sweden); Natural History Museum, University of Oslo (Norway); Zoological Museum, University of Copenhagen (Denmark); Institute of Zoology (Almaty, Kazakhstan); Schmalhausen Institute of Zoology (Kiev, Ukraine); Zoological Museum of Odessa University (Odessa, Ukraine); National Museum of Natural History (Sofia, Bulgaria); Ataturk University (Erzurum, Turkey); Institute of Zoology, Chinese Academy of Sciences (Beijing, China); Southwest Agricultural University (Chongqing, Sichuan, China); National Museum of Natural History, Smithsonian Institution (Washington, DC, USA); Field Museum of Natural History (Chicago, IL, USA); California Academy of Sciences (San Francisco, CA, USA); Canadian National Collection (Ottawa, ON, Canada); and Strickland Museum, University of Alberta, Alberta (Edmonton, AB, Canada). The personal collections of numerous amateur and professional entomologists were also studied. In total, many tens of thousands of specimens have been studied. It is impossible to specify more exactly the number of individuals studied, since many species in collections are often represented in very large series.

Standard methods for the comparative morphological study were applied when treating the material. Measurements of body length were made under a LOMO MBS 10 stereomicroscope using an ocular-micrometer and were taken from the anterior margin of

the clypeus to the elytral apex (strongly deviating data from the literature are enclosed in square brackets); the length and width of metepisterna were measured along their inner and anterior margins, respectively. The aedeagi and female genitalia were examined in glycerin and then placed in plastic microtubes or embedded in Euparal; in some cases, they were glued dry on a piece of hard paper under the beetle.

Drawings were prepared by using an ocular grid (10 × 10 squares) attached to the abovementioned stereomicroscope. The photographs were taken with a Canon EOS 6 D camera with a Canon MP-E 65 mm objective lens and subsequently processed using the Helicon Focus 6 software and optimized with Photoshop® CS2; the habitus photograph of the syntype of *Harpalus salinulus* Reitter, 1900 was taken with a Canon PowerShot A710 IS camera and also optimized with Photoshop® CS2.

The supraspecific taxa and species within subgenera are listed according to the assumed species relationships. Only valid names of species are given. For synonyms and a more detailed distribution, see the Palaearctic Catalogue of Coleoptera [51] for Palaearctic species and the Catalogue of Geadephaga of America, north of Mexico [82], for Nearctic species; full synonymy is also present in the Carabcat database [2]. Only the most important and mostly recent works on the taxonomy of the subgenera and other groups of Harpalus are listed in the relevant sections for each taxon.

## 3. Results

### 3.1. Genus Harpalus Latreille, 1802

*Harpalus* Latreille, 1802 [10] (p. 92) (as a genus). Type species: *Carabus proteus* Paykull, 1790 (=*C. affinis* Schrank, 1781), designated by Andrewes [12], tentatively accepted as type species.

*Harpaleus* Billberg, 1820 [121] (p. 23) (unjustified emendation).

### 3.1.1. Diagnosis

Body small to large (length 3.9–28.0 mm), glabrous or pubescent, eyes hairless. Paraglossae setaceous at margins (at least with one seta on each side). Epilobes of mentum more or less narrow, with inner margin more or less straight, not or at most indistinctly angular. Fronto-ocular furrows absent. Metacoxa in most species without posteromedial setigerous pore. Metatarsomere 1 shorter than 2 and 3 combined. In male, protarsomeres 1–4 and mesotarsomeres 1–4 or 2–4 dilated and with two rows of adhesive scales ventrally. Gonocoxite in most species with more or less long and fine setae on outer margin. Aedeagus asymmetrical, with apical orifice of median lobe in most species shifted to left, with or without apical capitulum.

### 3.1.2. Morphology

Size and body shape. Members of the genus are typically medium size; in most species, the body length is in the range of 7–11 mm. The smallest species is *H.* (*Actephilus*) *acupalpoides* (3.9–4.8 mm), the largest is *H.* (*Megapangus*) *caliginosus* (up to 28.0 mm).

Body of imago generally is somewhat robust, more or less parallel-sided (the width of the pronotum not much less than the width of the elytra), moderately convex, often wide and flattened, sometimes quite strongly convex, less often slender, with a relatively small pronotum and elytra elongate, rounded laterally. In some species, the body is oval, *Amara*-shaped, with widely rounded sides.

Color. The coloration of the body is variable, but brown and black tones predominate. Many species are characterized by the more or less pronounced metallic luster (especially green or blue) on the dorsum. In some species, a metallic tinge appears only in certain parts of the range. Like most other ground beetles, the metallic luster in *Harpalus* is primarily characteristic of species and populations living in humid biotopes. In arid conditions, the color is usually dark, without a metallic tinge. An iridescence of the dorsum is not characteristic of *Harpalus*, but is sometimes present in some members of the subgenus *Zangoharpalus*. Some species, or some populations within a species, are characterized

by a lack of pigmentation of the integument, resulting in the formation of a light color, various shades of yellow and red. Elytra are most often depigmented. In some species, due to partial depigmentation of the elytra, a pattern of light irregular stripes is formed. Sometimes wings are visible through the light-colored elytra, which makes the elytra appear spotted.

The coloration of the palpi and antennae varies within the genus from light yellow to black and is usually species specific. The coloration of the antennae is often bicolor; for example, the basal antennomeres are lighter than the more distal ones.

The color of the legs also varies from light yellow to black depending on the species or individually. Often, the legs are only partially infuscated, such as only the femora, or the femora together with the tibiae apically, or only tarsi. In some species, there is a fairly clear geographical pattern in the variability in the leg color, when populations with light and dark legs occupy different parts of the species range.

Microsculpture. The microsculpture of the integument is evident at least in certain parts of the body; it is often present everywhere, although the degree of its development in many species varies individually or geographically. In most species, the dorsal microsculpture, if developed, consists of isodiametric meshes; in some species, the meshes on the pronotum are slightly transverse; rarely, the microsculpture even on the elytra consists of thin transverse lines. The distribution of the dorsal microsculpture reveals a number of regularities. An indistinct microsculpture, and often its complete absence, is observed primarily in the central part of the frons and vertex, as well as in the central part of the disc of the pronotum and elytra; along the margins of the head, especially behind the supraorbital pore, as well as along the margins of the pronotum and elytra, the microsculpture is more distinct. In most species, the microsculpture on the elytra is more distinct than on the pronotum, and the meshes on the pronotum are more distinct than on the head. Generally, if a microsculpture is absent on the elytral disc, then it also absent on the pronotal disc, and certainly on the frons or vertex.

Head. The head is relatively large (although always narrower than the pronotum even including the eyes), short, sometimes thickened. It is usually narrowed behind the eyes; in some species the neck constriction is almost absent. The eyes are always developed, of medium size, rounded in lateral view, moderately convex, never pubescent (unlike *Ophonus*). Antennae are moderately long, usually reach the base of the pronotum and often extend beyond it; more rarely, they are shorter and do not reach base of pronotum. Antennomeres are almost always elongate (sometimes very slightly longer than wide), rounded in cross section in antennomeres 1–3, more or less laterally flattened, beginning from antennomere 4. Antennomere 1 (scapus) is the thickest and longest, antennomere 2 is the shortest; the remaining antennomeres are approximately equal in length, although, as a rule, antennomeres 3 and 11 are slightly longer than the other. Antennae are always covered with dense and short pubescence beginning from antennomere 3 (in *Cephalophonus*, sparser pubescence is also present on the two basal antennomeres).

The clypeus (Figure 1) has a slightly, arcuately emarginate apical margin, always covering dorsally the membranous base of the labrum; most species have one setigerous pore at the anterior angles; some species, for example among *Harpalophonus* (Figure 1c–h), have additional setigerous pores (from one to five on each side) located medially of the standard pair.

The frontal foveae are small, often punctiform, and, unlike many Selenophori, without fronto-ocular furrows (prolongations). The dorsal surface of the head in most species is impunctate and glabrous, but the genae are probably always finely punctate and setose. In some members, there are punctures and pubescence on the tempora, clypeus, near frontal foveae and around the supraorbital setigerous pores; in some species, punctation and pubescence cover almost the entire surface of the front and vertex.

Mouth parts. The labrum (Figure 2a) is transverse, with rounded apical angles and a straight or slightly notched apical margin; six setigerous pores along apical margin and several short marginal setae at the apical angles are always present. The dorsal surface in

most species is impunctate and glabrous, rarely (for example, in *Cephalophonus*) with fine punctures and short setae.

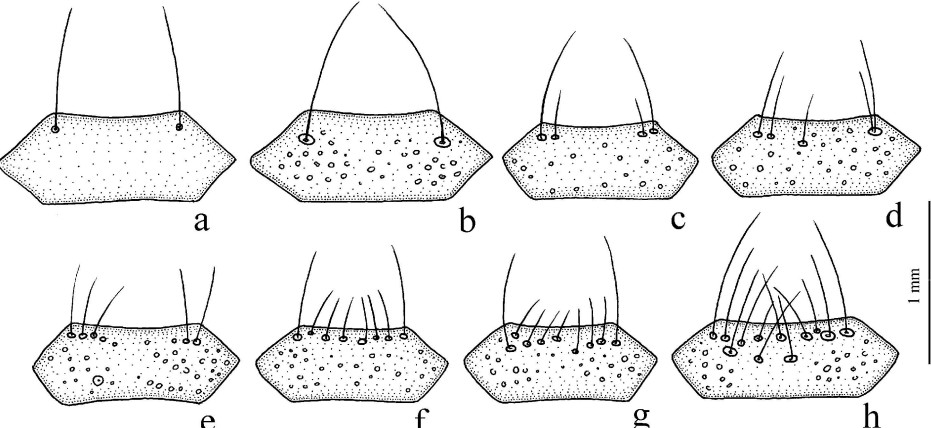

**Figure 1.** Clypeus of *Harpalus* species: (**a**) *H.* (*Haploharpalus*) *brevicornis*; (**b**) *H.* (*Harpalophonus*) *hospes*; (**c–h**) *H.* (*Harpalophonus*) *terrestris*.

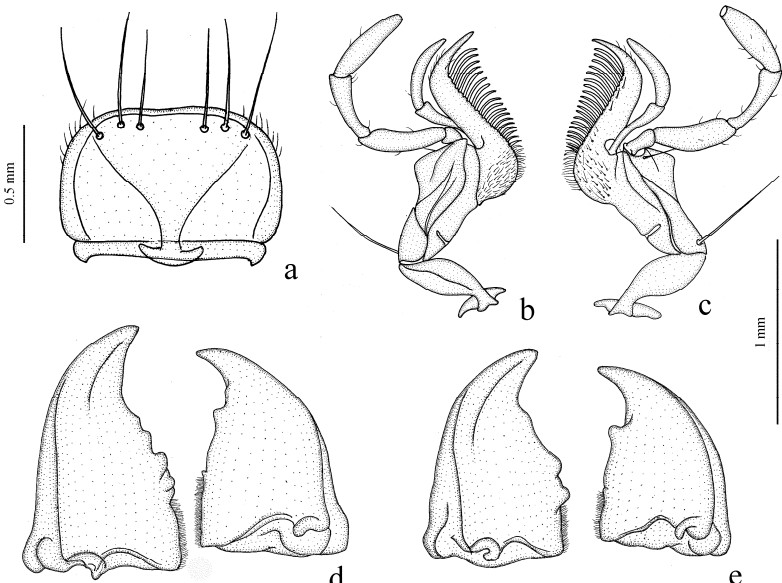

**Figure 2.** Mouth parts of *Harpalus* species: (**a–c,e**) *H.* (*Harpalus*) *affinis* (labrum, maxillae, and mandibles); (**d**) *H.* (*Caucasoharpalus*) *aeneipennis* (mandibles).

A characteristic feature of *Harpalus*, like most other members of the tribe Harpalini, is the presence of short and robust mandibles with a blunt apex, adapted for grinding solid particles of plants, mainly cereal seeds (Figure 2d,e). The outer scrobe of the mandibles in most species is glabrous, rarely with a few short setae.

Maxillae (Figure 2d,c) are not specialized and have a structure with typical characteristics of ground beetles. The last segment of the maxillary palpi is fusiform, slightly truncated at the apex.

The labium (Figure 3) includes some important characters for taxonomy of the genus. The submentum and mentum of most species are separated by a complete, transverse suture, rarely fused completely or partially (Figure 4). The submentum bears on each side one long ("gular") seta, in some species an additional shorter seta laterally; in addition, the surface is often covered with scattered short and fine setae. The mentum is transverse, deeply emarginate, generally with a more or less prominent median tooth. The degree of its development varies considerably in different species, and sometimes even within the

same species; rarely, a tooth is completely absent. The ventral surface of the mentum at the base of the tooth (if it is absent, then in the middle of the anterior margin of the mentum) has two setae, which are somewhat shorter than the inner setae on the submentum. The epilobes of the mentum are narrow, only slightly widened apically and in most species with a more or less straight inner margin. In members of *Cryptophonus* and some other species (Figure 3d,k,p), the epilobes are slightly wider than in most other *Harpalus*, and their inner margin is sometimes slightly angularly curved before the apex; however, the width of the epilobes and the angularity of their inner margin in *Harpalus* is always not as noticeable as, for example, in *Ophonus* and *Acinopus*. The ligular sclerite is narrow or slightly widened at apex, rather flat, usually without a transverse apical plate at the apex (a small apical plate, however, is present in *Cephalomorphus*) and with two ventroapical setae, usually directed anteriorly; some species also have one or several short erect setae on the dorsal surface (Figure 3p). In some taxa, for example, in the subgenera *Cephalophonus*, *Loboharpalus* and *Cryptophonus*, the presence of such setae is not constant. The paraglossae are membranous, in the form of two wide lobes, rounded apically and laterally, always with marginal setae (at least one seta on each side present). The setaceous paraglossae serve as a good diagnostic feature, making it possible to reliably distinguish members of *Harpalus* from those of some other similar genera (for example, *Trichotichnus*, *Nipponoharpalus*, *Anisochirus* and *Ophonus*). The basal labial palpomere in *Harpalus* is usually more or less cylindrical, in *Cryptophonus*, *Megapangus* and some species of *Pseudoophonus*, with an oblique (almost straight in some *Pseudoophonus*) longitudinal carina on its ventral surface (Figure 3e,h,i,j,l). The penultimate palpomere at the anterior margin closer to the apex bears two relatively thick and long setae and several (two to ten) finer and shorter setae; the apical palpomere is fusiform, slightly truncate at the very apex.

Thorax. The pronotum is transverse, in most species narrower than elytra; sometimes the width of the pronotum is almost equal to the width of the elytra or slightly exceeds it. The shape of the pronotum is variable, discoidal, rectangular trapezoid, or cordate, most often widest before the middle, less often in the middle or at the base. The apical and basal margins are usually arcuately emarginated, sometimes almost straight; the basal margin often has two notches, one on each side of the middle; the lateral margins (sides) are usually more or less rounded, often rectilinear or sinuate basally. The basal angles varies from widely rounded to rectangular or even (very rarely) acutangular. The basal and lateral margins are bordered; often the border along the basal margin is interrupted in the middle; in some species, border is obliterate laterally or completely absent. The border along the apical margin usually widens towards the middle and generally is more or less widely interrupted in the middle part. The disc has two basal, paramedial depressions (foveae); they are never very deep, often superficial or indistinct; their shape varies in different species from wide and flat to very small, like an oval or narrow pit. Many species have more or less pronounced depressions (lateral depressions) along the lateral margins, which are always widened basally; they usually begin in front of the middle of the pronotum and reach the base, but in some species, they begin just at the apical angles and sometimes are wide along entire length. The lateral depressions are either separated from the basal foveae by a convexity or fused with them, forming a common laterobasal depression on each side. The median line is very fine, does not reach the apical margin, and in most species even the basal margin. The anterior transverse depression is usually present, but very small and shallow. Both the median line and the anterior transverse depression are highly variable within one species. The sides of the pronotum (Figure 5) at some distance from the lateral margins, usually before the middle, less often exactly in the middle, bear a marginal setigerous pore (Figure 5a). Some species have additional such pores (up to seven), usually located in the apical half of the pronotum, but sometimes along its entire lateral margin (Figure 5b–f,h). As an aberration, additional setigerous pores (one or two) can also be found in species that normally have only one marginal pore; as a rule, such pores are present only on one side, less often on both sides. *Harpalus* (*Licinoderus*) *chobautianus* usually has one additional marginal pore on each side in front of the basal angles (Figure 5d). *Harpalus*

(*Loboharpalus*) *rubefactus*, in addition to the marginal setigerous pores, has also a row of very short setae in the anterior angles, directly on their edge (Figure 5g,h). Some species of *Artabas* and *Loxophonus* have rather large setigerous pores on the pronotal disc at its anterior margin between the median line and the apical angles as well as in the basal foveae (Figure 5f). The apical edge of the pronotum, in the area of its contact with the head, always has a rather dense row of short setae. Similar setae are also present in many species on the basal edge of the pronotum, forming a dense fringe of setae; setae are usually absent on the basal edge laterally (reaches only to the basal depressions). The surface of the pronotum may be smooth or more or less punctate, glabrous or covered with setae. The punctation is often limited only to the areas at basal foveae and within lateral depressions; often, it also extends to the entire base of the pronotum and can reach the anterior angles along the lateral margins. In some species, it is present along the anterior margin and even in the central part of the pronotal disc, but it is never regular and uniform. The punctation is almost always coarser and denser closer to the margins of the pronotum and especially coarse and dense in the region of the basal foveae. In some species, the punctures, primarily in the area of the basal foveae and lateral depressions, merge with each other, forming sinuous wrinkles.

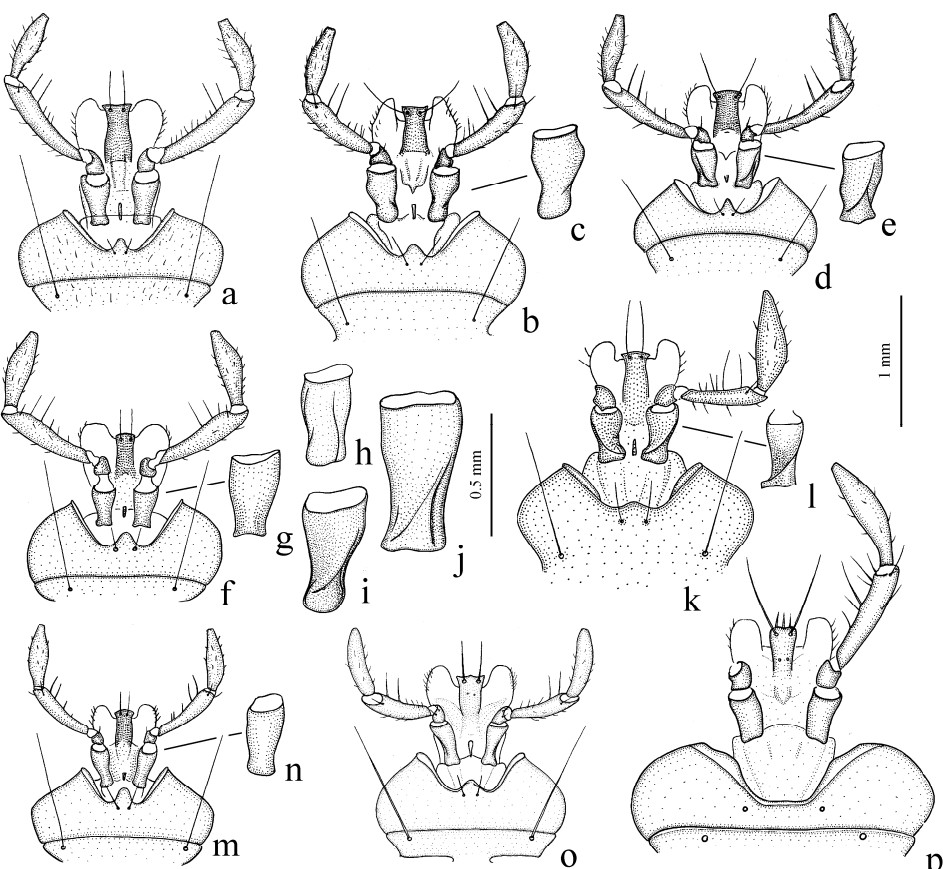

**Figure 3.** Labrum of *Harpalus* species (**c**,**e**,**g**–**j**,**m**,**o**, basal palpomere): (**a**) *H.* (*Cephalophonus*) *cephalotes* (from [104]); (**b**,**c**) *H.* (*Pseudoophonus*) *rufipes*; (**d**,**e**) *H.* (*Pseudoophonus*) *jureceki*; (**f**,**g**) *H.* (*Semiophonus*) *signaticornis*; (**h**) *H.* (*Pseudoophonus*) *eous*; (**i**) *H.* (*Pseudoophonus*) *roninus*; (**j**) *H.* (*Megapangus*) *caliginosus*; (**k**,**l**) *H.* (*Cryptophonus*) *grilli* (from [47], with modifications); (**m**,**n**) *H.* (*Anophonus*) *cyanopterus*; (**o**) *H.* (*Harpalus*) *affinis*; (**p**) *H.* (*Opadius*) *cordatus* (from [106]) (setae on mentum and submentum not shown).

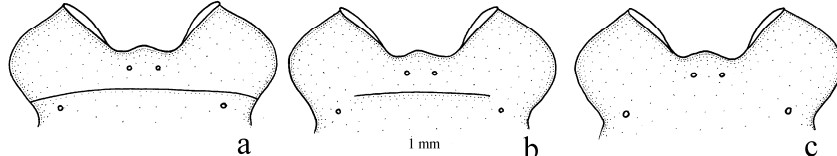

**Figure 4.** Mentum and submentum of *Harpalus* species (from [47]): (**a**,**b**) *H.* (*Cryptophonus*) *litigiosus*; (**c**) *H.* (*C.*) *tenebrosus*.

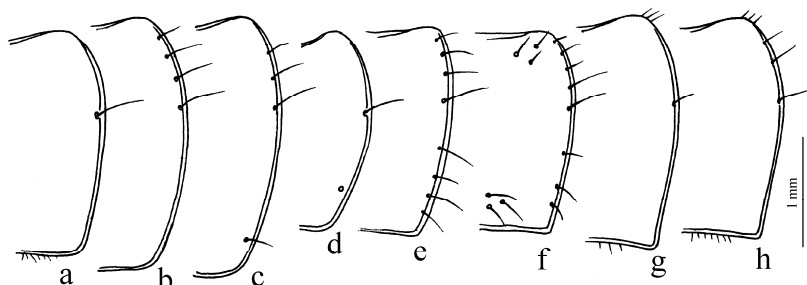

**Figure 5.** Chaetotaxy of pronotum of *Harpalus* species: (**a**) *H.* (*Cryptophonus*) *tenebrosus*; (**b**,**c**) *H.* (*Pseudoophonus*) *eous*; (**d**) *H.* (*Licinoderus*) *chobautianus*; (**e**,**f**) *H.* (*Artabas*) *punctatostriatus*; (**g**,**h**) *H.* (*Loboharpalus*) *rubefactus*.

The scutellum is always clearly visible.

The elytra are usually rounded at the sides, widest behind the middle, in some species almost parallel-sided or wedge-shaped. The base of the elytra is usually wider than the base of the pronotum, sometimes equal to it or narrower. The basal border is always complete, moderately wide or relatively narrow, connected to the lateral margin either at an angle or in a gentle arc. The shape of the humerus is varied from widely rounded to angular, almost rectangular; often, the humeral angle at the apex is developed as a denticle. The presence and size of the denticle is not related to the degree of development of the wings, as is the case in some other groups of beetles, and at least in many obligate wingless species, the denticle is well developed, while in flying species, it is completely absent. The lateral elytral margin is often sinuate apically (preapical sinuation). The elytral epipleuron usually only reaches the beginning of the preapical sinuation; in some species, it gradually narrows towards the apex and, not reaching it, gradually disappears, in others, before the preapical sinuation, it breaks rather steeply, bending at an angle to the lateral margin. In the latter case, a more or less sharp protrusion (preapical denticle or tooth) forms on the lateral margin, limiting the preapical sinuation from the outside. In a number of species, the lateral margin of the elytra in front of the apex, without forming a sinuation, is bent at an angle, so that the elytra at the apex look obliquely or almost transversely truncated.

Each elytron has eight striae that divide the surface into nine intervals. The striae are always complete, not ended before apex, superficial or more or less deepened (always deeper near apex), usually smooth, and in some species, finely punctate. An abbreviate (parascutellar) striole in *Harpalus*, as in all Harpalini, is actually the base of stria 1 (see [122]); it is always located in the actual interval 2 and connected basally with the base of stria 2; in most cases it is well developed, but in some species, it is greatly shortened or breaks up into several short segments; the apex of the parascutellar striole can be connected with stria 1, which actually consists of the real scutellar striole connected with the apical portion of stria 1. Depending on the depth of the elytral striae, the intervals can be flat or convex.

The elytral chaetotaxy deserves special attention, since the number and location of the setigerous pores (punctures) vary considerably within the genus (Figure 6).

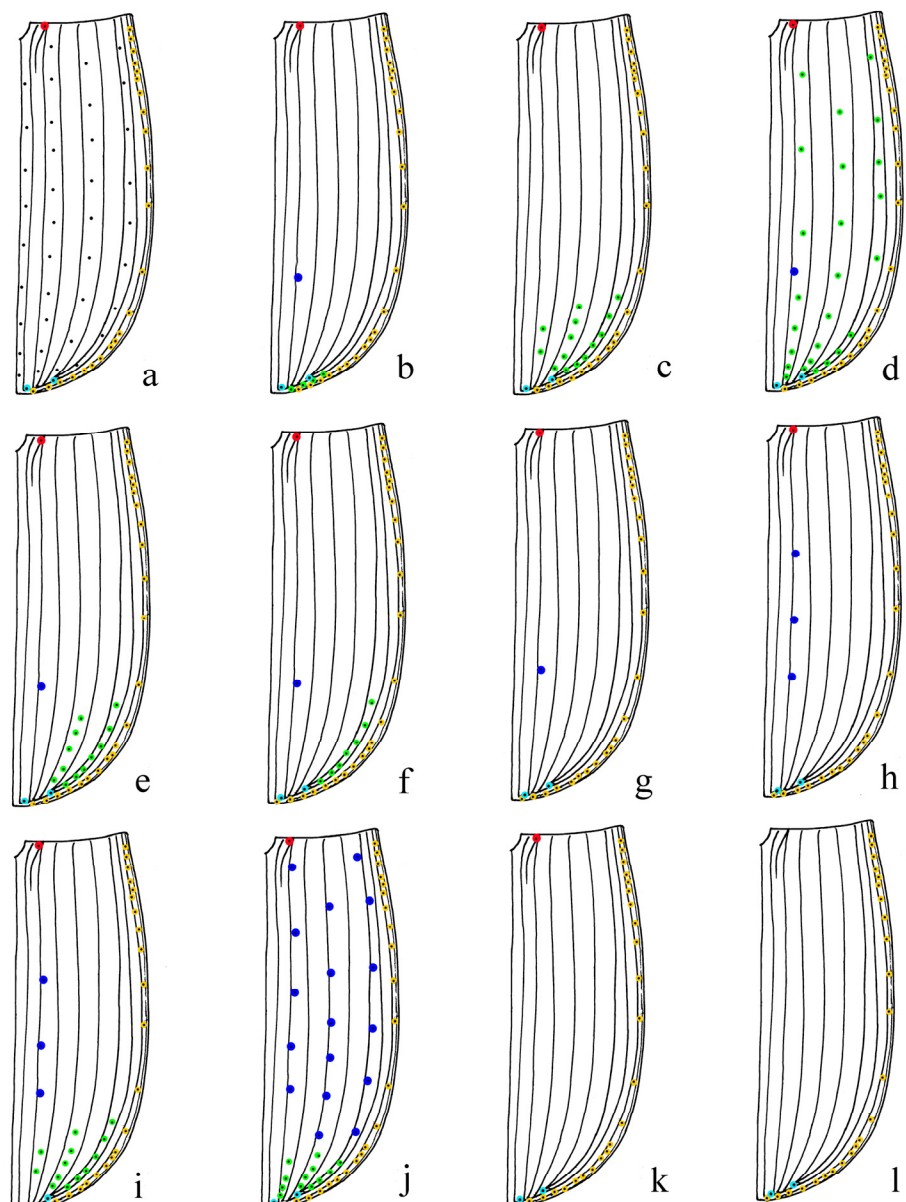

**Figure 6.** Chaetotaxy of elytron (right) of *Harpalus* species (scheme): (**a**) *H.* (*Cephalomorphus*) *capito*; (**b**) *H.* (*Cryptophonus*) *litigiosus*; (**c**) *H.* (*Cycloharpalus*) *pulvinatus*; (**d**) *H.* (*Pheuginus*) *davidianus*; (**e**) *H.* (*Baryharpalus*) *dimidiatus*; (**f**) *H.* (*Amblystus*) *rubripes*; (**g**) *H.* (*Proteonus*) *distinguendus*; (**h**) *H.* (*Hyloharpalus*) *laevipes*; (**i**) *H.* (*Brachyharpalus*) *autumnalis*; (**j**) *H.* (*Hypsinephus*) *salinus*; (**k**) *H.* (*Aristoharpalus*) *ingenuus*; (**l**) *H.* (*Actephilus*) *picipennis*; azure dots = pores in apical part of stria 7; yellow dots = umbilical pores of interval 9; red dots = basal (parascutellar) pores; blue dots = discal pores; green dots = preapical pores.

Several groups of setigerous pores can be distinguished:

1.  The setigerous pores in the apical part of stria 7 (Figure 6, azure dots). There are always two of them and they are located close to each other. One of them, preapical, is larger and is always clearly visible; the second, apical, usually very small, with a very short seta and visible only at high magnification (×50).

2.  The umbilical setigerous pores of interval 9, or the umbilicata series sensu Jeannel [123] (Figure 6, yellow dots). In *Harpalus*, they are characterized by a developed state: the pores are concentrated basally and apically, forming two groups. Usually, there are four to six basal and six to eight preapical pores. These pores look like nodules

extending from stria 8. In addition to these main marginal pores, there are many additional ones in interval 9. They are usually smaller, not associated with stria 8, and probably of secondary origin. In most species, additional pores are also concentrated basally and apically, but without a distinct gap between them. Additional pores largely mask the pattern of the distribution of the main marginal pores, and therefore the possibility of using the latter as a taxonomic feature is limited. In some taxa (for example, in *Cryptophonus* and *Zangoharpalus*), there are very few additional pores.

3.  The setigerous pore at the base of stria 2 (basal, or parascutellar pore) (Figure 6, red dots). It is present in the vast majority of species, but in some, it is constantly or occasionally absent.
4.  The discal setigerous pores (Figure 6, blue dots). They are located mainly in interval 3, less often also in intervals 5 and 7; in intervals 5 and 7, discal pores may be present only if there are pores in interval 3. The pores, as a rule, are connected with adjacent striae, and some may extend into adjacent intervals 2, 4 or 6. Most species have only one discal pore in interval 3 adjacent to stria 2 in the posterior third of the elytron (Figure 6b,d–g). Some species have several (up to eight) discal pores in interval 3 (Figure 6h,i), and only in *Hypsinephus*, discal pores are present in intervals 3, 5 and 7 (Figure 6j). On the other hand, in a number of species, discal pores are completely absent or their presence is not constant (Figure 6c,k,l).
5.  The preapical setigerous pores (Figure 6, green dots) (not to be confused with setigerous pores in the apical part of stria 7). They are present in many species in odd intervals, although in most species only in intervals 5 and 7 (more rarely also in interval 3), and in interval 5, pores may be present only if present in interval 7, and accordingly in interval 3 only if pores are present in interval 5 (Figure 6c–f,i,j). Some species of *Cryptophonus* have preapical pores in interval 8 (Figure 6b). Preapical pores are constantly present in some species and only present in some individuals in other species. In *H.* (*Artabas*) *suturangulus*, *H.* (*Artabas*) *szalliesi* and *H.* (*Pheuginus*) *davidianus*, the preapical pores often extend a considerable distance from the apex of the elytra, in some cases reaching the basal border (Figure 6d). In these species, such "preapical" pores resemble discal pores; however, in contrast to the discal pores, they are always located in the middle of the interval, without touching the striae, and are closer together towards the apex of the elytra. The independence of the discal and preapical pores is also evidenced by the fact that both *H.* (*Artabas*) *suturangulus* and *H.* (*Pheuginus*) *davidianus* also have a typical discal pore near stria 2 in interval 3.

The homology of the rows of setigerous pores on odd intervals (1, 3, 5 and 7) in *Cephalomorphus* (Figure 6a) is not clear. Since these pores are located in the middle of the intervals, they are unlikely to be homologous (identical) to the discal pores. Possibly, these pores are homologous to preapical pores of other *Harpalus* and pores on the elytral disc of many *Ophonus*.

Each type of pore (parascutellar, discal and preapical) varies independently, and their specific combination is often a reliable characteristic for species and also as evidence for membership in a particular supraspecific taxa. The disappearance of pores often correlates with small sizes. Discal pores are also often absent in burrowing forms.

In addition to these regular setigerous pores described above, the elytral intervals may also be covered with numerous punctures and short setae. Such punctation and pubescence, if present, in most cases are limited only to outer intervals and apices, rarely covering the entire surface of elytra. The punctation at the outer margins of the elytra is always denser and usually finer than in the middle part of the disc. Particularly sparse and coarse punctures occur in odd intervals. In some species, the setigerous punctures are organized in longitudinal rows along the striae. In a few species, the punctation and pubescence are limited to only a small area at the base of elytra on both sides of scutellum. In those species in which the basal edge of the pronotum bears a row of short setae, erect setae are often present on the basal border of the elytra. Such setae can be present in taxa with pubescent elytra and in taxa with glabrous elytra. The setae on the basal border of the elytra seem to

be functionally related to the setae on the basal edge of the pronotum. When the pronotum moves, they apparently come into contact with each other and thus register its position relative to the elytra.

Flight wings. Most species of *Harpalus* have fully developed and normally functioning flight wings of a structure common to ground beetles. In some species, the wings are more or less shortened or almost completely reduced. Winglessness is especially common among mountain and desert species. The obligatory reduction in the wings is typically accompanied by a shortening of the metepisterna. In some species, wing polymorphism is known, i.e., the presence of developed wings only in some individuals of the species, while their absence or significant reduction in others. In wing polymorphism, wing reduction usually does not cause a corresponding shortening of the metepisterna, and wingless individuals may not differ in appearance from winged individuals.

The intercoxal prosternal process is not bordered apically, but has several apical setae, and in some species, there are also several marginal setae at some distance from the apex. The anterior region and sides of the prosternum are always more or less punctate and pubescent; in some species, setae are also present in the medial part, including the prosternal process. In some species, the setae in the anterior part of the prosternum are rather long. Proepisterna are usually impunctate and glabrous, occasionally finely punctate and pubescent.

The shape of the metepisterna (Figure 7) is varied and largely depends on the degree of development of the wing muscles, since the main flight muscles are attached to them.

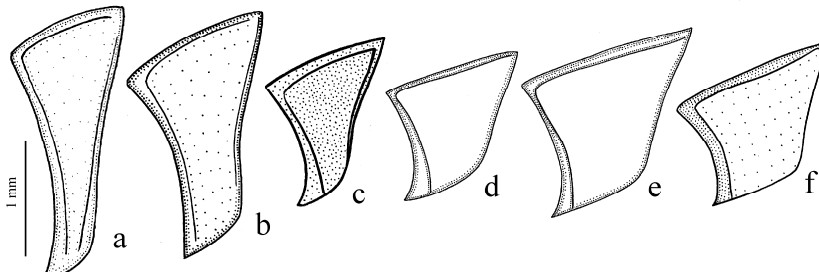

**Figure 7.** Left metepisternum of *Harpalus* species: (**a**) *H.* (*Cryptophonus*) *tenebrosus* (from [47]); (**b**) *H.* (*Harpalus*) *affinis* (from [86]); (**c**) *H.* (*Mesoharpalus*) *mitridati* (from [90]); (**d**) *H.* (*Aristoharpalus*) *ingenuus* (from [104]); (**e**) *H.* (*A.*) *arcuatus* (from [104]); (**f**) *H.* (*Brachyharpalus*) *reflexus*.

In most species with well-developed wings, the metepisterna are elongate (their inner margin longer than the anterior margin) and strongly narrowed posteriorly; in species with shortened wings, and especially in obligate wingless species, in which winglessness is a constant feature of the species, the metepisterna are shortened and less strongly narrowed posteriorly. With an extreme degree of reduction in the wings, the metepisterna are transverse and only slightly narrowed posteriorly.

Legs. The members of *Harpalus*, compared to most other ground beetles, have relatively short and thick legs. Shortened legs usually correlate with a robust body, and in species with a more slender body, the legs are also relatively long and thin.

The procoxae in most cases are without setae (sometimes with short and thin setae on the posterior margin), and the mesocoxae are with numerous short and thick setae. The metacoxae typically bear only two setigerous pores, one in the region of the posterior outer angle and the other more latero-anteriorly (Figure 8a). Only a few species also have an additional posteromedial pore (Figure 8c), which is located at mesal margin of metacoxa posteriorly and constantly present, for example, in Acinopi and Ophoni. Many species also have one or more additional setigerous pores in the medial part of the metacoxa (Figure 8b,d); more rarely, the entire surface of metacoxae is finely punctate and pubescent.

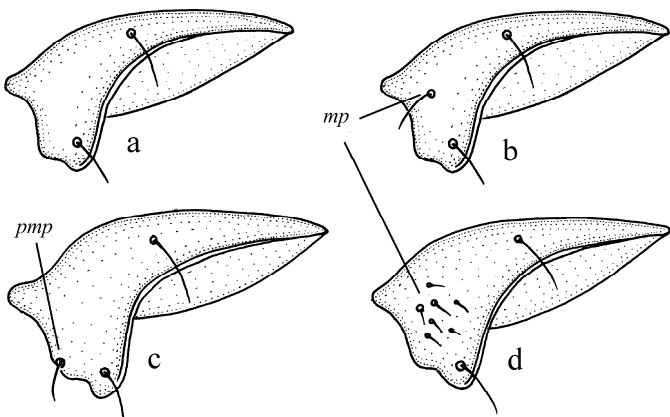

**Figure 8.** Left metacoxa of *Harpalus* species (scheme): (**a**) *H.* (*Ooistus*) *anxius*; (**b**) *H.* (*O.*) *amariformis*; (**c**) *H.* (*Semiophonus*) *signaticornis*; (**d**) *H.* (*Harpalus*) *affinis*; *pmp* = posteromedial pore; *mp* = medial pores.

Pro-, meso-, and metatrochanters bear one setigerous pore each; in some species, the metatrochanters also have several additional pores, which, together with the common pore, form one row along the posterior margin.

The femora are relatively thick, with numerous setigerous pores forming regular rows. The number of setae located ventrally along the posterior and anterior margins of the metafemora is often used as a taxonomic character. The number of setae along the anterior margin varies from zero to fifteen; there are three or more setae on the posterior margin (up to nineteen), and rarely only two setae.

The meso- and metatibiae are somewhat uniform within the genus, varying mainly in relative length and width.

The structure of the protibia (Figures 9 and 10a,b) includes many important characters both for species diagnosis and for elucidating the relationship between taxa within the genus. The apical spur is lanceolate in most species, but in some species of *Pseudoophonus*, it is dentate at the margins. The protibia usually has three to nine preapical spines on the outer margin (ten to sixteen spines in species of *Loboharpalus*) and one to three (in some taxa up to six) ventroapical spines forming a transverse row along the apical margin.

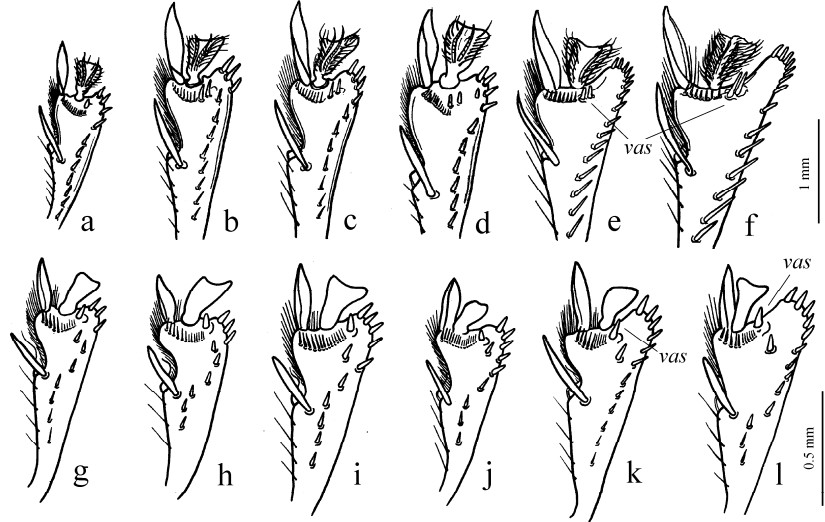

**Figure 9.** Left protibia of *Harpalus* species in ventral view (from [54,89], with modifications): (**a**) *H.* (*Haploharpalus*) *froelichi*; (**b,c**) *H.* (*H.*) *melaneus*; (**d**) *H.* (*H.*) *alajensis*; (**e**) *H.* (*H.*) *brevicornis*; (**f**) *H.* (*H.*) *hirtipes*; (**g**) *H.* (*Actephilus*) *masoreoides*; (**h**) *H.* (*A.*) *pumilus*; (**i**) *H.* (*A.*) *picipennis*; (**j**) *H.* (*A.*) *pusillus*; (**k**) *H.* (*A.*) *longipalmatus* (female); (**l**) same, male; *vas* = ventroapical spines.

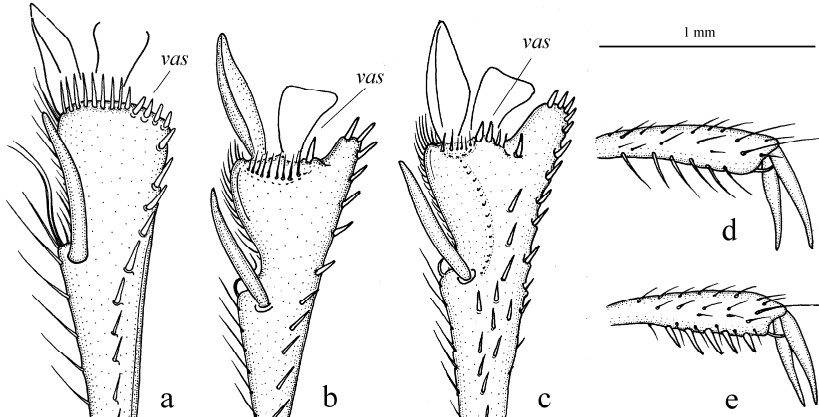

**Figure 10.** Left protibia in ventral view and tarsomere 5 in lateral view of *Harpalus* species: (**a**) *H.* (*Opadius*) *cordatus* (from [106]); (**b**) *H.* (*Acardystus*) *flavescens*; (**c**) *H.* (*Loboharpalus*) *platynotus*; (**d**) *H.* (*Pseudoophonus*) *rufipes* (from [113]); (**e**) *H.* (*Platus*) *calceatus* (from [113]); *vas* = ventroapical spines.

The protibiae of many species are adapted to burrowing. Adaptive changes affect mainly the apical part of the protibia, which in specialized forms is markedly widened and flattened, and its outer angle is elongated in the form of a blade (Figures 9f,l and 10b,c). A comparative morphological analysis shows that the process of specialization of the protibia for burrowing in different taxa of *Harpalus* proceeded in at least two different ways, with a significant similarity of the results (Figure 10a–c).

The first way is more common; already in the initial stages, the widening and flattening of the protibia is accompanied by a bending of the ventral longitudinal row of spines towards the outer margin (as in Figure 9h,i). In the next stage, the ventral row is connected to the preapical spines of the outer margin. This stage corresponds to the structure of the protibia, for example, in *H.* (*Actephilus*) *pusillus* (Figure 9j). With further transformations, the spines of the ventral row between the junction of two rows and the apical ventroapical spines atrophy, as a result of which a single row is formed from two rows of spines, with its basal part on the ventral surface of protibia and apical part on the outer margin. This structure of the protibia is observed, for example, in *H.* (*Actephilus*) *longipalmatus*, *H.* (*Acardystus*) *flavescens* and *H.* (*Opadius*) *cordatus*, as well as in most species of the subgenus *Haploharpalus*, and the modification of the rows of spines is not necessarily accompanied by transformation of the outer angle of the protibia in the form of a blade (Figures 9a,c–f,k,l and 10a,b). The process of specialization of the protibia for burrowing in the subgenus *Loboharpalus* (Figure 10c) followed a different path. The widening and flattening of the apical part of protibia did not correlate with the corresponding modification of the rows of spines, so that the spines on the ventral surface of the protibia and on its outer margin retained a separate position. The protibia of *Plectralidus* probably represent another way of specialization; its ventroapical margin forms a lamella extended downward.

The tarsi are relatively short. Metatarsomere 1 is slightly longer than metatarsomere 2, but always shorter than metatarsomeres 2 and 3 combined. Numerous setae and spines are located on the sides and on the lower surface of the tarsomeres. Tarsomere 5 usually has only thin setae ventrally (Figure 10d), but *Platus* also has two longitudinal rows of spines (Figure 10e). The dorsal side of tarsi in most species is impunctate and glabrous, sometimes with scattered punctures and fine short setae, and in some taxa, for example in *Pseudoophonus*, with dense punctures and pubescence. Claws are strongly curved, never serrated.

The abdomen has a structure common to most ground beetles; its ventral side is formed by six visible sternites (ventrites) (corresponding to true segments III–VIII), the dorsal side is formed by eight tergites (II–IX). The visible abdominal sternites 3, 4 and 5 bear two setigerous pores at the posterior margin; the last visible sternite bears four marginal setigerous pores in both sexes (two on each side). In addition, all or only the visible sternites

3–5 may be covered with fairly numerous additional (accessory) setae. In some species, the additional setae are very short and form a continuous dense pubescence; often very short and fine setae are present between metacoxae and at base of visible sternite 3 medially.

Male genitalia. The aedeagus is rather strongly sclerotized; it is asymmetric and consists of the median lobe (penis) and a pair of parameres attached to it. Since the aedeagus lies in the abdomen on the right side, its right paramere is always smaller and narrower than the left one. The shape of the parameres is fairly constant not only within the genus *Harpalus* but also within the Harpalini.

The median lobe (Figure 11) is tubular, more or less strongly curved ventrally; its base is distinctly enlarged, forming the basal bulb delimited from the rest of the median lobe by a constriction. On the ventral side of the basal bulb, there is a process, to which the parameres are attached, and proximally from the process, also on the ventral side, there is a more or less wide opening (basal orifice, or ostium), into which the ejaculatory canal enters. The apical part of the ventral surface of the medial lobe of many species has short spines; in some species, the surface is serrate. The median lobe at some distance from the apex has an oval-shaped apical orifice. In most species, the apical orifice is more or less shifted to the left, but in *H.* (*Harpalobius*) *fuscipalpis*, *H.* (*Nephoharpalus*) *pallidipennis*, *H.* (*Smirnovia*) *kandaharensis*, in many species of *Artabas* and in some Afrotropical species, it is located almost dorsally. The apical orifice in the proximal direction reaches approximately the middle of the penis, but sometimes it occupies a much larger area, almost reaching the basal bulb; it is rarely reduced to a small oval opening located apically (Figure 11e,f).

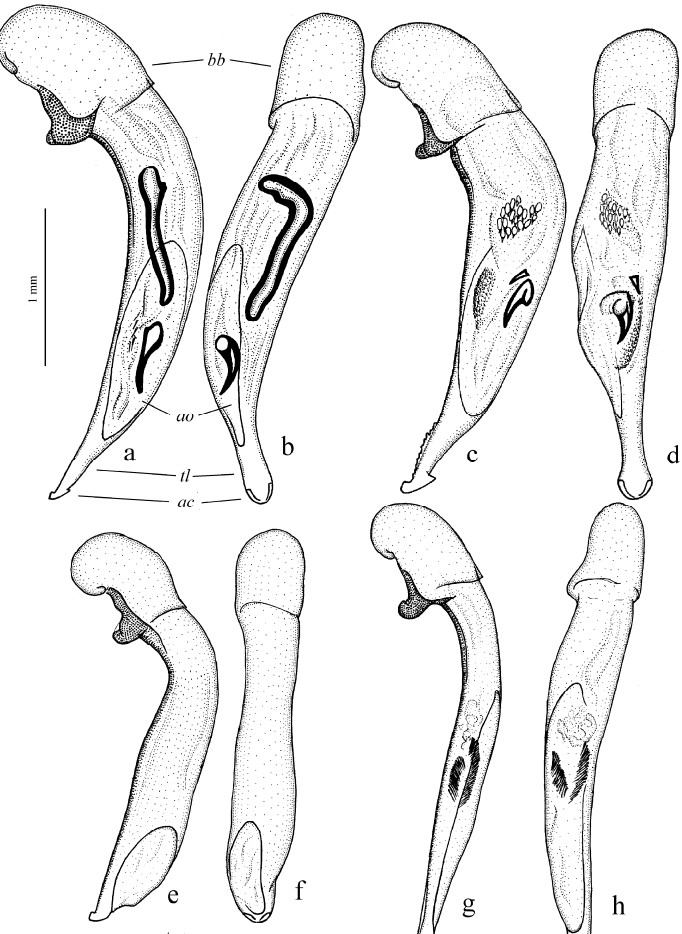

**Figure 11.** Median lobe of aedeagus of *Harpalus* species, lateral and dorsal views: (**a**,**b**) *H.* (*Cryptophonus*) *melancholicus* (from [47], with modifications); (**c**,**d**) *H.* (*Amblystus*) *rufipalpis*; (**e**,**f**) *H.* (*Artabas*) *punctatostriatus*; (**g**,**h**) *H.* (*Nephoharpalus*) *pallidipennis*; *ac* = apical capitulum; *ao* = apical orifice; *bb* = basal bulb; *tl* = terminal lamella.

The apical part of the median lobe from the apex to the distal edge of the apical orifice is the terminal lamella. The latter is very diverse in shape; it can be very short and wide, flattened dorsoventrally, or more or less narrow and elongated. The apex of the terminal lamella is usually thickened, forming the apical capitulum (or apical disc) (Figure 12). The presence of an apical capitulum is characteristic of most *Harpalus*, but it is absent in *Semiophonus*, *Zangoharpalus*, *Cephalophonus* and some *Pseudoophonus* (Figure 12o–t); it is also almost undeveloped in *Loboharpalus* and in many Afrotropical species. The apical capitulum typically has the appearance of a horseshoe-shaped thickening along the apical margin of the terminal lamella ("serpent head" in Jeannel's terminology [14]) (Figure 12u–x). The lateral parts of such a capitulum usually protrude dorsally on both sides in the form of two denticles. The middle part of the capitulum almost always protrudes ventrally, is often pointed, and in many species forms a small hook at the apex. All other variants of the apical capitulum found in *Harpalus* can be deduced from this basic type by the uneven growth of its various parts. In a number of species, the dorsal denticles fuse with each other to form a single protrusion on the dorsal side of the median lobe, resulting in a disc-shaped (discoidal) apical capitulum (Figure 13).

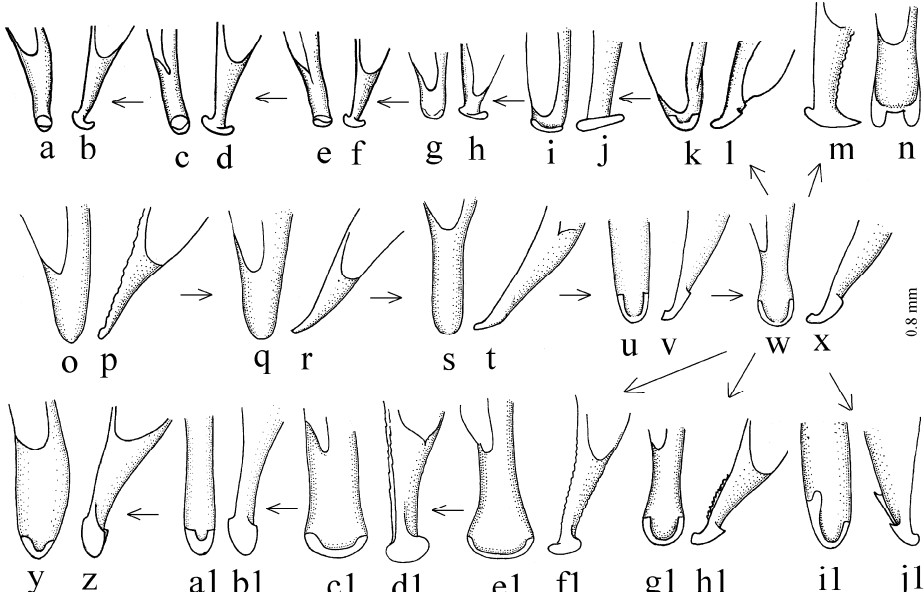

**Figure 12.** Apical capitulum of aedeagus of *Harpalus* species, dorsal (**left**) and lateral (**right**) views: (**a,b**) *H.* (*Mesoharpalus*) *zhdankoi*; (**c,d**) *H.* (*M.*) *gisellae*; (**e,f**) *H.* (*M.*) *ovtshinnikovi*; (**g,h**) *H.* (*Nephoharpalus*) *smaragdinus*; (**i,j**) *H.* (*N.*) *pallidipennis*; (**k,l**) *H.* (*Artabas*) *dispar*; (**m,n**) *H.* (*Heteroharpalus*) *tiridates*; (**o,p**) *H.* (*Zangoharpalus*) *praticola*; (**q,r**) *H.* (*Cephalophonus*) *cephalotes*; (**s,t**) *H.* (*Cryptophonus*) *tenebrosus*; (**u,v**) *H.* (*Homaloharpalus*) *modestus*; (**w,x**) *H.* (*Amblystus*) *sulphuripes*; (**y,z**) *H.* (*Heteroharpalus*) *tithonus*; (**a1,b1**) *H.* (*H.*) *metallinus*; (**c1,d1**) *H.* (*Amblystus*) *morvani*; (**e1,f1**) *H.* (*A.*) *indicola shogranensis*; (**g1,h1**) *H.* (*A.*) *rufipalpis*; (**i1,j1**) *H.* (*Hypsinephus*) *lumbaris*. The arrows show the possible direction of evolutionary transition.

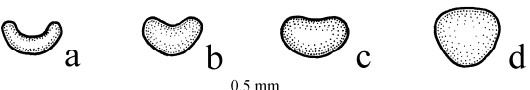

**Figure 13.** Apical capitulum of aedeagus of *Harpalus* species, caudal view (from [104]): (**a**) *H.* (*Baryharpalus*) *karamani*; (**b**) *H.* (*B.*) *caspius*; (**c**) *H.* (*B.*) *murzini*; (**d**) *H.* (*B.*) *dimidiatus*.

The apical orifice leads to the internal sac, which is attached to the edges of the apical orifice and is located inside the median lobe. During copulation, it turns outward. The membranous walls of the internal sac in the inverted state close the apical orifice from the inside, forming a membranous surface (preputial field). The walls of the internal

sac from the inside bear sclerotized spines developed to varying degrees, which together form the so-called armament of the internal sac. When the internal sac is everted, they are sticking out. Compared to other ground beetles, the armament of the internal sac of *Harpalus* is generally well developed. Spines can be flat or oval in cross section, very small, microscopic, or quite large, in the form of heavy spines or teeth, often with a sclerotized sole at the base (see e.g., Figure 6c,d), as first described by Antoine [40]. The spines are often joined together, forming separate groups or spiny patches (fields), or grouped into more complex structures. However, *Harpalus* does not have more than two, more or less large, single (separate) spines [46]. Some taxa are characterized by the complete absence of any sclerotized elements in the internal sac.

The evolution of the sclerotized structures of the internal sac in *Harpalus* followed both the path of gradual complication and the path of reduction, up to their complete disappearance. Among *Harpalus*, there are examples when, within the same group of closely related species characterized by a well-developed armament of the internal sac, there are also members with its complete absence. It is interesting to note that, for example, in *Haploharpalus*, sympatric species have the abundant armament, while in species that have been isolated geographically for a long time (*H. melaneus*, *H. alajensis* and *H. alpivagus*), the armament of the internal sac is somewhat reduced [54]. Presumably, the development of specific sclerotized elements in the internal sac is one of the mechanisms preventing hybridization between representatives of closely related species.

Female genitalia. In the females of *Harpalus*, as in most other ground beetles, the genitalia are part of the ovipositor and consist of the paired laterotergites (hemisternites), gonosubcoxites (basal stylomeres) and gonocoxites (apical stylomeres) (Figure 14).

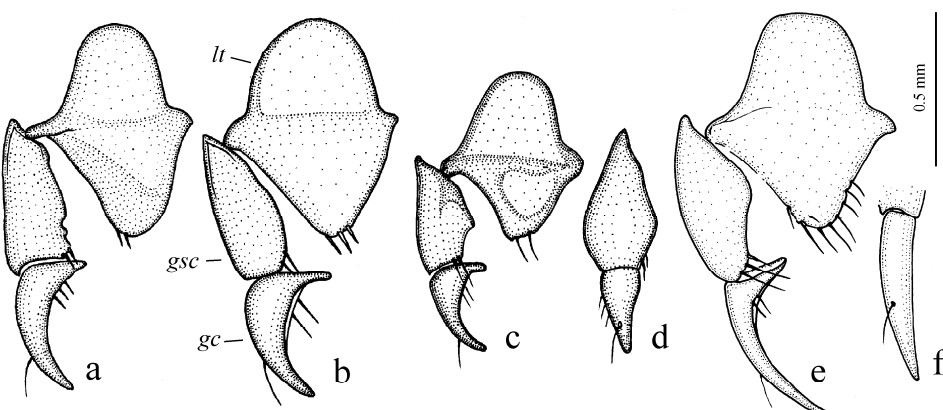

**Figure 14.** Female genitalia of *Harpalus* species, ventral view (**d**,**f**, lateral view): (**a**) *H.* (*Asioharpalus*) *parasinuatus* (from [112]); (**b**) *H.* (*Homaloharpalus*) *vernicosus* (from [112]); (**c**,**d**) *H.* (*Actephilus*) *minutulus* (from [111]); (**e**,**f**) *H.* (*Loboharpalus*) *platynotus*; *lt* = laterotergite; *gsc* = gonosubcoxite; *gs* = gonocoxite.

The gonosubcoxite has the appearance of an elongated plate, while the gonocoxite is pointed at the apex and resembles a beak or a claw in shape. In most species, the gonocoxite is short and rather wide at the base; in species of *Loboharpalus* and *Acardystus*, it is relatively narrow and long (Figure 14e,f). The laterotergite usually has several (two to eight) short setae apically; the gonosubcoxite has somewhat fewer (two to four) short setae on the outer distal corner; and the gonocoxite is with fine setae on the outer margin. Like most other Harpalini, in *Harpalus*, the gonocoxite has two characteristic fine, closely inserted setae (nematiform setae) on its convex inner margin near the apex.

Sexual dimorphism (Figure 15). This is common to all *Harpalus* and expressed in different parts of the body. It can manifest itself in color, size, shape and structural features. Almost all secondary sexual characters, according to their functional purpose, can be combined into three main groups: (A) characters associated with the development of adaptations that contribute to successful copulation; (B) characters associated with the de-

velopment of adaptations to oviposition; and (C) features associated with the development of eggs in the body of females.

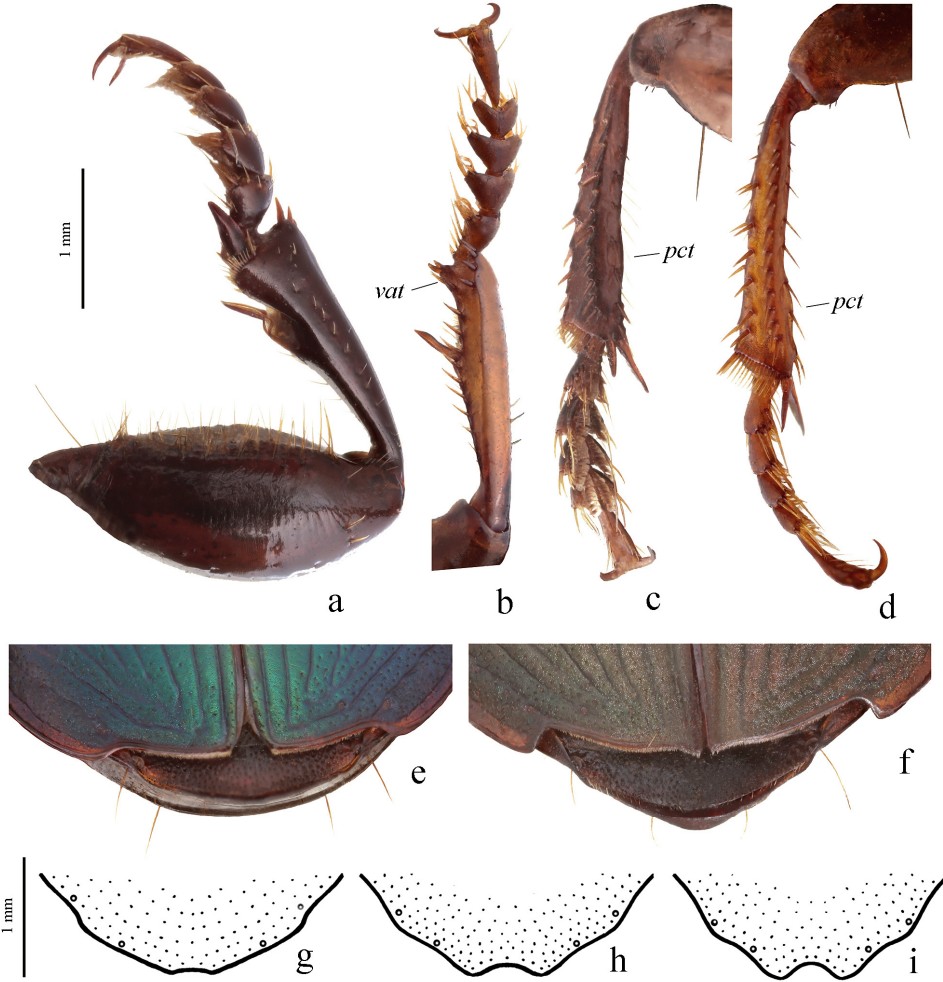

**Figure 15.** Sexual dimorphism in *Harpalus* species: (**a**) fore leg of male of *H.* (*Heteroharpalus*) *metallinus*; (**b**) protibia and protarsus of male of *H.* (*Harpalus*) *affinis*; (**c**) mesotibia and mesotarsus of male of *H.* (*Harpalus*) *affinis*; (**d**) same of *H.* (*Hyloharpalus*) *laevipes*; (**e**) apices of elytra and abdomen of male of *H.* (*Harpalus*) *affinis*; (**f**) same, female; (**g**) apex of last abdominal sternite of male of *H.* (*H.*) *glasunovi* (ventral view); (**h**) same, *H.* (*H.*) *affinis*; (**i**) same, *H.* (*H.*) *caeruleatus*; *vat* = ventroapical tubercle; *pct* = preapical callous thickening.

(A) Characters associated with the development of morphological adaptations that contribute to successful copulation. They are typical mainly for males.

(i) Features associated with the development of facilities used to search for a female.

Males of all species of *Harpalus* differ from females in their greater development of the sense organs; in particular, they have longer antennae.

(ii) Features associated with the development of facilities for grasping the female during copulation.

This is the most common and visible type of secondary sexual characters in *Harpalus*. In males, these features are developed on the legs. In the vast majority of species, the pro- and mesotarsomeres 1–4 of males are widened and bear ventral adhesive scales, widened at the apex and arranged in two longitudinal rows; tarsomere 1 is usually less widened than subsequent tarsomeres and bears fewer adhesive scales, often located only apically; in some species, mesotarsomere 1 is not widened and lacks adhesive scales. In most species, the male femora, especially of fore and middle legs, are markedly thickened; in species of the *Heteroharpalus* **subg. n.**, the profemora, in addition, have a dense brush of setae along

the inner (ventral) margin (Figure 15a). In most members of the *Harpalus*, *Caloharpalus* and *Hyloharpalus* subgroups, the mesotibiae have a small preapical callous thickening (swelling) on inner (ventral) margin (Figure 15c,d). Differences in the form of the ventroapical tubercle of the protibia, which is always more strongly developed in males (Figure 15b) than in females, probably belong to the same type of secondary sexual characters. In females, the elytra, and often the pronotum, are always characterized by a more strongly developed microculture, which probably facilitates the fixation of the male on the female during copulation ("provide a better surface for grasping") [16] (p. 227). The microsculpture on the elytra of the female is always distinct and is present even when it is absent or little evident in the male. The features of the microsculpture are associated with sexual dimorphism in the coloration and intensity of the sheen of the dorsum (in almost all species, females are less glossy).

(iii) Features associated with the development of facilities that contribute to the correct orientation of the median lobe of aedeagus during copulation.

This type includes the formation of an emargination at the apex of the last abdominal sternite in males of the subgenera *Harpalus*, *Harpalophonus*, *Harpalotypsis* and some others (Figure 15g–i). During copulation, when the median lobe is pushed outward and rotated down 90°, its base enters this emargination.

(B) Characters associated with the development of adaptations to oviposition.

Found only in females, as a rule, species, that lay eggs in a dense substrate (soil). When laying eggs, the female immerses the apex of the abdomen into the soil, while the apex of the elytra rests on the surface of the soil, and the abdomen itself is extended. The introduction of the abdomen into the soil is carried out due to the work of the gonocoxites; the legs and apex of the elytra provide the support and adhesion to the soil necessary for this. Therefore, in females, the sutural angle of the elytra is usually sharper than in males and often more or less strongly elongated in the form of a spine. An additional support for the working gonocoxites is also provided by the apex of the last abdominal sternite, which abuts against the wall of the formed hollow. The apex of the last abdominal sternite in females is always somewhat thicker than in males and often angular; in some taxa, this thickening is rather strongly developed (Figure 15f). In females of *H.* (*Cephalophonus*) *cephalotes* and *H.* (*Megapangus*) *caliginosus*, the apex of the anal tergite is pointed (Figure 16).

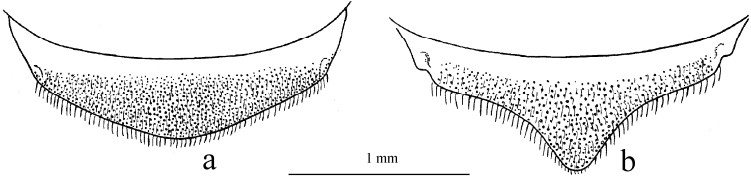

**Figure 16.** Anal tergite of *Harpalus* (*Cephalophonus*) *cephalotes* (from [104]): (**a**) male; (**b**) female.

The formation of a preapical sinuation of the elytra, which is always more developed in females and often has a denticle or acute tooth at the base, is probably also associated with the a support-fixation function of the elytra (Figure 15f). On the one hand, the preapical sinuation makes the apex of the elytra narrower and sharper; on the other hand, the formed marginal angulations at base of the preapical sinuations provide additional support and fixation on the sides of the elytra. In females of some species, the apical spurs of the metatibia are spoon-shaped, which is also probably associated with an increase in the support-fixation function of these spurs. It is also possible that females use the thickened apex of the last abdominal sternite together with the widened apical spurs of metatibiae for burrowing after oviposition.

(C) Features associated with the development of eggs in the abdomen of females.

Sexual dimorphism of this type is manifested in different body proportions in males and females. Thus, the elytra of females in almost all species of *Harpalus* are relatively longer and wider than those of males.

Some manifestations of sexual dimorphism in *Harpalus* still require explanation. Thus, it remains unclear what causes the differences in the shape of the outer angle of the protibia with fossorial adaptations in males and females of some species, for example, in *Haploharpalus* and *Actephilus* (Figure 9k,l). The outer angle of the protibia in males of these subgenera are usually more elongated than in females; this is especially noticeable in *H.* (*Haploharpalus*) *zabroides*; as a result, the fore legs of males are more adapted to digging. Another example of unclear sexual dimorphism is that females in most *Harpalus* species have relatively larger heads, which is obviously associated with the development of more massive muscles to power the mouthparts. The reason for this may be both the greater consumption of roughage food by females, such as hard-coated seeds necessary for egg maturation, and the use of the head by females when digging a hole before laying eggs or when creating a chamber with food reserves.

3.1.3. Composition and Distribution

The genus *Harpalus* includes over 400 described species. According to the recent data [8,55], it ranges over the whole Holarctic region from the southern tundra in the north to the Sahara, Himalayas and northern Mexico in the south; the species are also distributed in east and south Africa, and a few Palaearctic species of the subgenera *Pseudoophonus* and *Zangoharpalus* enter the mountainous areas of the northern Oriental region within India, Myanmar (south to Tenasserim), northern Laos, northern Thailand and northern Vietnam, but species endemic to the Oriental region are not known. The range of the genus comprises the Azores, Canaries, Madeira, Cabo Verde, Saint Helena, Reunion, Taiwan, Hainan, Ryukyus and Aogashima islands. In Mexico, the species of *Harpalus* occur in temperate regions, south to the Trans-Volcanic Sierra in the vicinity of Mexico City [80] (Figure 17). Several Palaearctic and Afrotropical species were introduced to Australia and New Zealand [55,124–126]. The species described in the past as *Harpalus* from areas outside the Holarctic and Afrotropical regions have all been found to belong to other genera.

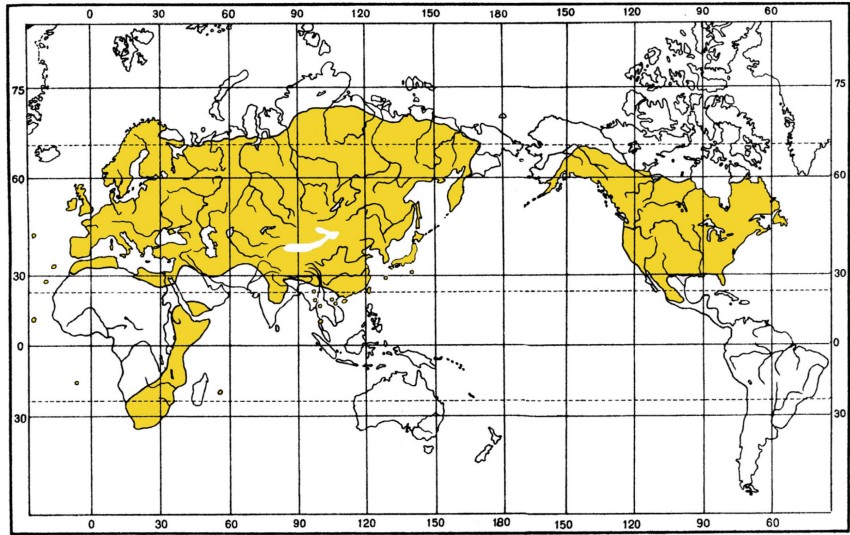

**Figure 17.** Distribution of the genus *Harpalus*.

The genus is divided here into 70 subgenera combined into 19 subgroups and ten groups.

*3.2. Key to Subgenus Groups of Harpalus*

1. Pronotum not bordered basally. Body densely pubescent, including labrum, tempora, external scrobe of mandibles, antennomeres 1 and 2 and tarsi dorsally . . . . . . . . . . . . . . . . . . . ...................... . . . . . . ....... . . . . . . ..... . ....... . . *Cephalophonus* group (subg. *Cephalophonus*).

- Pronotum bordered basally, throughout or partly; if not bordered, then at least head glabrous dorsally; body either pubescent or glabrous, antennae densely pubescent beginning from antennomere 3 . . . . . . . . . . . . . . . . . . . . . . . . . . . . . . . . . . . . . . . . . . . . . . . . . . ...... 2.

2. Mentum and submentum fused completely or only medially, rarely (in some specimens of *H. litigiosus*) separated by complete transverse suture (Figure 4), but in this case elytral interval 8 with a row of setigerous pores (sometimes only one pore) before apex. Labial basal palpomere with distinct oblique carina ventrally (Figure 3k,l). Mesotarsomere 1 in male not dilated and without adhesive scales ventrally, rarely (in *H. melancholicus*) with a pair of scales at apex, but in this case elytral interval 8 with a row of pores (sometimes only one pore) before apex . . . . . . . . . . . . . . . . . . . . . . . . . . . . . . . . . . . . . . . . . . . . . . . . . . . . . . . . . . . . . . . . . . . . . . . . . . . . . *Cryptophonus* group (subg. *Cryptophonus*).

- Mentum and submentum separated by complete, transverse suture (as in Figure 3b,d,f,m,o,p (occasionally fused in some *Afroharpalus* **subg. n.**). Elytra without pores in elytral interval 8 apically. Labial basal palpomere usually without oblique carina ventrally (as in Figure 3b,c,f,g,m–p), rarely (in *Megapangus* and some species of *Pseudoophonus*) with oblique (as in Figure 3d,e,i,j) or straight carina (as in Figure 3h). Mesotarsomere 1 dilated or not dilated, with or without adhesive scales ventrally . . . . . . . . . . . . . . . . . . . . . . . . . . . . . . . . . 3.

3. Metacoxa with a posteromedial setigerous pore (Figure 8c). Tempora distinctly pubescent, elytra densely punctate and pubescent . . . . . . . . . . . . . . . . . . . . . . . . . . . . . . . . . . . . . . . . . . . . . . . . . . . . . . . . . . . . . . . . . . . . . . . . . . . . *Semiophonus* group (subg. *Semiophonus*).

- Metacoxa without posteromedial setigerous pore (Figure 8a,b,d); if setigerous posteromedial pore present (in some *Afroharpalus* **subg. n.** and *Hypsinephus*), then tempora and elytra glabrous; in other cases, tempora and elytra pubescent or glabrous . . . . . . . . . . . 4.

4. Tarsi more or less densely setose dorsally, rarely (in some North American species) metatarsi almost glabrous. Protibia with two to four ventroapical spines arranged in a transverse row. Last visible abdominal sternite more or less rounded along posterior margin; its apex in males without distinct emargination, in females, at most only hardly swollen. Body black or brown, without metallic tinge. . . . . . . . . . . . . . . . *Pseudoophonus* group.

- Tarsi glabrous dorsally; if tarsi setose, then either protibia with one ventroapical spine (as in Figure 9g) or apex of last abdominal sternite with pronounced sexual dimorphism: in males, with a distinct emargination; in females notably swollen and slightly angularly expanded (as in Figure 15f), and body often with a metallic tinge on dorsum; if tarsi glabrous, number of ventroapical spines of protibia, shape of last visible abdominal sternite and coloration various. . . . . . . . . . . . . . . . . . . . . . . . . . . . . . . . . . . . . . . . . . . . . . . . . . . . . . . . . . . . . . . . . . . . . . . . . . . . . 5.

5. Protibia with four to six ventroapical spines arranged in a transverse row (rarely with three spines, for example in some specimens of North American *Opadius*, but in this case preapical spines on outer margin of protibia arranged in a single row together with spines on ventral surface of tibia) (as in Figure 10a) . . . . . . . . . . . . . . . . . . . . . . . . . . . . . ....... 6.

- Protibia with one to three ventroapical spines (occasionally four in North American *H. fraternus* and the West Mediterranean *H. numidicus*); if protibia with three or four ventroapical spines, then preapical spines on outer margin of protibia isolated from spines on ventral side of tibia and not arranged with them in a single row (as in Figures 9b and 10c) . . . . . . . . . . . . . . . . . . . . . . . . . . . . . . . . . . . . . . . . . . . . . . . . . . . . . . . . . . . . . . . . . . . . . . . . . . . . 8.

6. Head distinctly punctate dorsally. Elytra with one discal pore in interval 3 and with short row of preapical setigerous pores at least in intervals 5 and 7 (as in Figure 6e). Two penultimate abdominal sternites with numerous additional long setae . . . . . . . . . . . . . . . . . . . . . . . . . . . . . . . . . . . . . . . . . . . . . . . . . . . . . . . . . . ...... *Glanodes* group.

- Head impunctate dorsally. Elytra without discal pore in interval 3 and at most with a few preapical setigerous pores in interval 7. Two penultimate abdominal sternites glabrous or with a few rather short additional setae. . . . . . . . . . . . . . . . . . . . . . . . . . . . . . . . . . .......... 7.

7. Protibia with seven to nine, rarely six, preapical spines on outer margin. Elytral preapical sinuation very shallow or absent . . . . . . ..... *Megapangus* group (subg. *Megapangus*)

- Protibia with four or five preapical spines on outer margin. Elytral preapical sinuation deep, often with a denticle at its base . . . . . . . . ............. *Plectralidus* group (subg. *Plectralidus*).

8. Protibia (Figure 10c) strongly flattened apically, with at least ten preapical spines on outer margin isolated from spines on ventral surface of tibia and not formed with them a single row. Mesotarsomere 1 of male not dilated and without adhesive scales ventrally . . . . . . . . . . . . . . . . . . . . . . . . . . . . . . ..... . . . .......... *Loboharpalus* group (subg. *Loboharpalus*).

- Protibia flattened or not, with a smaller number of preapical spines on outer margin or preapical spines on outer margin gone to ventral surface of tibia and forming a single row with spines of ventral surface. Mesotarsomere 1 of male dilated or not dilated, with or without adhesive scales ventrally . . . . . . . . . . . . . . . . . . . ..... . . . . . . . . . . . . . . . . . . ............. 9.

9. Median lobe of aedeagus with a wide and thin terminal lamella, strongly flattened at apex and without apical capitulum (as in Figures 12o,p and 22e,f). Mesotarsomere 1 of male without adhesive vestiture ventrally, at most (sometimes in *H. praticola*) with a pair of small adhesive scales apically, but in this case, elytral microsculpture consisting of thin transverse lines . . . . .... . . . ... .... . ................... . . . . . *Zangoharpalus* group (subg. *Zangoharpalus*).

- Shape of terminal lamella of median lobe of aedeagus various, but with more or less developed apical capitulum (as in Figure 12a–n,u–j1); in many *Afroharpalus* **subg. n.**, without apical capitulum (as in Figure 24c,d). Mesotarsomere 1 in male, in most cases, with adhesive vestiture ventrally; very rarely, in some Palaearctic species (for example, in *H. saxicola* and some specimens of *H. angulatus*), adhesive vestiture on mesotarsomere 1 strongly reduced or absent. Elytral microsculpture, if present, consisting of isodiametric, sometimes weakly transverse meshes . . . . . . . . . . . . . . . . ..... . . . . . . . . . . ...... *Harpalus* group.

*3.3. Subgeneric Classification*

3.3.1. *Cephalophonus* Group

Diagnosis. Same as for the subgenus.

Composition and distribution. The monobasic group, including only one western Palaearctic subgenus.

Subgenus *Cephalophonus* Ganglbauer, 1892

*Cephalophonus* Ganglbauer, 1892 [36] (pp. 340, 345) (as a subgenus of *Ophonus* Dejean, 1821). Type species: *Harpalus cephalotes* Fairmaire et Laboulbène, 1875, by monotypy.

Diagnosis. Large size (length 12.0–14.0 mm). Body moderately convex, elongate, brownish yellow or reddish brown to black, without metallic luster. Surface densely punctate and pubescent, including labrum, tempora, external scrobe of mandibles, basal antennomeres and tarsi dorsally. Mentum and submentum separated by complete transverse suture; labial basal palpomere more or less cylindrical, without carina on ventral side (Figure 3a). Pronotum not bordered along basal margin, with one lateral seta on each side. Elytra without discal setigerous pores (at least pores invisible against background of dense punctation of intervals); preapical sinuation present, not deep. Metacoxa without posteromedial setigerous pore. Protibia with two ventroapical spines and with three to five preapical spines on outer margin, isolated from spines on ventral surface of tibia; ventroapical tubercle in male not developed; apical spur of protibia simple, lanceolate. Tarsomere 5 with thin setae ventrally; male mesotarsomere 1 with adhesive scales ventrally. Apex of last visible abdominal sternite slightly emarginate in male and pointed in female (Figure 16). Last visible tergite in male angularly rounded apically, in female, emarginate bilaterally, with apex projecting angularly posteriorly and narrowly rounded. Terminal lamella of median lobe of aedeagus flat, without apical capitulum; internal sac with short and broad spine at apex of median lobe (Figure 18a,b).

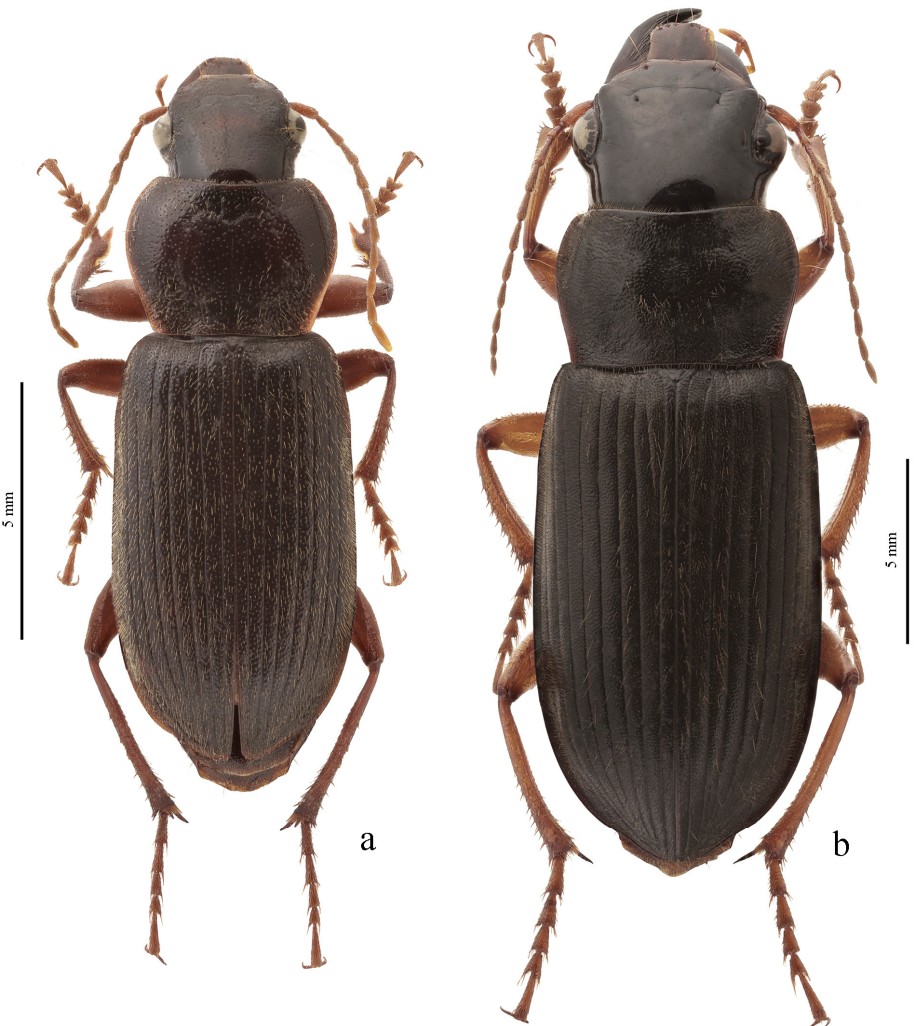

**Figure 18.** Habitus of *Harpalus* species: (**a**) *H.* (*Cephalophonus*) cephalotes; (**b**) *H.* (*Cephalomorphus*) capito.

Composition and distribution. The subgenus includes only *H. cephalotes* Fairmaire et Laboulbène, 1875 (Figures 18a and 19a,b) with two subspecies from southern Europe and west Asia: the nominotypical one and *H. c. somcheticus* Schauberger, 1933.

Ecology. The single representative of this subgenus occurs in arid and semiarid areas, in open habitats, often on saline soils.

Remarks. Because of the densely punctate and pubescent body, *Cephalophonus* has been described as a subgenus of the genus *Ophonus*. Until recently, most authors also included this monotypical taxon in the genus *Ophonus* (e.g., [42,127]). However, *Cephalophonus* has all the diagnostic features of the genus *Harpalus* [24,104,128].

The monobasic *Cephalophonus* group belongs to the pseudoophonoid subgeneric complex (phylogenetic stock), which also comprises the subgenera *Cephalomorphus*, *Pseudoophonus*, *Platus*, *Megapangus* and *Plectralidus* [8,104,113,128]. The species of this complex are characterized by two to six ventroapical spines on protibia arranged in a transverse row and generally by the absence of discal setigerous pore on interval 3 (in *Cephalomorphus*, with rows of pores in the middle of each interval 1, 3, 5 and 7 along their entire length). *Cephalophonus* is most similar to some members of the *Pseudoophonus* group in having body and tarsi dorsally punctate and pubescent, pronotal basal margin not bordered (as in *Cephalomorphus*), elytral interval 3 without dorsal setigerous pore and median lobe of aedeagus without or with a hardly recognizable apical capitulum. *Cephalophonus* easily differs from them as well as from other congeners in having distinctly punctate and pubescent head, setose basal antennomeres and modified anal sternum and tergum. The latter character is known also

for one of the two species of *Megapangus* (*H. caliginosus*), but this species has body and tarsi dorsally glabrous, and antennae pubescent from antennomere 3.

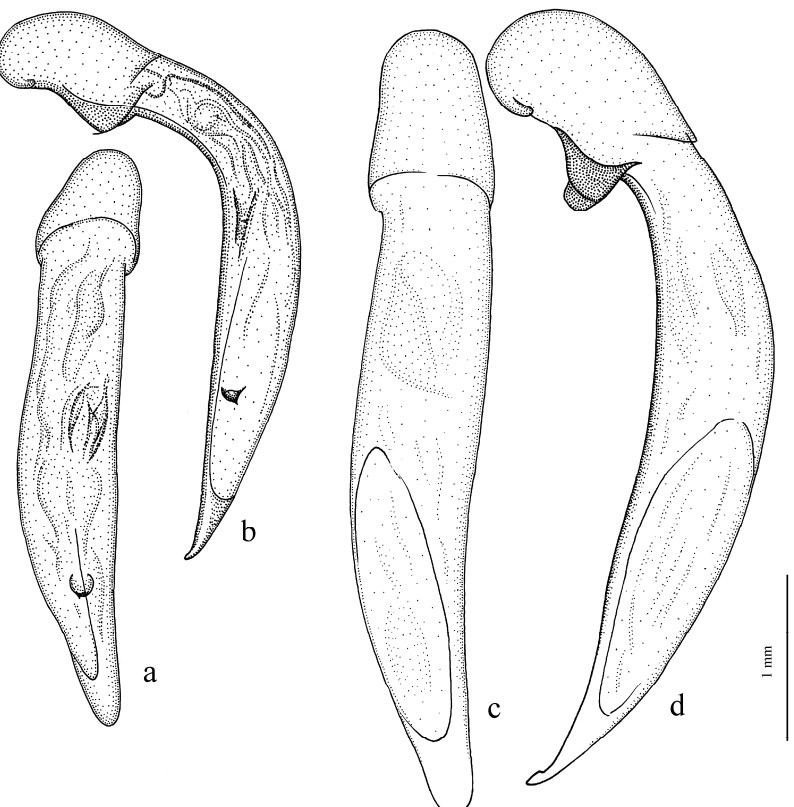

**Figure 19.** Median lobe of aedeagus of *Harpalus* species, dorsal and lateral views: (**a**,**b**) *H.* (*Cephalophonus*) *cephalotes* (from [104]); (**c**,**d**) *H.* (*Cephalomorphus*) *capito* (from [113]).

A redescription of the subgenus *Cephalophonus* and its single species with two subspecies was published by Kataev [104].

### 3.3.2. *Pseudoophonus* Group

Diagnosis. Brown to black, without metallic luster. Body dorsally, particularly elytra, in most species more or less widely punctate and pubescent, in some species impunctate and glabrous; head glabrous, impunctate or with very fine punctures. Tempora generally glabrous, rarely (in one species of *Pseudoophonus*) setose. Antennae pubescent from antennomere 3. Mentum and submentum separated by complete transverse suture; labial basal palpomere generally without carina on ventral side, in some species with straight or oblique carina (Figure 3b–e,h,i,). Pronotum bordered or not bordered along basal margin, with one or several lateral setae on each side. Elytra usually without discal pore on interval 3 (very rarely present as aberration), with or without rows of setigerous pores in the middle of each interval 1, 3, 5 and 7 along their entire length; interval 7 in some species with preapical pores; subapical sinuation absent or shallow. Metacoxa without posteromedial setigerous pore. Protibia with two to five ventroapical spines arranged in a transverse row and with three to seven preapical spines on outer margin, isolated from spines on ventral surface of tibia; ventroapical tubercle in male not developed; apical spur simple, lanceolate, or dentate at margins (in some species of the subgenus *Pseudoophonus*). Tarsi more or less densely pubescent dorsally, rarely (in some North American species of *Pseudoophonus*) metatarsi almost glabrous; male mesotarsomere 1 with or without adhesive scales ventrally. Last visible abdominal sternite and tergite generally without pronounced sexual dimorphism (in some species apex of sternite slightly swollen in female and slightly truncate in male, and apex of tergite slightly bent dorsally in female). Median lobe of aedeagus with or

without apical capitulum; internal sac either without sclerotic elements or with more or less developed armament consisting of spiny patches and, more rarely, one or two moderately large spines.

Composition and distribution. This group includes one Holarctic and two Palaearctic subgenera.

Remarks. This group corresponds to the subgenus *Pseudoophonus* in the understanding of many recent authors (e.g., [1,7,8,29,51], etc.). *Pseudoophonus*, including *Cephalomorphus* and *Platus*, is often also considered a separate genus (e.g., [42–44,129]) or as a subgenus of the genus *Ophonus* in the earlier literature (e.g., [14,40,130]). It is included in the genus *Harpalus*, since the characters of *Pseudoophonus* fully correspond to the diagnosis of this genus, and it is not possible to unequivocally separate it from *Harpalus* if all species of these taxa are considered, based on some specific characters. The known larvae of *Pseudoophonus*, *Cephalomorphus* and *Platus* are also very close in their morphology to those of other *Harpalus* and differ significantly, for example, from the larvae of *Ophonus* [41]. According to molecular data [44], *Pseudoophonus* is also much closer to other *Harpalus* than to *Ophonus*.

The members of *Pseudoophonus* group, like *Cephalophonus*, are characterized by tarsi setose dorsally, but distinguished from it by a head and two basal antennomeres glabrous.

The *Pseudoophonus* group, like most of the other members of the pseudoophonoid complex, demonstrates very high variability in some of its distinctive features, with a mosaic set of plesiomorphic and apomorphic character states, which are usually more constant in most of the groups and subgenera [113]. Such a characteristic of this complex creates significant difficulties in trying to find apomorphies common to all its members. It apparently means that members of the pseudoophonoid complex seem to be rather ancient taxa which evolved early from other *Harpalus* members. It is hypothesized that a wide variation in basal diagnostic features is a characteristic of taxa located at the basis of large phylogenetic branches [46,131,132]. The status of such taxa is always a subject of discussions and disagreements among researchers.

Subgenus *Cephalomorphus* Tschitschérine, 1897

*Cephalomorphus* Tschitschérine, 1897 [133] (p. 45) (as a subgenus of *Ophonus* Dejean, 1821). Type species *Harpalus capito* Morawitz, 1862, by monotypy.

Diagnosis. Large size (length 17.3–24.0 mm). Body moderately convex, elongate, with large head. Head impunctate, pronotum and elytra dendely punctate and setose. Ligular sclerite with a small apical plate. Pronotum with one to three lateral setae on each side, not bordered along basal margin, with sparsely setose basal edge. Elytra with glabrous basal border and with rows of pores on each interval 1, 3, 5 and 7 along their entire length in middle of intervals (Figure 6a). Protibia with four (more rarely three or five) ventroapical spines arranged in a transverse row. Tarsomere 5 without spines ventrally, only with usual thin setae (as in Figure 10d). Male mesotarsomere 1 with adhesive scale ventroapically. Abdominal sternites densely setose. Aedeagus without sclerotic elements in internal sac (Figure 19c,d).

Composition and distribution. This subgenus includes only *H. capito* Morawitz, 1862 (Figures 18b and 19c,d), distributed in the southern part of Russian Far East, Korea, the northeastern and eastern parts of China and in Japan.

Ecology. The single species of this subgenus is found in various open habitats, usually on the edge of the forests, often also in destroyed antropogenic biotopes and agricultural fields.

Remarks. In pronotum not bordered along basal margin, *Cephalomorphus* is similar to *Cephalophonus*, but differs from it in having head impunctate and glabrous, elytra with rows of setigerous pores on intervals 1, 3, 5 and 7 in middle of intervals, protibia with three to five ventroapical spines and the last visible abdominal sternite and tergite without pronounced sexual dimorphism.

A detailed redescription of the subgenus *Cephalomorphus* and its single species was published by Habu [29].

Subgenus *Pseudoophonus* Motschulsky, 1844

*Holosus* Fisher von Waldheim, 1829 [134] (p. 21) (as a genus) (nomen oblitum [135]). Type species *Carabus ruficornis* Fabricius, 1775 (=*C. rufipes* DeGeer, 1774), designated by Bousquet [135].

*Pseudoophonus* Motschulsky, 1844 [34] (p. 196, tabl. between pp. 196 and 197) (as a genus) (nomen protectum [135]). Type species *Carabus ruficornis* Fabricius, 1775 (=*C. rufipes* DeGeer, 1774), designated by Motschulsky [61].

*Pseudophonus* Motschulsky in Ménétriés, 1848 [136] (p. 37) (unjustified emendation).

*Erpeinus* Motschulsky, 1844 [34] (tabl. between pp. 196 and 197, 197) (as a subgenus of *Harpalus* Latreille, 1802). Type species *Harpalus pastor* Motschulsky, 1844, designated by Noonan [6].

*Empeirus* Motschulsky, 1844 [34] (p. XI) (nomen pro *Erpeinus* Motschulsky, 1844). Type species *Harpalus pastor* Motschulsky, 1844, designated by Noonan [6] (pro *Erpeinus*) [100].

*Migadophonus* Tschitschérine, 1897 [133] (p. 47) (as a subgenus of *Ophonus* Dejean, 1821). Type species *Ophonus aenigma* Tschitschérine 1897, by monotypy.

*Empeinus*: Bousquet, 2012 [82] (p. 1095) (print error).

Diagnosis. Medium-sized to large (length 8.0–23.0 mm). Body moderately convex, somewhat wide or elongate. Head medium-sized, impunctate or with very fine punctures dorsally. Pronotum and elytra either more or less widely punctate and pubescent or impunctate and glabrous. Ligular sclerite without apical plate. Pronotum with one or several (Figure 5b,c) lateral setae on each side; basal margin bordered, in some species border interrupted or indistinct medially or laterally; basal edge generally glabrous. Elytra generally lacking discal setigerous pore, occasionally with one discal pore on interval 3; interval 7 in some species with preapical pores; basal border glabrous or setose. Protibia with two (rarely three) ventroapical spines arranged in a transverse row. Tarsomere 5 without spines ventrally (as in Figure 10d), only with thin setae. Male mesotarsomere 1 with or without adhesive scale ventrally. Abdominal sternites with or without additional setae. Internal sac of aedeagus either without sclerotic elements or with more or less developed armament consisting of spiny patches and, more rarely, one or two moderately large spines.

Composition and distribution. This subgenus comprises 38 species distributed in the Holarctic region, with a few species also occurring in the northern part of the Oriental region. The Palaearctic and Nearctic faunas have no common species and are taxonomically separated from each other.

The Palaearctic fauna includes 26 species, divided tentatively into five species groups (originally designated as subgroups [113]):

(1) The *rufipes* group: *H. rufipes* (DeGeer, 1774) (Figure 20a,b), *H. griseus* (Panzer, 1796), *H. jureceki* (Jedlička, 1928), *H. eous* Tschitschérine, 1901, *H. roninus* Bates, 1873, *H. ussuriensis* Chaudoir, 1863 (Figure 20c,d) (with the subspecies *H. u. vicarius* Harold, 1878), *H. aenigma* (Tschitschérine, 1897), and *H. pseudophonoides* Schauberger, 1930. This group is characterized by the elytra densely pubescent throughout and the apical spur of protibia not dentate at margins.

(2) The *pastor* group: *H. fokienensis* Schauberger, 1930, *H. pastor* Motschulsky, 1844 (with the subspecies *H. p. niigatanus* Schauberger, 1929), *H. simplicidens* Schauberger, 1929 and *H. coreanus* (Tschitschérine, 1895). This group is characterized by the elytra pubescent at most on lateral intervals, the pronotum with basal angles sharp at apex, often denticulate, and the apical spur of protibia not dentate at margins.

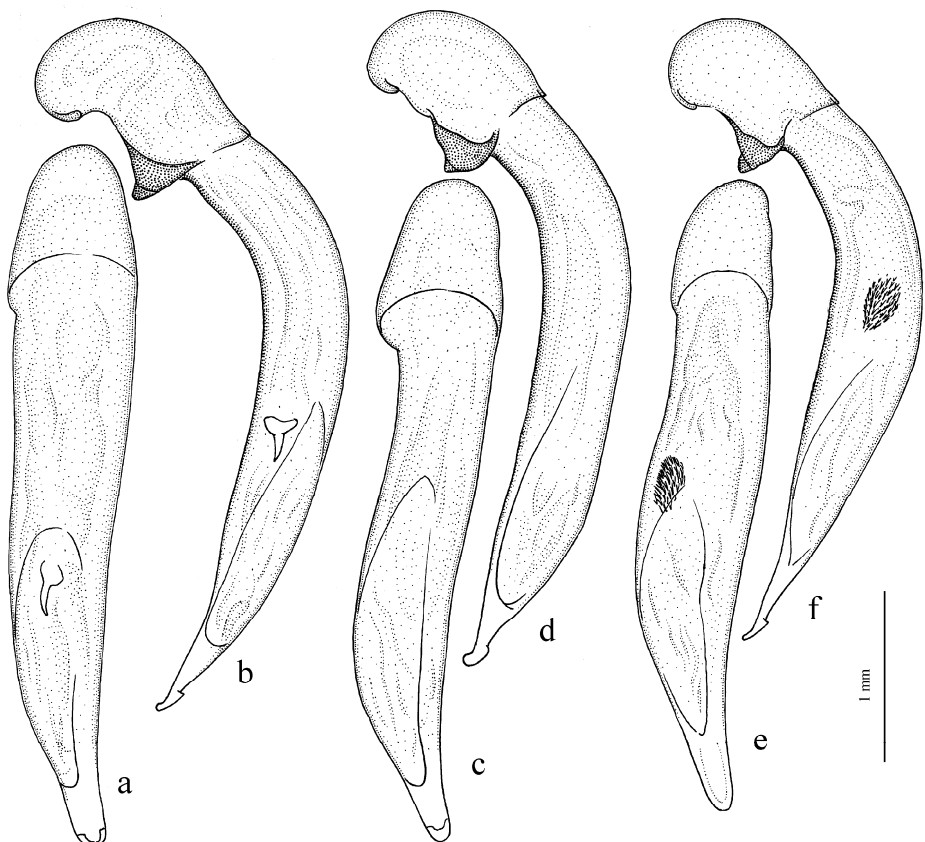

**Figure 20.** Median lobe of aedeagus of *Harpalus* species, dorsal and lateral views (from [113]): (**a**,**b**) *H.* (*Pseudoophonus*) *rufipes*; (**c**,**d**) *H.* (*P.*) *ussuriensis*; (**e**,**f**) *H.* (*Platus*) *calceatus*.

(3) The *tridens* group: *H. tridens* Morawitz, 1862 and *H. suensoni* Kataev, 1997. These two species are characterized by the elytra pubescent at most on lateral intervals, the pronotum with basal angles sharp at apex, often denticulate, and the apical spur of protibia markedly dentate at margins.

(4) The *sinicus* group: *H. sinicus* Hope, 1845, *H. meridianus* Andrewes, 1923, *H. pseudohauserianus* Kataev, 2001, *H. davidi* (Tschitschérine, 1897), *H. sericatus* (Tschitschérine, 1906), *H. babai* Habu, 1973, *H. indicus* Bates, 1891 (with the subspecies *H. i. orientalis* Kataev, 2014), *H. hauserianus* Schauberger, 1929, and *H. meghalayensis* Kataev, 2001. This group is characterized by the elytra pubescent at most on lateral intervals, the pronotum with basal angles more or less widely rounded at apex, and the apical spur of protibia not or markedly dentate at margins.

(5) The *singularis* group: *H. singularis* Tschitschérine, 1906, *H. azumai* Habu, 1968, and *H. aogashimensis* (Habu, 1957). These three species are recognizable by having the elytra pubescent at most on lateral intervals, the peculiar shape of pronotum with distinct obtuse basal angles, blunt at apex, the apical spur of protibia not dentate at margins and the peculiar armament of the internal sac of aedeagus with two large spines.

The status and position of *H.* (*Pseudoophonus*) *disimuciulus* Huang, Lei, Yan et Hu, 1996, described from Sichuan, are obscure [113].

Among the Palaearctic species, *H. rufipes* (DeGeer, 1774) is distributed in the western Palaearctic, *H. griseus* (Panzer, 1796) has a trans-Palaearctic distribution, and all other species occur in eastern Asia.

The Nearctic fauna includes twelve endemic species, divided into three species groups [78,82]:

(1) The *compar* group: *H. actiosus* Casey, 1914, *H. compar* LeConte, 1848, *H. erythropus* Dejean, 1829, *H. paratus* Casey, 1924, and *H. vagans* LeConte, 1865.

(2) The *pensylvanicus* group: *H. pensylvanicus* (DeGeer, 1774), *H. liobasis* Chaudoir, 1868, *H. texanus* Casey, 1914, and *H. protractus* Casey, 1914.

(3) The *rufipes* group: *H. faunus* Say, 1823 and *H. hatchi* Ball et Anderson, 1962.

The position of *H. poncei* Will, 2001 is regarded as incertae sedis [82].

With the exception of the trans-American *H. pensylvanicus* (DeGeer, 1774), all other species are distributed mainly in the eastern regions, to south up to northern Mexico and Florida.

Ecology. Species of this subgenus inhabit various open landscapes, mainly meadows and light deciduous forests; many species are common in agricultural fields and disturbed antropogenic habitats.

Remarks. Within the *Pseudoophonus* group, this subgenus is recognized by pronotum bordered along basal margin, elytra without rows of setigerous pores on odd intervals and tarsomere 5 without strong spines ventrally.

The American species were revised by Ball and Anderson [78]. The Palaearctic species were revised by Schauberger [66,67,137], Habu [28,29], Kataev [94,97] and Kataev and Liang [113].

Subgenus *Platus* Motschulsky, 1844

*Platus* Motschulsky, 1844 [34] (tabl. between pp. 196 and 197, 197) (as a subgenus of *Harpalus* Latreille, 1802). Type species *Harpalus calcitrapus* Motschulsky, 1844 (=*Carabus calceatus* Duftschmid, 1812), designated by Noonan [6].

*Pardileus* Gozis, 1882 [138] (p. 289) (as a genus). Type species *Carabus calceatus* Duftschmid, 1812, by monotypy.

*Neopardileus* Habu, 1954 [139] (p. 283) (as a subgenus of *Ophonus* Dejean, 1821). Type species: *Ophonus itoshimanus* Habu, 1954 (=*Carabus calceatus* Duftschmid, 1812), by original designation.

Diagnosis. Large size (length 12.0–14.2 mm). Body moderately convex, elongate. Head medium-sized, impunctate. Ligular sclerite without apical plate. Pronotum punctate and sparsely setose laterally and basally, with one lateral seta on each side; basal margin completely bordered, with glabrous basal edge. Elytra punctate and pubescent laterally, with setose basal edge and without rows of setigerous pores on odd intervals; interval 7 with one or two preapical setigerous pores. Protibia with three (rarely two) ventroapical spines arranged in a transverse row. Tarsomere 5 with strong spines ventrally (in addition to thin setae) (Figure 10e). Male mesotarsomere 1 with or without adhesive scale ventrally. Abdominal sternites densely setose. Aedeagus with a spiny patch in internal sac (Figure 20e,f).

Composition and distribution. This subgenus includes only the trans-Palaearctic *H. calceatus* (Duftschmid, 1812) (Figure 20e,f) distributed from Portugal to Japan.

Ecology. The single representative of this subgenus occurs in open, rather dry habitats and is particularly common in agricultural fields.

Remarks. Distinctly differs from all other congeners in having strong spines on the tarsomere 5 ventrally. This is a unique feature among *Harpalus*, since in all other members of the genus, tarsomere 5 has only thin setae ventrally.

A detailed redescription of this subgenus and its single species was published by Habu (1973).

3.3.3. *Megapangus* Group

Diagnosis. Same as for the subgenus.

Composition and distribution. A monobasic group, including only one Nearctic subgenus.

Subgenus *Megapangus* Casey, 1914

*Megapangus* Casey, 1914 [71] (p. 71) (as a subgenus of *Harpalus* Latreille, 1802). Type species *Carabus caliginosus* Fabricius, 1775, by monotypy.

Diagnosis. Very large size (length 17.5–28.0 mm). Body moderately convex, stout, wide or somewhat elongate, dark brown to black, without metallic luster, glabrous dorsally. Head impunctate dorsally, at most covered with very fine micropunctures behind eyes. Tempora glabrous. Antennae pubescent from antennomere 3. Mentum and submentum separated by complete transverse suture; labial basal palpomere with more or less distinct oblique carina on ventral side (Figure 3j). Pronotum with one lateral seta on each side, bordered along basal margin (border occasionally indistinct in some places) amd with setose basal edge. Elytra finely punctate laterally, glabrous or densely pubescent on basal border and basal area near scutellum, without discal pores on interval 3; subapical sinuation absent or very shallow. Metacoxa without posteromedial setigerous pore. Protibia usually with four (occasionally five) ventroapical spines arranged in a transverse row and with seven to nine (rarely six) preapical spines on outer margin, isolated from spines on ventral surface of tibia; ventroapical tubercle in male not developed; apical spur simple, lanceolate. Tarsi glabrous dorsally; tarsomere 5 without spines ventrally, only with thin setae; male mesotarsomere 1 with adhesive scales ventrally. Abdominal sternites finely punctate laterally (throughout on last visible sternite), glabrous; last visible sternite without pronounced sexual dimorphism (slightly thickened apically in female); apex of last visible tergite of female emarginate at sides and bent dorsally (in *H. caliginosus* more markedly than in *H. katiae*). Median lobe of aedeagus with distinct apical capitulum; internal sac with a group of small spines.

Composition and distribution. The subgenus includes two species: the trans-American *H. caliginosus* (Fabricius, 1775) and *H. katiae* Battoni, 1985 (Figure 21a) known from the southwestern United States.

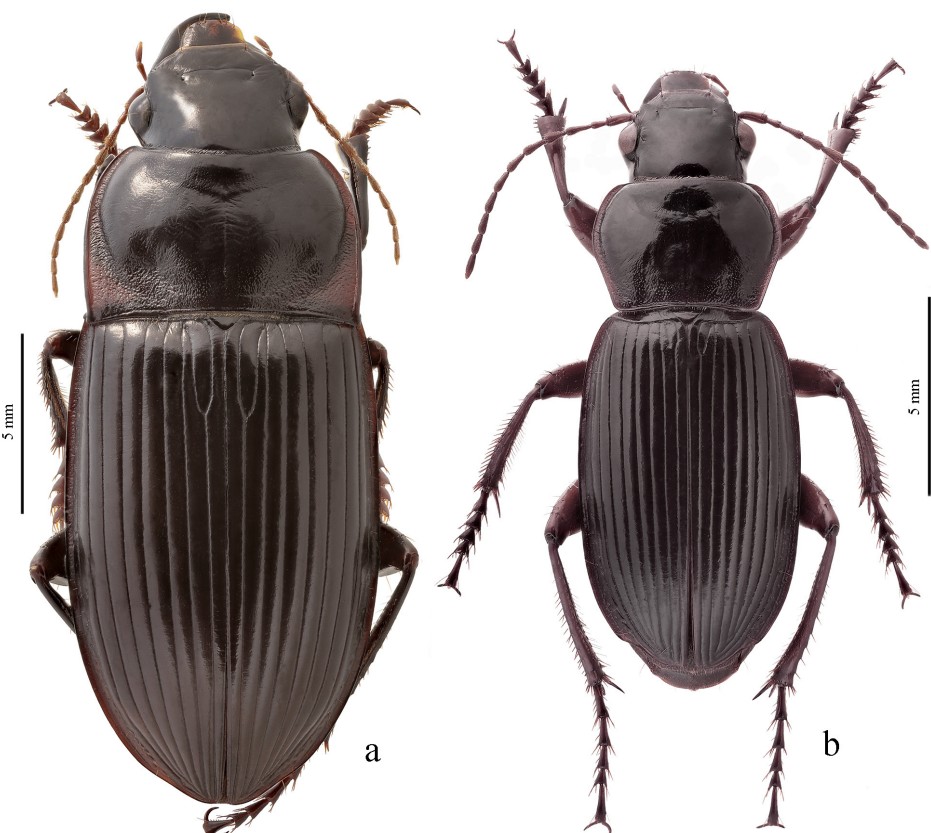

**Figure 21.** Habitus of *Harpalus* species: (**a**) *H.* (*Megapangus*) *katiae*; (**b**) *H.* (*Plectralidus*) *retractus*.

Ecology. Both species of this subgenus live in open, rather dry biotopes with sparse vegetation, usually on sandy soil, both in natural landscapes, for example prairies or light deciduous forests, and in agricultural fields; often along roads [140].

Remarks. In combination of characters, this subgenus is very similar to subgenera of the *Pseudoophonus* group, differing from them mainly in tarsi glabrous dorsally. *Megapangus* also differs from the subgenera *Pseudoophonus* and *Platus* in a greater number of ventroapical spines on protibia (four or five versus two or three) and from *Cephalomorphus* in glabrous elytra without rows of setigerous pores on odd intervals. For differences from *Plectralidus*, see remarks to this subgenus.

The subgenus was revised by Will [141].

### 3.3.4. *Plectralidus* Group

Diagnosis. Same as for the subgenus.
Composition and distribution. A monobasic group, including only one Nearctic subgenus.

### Subgenus *Plectralidus* Casey, 1914

*Plectralidus* Casey, 1914 [71] (p. 72) (as a subgenus of *Harpalus* Latreille, 1802). Type species *Harpalus eraticus* Say, 1923, designated by El-Moursy [142].

Diagnosis. Large size (length 11.0–16.0 [18.0] mm). Body moderately convex, elongate, brownish yellow to almost black, without metallic luster, glabrous dorsally. Head impunctate, with glabrous tempora. Antennae pubescent from antennomere 3. Mentum and submentum separated by complete transverse suture; epilobes of mentum with short setae; labial basal palpomere more or less clearly cylindrical, without oblique carina ventrally. Pronotum with one lateral seta, bordered along entire length of basal margin and with setose basal edge. Elytra impunctate, densely pubescent on basal border and basal area near scutellum, without discal pores on interval 3; subapical sinuation very deep, often with denticle at its base. Metacoxa without posteromedial setigerous pore. Protibia with four to six ventroapical spines arranged in a transverse row along anterior protibial margin; latter forming a lamella extended downward; outer margin of protibia with four or five preapical spines isolated from spines on ventral surface; apical spur simple, lanceolate. Tarsi glabrous dorsally; arsomere 5 with thin setae ventrally; male mesotarsomere 1 with adhesive scales ventrally. Abdominal sternites with several additional long setae; last visible sternite and tergite without pronounced sexual dimorphism, their apices in both sexes more or less rounded and not swollen. Median lobe of aedeagus with distinct apical capitulum; internal sac with a large separate spine.

Composition and distribution. The subgenus comprises two Nearctic species: *H. eraticus* Say, 1823 and *H. retractus* LeConte, 1863 (Figure 21b).

Ecology. Both species of this subgenus occur in open, rather dry habitats, usually on sandy soil with sparse vegetation, sometimes on pure sand.

Remarks. *Plectralidus* is similar to *Megapangus* in having glabrous dorsum, elytra without discal pore, tarsi glabrous dorsally and protibia with at least four ventroapical spines, but readily distinguished from the latter subgenus by a smaller number of preapical spines on outer margin of protibia, and elytra impunctate on lateral intervals and with deeper preapical sinuation. In addition, ventroapical margin of protibia in *Plectralidus* forms a lamella extended downward.

According to Noonan [16], *Plectralidus* and *Megapangus* form a monophyletic group based on the presence of setae on the basal elytral border. According to my data, only *Plectralidus* and only one of the two species of *Megapangus* (*H. caliginosus*) have such setae. The second species of *Megapangus* (*H. katiae*), unknown to Noonan, has a glabrous basal elytral border. Meanwhile, the *Megapangus* group and the *Plectralidus* group, in my opinion, may be closely related on the basis of their common four to six ventroapical spines on the protibia. This character, however, unlikely to be a synapomorphy, since within the pseudoophonoid subgeneric complex, *Cephalomorphus* also usually has four ventroapical spines.

The subgenus *Plectralidus* was revised by El-Moursy [142].

### 3.3.5. *Loboharpalus* Group

Diagnosis. Same as for the subgenus.

Composition and distribution. The monobasic group, including only one East Asian subgenus.

### Subgenus *Loboharpalus* Schauberger, 1932

*Loboharpalus* Schauberger, 1932 [143] (p. 174) (as a subgenus of *Harpalus* Latreille, 1802). Type species *Harpalus platynotus* Bates, 1873, by original designation.

Diagnosis. Medium-sized to large (length 9.0–16.0 mm). Body convex, stout, somewhat wide, brownish yellow to black, without metallic luster, glabrous. Head impunctate dorsally, with tempora not setose. Antennae pubescent from antennomere 3. Mentum and submentum separated by complete transverse suture; labial basal palpomere more or less cylindrical, without oblique carina ventrally. Pronotum with one lateral seta on each side (in *H. rubefactus* also with several very short marginal setae in apical angles and occasionally with one or two additional lateral setae: Figure 5g,h), bordered along entire length of basal margin and with setose basal edge. Elytra with glabrous basal border and generally with one discal setigerous pore on interval 3; subapical sinuation very shallow, without denticle at its base. Protibia (Figure 10c) with two or three ventroapical spines (in addition to several much thinner spines located laterally); its outer margin strongly flattened and with at least ten preapical spines isolated from longitudinal row of spines on ventral surface; ventroapical tubercle in male more or less prominent; apical spur simple, lanceolate. Metacoxa without posteromedial setigerous pore. Tarsi glabrous dorsally; tarsomere 5 with thin setae ventrally; male mesotarsomere 1 not dilated and without adhesive scales on ventral side. Abdominal sternites glabrous or with additional short setae. Last visible abdominal sternite and tergite without pronounced sexual dimorphism, their apices in both sexes more or less rounded and not swollen. Female genitalia with a very long and narrow gonocoxite (Figure 14e,f). Median lobe of aedeagus with weakly prominent apical capitulum; internal sac without sclerotic elements (as in Figure 23a,b).

Composition and distribution. This subgenus includes two east Asian species: *H. rubefactus* Bates, 1873 (Figures 22a and 23a,b) (with the subspecies *H. r. bachmayeri* Mlynář, 1979) and *H. platynotus* Bates, 1873.

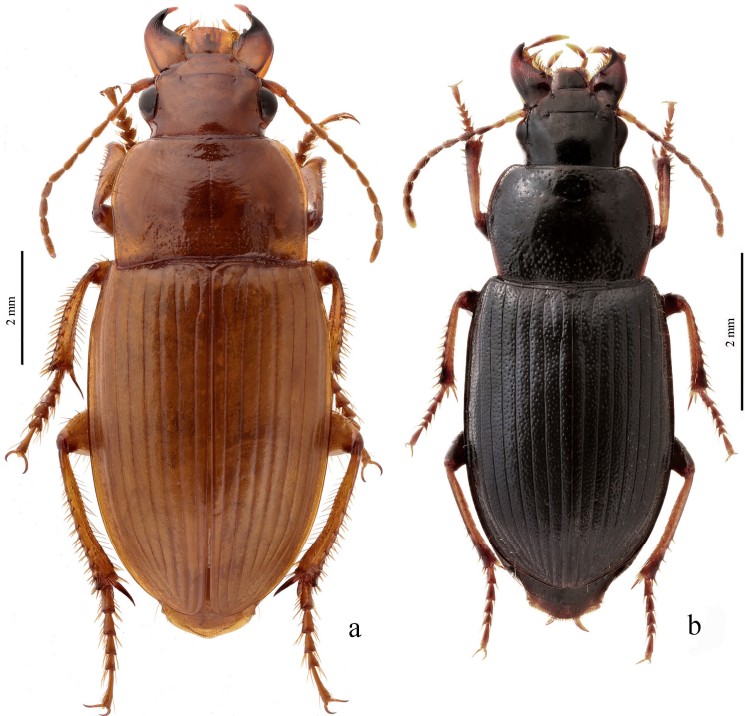

**Figure 22.** Habitus of *Harpalus* species: (**a**) *H.* (*Loboharpalus*) *rubefactus*; (**b**) *H.* (*Semiophonus*) *signaticornis*.

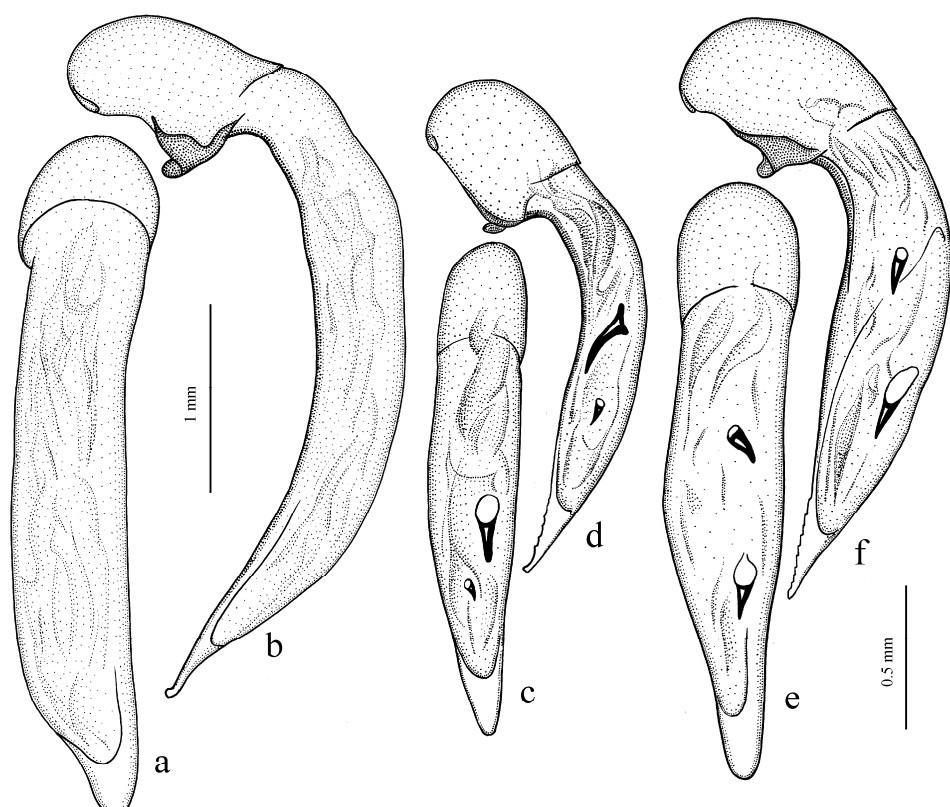

**Figure 23.** Median lobe of aedeagus of *Harpalus* species, dorsal and lateral views: (**a**,**b**) *H.* (*Loboharpalus*) *rubefactus*; (**c**,**d**) *H.* (*Semiophonus*) *signaticornis* (from [46]); (**e**,**f**) *H.* (*Zangoharpalus*) *praticola* (from [46]).

Ecology. Both representatives of this subgenus occur on sandy seashores and sandy banks of large rivers and have developed morphological adaptations for digging into the ground.

Remarks. The main distinctive features of *Loboharpalus* include the adaptive structure of protibia with outer margin strongly flattened and with at least ten preapical spines isolated from longitudinal row of spines on ventral surface (Figure 10c) and the mesotarsomere 1 of male not dilated and without adhesive vestiture ventrally. Since the structure of the aedeagus in *Loboharpalus* with a weakly protruding apical capitulum and without sclerotic elements in the internal sac resembles that of the aedeagus in some *Pseudoophonus*, this subgenus appears to be an early isolated specialized derivative of the common ancestor with the pseudoophonoid subgeneric complex.

Some authors ([6,29], etc.) considered *Loboharpalus* as a synonym of *Acardystus*, but their similarity is convergent, since they differ markedly both in the imaginal [143,144] and larval [145] morphology.

This subgenus was revised by Schauberger [143], Habu [29] and Mlynář [75,76].

### 3.3.6. *Semiophonus* Group

Diagnosis. Same as for the subgenus.

Composition and distribution. A monobasic group, including only one western Palaearctic species.

### Subgenus *Semiophonus* Schauberger, 1933

*Semiophonus* Schauberger, 1933 [24] (p. 131) (as a subgenus of *Harpalus* Latreille, 1802). Type species *Carabus signaticornis* Duftschmidt, 1812, by monotypy.

Diagnosis. Medium-sized (length 5.6–7.5 mm). Body moderately convex, elongate, dark brown to black, without metallic luster. Head impunctate and glabrous dorsally, with setose tempora. Pronotum and elytra punctate and pubescent. Antennae pubescent from antennomere 3. Mentum and submentum separated by complete transverse suture; labial basal palpomere more or less clearly cylindrical, without oblique carina ventrally (Figure 3f,g). Pronotum with one lateral seta on each side, bordered along entire length of basal margin and with glabrous basal edge. Elytra with one discal pore on interval 3 and with very sparsely setose basal border; subapical sinuation shallow. Metacoxa with a posteromedial setigerous pore (Figure 8c). Protibia with one ventroapical spine and with three preapical spines on outer margin, isolated from longitudinal row of spines on ventral surface of tibia; ventroapical tubercle in male not developed; apical spur simple, lanceolate. Tarsi glabrous dorsally; tarsomere 5 with thin setae ventrally; male mesotarsomere 1 with adhesive scales ventrally. Abdominal sternites without additional setae. Last visible sternites and tergite without pronounced sexual dimorphism, their apices in both sexes more or less rounded and not swollen. Median lobe of aedeagus with thin terminal lamella lacking apical capitulum; internal sac with two separate spines (Figure 23c,d).

Composition and distribution. This subgenus includes only *H. signaticornis* (Duftschmid, 1812) (Figures 22b and 23c,d) distributed in western Eurasia from the Iberian Peninsula to the Baikal region.

Ecology. The single species of this subgenus occurs in various dry open habitats, mainly in dry meadows and grasslands.

Remarks. This subgenus is recognizable by the combination of the following main distinctive features: head glabrous, with setose tempora, pronotum and elytra punctate and pubescent, metacoxa with a posteromedial setigerous pore and tarsi glabrous dorsally.

*Semiophonus* was originally described as related to the subgenus *Harpalophonus* and is sometimes considered as a separate genus [14,43] or, more often, as belonging to the genus *Ophonus* (e.g., [42,126,130]). In my opinion [146], it should be assigned to the genus *Harpalus*, since it has almost all the diagnostic features of this genus, including setose paraglossae. Like *Ophonus*, *Semiophonus* possesses a posteromedial setigerous pore on metacoxa; however, this pore is found, for example, in some Afrotropical species of *Harpalus*, for example, *H.* (*Afroharpalus*) *frater*, *H.* (*A.*) *kibonoti*, *H.* (*A.*) *merkli*, 2021 and in the Palaearctic *H.* (*Hypsinephus*) *salinus*, but it is absent in closely related species *H.* (*H.*) *lumbaris*. The described larva of *Semiophonus* is also structurally very close to *Harpalus* larvae and markedly different from larvae of *Ophonus* [147].

3.3.7. *Zangoharpalus* Group

Diagnosis. Same as for the subgenus.

Composition and distribution. The monobasic group, including only one Eastern Palaearctic subgenus.

Subgenus *Zangoharpalus* Huang, 1998

*Zangoharpalus* Huang, 1998 [148] (pp. 201, 203) (as a subgenus of *Harpalus* Latreille, 1802). Type species *Harpalus yadongensis* Huang, 1998 (=*H. praticola* Bates, 1891), by original designation.

Diagnosis. Small- to medium-sized (length 4.5–8.5 mm). Body moderately convex, somewhat wide or elongate, dark brown to black, often with metallic tinge on dorsum, glabrous dorsally. Head impunctate, with tempora not setose. Antennae pubescent from antennomere 3. Mentum and submentum separated by complete transverse suture; labial basal palpomere more or less clearly cylindrical, without oblique carina ventrally. Pronotum punctate or impunctate basally, with one lateral seta on each side, bordered along entire length of basal margin and with glabrous basal edge. Elytra with glabrous basal border, with or without one discal setigerous pore on interval 3; in most species impunctate, in *H. praticola* with very fine punctures on lateral intervals; subapical sinuation shallow;

microsculpture, if present, isodiametric to clearly transverse. Metacoxa without posterome-dial setigerous pore. Protibia with one ventroapical spine and with four preapical spines on outer margin, isolated from spines on ventral surface; ventroapical tubercle in male not de-veloped; apical spur simple, lanceolate. Tarsi glabrous dorsally; tarsomere 5 with thin setae ventrally; male mesotarsomere 1 of male in most species without adhesive scales ventrally, more rarely (in *H. praticola*) with a pair of small scales apically. Abdominal sternites without additional setae. Last visible sternites and tergite without pronounced sexual dimorphism, their apices in both sexes more or less rounded and not swollen. Median lobe of aedeagus with wide and thin terminal lamella lacking apical capitulum (as in Figure 23e,f); internal sac with two separate spines and in most species also with one or several spiny patches.

Composition and distribution. This subgenus comprises five species, the distribution of which is mainly restricted to the eastern Palaearctic and southeast Asia, including the Himalayan region: *H. tinctulus* Bates, 1873 (with the subspecies *H. t. luteicornoides* Breit, 1913), *H. pseudotinctulus* Schauberger, 1932, *H. microdemas* Schauberger, 1932, *H. mariae* Kataev, 1997, and *H. praticola* Bates, 1891 (Figures 23e,f and 24a). Three of them (*H. tinctulus*, *H. pseudotinctulus* and *H. praticola*) enter the northern part of the Oriental region.

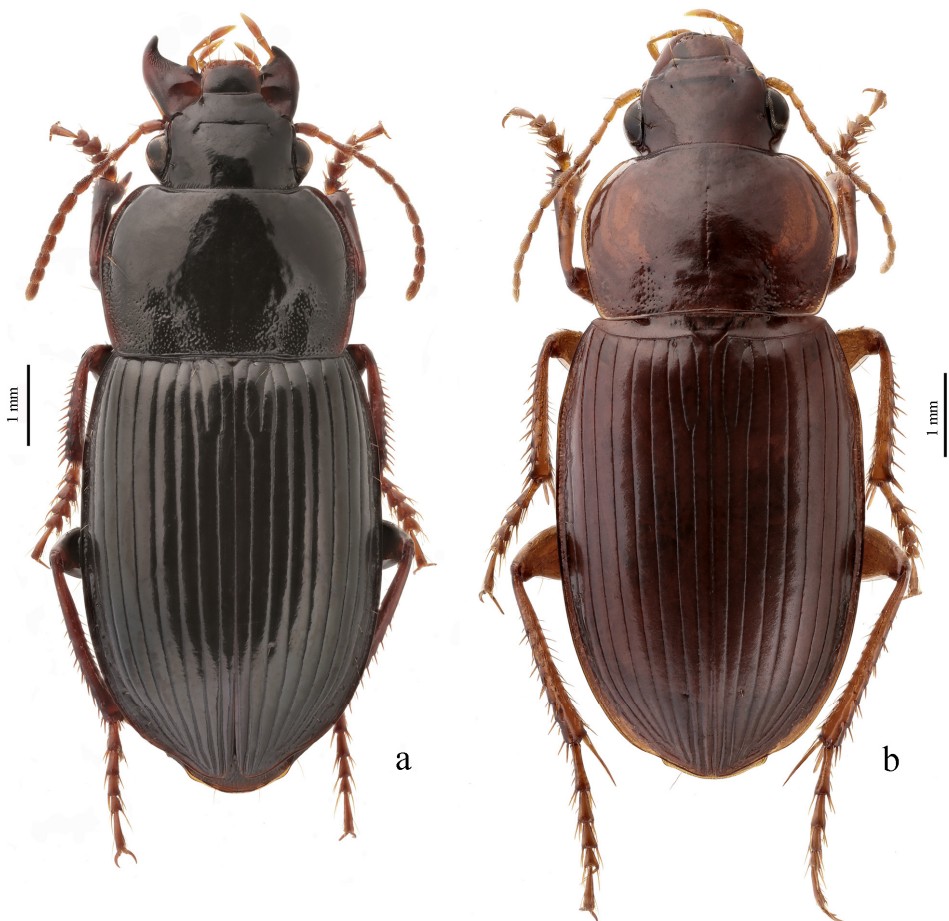

**Figure 24.** Habitus of *Harpalus* species: (**a**) *H.* (*Zangoharpalus*) *praticola*; (**b**) *H.* (*Cryptophonus*) *fulvus*.

Ecology. The species of this subgenus occur in different open habitats, usually at forest edges, often also in agricultural fields.

Remarks. The main distinctive features of *Zangoharpalus* are median lobe of aedeagus with thin and flat terminal lamella lacking apical capitulum and mesotarsomere 1 in male generally without adhesive vestiture ventrally.

The *Zangoharpalus* group and the *Semiophonus* group possibly form a monophyletic unit. The members of both groups are very similar in the structure of the aedeagus without a pronounced apical capitulum, with a particularly strong similarity in the genitalia between

*H.* (*Semiophonus*) *signaticornis* and *H.* (*Zangoharpalus*) *praticola* (Figure 23c–f). Considering that the external morphology of these species is also rather similar, there are reasons to consider these subgeneric groups to be related [46].

A revision of the subgenus *Zangoharpalus* (as the *tinctulus* species group) was published by Kataev [94].

### 3.3.8. *Cryptophonus* Group

Diagnosis. Same as for the subgenus.

Composition and distribution. This monobasic group includes only one subgenus with a western Palaearctic and Afrotropical distribution.

Subgenus *Cryptophonus* Brandmayr et Zetto Brandmayr, 1982

*Cryptophonus* Brandmayr et Zetto Brandmayr, 1982 [149] (p. 81) (as a subgenus of *Ophonus* Dejean, 1821). Type species *Harpalus tenebrosus* Dejean, 1829, by original designation.

Diagnosis. Medium-sized to large (length 7.9–12.5 mm). Body moderately convex, somewhat wide or elongate, brownish yellow to black, glabrous dorsally. Head impunctate, with tempora not setose. Antennae pubescent from antennomere 3. Mentum and submentum in most species completely fused (Figure 4c), in *H. litigiosus* either fused only laterally or separated from each other by complete transverse suture (Figure 4a,b); epilobes of mentum in some species slightly widened apically and their inner margin slightly angulate; labial basal palpomere with oblique carina ventrally (Figure 3k,l). Pronotum more or less punctate basally, with one lateral seta on each side (in *H. schaumii* also with several shorter marginal setae in apical angles), bordered along entire length of basal margin and with glabrous or setose basal edge. Elytra with one discal pore on interval 3 and with shallow preapical sinuation; intervals 7 and 8 in some species with preapical pores (Figure 6b); basal border glabrous or with a few very short setae. Metacoxa in most species without posteromedial setigerous pore, in *H. agnatus* and occasionally in *H. idiotus* posteromedial setigerous pore present. Protibia in most species with one ventroapical spine, in *H. melancholicus* generally with two spines; outer margin of protibia with three preapical spines isolated from longitudinal row of spines on ventral surface; ventroapical tubercle in male not developed; apical spur simple, lanceolate. Tarsi either glabrous dorsally or with sparse setae in some tarsomeres; tarsomere 5 with thin setae ventrally; male mesotarsomere 1 in most species without adhesive scales, in *H. melancholicus* with a pair of small scales apically. Abdominal sternites with or without additional setae. Last visible abdominal sternite and tergite without pronounced sexual dimorphism, their apices in both sexes more or less rounded and not swollen. Apical capitulum of median lobe of aedeagus absent or very small, almost indistinct (as in Figure 25a,b); internal sac with two (occasionally one) separate spines and usually also with one or several small spiny patches.

Composition and distribution. This subgenus comprises ten species with distributional areas concentrated within the ancient Thetyan region, predominantly in the Mediterranean, including the Canary islands, also in the northeastern part of the Afrotropical region and in the Cabo Verde Archipelago: *H. tenebrosus* Dejean, 1829 (Figure 25a,b) (with the subspecies *H. t. paivanus* Wollaston, 1867), *H. idiotus* Bates, 1889, *H. grilli* Kataev, 2002, *H. litigiosus* Dejean, 1829, *H. agnatus* Reiche, 1847, *H. schaumii* Wollaston, 1864, *H. janinae* Jeanne, 1984, *H. cyrenaicus* Koch, 1939, *H. fulvus* Dejean, 1829 (Figure 24b) and *H. melancholicus* Dejean, 1829 (Figure 11a,b) (with the subspecies *H. m. reicheianus* Kataev, 2012).

Ecology. According to Machado [150], the distribution of at least one of the two endemic Canarian species, *H. janinae*, is associated precisely with the zone of relict laurel forests; the second species, *H. schaumii*, is ecologically more plastic and occurs both in semi-desert conditions of the coast and in moist laurel forests, however, preferring forest and meadow habitats of the mid-mountain belt. Other species for which ecological data are available occur mainly on sandy soils, some of them (*H. fulvus*, *H. melancholicus* and *H. litigiosus*) are inhabitants of the dunes of seacoasts.

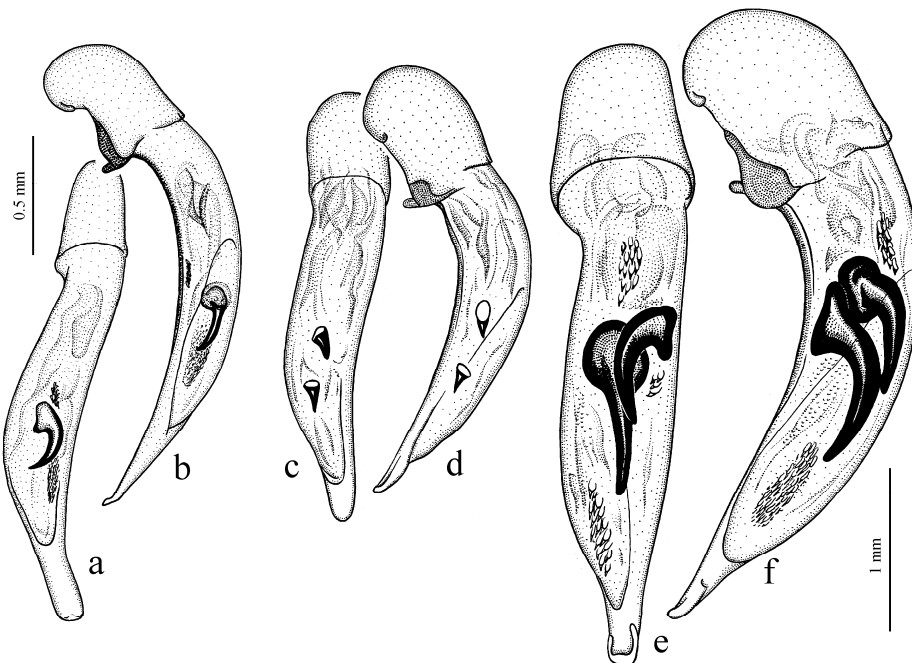

**Figure 25.** Median lobe of aedeagus of *Harpalus* species, dorsal and lateral views: (**a**,**b**) *H.* (*Cryptophonus*) *tenebrosus* (from [47]); (**c**,**d**) *H.* (*Afroharpalus*) *parvulus* (from [46]); (**e**,**f**) *H.* (*A.*) *gregoryi* (from [46]).

Remarks. All members of *Cryptophonus* have a typical "harpaloid" appearance and have always been considered members of the genus *Harpalus*. Meanwhile, some characteristics of the larval stage of this taxon are similar to those of *Ophonus* larvae and, according to Brandmayr et al. [41] and Brandmayr and Zetto Brandmayr [149], *Cryptophonus* should be included in this genus as a subgenus. Based on imaginal characters (particularly, the setose paraglossae, rather narrow epilobes of the mentum, absence of a posteromedial pore in metacoxa in most species, and glabrous body), *Cryptophonus* is treated as a member of the genus *Harpalus*, although a number of characters, such as the presence of an oblique carina on the basal labial palpomere, really bring *Cryptophonus* closer to *Ophonus*. The latter feature, however, is not exclusive to the genus *Ophonus*, but is also found in some *Pseudoophonus* and *Megapangus* [47,151]. The other most distinctive features of most *Cryptophonus* species are: mentum and submentum in most species completely fused, male mesotarsomere 1 without adhesive scales, and apical capitulum of median lobe of aedeagus absent or very small.

*Cryptophonus* seems to be one of the ancient taxa which, like *Pseudoophous* and other subgenera listed above, evolved rather early from other *Harpalus* members. This fact mostly explains the mosaic set of characters in *Cryptophonus*, with each of these being usually typical of different, younger, more specialized groups. The geographical distribution of *Cryptophonus*, including two species endemic to the Canary Islands, two species endemic to the western part of the Himalayas and one endemic to east Africa, also favors the ancient nature of this group [47].

The subgenus was revised by Kataev [47].

### 3.3.9. *Harpalus* Group

Diagnosis. Body glabrous or more or less densely pubescent. Head generally impunctate and glabrous, rarely finely punctate and setose dorsally; temporae in most members glabrous, in some species setose. Antennae pubescent from antennomere 3. Mentum and submentum separated by complete transverse suture (occasionally fused in some *Afroharpalus* **subg. n.**); labial basal palpomere more or less cylindrical, without carina on ventral side. Pronotum with one or several lateral setae on each side, bordered along

basal margin, with glabrous or setose basal edge. Elytra with one or several setigerous discal pores on interval 3, in some members also on intervals 5 and 7; in some members, occasionally or constantly, discal pores absent; intervals 5 and 7, more rarely also 3, in many members with short row of preapical setigerous pores; subapical sinuation variable from rather deep to very shallow or indistinct; basal border glabrous or setose. Metacoxa in most members without posteromedial setigerous pore, very rarely this pore present. Protibia with one to three (very rarely four) ventroapical spines arranged in a transverse row and generally with at least three (very rarely two) preapical spines on outer margin of tibia; preapical spines either isolated from spines on ventral surface or arranged with them in a single row; ventroapical tubercle in male absent or more or less prominent; apical spur simple, lanceolate. Dorsal side of tarsi glabrous or more or less densely pubescent; tarsomere 5 with thin setae ventrally; male mesotarsomere 1 generally with adhesive scales ventrally, rarely without them. Abdominal sternites with or without additional setae; last visible abdominal sternite and tergite of both sexes either similar in both sexes or with more or less pronounced sexual dimorphism (apex of last visible sternite truncate or emarginate in male, and narrowed and swollen in female). Median lobe of aedeagus in most species with distinct, oblique or transverse, apical capitulum, rarely without it; internal sac either without any sclerotic elements or with more or less developed armament consisting of spiny patches, groups of spines and one or two separate spines.

Composition and distribution. This group comprises 58 subgenera combined into 19 subgroups, which are distributed in the Holarctic and Afrotropical regions. The range of this subgeneric group, which includes most species of the genus, almost completely coincides with the generic range, but its representatives do not spread in the Cabo Verde Islands and the Oriental region.

Remarks. This group corresponds to the subgenus *Harpalus* sensu stricto in the understanding of many recent authors (e.g., [1,7,8,29,51], etc.) and comprises most species of the genus.

An analysis of the distribution of characters in this very diverse group shows that all its members can be divided into three subgeneric complexes or phyletic stocks: one Afrotropical stock, corresponding to the modern subgenus *Afroharpalus* **subg. n.**, and two Holarctic stocks, the *latus* stock and the *affinis* stock, respectively [8]. The *latus* stock comprises the *Hyloharpalus*, *Cordoharpalus*, *Amblystus*, *Actephilus*, *Acardystus*, *Psammoharpalus*, *Ooistus*, *Asioharpalus*, *Anamblystus* and *Harpalobius* subgroups; the *affinis* stock are the *Pheuginus*, *Pharalus*, *Hypsinephus*, *Mauriharpalus*, *Calloharpalus*, *Idioharpalus*, *Artabas* and *Harpalus* subgroups. The belonging of these subgroups to one or another stock is determined, first of all, by different initial types of sclerotized armament of the internal sac of the aedeagus, to which all the diversity of armaments observed within each of these two taxonomic complexes can be reduced. The members of the *latus* stock are characterized by a more complex type of the armament. It is present in its most complete form in species of the *Amblystus* subgroup (Figures 11c,d and 35e–h), where the following sclerotic elements are clearly distinguished: (1) two separate spines (sometimes one spine), (2) one or several groups of small spines, more rarely spiny patches, medially, and (3) a separate apical spiny patch, usually on the right side of the median lobe of the aedeagus. This type of armament corresponds to the type that Lindroth [15] described for the *fraternus* species group. During the evolution of the taxa of this stock, the main elements of this type were generally preserved, although in some taxa, the armament was simplified, and even individual cases of its complete loss can be traced (for example, in some representatives of the *Acardystus*, *Anamblystus* and *Harpalobius* subgroups). The *affinis* stock is characterized by a simpler initial armament of the internal sac. In its most complete form, it is observed in some representatives of the *Pheuginus* subgroup (Figure 50): (1) one or two separate spines, and (2) two spiny patches, usually located medially. In the course of the evolution of the taxa of this stock, in different phyletic lineages, a rather early loss of elements of this type armament, especially separate spines, is traced, although in some subgenera, for example,

in the *Caloharpalus* subgroup, new elements appeared, mainly spiny patches and groups of small or medium spines.

The differences between the two stocks are striking in ecology and distribution of the included taxa. The *latus* stock, in addition to the taxa typical for open landscapes, also includes all known forest inhabiting species of the *Harpalus* group. The species of open landscapes of this stock are mainly typical xerophilic species occurring in zonal steppe habitats. Among them, there is no species associated with saline soils and specialized desert forms. Only a few have adapted to living in almost pure sand along the banks of rivers [for example, *H.* (*Acardystus*) *flavescens* and *H.* (*Psammoharpalus*) *kozlovi*], less often seas [*H.* (*Amblystus*) *neglectus*]. In zoogeographical terms, in addition to the Palaearctic taxa, this stock also includes the majority of the Holarctic and Nearctic species. On the contrary, only taxa inhabiting open landscapes, including the Mediterranean type, both steppe and desert, belong to the *affinis* stock. Among them, species associated with azonal habitats clearly predominate, i.e., they are found in arid territories mainly along the banks of water bodies, in various wetter depressions, on saline soils and salt-marshes. The ranges of most of the species lie within the Palaearctic, concentrating in the Mediterranean region, western and middle Asia. The only exceptions are two species of the subgenus *Pharalus*, two or three species of the predominantly Palaearctic subgenus *Brachyharpalus* **subg. n.**, and the nominotypical subspecies of the Holarctic *H.* (*Harpalus*) *amputatus*, which all are distribured in North America. The latter species belongs to the nominotypical subgenus (as it is here treated), all other species of which are distributed in the Palaearctic. There is no doubt that the origin and subsequent evolution of these two stocks of the *Harpalus* group took place in completely different geographical and ecological conditions [8].

*Afroharpalus* Subgroup

Diagnosis. Same as for the subgenus.

Composition and distribution. A monobasic subgroup, including only one Afrotropical subgenus.

Subgenus *Afroharpalus* **subg. n.**

https://zoobank.org/urn:lsid:zoobank.org:act:A2DE73C8-D277-4E24-9D2E-ABB110F44F46
Type species *Harpalus fulvicornis* Thunberg, 1806.

Diagnosis. Size small to large (length 4.9–12.5 mm). Body flattened or moderately convex, somewhat wide or elongate, brownish yellow to black, with or without metallic luster on dorsum. Pronotum with one lateral seta on each side before middle and with glabrous or (more rarely) setose basal edge; surface in most species almost impunctate, rarely (e.g., in *H. massarti*) densely punctate basally. Elytra impunctate and glabrous, with glabrous basal border; interval 3 generally with one discal setigerous pore, in some species this pore absent. Metepisternum elongate or slightly wider than long. Metacoxa generally without posteromedial setigerous pore, in some species this pore present. Metafemur usually with three, sometimes four, setigerous pores along posterior margin. Protibia with one ventroapical spine and with three (more rarely two) preapical spines on outer margin, isolated from spines on ventral surface of tibia; ventroapical tubercle in male generally absent, rarely (e.g., in *H. fuscipennis*) more or less prominent. Male mesotibia without preapical callous thickening on inner margin. Tarsi glabrous dorsally. Abdominal sternites glabrous; last visible abdominal sternite without pronounced sexual dimorphism, its apex in both sexes more or less rounded and not swollen. Median lobe of aedeagus with comparatively long terminal lamella (its length greater than width) and usually without apical capitulum (in some species apical capitulum more or less developed, horseshoe-shaped); internal sac generally with small spiny patches, in some species also with one or two separate spines or without any sclerotic elements.

Etymology. The subgeneric name is based on a combination of *Africa* and the name of the carabid taxon *Harpalus*.

Composition and distribution. This subgenus comprises all Afrotropical *Harpalus* (about 55 described species), with the exception of most Madagascan species, which are treated as members of the separate genus *Anisochirus* Jeannel, 1946 [55], and with the exception of *H. agnatus*, which is treated as a member of the subgenus *Cryptophonus*. Among them, the majority (52 species) are distributed in east and south Africa; two species, *H. sanctaehelenae* Basilewsky, 1972, and *H. prosperus* Basilewsky, 1972, were described from Saint Helena Island, and one species, *H. rivalsi* Jeannel, 1948, seems to be endemic or introduced to Reunion Island [55]. There are no common species for east and south Africa, though, according to preliminary data, some east African species are replaced in south Africa by very close vicariant taxa.

Fauna of east Africa includes 16 species: *H. impressus* Roth, 1851, *H. asemus* Basilewsky, 1946, *H. pseudoasemus* Kataev, 2021, *H. merkli* Kataev, 2021, *H. meteorus* Basilewsky, 1946, *H. jeanneli* Basilewsky, 1946, *H. somereni* Basilewsky, 1946, *H. kibonoti* Alluaud, 1926, *H. gilgil* Basilewsky, 1946, *H. inconcinnus* Chaudoir, 1876, *H. procognatus* Lorenz, 1998, *H. frater* Chaudoir, 1876, *H. gregoryi* Alluaud, 1917 (Figure 25e,f), *H. clarkei* Kataev et Schmidt, 2020, *H. rougemonti* Clarke, 1973 and *H. baleensis* Clarke, 1973. Among these species, *H. asemus* and *H. inconcinnus* occur also in the southern part of the Arabian Peninsula (Yemen and Saudi Arabia).

Fauna of south Africa are more diverse and comprises 36 species: *H. fulvipennis* Chaudoir, 1843, *H. exiguus* Boheman, 1848, *H. minutissimus* Facchini, 2015, *H. nanniscus* Peringuey, 1896, *H. parvulus* Dejean, 1829 (Figure 25c,d), *H. fuscoaeneus* Dejean, 1829, *H. massarti* Burgeon, 1935, *H. natalicus* Peringuey, 1896, *H. diversicollis* Basilewsky, 1958, *H. nyassicus* Basilewsky, 1946, *H. venator* Boheman, 1848, *H. makhekensis* Basilewsky, 1958, *H. subaeneus* Boheman, 1848, *H. miles* Peringuey, 1896, *H. defector* Peringuey, 1896, *H. rufocinctus* Chaudoir, 1843, *H. sinuaticollis* Facchini, 2015, *H. fuscipennis* Wiedeman, 1825, *H. fulvicornis* Thunberg, 1806, *H. corrugatus* Basilewsky, 1958, *H. basuto* Basilewsky, 1958, *H. natalensis* Boheman, 1848 (Figure 26a), *H. parallelocollis* Facchini, 2003, *H. basilewskyi* Facchini, 2003, *H. rotundus* Facchini, 2003, *H. fimetarius* Dejean, 1829, *H. capicola* Dejean, 1829, *H. kmecoi* Facchini, 2003, *H. elliptipennis* Facchini, 2003, *H. hybridus* Boheman, 1848; *H. agilis* Peringuey, 1896, *H. dubius* Boheman, 1848, *H. lugubris* Boheman, 1848, *H. angustipennis* Boheman, 1848, *H. spurius* Peringuey, 1896 and *H. spretus* Peringuey, 1896. Two south African species, *H. parvulus* and *H. fulvicornis*, were introduced to Australia (the former species also to New Zealand). *Harpalus australasiae* Dejean, 1829, originally described from Australia, appears to have also been introduced from Africa and appears to be conspecific with either *H. fuscoaeneus* or *H. asemus* [55].

Ecology. As far as is known, all species of this group occur in various open habitats, mainly in mountainous areas.

Remarks. *Afroharpalus* **subg. n.** corresponds to the *fulvicornis* species group sensu Kataev [8,55]. This subgenus is in need of revision, so this review is preliminary. The only revision of the east and south African species was published by Basilewsky [13], but after that, 18 new species were described [55,119,152–156]. In addition, *Harpalus subphaedrus* Basilewsky, 2005 is transferred to the subgenus *Anisotarsus* Chaudoir, 1837, of the genus *Notiobia* Perty, 1830 (subtribe Anisodactylina) [103]. Although the Afrotropical species are somewhat variable in their morphology and appear to represent several different species groups, they are all included here in one subgenus since, in my opinion, they form a monophyletic unit, taxonomically isolated from all Holarctic congeners. The subgenus *Afroharpalus* **subg. n.** probably occupies a basal position in the *Harpalus* group and is characterized by an unmodified state of many morphological structures and generally by the absence of many apomorphies, which are observed in most of the Holarctic taxa; for example, the median lobe of many species lacks apical capitulum and has only two spines in internal sac; there are also no conspicuous sexual modifications of tibiae and last visible abdominal sternite. In structure of aedeagus, these species are very similar to members of *Pseudoophonus*, *Semiophonus*, *Zangoharpalus* and *Cryptophonus* [46]. Further study may demonstrate that *Afroharpalus* **subg. n.** warrants the status of a separate subgenus group.

Many Afrotropical species are very similar in their external morphology and can reliably be distinguished only by features of male genitalia.

*Hyloharpalus* Subgroup

Diagnosis. Body moderately convex, glabrous on dorsum; legs comparatively slender and long. Head smooth, at most very finely punctate dorsally. Pronotum with sides generally not sinuate basally (this character variable in *H. spadiceus*), with one lateral seta on each side and with glabrous basal edge; surface either more or less punctate (mainly basally) or almost smooth, only with a few punctures in basal foveae. Elytra glabrous, generally impunctate, more rarely punctate along sides or throughout, with glabrous basal border; humeral denticle present in most members; interval 3 with one or several discal setigerous pores, rarely without pore; intervals 7 and 5 without preapical pores. Metacoxa generally without additional setae, more rarely with one or several additional setae medially. Metafemur with two to six setigerous pores along posterior margin. Protibia with one or two ventroapical spines and with three (more rarely four) preapical spines on outer margin, isolated from spines on ventral surface of tibia; ventroapical tubercle in male of most species not developed. Male mesotibia in most species with more or less distinct preapical callous thickening on inner margin (Figure 15c). Tarsi glabrous, rarely (in *H. puetzi*) setose dorsally; metatarsomere 1 average for genus or slightly more elongate. Abdominal sternites generally glabrous, in a few species with short additional setae; last visible sternite without pronounced sexual dimorphism, its apex in both sexes more or less rounded and not swollen. Median lobe of aedeagus with moderately long terminal lamella and distinct horseshoe-shaped apical capitulum (modified in *H. laevipes*); internal sac generally with one separate spine, one to three groups of medium-sized spines and usually also with one or several spiny patches.

Composition and distribution. This subgroup comprises five subgenera with Holarctic, eastern Palaearctic and Nearctic distributions.

Ecology. Most species of this subgroup occur in forests or at forest edges.

Remarks. Many species of this subgroup are apterous, with short metepisterna, and have more or less distinct preapical callous thickening on inner margin of male mesotibia.

Subgenus *Hyloharpalus* **subg. n.**

https://zoobank.org/urn:lsid:zoobank.org:act:DDE495ED-4DED-4640-B61A-F757209C6F23
Type species *Harpalus laevipes* Zetterstedt, 1828.

Diagnosis. Medium-sized to large (length 8.0–11.0 [12.5] mm). Body wide or elongate, dark brown to black, without metallic luster. Pronotal disc densely punctate along base. Elytra impunctate; interval 3 with several, up to six, large foveate discal setigerous pores, occasionally (usually on one interval) with one such pore or without pore; abbreviate (parascutellar) striole long, with or without basal pore; preapical sinuation shallow or somewhat deep with small denticle at its base; microsculpture consisting of isodiametric or weakly transverse meshes. Metepisternum either elongate, longer than wide, or short and wide, wider than long. Metacoxa with or without additional setae medially. Protibia with one ventroapical spine; ventroapical tubercle in male not or only slightly prominent. Tarsi glabrous dorsally. Internal sac of aedeagus with two compact groups of medium-sized spines and a medium-sized separate spine.

Etymology. The subgeneric name is a combination of the Greek *húlē*, meaning "wood, forest", and the name of the carabid taxon *Harpalus*.

Composition and distribution. This subgenus includes the Holarctic *H. laevipes* Zetterstedt, 1828 (Figure 27a,b) and two east Asian species: *H. tibeticus* Andrewes, 1930 (Figure 26b), and *H. farkaci* Kataev et Wrase, 1995.

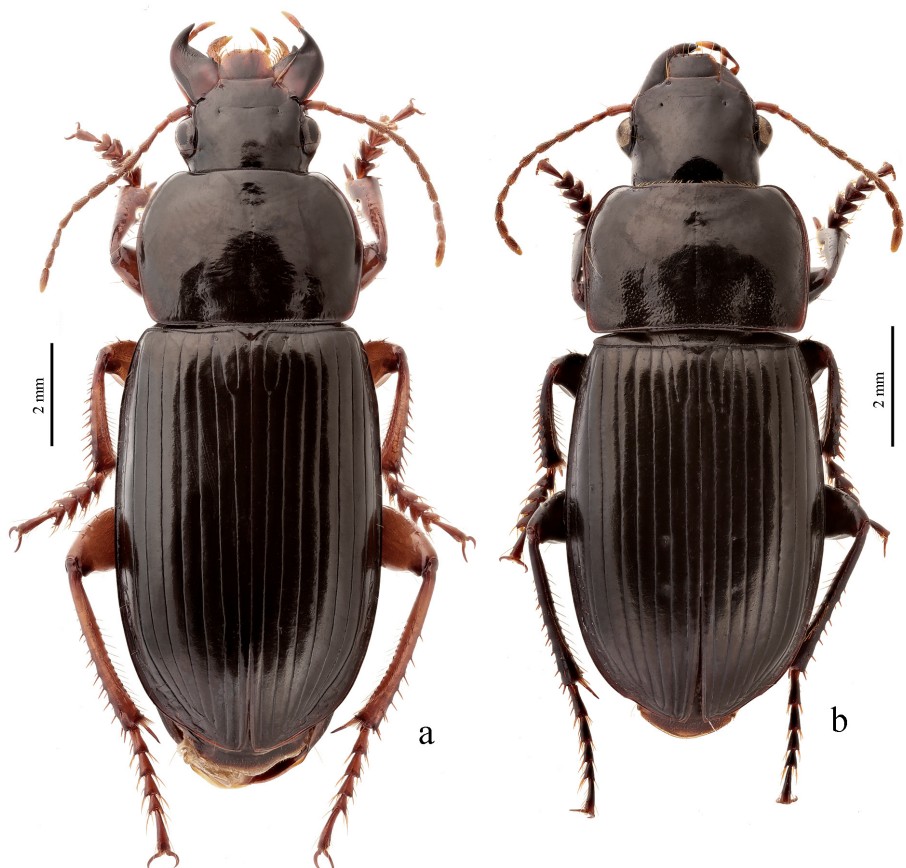

**Figure 26.** Habitus of *Harpalus* species: (**a**) *H.* (*Afroharpalus*) *natalensis*; (**b**) *H.* (*Hyloharpalus*) *tibeticus*.

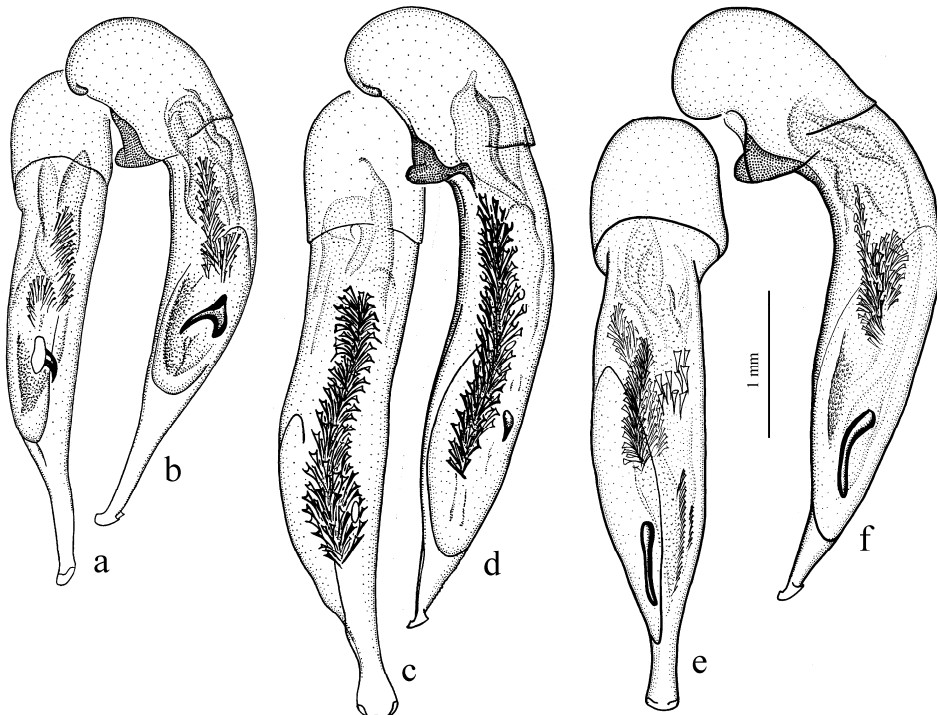

**Figure 27.** Median lobe of aedeagus of *Harpalus* species, dorsal and lateral views: (**a**,**b**) *H.* (*Hyloharpalus*) *laevipes*; (**c**,**d**) *H.* (*Sinoharpalus*) *puetzi* (from [116]); (**e**,**f**) *H.* (*Macroharpalus*) *major*.

Remarks. The species of this subgenus differ from the species of other subgenera of the *Hyloharpalus* subgroup in the presence of usually several large foveate discal setigerous pores on elytral interval 3 and in the characteristic male genitalia with two compact groups of medium-sized spines and a medium-sized separate spine in the internal sac (as in Figure 27a,b).

*Hyloharpalus* **subg. n.** corresponds to the *quadripunctatus* (=*laevipes*) species group sensu Kryzhanovskij et al. [56], sensu Kataev [8,108] and sensu Kataev & Wrase [116]. Noonan [16] treated *H. quadripunctatus* (=*H. laevipes*), the only species occurring in North America, as a member of the monobasic *quadripunctatus* group.

The genesis of this subgenus (as the *laevipes* species group) was published by Kataev [108].

## Subgenus *Sinoharpalus* **subg. n.**

https://zoobank.org/urn:lsid:zoobank.org:act:74486C54-2F24-4EA1-BC41-2A60FCE8107D
Type species *Harpalus puetzi* Kataev et Wrase, 1997.

Diagnosis. Medium-sized to large (length 9.2–12.4 mm). Body slightly elongate, dark brown to black, without metallic luster. Pronotum densely punctate along base. Elytra finely punctate or smooth; interval 3 generally with one (occasionally two) small discal setigerous pores; abbreviate (parascutellar) striole short (usually not longer than width of intervals 1 and 2 combined basally), with basal pore; preapical sinuation somewhat deep with small denticle at its base; microsculpture consisting of transverse or isodiametric meshes. Metepisternum short and wide, wider than long. Metacoxa with or without additional setae medially. Protibia with one ventroapical spine. Tarsi setose or glabrous dorsally. Internal sac of aedeagus with a large longitudinal compact group of medium-sized spines and a small or medium-sized separate spine.

Etymology. The subgeneric name is a combination of *Sina* (China) and the name of the carabid taxon *Harpalus*.

Composition and distribution. This group includes two described species from China: *H. puetzi* Kataev et Wrase, 1997 from Shaanxi (Figure 27c,d) and *H. hiekei* Kataev et Wrase, 2010 (Figure 28a) from Hubei and Sichuan. Several taxa of this subgenus also from China are still undescribed.

Remarks. This subgenus is very similar and apparently closely related to *Hyloharpalus* **subg. n.**, differing mainly in elytra with a short abbreviate striole and with generally one discal setigerous pore on interval 3, and in the presence of only one large longitudinal group of medium-sized spines in internal sac; also in some members, tarsi are more or less setose dorsally and elytra finely punctate throughout or along sides.

*Sinoharpalus* **subg. n.** corresponds to the *puetzi* species group sensu Kataev [8].

## Subgenus *Macroharpalus* **subg. n.**

https://zoobank.org/urn:lsid:zoobank.org:act:13FA9D2B-DF5F-47C3-8513-A4D675E4AEF4
Type species *Erpeinus major* Motschulsky, 1850.

Diagnosis. Comparatively large (length 10.0–13.7 [15.5] mm). Body stout, wide, dark brown to black, without metallic luster. Pronotal disc almost smooth, with very fine, occasionally indistinct micropunctation along base. Elytra impunctate; interval 3 with one discal setigerous pore; preapical sinuation shallow, without denticle at its base; microsculpture isodiametric. Metepisternum elongate, longer than wide. Metacoxa without additional setae medially. Metafemur with three (occasionally two and four) setigerous pores along posterior margin. Protibia with one or two ventroapical spines. Tarsi glabrous dorsally. Internal sac of aedeagus with one large separate spine, one to three groups of medium-sized spines, and a spiny patch apically.

Etymology. The subgenus name is a combination of the Greek *makrós*, meaning "big, large", and the name of the carabid taxon *Harpalus*.

Composition and distribution. This subgenus includes the eastern Palaearctic *H. major* (Motschulsky, 1850) (Figure 27e,f) and three Nearctic species: *H. providens* Casey, 1914, *H. animosus* Casey, 1924 (Figure 28b), and *H. laticeps* LeConte, 1850.

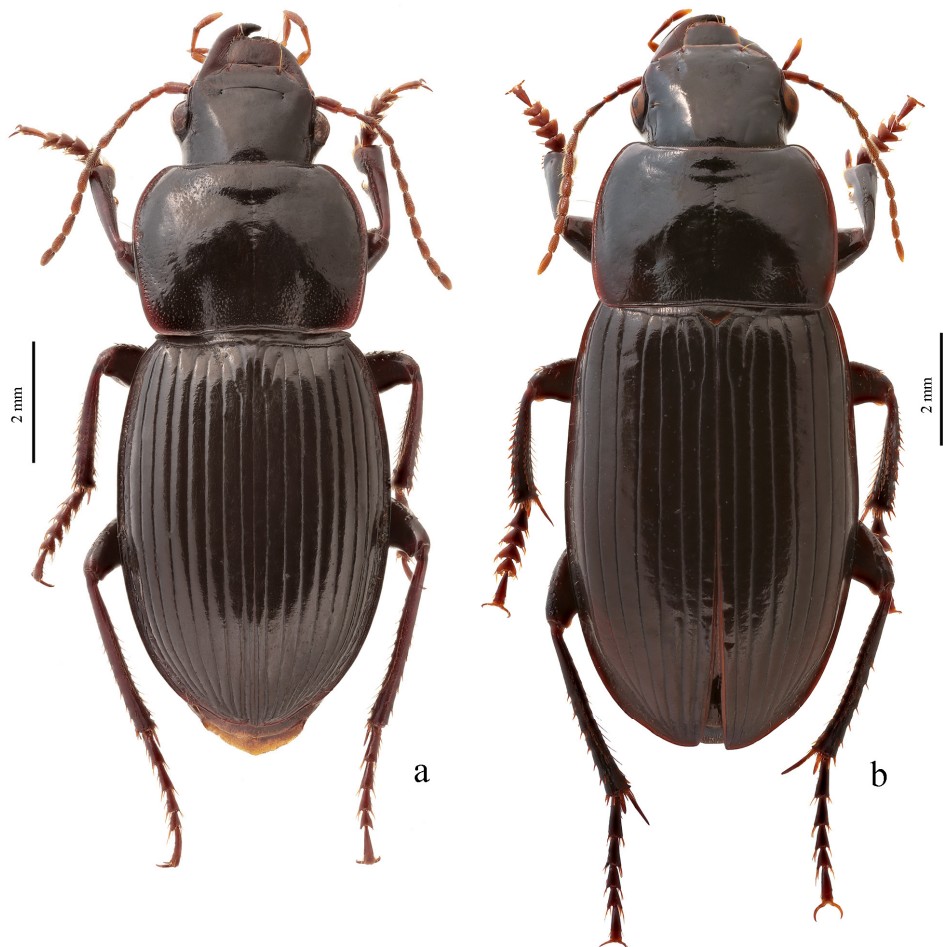

**Figure 28.** Habitus of *Harpalus* species: (**a**) *H.* (*Sinoharpalus*) *hiekei*; (**b**) *H.* (*Macroharpalus*) *animosus*.

Ecology. The most species of this subgenus occur mainly in forests; *H. animosus* lives in mountains, subalpine and alpine zones, where it occurs in meadows, pastures, moraines and forest clearings [140].

Remarks. This subgenus can be recognized among other subgenera of this subgroup in large stout body with almost smooth (without distinct punctation) pronotum.

*Macroharpalus* **subg. n.** corresponds to the Palaearctic *obesus* species group sensu Kryzhanovskij et al. [56], and to the *major* species group sensu Kataev [8], including both Palaearctic and Nearctic species. Lindroth [15] included the Nearctic species of this subgenus in the *fraternus* species group (=*Euharpalops*), which he considered very widely. Noonan [16] included *H. viduus* (=*H. providens*) and *H. animosus* in the *viduus* species group and treated *H. laticeps* as a member of the closely related, monobasic *laticeps* group based on some features of the male genitalia. I prefer to consider these three species as members of one subgenus. The Palaearctic *H. major* is very similar to the Nearctic *H. providens* both in external features and male genitalia.

Subgenus *Meroharpalus* **subg. n.**

https://zoobank.org/urn:lsid:zoobank.org:act:C76A4DC1-238F-4BA4-AC4E-A5C30B0126CF
Type species *Harpalus fulvilabris* Mannerheim, 1853.

Diagnosis. Medium-sized (length 7.4–9.5 [10.8] mm). Body elongate, dark brown to black, with metallic luster on dorsum. Pronotal disc densely punctate along base. Elytra impunctate; interval 3 either without discal setigerous pore or with one, occasionally two, pores; basal pore present; abbreviate (parascutellar) striole comparatively long; preapical sinuation shallow, without denticle at its base; microsculpture isodiametric, in male highly obliterate on disc. Metepisternum elongate or as long as wide. Metacoxa without additional setae medially. Protibia with one ventroapical spine; ventroapical tubercle in male not developed or slightly prominent. Tarsi glabrous dorsally. Internal sac of aedeagus with one or two medium-sized separate spines, one or two groups of medium-sized spines and a spiny patch apically.

Etymology. The subgeneric name is a combination of the Greek *méros*, meaning "part, portion", and the name of the carabid taxon *Harpalus*.

Composition and distribution. This subgenus includes two Nearctic species: *H. fulvilabris* Mannerheim, 1853 (Figure 29a), and *H. megacephalus* LeConte, 1848.

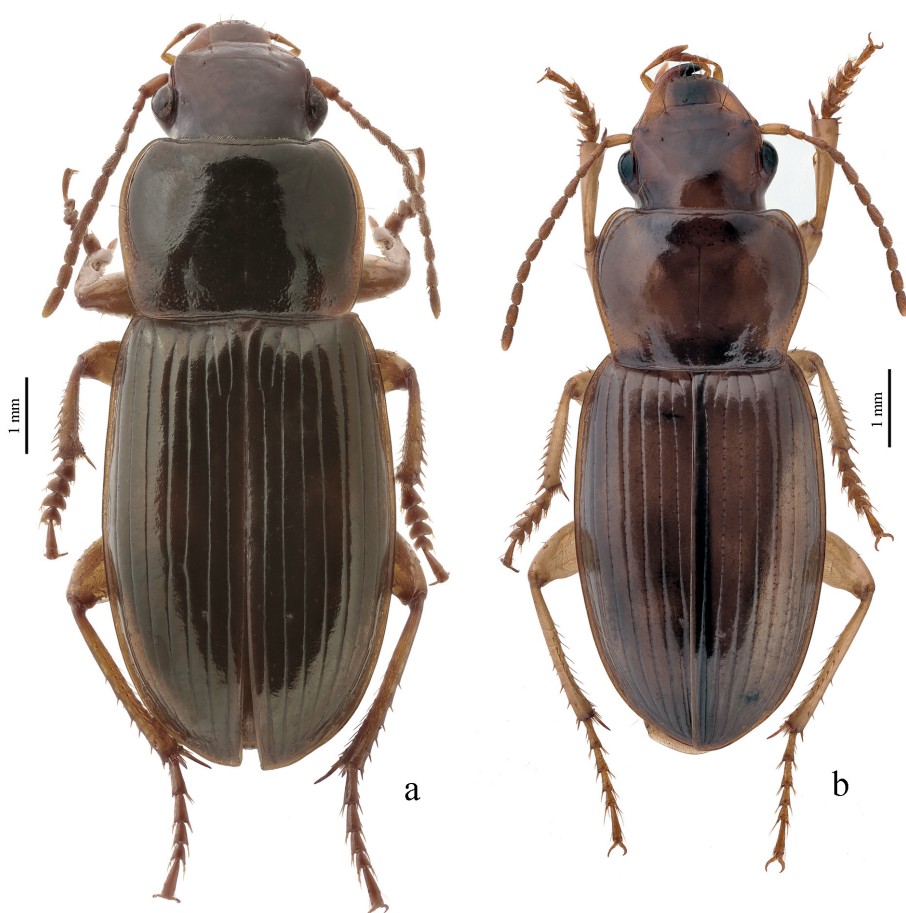

**Figure 29.** Habitus of *Harpalus* species: (**a**) *H. (Meroharpalus) fulvilabris*; (**b**) *H. (Ameroharpalus) spadiceus* (teneral).

Remarks. The species of this subgenus are recognizable among other members of the *Hyloharpalus* subgroup by the metallic luster on dorsum and the characteristic male genitalia.

This subgenus corresponds to the *fulvilabris* species group sensu Lindroth [15]. Noonan [16] included *H. fulvilabris* together with *H. spadiceus* in the *spadiceus* group and treated *H. megacephalus* together with *H. nigritarsis* as members of the separate *nigritarsis* group; according to my data, the latter species in his interpretation is actually a complex of several taxa. The position of *H. megacephalus*, known to me only from one female, needs further study.

Subgenus *Ameroharpalus* **subg. n.**

https://zoobank.org/urn:lsid:zoobank.org:act:F62503C7-BB56-4F6D-8409-03FEC1DD96DC
Type species *Harpalus spadiceus* Dejean, 1829.

Diagnosis. Medium-sized (length 7.7–9.0 [10.5] mm). Body elongate, dark brown to black, without metallic luster. Pronotal disc with very fine, often more or less obliterate punctation along base. Elytra impunctate; interval 3 with one discal setigerous pore; abbreviate (parascutellar) striole short or moderately long, with basal pore; preapical sinuation shallow, without denticle at its base; microsculpture clearly transverse, in male highly obliterate on disc. Metepisternum short and wide, wider than long. Metacoxa without additional setae medially. Protibia with one ventroapical spine; ventroapical tubercle in male not prominent. Tarsi glabrous dorsally. Internal sac of aedeagus with a large group of medium-sized spines, one medium-sized separate spine and a spiny patch apically.

Etymology. The subgeneric name is based on a combination of *America* and the name of the carabid taxon *Harpalus*.

Composition and distribution. This subgenus includes only the Nearctic *H. spadiceus* Dejean, 1829 (Figure 29b), distributed in the mountains of the east of the continent (Appalachians and Black Mountains). This wingless species is represented by several geographical forms, the status of which requires further study.

Remarks. This subgenus corresponds to the *spadiceus* species group sensu Lindroth [15]. The only included species of this subgenus differs from the species of *Meroharpalus* **subg. n.** in clearly transverse elytral microsculpture, metepisternum wider than long and dorsum dark, without metallic luster; in addition, its pronotum is with finer, often more or less obliterate punctation along base.

*Amblystus* Subgroup

Diagnosis. Body moderately convex or flattened. Head impunctate or punctate, glabrous or sparsely setose on dorsum; legs comparatively slender and long. Pronotum with sides sinuate or not sinuate basally, with one lateral seta on each side (in *H. chobautianus* with an additional setigerous pore before basal angles) and with basal edge glabrous or setose; surface in most species more or less coarsely and densely punctate, rarely impunctate. Elytra either impunctate and glabrous or punctate and pubescent; basal border glabrous or setose, humeral denticle in most species present; interval 3 generally with one discal setigerous pore; intervals 7 and 5 with or without preapical pores. Metacoxa generally without additional setae medially, rarely (for example, in *H. neglectus* with additional setae). Metafemur with three to ten setigerous pores along posterior margin. Protibia with one ventroapical spine and with three or four preapical spines on outer margin, isolated from spines on ventral surface of protibia; ventroapical tubercle in male not prominent. Male mesotibia without preapical callous thickening on inner margin. Tarsi glabrous dorsally. Abdominal sternites usually glabrous, more rarely with additional long setae; last visible abdominal sternite without pronounced sexual dimorphism, its apex in both sexes more or less rounded and not swollen, more rarely slightly blunted in male. Median lobe of aedeagus with a moderately long terminal lamella and a distinct horseshoe-shaped apical capitulum; internal sac usually with two (rarely one) more or less large separate spines, also a group of medium-sized or small spines and generally one or several spiny patches.

Composition and distribution. This subgroup comprises six Palaearctic subgenera, with most species distributed in the western part of the Palaearctic, mainly in mountainous regions; one monobasic subgenus is endemic to mountains in the Chinese province of Sichuan.

Remarks. This subgroup is very similar to *Hyloharpalus* subgroup in many distinctive characters listed in their diagnoses, but male mesotibia lacks preapical callous thickening on inner margin. Since this feature is somewhat variable within *Hyloharpalus* subgroup, the relationship and composition of these two subgroups need further study.

The *Amblystus* subgroup includes many morphologically distinct mountain wingless species, the ranges of which are sometimes significantly distant from each other.

Subgenus *Drymoharpalus* **subg. n.**

https://zoobank.org/urn:lsid:zoobank.org:act:1A461C7C-7AE7-4DF1-802F-7CAA93CFCA32
Type species *Harpalus atratus* Latreille, 1804.

Diagnosis. Comparatively large size (length 10.0–13.0 mm). Head impunctate and glabrous. Body moderately wide, dark brown to black, without metallic luster. Pronotal basal edge glabrous. Elytra impunctate and glabrous, with glabrous basal border and distinct humeral denticle; preapical sinuation shallow; intervals 7 and 5 without preapical pores. Metepisternum elongate, longer than wide. Abdominal sternites glabrous. Internal sac of aedeagus with one large separate spine and two groups of small spines medially (Figure 32a,b).

Etymology. The subgeneric name is a combination of the Greek *drymós*, meaning "wood, forest", and the name of the carabid taxon *Harpalus*.

Composition and distribution. This subgenus includes only one winged species, *H. atratus* Latreille, 1804 (Figures 30a and 32a,b), from the western Palaearctic.

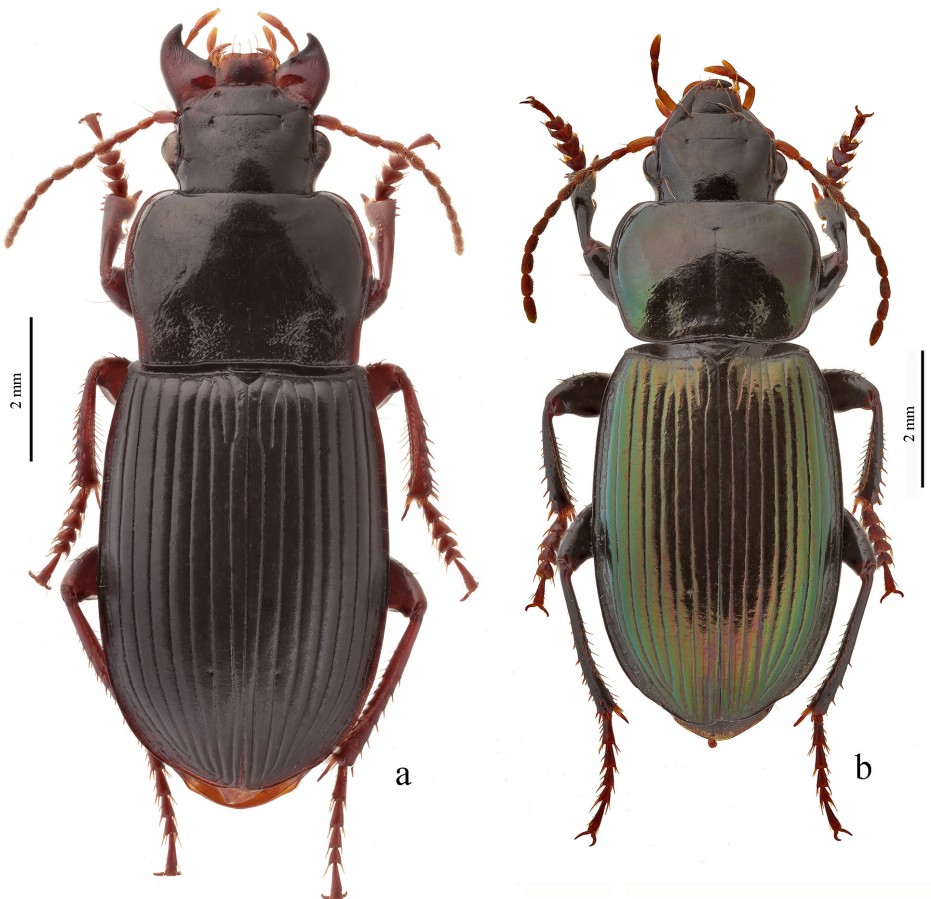

**Figure 30.** Habitus of *Harpalus* species: (**a**) *H.* (*Drymoharpalus*) *atratus*; (**b**) *H.* (*Caucasoharpalus*) *aeneipennis*.

Ecology. The only representative of this subgenus usually occurs in deciduous forests.

Remarks. The monobasic *Drymoharpalus* **subg. n.** is similar to members of *Epiharpalus* in general habitus and having glabrous basal edge of pronotum and abdominal sternites, but markedly differs from it in elongate metepisternum and only one separate spine in internal sac of aedeagus.

This subgenus corresponds to the *atratus* species group sensu Kryzhanovskij et al. [56] and sensu Kataev [8,109].

Subgenus *Epiharpalus* Reitter, 1900

*Epiharpalus* Reitter, 1900 [38] (pp. 75, 80) (as a subgenus of *Harpalus* Latreille, 1802). Type species *Harpalus punctipennis* Mulsant, 1852, designated by Antoine [40].
*Harpaloxys* Reitter, 1900 [38] (pp. 75, 94) (as a subgenus of *Harpalus* Latreille, 1802). Type species: *Harpalus cardioderus* Putzeys, 1872 (=*H. ebeninus* Heyden, 1870), designated by Noonan [6].
*Epharpalus*: Jakobson, 1907 [63] (p. 378) (print error).

Diagnosis. Medium-sized to moderately large (length 9.4–11.8 mm). Head impunctate and glabrous. Body moderately wide, dark brown to black, without metallic luster. Pronotal basal edge glabrous. Elytra either coarsely punctate and pubescent (laterally and apically) or impunctate and glabrous; basal border glabrous; humeral denticle present; preapical sinuation shallow or moderately deep, at most with traces of obtuse denticle at its base; intervals 7 and 5 without preapical pores. Metepisternum short and wide, wider than long. Abdominal sternites glabrous. Internal sac of aedeagus with two large separate spines, one or two groups of small spines medially and generally three spiny patches.

Composition and distribution. This subgenus includes two wingless species from west Europe: *H. ebeninus* Heyden, 1870 (Figures 31b and 32c,d), endemic to the Cantabrian Mountains in Spain, and *H. punctipennis* Mulsant, 1852 (Figures 31a and 32e,f), endemic to the Alpes-Maritimes on the border of France and Italy.

Ecology. The species occurs in mountain meadows and grasslands.

Remarks. Although the two included species are distinguished by elytral punctation, they are very similar in other characters and apparently closely related. So, they are included in one subgenus (the *punctipennis* species group sensu Kataev [8]).

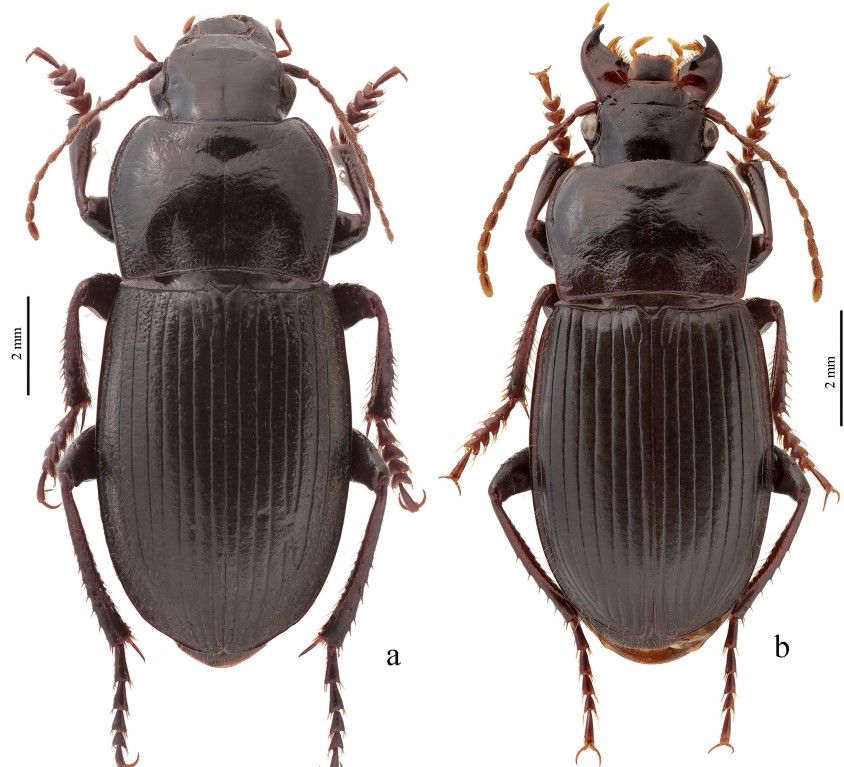

**Figure 31.** Habitus of *Harpalus* species: (**a**) *H. (Epiharpalus) punctipennis*; (**b**) *H. (E.) ebeninus*.

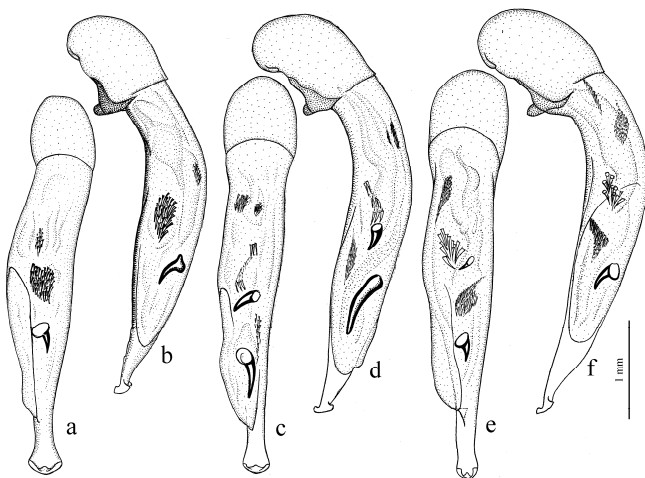

**Figure 32.** Median lobe of aedeagus of *Harpalus* species, dorsal and lateral views: (**a,b**) *H. (Drymoharpalus) atratus*; (**c,d**) *H. (Epiharpalus) ebininus*; (**e,f**) *H. (E.) punctipennis*.

Subgenus *Caucasoharpalus* **subg. n.**

https://zoobank.org/urn:lsid:zoobank.org:act:82E3273E-A83F-47FE-8DE5-1C9D8B364C35
Type species *Omaseus aeneipennis* Faldermann, 1836.

Diagnosis. Medium-sized (length 7.3–10.6 mm). Body slightly elongate, moderately convex, dark brown to black, with or without metallic luster. Head impunctate and glabrous. Pronotal basal edge glabrous. Elytra impunctate and glabrous, with glabrous basal border and prominent humeral denticle; preapical sinuation deep, with distinct denticle at its base; intervals 7 and 5 with or without preapical pores. Metepisternum short and wide, wider than long. Abdominal sternites glabrous. Internal sac of aedeagus with two separate closely spaced spines apically, one to three groups of small or medium-sized spines medially and a spiny patch apically.

Etymology. The subgeneric name is based on a combination of *Caucasus* and the name of the carabid taxon *Harpalus*.

Composition and distribution. This subgenus includes two wingless species from the West and Lesser Caucasus: *H. aeneipennis* (Faldermann, 1836) (Figures 30b and 33a,b) and *H. chrysopus* Reitter, 1887 (Figure 33c,d) (with the subspecies *H. ch. abasinus* Rost, 1891 and *H. ch. contumax* Lutshnik, 1933). Several taxa of this subgenus have not yet been described.

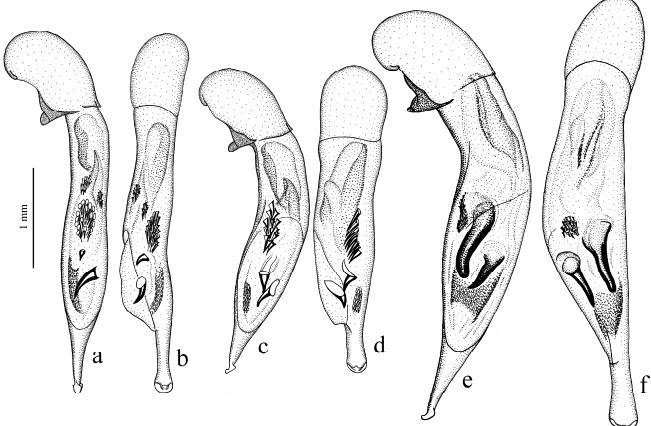

**Figure 33.** Median lobe of aedeagus of *Harpalus* species, dorsal and lateral views: (**a,b**) *H. (Caucasoharpalus) aeneipennis*; (**c,d**) *H. (C.) chrysopus*; (**e,f**) *H. (Calathoderus) potanini* (from [112]).

Ecology. Both species live in the mountains, both in the forest belt and higher, in alpine and subalpine meadows.

Remarks. This subgenus is morphologically very similar to *Epiharpalus*, but differs from it in the impunctate elytra combined with a deep preapical sinuation having a distinct denticle at the base and the characteristic male genitalia.

*Caucasoharpalus* **subg. n.** corresponds to the *aeneipennis* species group sensu Kryzhanovskij et al. [56] and sensu Kataev [8,109].

Subgenus *Calathoderus* **subg. n.**

https://zoobank.org/urn:lsid:zoobank.org:act:59D5F177-4CDF-44AF-A4C1-135BB4FDDA05
Type species *Harpalus potanini* Tschitschérine, 1906.

Diagnosis. Large-sized (length 10.8–14.0 mm). Body moderately wide and flat, black, without metallic luster; legs comparatively long and slender, with long and narrow, almost parallel-sided metatarsomeres. Head impunctate and glabrous. Pronotal basal edge glabrous. Elytra impunctate and glabrous, with glabrous basal border and without humeral denticle; preapical sinuation rather deep, with traces of obtuse denticle at its base; intervals 7 and 5 without preapical pores. Metepisternum short and wide, wider than long. Abdominal sternites glabrous. Internal sac of aedeagus with two large separate closely spaced spines, a small spiny patch medially and a larger spiny patch apically (Figure 33e,f).

Etymology. The subgeneric name is based on a combination the name of the carabid taxon *Calathus* and the Greek *dére*, meaning "neck/notum".

Composition and distribution. This subgenus includes one wingless species, *H. potanini* Tschitschérine, 1906 (Figures 33e,f and 34a), known from the mountains of the Chinese province of Sichuan.

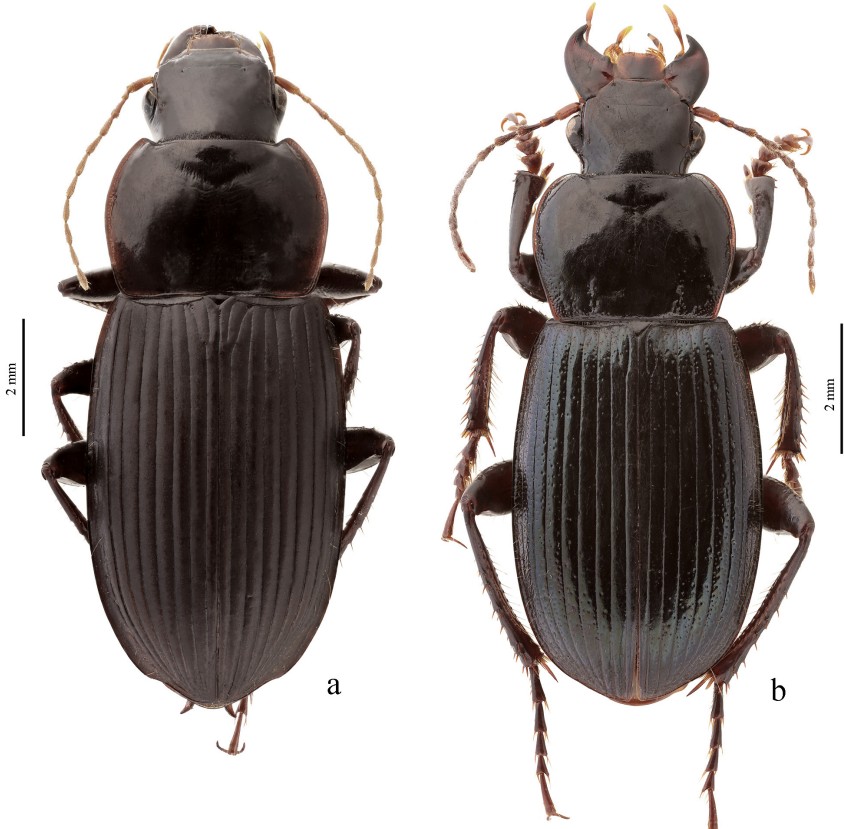

**Figure 34.** Habitus of *Harpalus* species: (**a**) *H.* (*Calathoderus*) *potanini*; (**b**) *H.* (*Licinoderus*) *franzi*.

Remarks. The position of this subgenus requires further study. It is remarkable in the appearance of the only included species, resembling some members of *Calathus* Bonelli, 1810, but similar in the morphology and probably related to *Caucasoharpalus* **subg. n.**

*Calathoderus* **subg. n.** shares with the latter subgenus almost all its distinctive characters listed in the diagnosis, differing mainly in the larger and less convex body, the longer legs, the absence of prominent humeral denticle on elytra and in the characteristic male genitalia.

*Calathoderus* **subg. n.** corresponds to the *potanini* species group sensu Kataev and Liang [112] and sensu Kataev [8,109].

Subgenus *Licinoderus* Sainte-Claire Deville, 1905

*Licinoderus* Sainte-Claire Deville, 1905 [157] (p. 114) (as a genus). Type species *Licinoderus chobauti* Sainte-Claire Deville, 1905, by monotypy.

*Neoharpalus* Mateu, 1954 [158] (p. 4) (as a subgenus of *Harpalus* Latreille, 1802). Type species *Harpalus franzi* Mateu, 1954, by monotypy.

*Baeticoharpalus* Serrano et Lecina, 2009 [159] (p. 194) (as a subgenus of *Harpalus* Latreille, 1802). Type species *Harpalus lopezi* Serrano et Lecina, 2009, by original designation.

Diagnosis. Medium-sized (length 8.0–10.4 mm). Body slightly elongate, brown to black, with or without metallic luster on dorsum. Head impunctate and glabrous, or with fine punctures and very sparse setae dorsally; tempora setose or glabrous. Pronotal basal edge glabrous or setose. Elytra punctate and pubescent, with setose basal border and distinct humeral denticle; preapical sinuation shallow, without denticle at its base; interval 7 and 5 with or without preapical pores. Metepisternum short and wide, wider than long. Abdominal sternites with additional long setae. Internal sac of aedeagus with one or two large spines, one or two groups of small spines medially and generally a spiny patch apically.

Composition and distribution. This subgenus includes three wingless species from the Iberian Peninsula: *H. chobautianus* Lutshnik, 1922 (Figure 35a,b), endemic to the Pyrenees, *H. franzi* Mateu, 1954 (Figures 34b and 35c,d), endemic to the Cantabrian Mountains, and *H. lopezi* Serrano et Lecina, 2009, endemic to the Baetic Mountains.

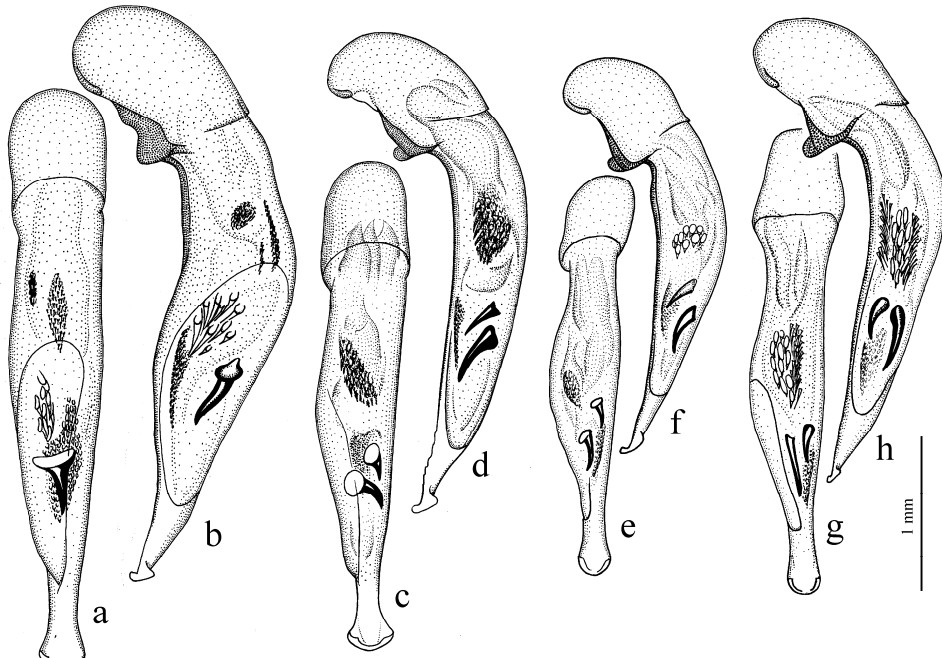

**Figure 35.** Median lobe of aedeagus of *Harpalus* species, dorsal and lateral views: (**a**,**b**) *H.* (*Licinoderus*) *chobautianus* (from [99]); (**c**,**d**) *H.* (*L.*) *franzi* (from [99]); (**e**,**f**) *H.* (*Amblystus*) *sulphuripes*; (**g**,**h**) *H.* (*A.*) *rubripes*.

Ecology. All three species occur in open mountain habitats.



Remarks. The species of *Licinoderus* well differs from other members of this subgroup in having punctate and pubescent elytra combined with setose basal border and abdominal sternites with additional long setae.

*Licinoderus* is usually considered as a separate genus (e.g., [43,160]), although it has all the features of *Harpalus* and should be included in this genus [99]. Jeanne [160] synonymized *Neoharpalus* and *Licinoderus*. The synonymy of *Baeticoharpalus* and *Licinoderus* was stated by Kataev [109].

This subgenus corresponds to the *chobautianus* species group sensu Kataev [8,109].

Subgenus *Amblystus* Motchulsky, 1864

*Amblystus* Motchulsky, 1864 [61] (p. 209) (as a genus). Type species *Carabus rubripes* Duftschmidt, 1812, by original designation.
*Harpaloderus* Reitter, 1900 [38] (pp. 76, 100) (as a subgenus of *Harpalus* Latreille, 1802). Type species *Harpalus sulphuripes* Germar, 1824, designated by Habu [29].

Diagnosis. Medium-sized (length 6.4–11.9 mm). Body moderately wide or elongate, dark brown to black, often with metallic luster on dorsum. Head impunctate and glabrous. Pronotal basal edge setose. Elytra impunctate and glabrous, with a more or less prominent humeral denticle; basal border generally glabrous, in some species setose; intervals 7 and 5 often with preapical pores. Metepisternum either elongate, longer than wide, or short and wide, wider than long. Abdominal sternites in most species with long additional setae, in some species glabrous. Internal sac of aedeagus with two more or less large separate spines, one or two groups of small spines medially and a spiny patch apically.

Composition and distribution. This subgenus comprises 17 winged and wingless species, distributed mainly in the western Palaearctic, with two centers of species diversity—one, larger, in the Mediterranean region, mainly in its western part, and the second, much smaller, in the western part of the Himalaya region; one species, *H. rubripes* (Duftschmid, 1812) (Figure 35g,h), has a trans-Palaearctic distribution.

The Mediterranean center of diversity includes 12 species, both winged and wingless: *H. rufipalpis* Sturm, 1818 (Figure 11c,d) (with the subspecies *H. r. montanellus* Mateu, 1953, *H. r. machadoi* Jeanne, 1970, and *H. r. lusitanicus* Schatzmayr, 1943), *H. wagneri* Schauberger, 1926, *H. nevadensis* K. et J. Daniel, 1898, *H. rufitarsoides* Schauberger, 1934, *H. dissitus* Antoine, 1931, *H. wohlberedti* Emden et Schauberger, 1932 (Figure 36a), *H. bellieri* Reiche, 1861, *H. decipiens* Dejean, 1829 (with the subspecies *H. d. correiroi* Schatzmayr, 1943 and *H. d. latianus* Schauberger, 1923), *H. honestus* (Duftschmid, 1812) (with the subspecies *H. h. creticus* Maran, 1934), *H. sulphuripes* Germar, 1824 (Figure 35e,f) (with the subspecies *H. s. goudotii* Dejean, 1829), *H. neglectus* Audinet-Serville, 1821 (with the subspecies *H. n. mayeti* Verdier, Quezel et Rioux, 1951 and *H. n. alluaudi* Antoine, 1922) and *H. attenuatus* Stephens, 1828.

The Himalayan center of diversity comprises four apterous species: *H. indicola* Bates, 1878 (with the subspecies *H. i. uriensis* Schauberger, 1933, *H. i. kashmirensis* Bates, 1889, *H. i. kirschenhoferi* Kataev, 2002 and *H. i. shogranensis* Kataev, 2002), *H. calciatii* Della Beffa, 1931, *H. hartmanni* Kataev, 2002 and *H. morvani* Kataev, 2002.

Ecology. The species of this subgenus inhabit predominantly the mountains, more rarely the lowlands, occurring in various open biotopes. *Harpalus neglectus* is a psammophilous species, occurring usually in dunes near the coast.

Remarks. The species of *Amblystus* are readily distinguished from other members of this subgroup in having setose pronotal basal edge combined with impunctate and glabrous elytra.

The species of this subgenus, especially the western Mediterranean taxa, require revision, so the exact status of some forms is not yet entirely clear.

This subgenus corresponds to the *Harpalus rufitarsis* (=*H. rufipalpis*) species group (=*Harpaloderus*) sensu Schauberger [64,161,162], who published the revision of the Mediter-

ranean and European species of this subgenus known in that time. The Himalayan species (as the *honestus* group) were revised by Kataev [98].

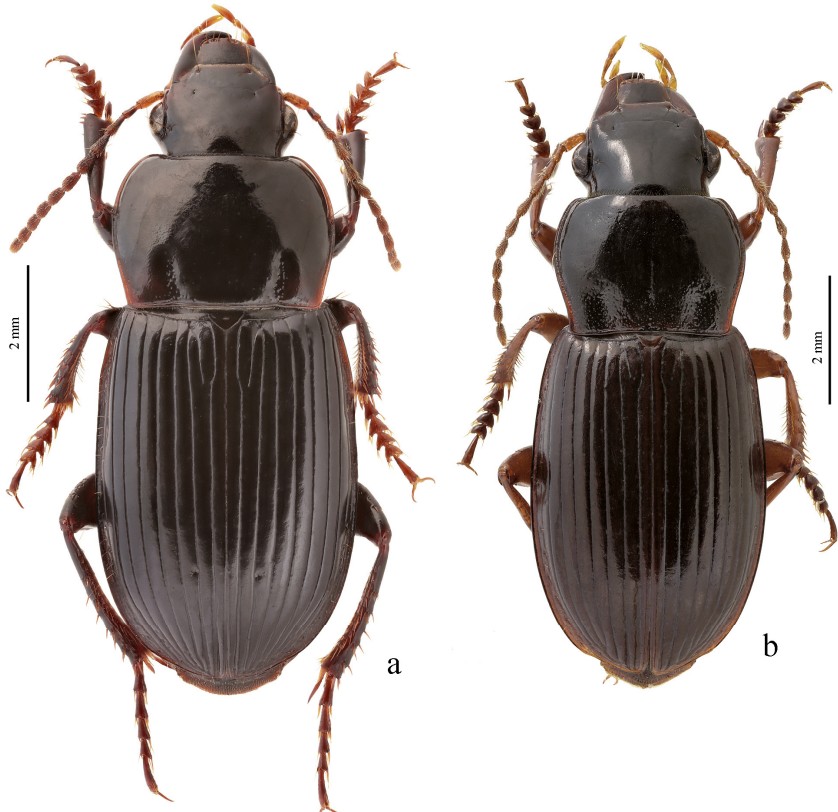

**Figure 36.** Habitus of *Harpalus* species: (**a**) *H. (Amblystus) wohlberedti*; (**b**) *H. (Cordoharpalus) cordifer*.

*Cordoharpalus* Subgroup

Diagnosis. Same as for the subgenus.

Composition and distribution. A monobasic subgroup, including only one Nearctic subgenus.

Subgenus *Cordoharpalus* Hatch, 1949

*Cordoharpalus* Hatch, 1949 [74] (p. 87) (as a subgenus of *Harpalus* Latreille, 1802). Type species *Harpalus cordifer* Notman, 1919, by original designation.

Diagnosis. Medium-sized (length 7.5–9.0 [9.8] mm). Head micropunctate dorsally, glabrous. Body moderately convex, somewhat elongate, dark brown to black, without metallic luster. Pronotum with sides sinuate basally, with one lateral seta on each side and with glabrous, not setose, basal edge; surface coarsely punctate along base. Elytra impunctate and glabrous, without humeral denticle, with glabrous basal border; interval 3 without discal setigerous pore; intervals 7 and 5 without preapical pores. Metacoxa without additional setae medially. Metafemur with two or three setigerous pores along posterior margin. Protibia with three ventroapical spines and with three (more rarely four) preapical spines on outer margin, isolated from spines on ventral surface of tibia; ventroapical tubercle in male not prominent. Male mesotibia without preapical callous thickening on inner margin. Tarsi glabrous dorsally. Abdominal sternites glabrous; last visible sternite without pronounced sexual dimorphism, its apex in both sexes more or less rounded and not swollen. Median lobe of aedeagus with moderately long terminal lamella and distinct horseshoe-shaped apical capitulum; internal sac with two medium-sized separate spines and a group of narrow, medium-sized spines.

Composition and distribution. The only representative of this subgenus, *H. cordifer* Notman, 1919 (Figure 36b), is distributed along the northwestern coast of North America from Alaska to Oregon.

Ecology. This species inhabits mainly forests and their glades, sometimes thickets or shrubs in fields, occurring in shaded or open localities, with moderately dry soil covered with dense leaf litter and humus [140].

Remarks. In combination of characters, the single subgenus of this subgroup is similar to the subgenera of the *Hyloharpalus* and *Amblystus* subgroups but differs from them in having three ventroapical spines on protibia. It is also distinguished from most of the members of the *Hyloharpalus* subgroup by having two separate spines in the internal sac of aedeagus.

The subgenus *Cordoharpalus* corresponds to the monobasic *cordifer* species group sensu Lindroth [15] and sensu Kataev [8]. Noonan [16] also included in it *H. tadorcus* Ball, 1972 (=*H. cordatus*), the type species of the subgenus *Opadius*, without a mentioning of the name of this subgenus in his revision. The relationships of these taxa were discussed by Kataev [106].

*Actephilus* Subgroup

Diagnosis. Body more or less convex and wide; legs relatively short. Head impunctate and glabrous. Pronotum generally impunctate, with sides more or less rounded, not sinuate basally, with one lateral seta on each side and with setose or glabrous basal edge. Elytra impunctate and glabrous, with glabrous basal border and with or without humeral denticle; interval 3 with one to three discal setigerous pores or without pore; intervals 7 and 5 generally without preapical pores (such pores occasionally present in *H. pseudoserripes*). Metacoxa with or without additional setae medially. Metafemur with four to ten setigerous pores along posterior margin. Protibia with one ventroapical spine and with three to seven preapical spines on outer margin either isolated from spines on ventral surface of tibia or arranged with them in a single row; ventroapical tubercle in male not developed. Male mesotibia without preapical callous thickening on inner margin. Tarsi glabrous ventrally. Abdominal sternites glabrous, more rarely with additional scattered long setae; last visible sternite without pronounced sexual dimorphism, its apex in both sexes more or less rounded and not swollen. Median lobe of aedeagus with somewhat short or moderately long terminal lamella and with a distinct horseshoe-shaped apical capitulum; internal sac usually with one or two large separate spines, also one to three groups of medium-sized spines and several spiny patches.

Composition and distribution. This subgroup comprises two Palaearctic subgenera.

Ecology. The members of this subgroup are confined to open habitats and most numerous in the steppe zone. They occur on dry, usually sandy or gravelly soil with sparse vegetation.

Remarks. The members of this subgroup are similar to those of the *Hyloharpalus* and *Amblystus* subgroups in the structure of the aedeagus but are generally characterized by a more or less convex and wide body with relatively short legs; pronotum in most species impunctate. The legs of some species are adapted for burrowing into the soil.

Subgenus *Actephilus* Stephens, 1833

*Actephilus* Stephens, 1833 [163] (column 11) (as a genus). Type species *Carabus vernalis* Paykull sensu Duftschmid, 1812 (=*Harpalus pumilus* Sturm, 1818), designated by Westwood [164].

*Actophilus* Agassiz, 1846 [165] (pp. 6, 7) (unjustified emendation).

*Euxenus* Gistel, 1856 [166] (p. 359) (as a genus). Type species *Carabus vernalis* Paykull sensu Duftschmid, 1812 (=*Harpalus pumilus* Sturm, 1818), by monotypy.

Diagnosis. Small-sized (length 3.9–6.6 mm). Body somewhat stout, brown to black, without metallic luster on dorsum. Pronotal basal edge setose. Elytra in most species

without parascutellar pore, rarely (in *H. masoreoides*) such pore present; interval 3 either without discal setigerous pores or with one to three pores; humeral denticle present or absent. Metacoxa either without additional setigerous pores medially or with one or two such pores. Terminal lamella of aedeagus longer than wide.

Composition and distribution. This group includes eleven species distributed mainly over the moderately arid areas of Eurasia from the Pyrenees in the west to the northern part of the Korean Peninsula in the east, with most species concentrated in southern Siberia, Mongolia and China: *H. picipennis* (Duftschmid, 1812), *H. pumilus* Sturm, 1818 (Figures 37a and 38a,b), *H. lutshniki* Schauberger, 1932, *H. masoreoides* Bates, 1878, *H. pusillus* (Motschulsky, 1850), *H. acupalpoides* Reitter, 1900, *H. michaili* Kataev, 1990, *H. longipalmatus* Mordkovitsh, 1969, *H. minutulus* Kataev et Liang, 2004, *H. alexandrae* Kataev, 1990 and *H. sushenicus* Kataev, 1990.

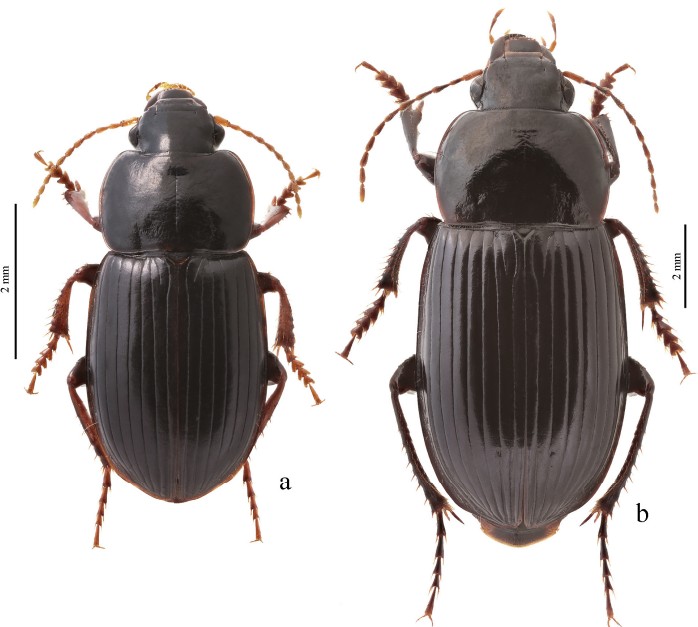

**Figure 37.** Habitus of *Harpalus* species: (**a**) *H.* (*Actephilus*) *pumilus*; (**b**) *H.* (*Isoharpalus*) *pseudoserripes*.

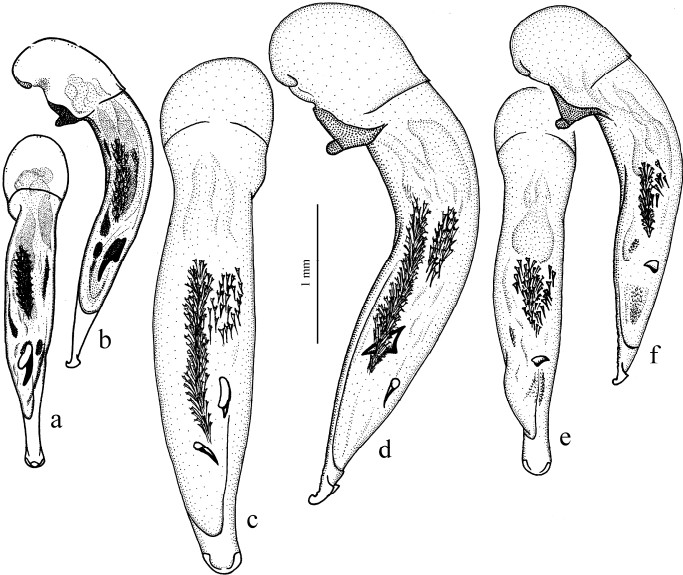

**Figure 38.** Median lobe of aedeagus of *Harpalus* species, dorsal and lateral views: (**a,b**) *H.* (*Actephilus*) *pumilus* (from [89]); (**c,d**) *H.* (*Isoharpalus*) *serripes*; (**e,f**) *H.* (*I.*) *flavicornis*.

Ecology. Species of this group prefer dry open habitats, mostly with sandy soil.

Remarks. In shaping the easily recognizable appearance of members of this subgenus, morphological adaptations played a significant role, facilitating the digging of beetles in light, mostly sandy soil (small size, compact body shape, widely rounded basal angles of the pronotum, shortened antennae and flattened protibiae—Figure 9g–l). The absence of discal and parascutelar pores on the elytra in many species may also be associated with a burrowing mode of life.

The species of this subgenus were revised by Schauberger [68], and Kataev [89] (as the *pumilus* species group), with a subsequent addition [111].

### Subgenus *Isoharpalus* subg. n.

https://zoobank.org/urn:lsid:zoobank.org:act:877F32E3-F623-4275-B781-56AD369632EC
Type species *Carabus serripes* Quensel, 1806.

Diagnosis. Medium-sized to comparatively large (length 6.6–12.2 mm). Body stout, dark brown to black, without metallic luster or with weak bluish metallic tinge on dorsum. Pronotal basal edge setose or glabrous. Elytra with a basal (parascutellar) pore, one discal setigerous pore on interval 3 and generally with a humeral denticle. Terminal lamella of aedeagus somewhat short, at most slightly longer than wide.

Etymology. The subgeneric name is a combination of the Greek *isos*, meaning "equal", and the name of the carabid taxon *Harpalus*.

Composition and distribution. This subgenus includes six Palaearctic species. Four of them are distributed mainly in the Thetyan region: *H. serripes* Quensel, 1806 (Figure 38c,d) (with the subspecies *H. s. ernsti* Kataev, 1995), *H. pseudoserripes* Reitter, 1900 (Figure 37b), *H. politus* Dejean, 1829 (with the subspecies *H. p. vasilinini* Lutshnik, 1916) and *H. flavicornis* Dejean, 1829 (Figure 38e,f) (with the subspecies *H. f. tingens* Reitter, 1900). The taxonomically more separated *H. vanemdeni* Schauberger, 1932 and *H. beneshi* Kataev et Wrase, 1997 are known from China. The true taxonomic position of these two species needs further study.

Ecology. Most members of this subgenus are moderately xerophilous species occurring mostly in dry meadow and steppe habitats.

Remarks. *Isoharpalus* subg. n. corresponds to the *serripes* species group sensu Kryzhanovskij et al. [56] and sensu Kataev [8] together with the *beneshi* species group sensu Kataev [8]. In combination of characters, this subgenus is very similar to *Actephilus*, differing from it mainly in larger body size, longer antennae, and a shorter terminal lamella of the aedeagus; elytra are with a parascutellar pore.

### *Acardystus* Subgroup

Diagnosis. Body large- or medium-sized, more or less convex. Head impunctate and glabrous. Pronotum with one lateral seta on each side and with setose basal edge; surface usually impunctate, more rarely with fine punctures basally. Elytra impunctate and glabrous, with basal border glabrous or setose; humeral denticle generally prominent; interval 3 with one discal setigerous pore or without it; intervals 7 and 5 without preapical pores. Metacoxa with or without additional setae medially. Metafemur with four to twelve setigerous pores along posterior margin. Protibia notably widened and flattened apically, with one to three (rarely four) ventroapical spines and with five to six preapical spines on outer margin, forming a single row with spines on ventral surface of tibia or isolated from them; ventroapical tubercle in male absent or slightly prominent. Male mesotibia without preapical callous thickening on inner margin. Tarsi glabrous dorsally; length of metatarsomere 1 average for genus. Abdominal sternites generally with additional long setae; last visible sternite without pronounced sexual dimorphism, its apex in both sexes more or less rounded and not swollen. Median lobe of aedeagus with elongate terminal lamella and with a horseshoe-shaped apical capitulum; internal sac with one or two more or less large separate spines (absent in few species) and with two to three spiny patches.

Composition and distribution. This subgroup comprises one Nearctic and two Palaearctic subgenera.

Ecology. Most species of this subgroup occur in dry open habitats and are most common in the steppe and forest-steppe zones.

Remarks. Most members of this subgroup are characterized by a development of adaptations for burrowing, of which the most noticeable are characteristic changes in the structure of the protibia.

Subgenus *Euharpalops* Casey, 1924

*Euharpalops* Casey, 1924 [72] (p. 116) (as a genus). Type species *Euharpalops wadei* Casey 1924 (=*H. fraternus* LeConte, 1852), by original designation.
*Euharpalus*: Hatch, 1953 [167] (p. 170) (print error).

Diagnosis. Large-sized (length 10.0–14.5 [15.0] mm). Body stout, wide, dark brown to black, without metallic luster. Elytral basal border very finely setose (setae very short and sometimes barely visible); interval 3 with one discal setigerous pore (occasionally without pore). Protibia with generally three, rarely four ventroapical spines, and with five to six preapical spines on outer margin, isolated from spines on ventral surface of tibia. Internal sac of aedeagus with one large separate spine. Gonocoxite in female moderately wide and notably curved.

Composition and distribution. This group includes five Nearctic species: *H. fraternus* LeConte, 1852 (Figure 39a), *H. lecontei* Casey, 1914, *H. reversus* Casey, 1924, *H. lewisii* LeConte, 1865 and *H. alienus* Bates, 1878.

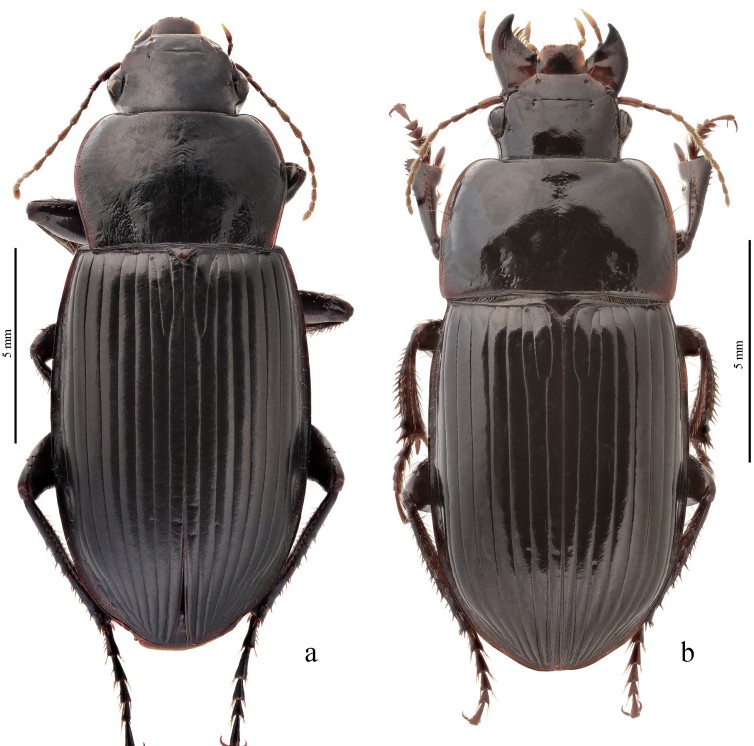

**Figure 39.** Habitus of *Harpalus* species: (**a**) *H.* (*Euharpalops*) *fraternus*; (**b**) *H.* (*Haploharpalus*) *macronotus*.

Ecology. The members of this group are most common in dry open habitats, usually on sandy soil covered with sparse vegetation. *Harpalus lewisii* occurs also in open forests, subalpine and alpine meadows [140].

Remarks. This subgenus is similar to *Haploharpalus* in many characters including stout body, setose basal elytral border and shape of gonocoxite but differs in less-specialized

protibia with generally three ventroapical spines and with preapical spines on outer margin isolated from spines on ventral surface of tibia.

Lindroth [15] included species of this subgenus in the *fraternus* species group together with some Nearctic species of the subgenera *Hyloharpalus* **subg. n.** and *Macroharpalus* **subg. n.**, based exclusively on the structures of the internal sac of the aedeagus. However, similar type of the armament of the internal sac is distributed among the Palaearctic species more widely and occurs with some modifications in many species of the *Hyloharpalus*, *Amblystus*, *Actephilus*, *Acardystus* and *Ooistus* subgroups. Noonan [16] considered *H. fraternus* and *H. funerarius* Csiki, 1932 (=*H. reversus*) as members of the *fraternus* subgroup (=*Euharpalops*) of the *fraternus* group and regarded *H. lewisi* as a member of the related, monobasic *lewisi* group.

The subgenus *Euharpalops*, as it is treated here, corresponds to the *fraternus* species group sensu Kataev [8].

## Subgenus *Haploharpalus* Schauberger, 1926

*Haploharpalus* Schauberger, 1926 [64] (pp. 44, 45) (as a subgenus of *Acardystus* Reitter, 1908). Type species *Harpalus froelichi* Sturm 1818, designated by Habu [29].

Diagnosis. Medium-sized to large (length 7.5–15.0 mm). Body stout, wide, dark brown to black, without metallic luster. Elytral basal border generally setose, rarely (in *H. froelichi*) glabrous; interval 3 with one discal setigerous pore. Protibia with one or two, very rarely three, ventroapical spines, and with at least five preapical spines on outer margin, forming in most species a single row with spines on ventral surface of tibia (in some specimens of *H. melaneus* spines of outer margin isolated from spines on lower surface). Internal sac of aedeagus with one or two separate spines, rarely without spines. Gonocoxite in female somewhat wide and notably curved.

Composition and distribution. This group comprises twelve Palaearctic species, most of which are widely distributed over steppe and forest-steppe zones of Eurasia, with the center of species diversity in south Siberia and Mongolia: *H. froelichii* Sturm, 1818 (Figure 40a,b), *H. raphaili* Kataev, 1997, *H. hirtipes* (Panzer, 1796), *H. zabroides* Dejean, 1829, *H. tichonis* Jacobson, 1907, *H. macronotus* Tschitschérine, 1893 (Figure 39b), *H. brevis* Motschulsky, 1844, *H. brevicornis* Germar, 1824 (Figure 39c,d), *H. corporosus* (Motschulsky, 1861), *H. alpivagus* Tschitschérine, 1899, *H. alajensis* Tschitschérine, 1898, and *H. melaneus* Bates, 1878 (with the subspecies *H. m. sherpicus* Kataev, 2002 and and *H. m. stoetzneri* Schauberger, 1933).

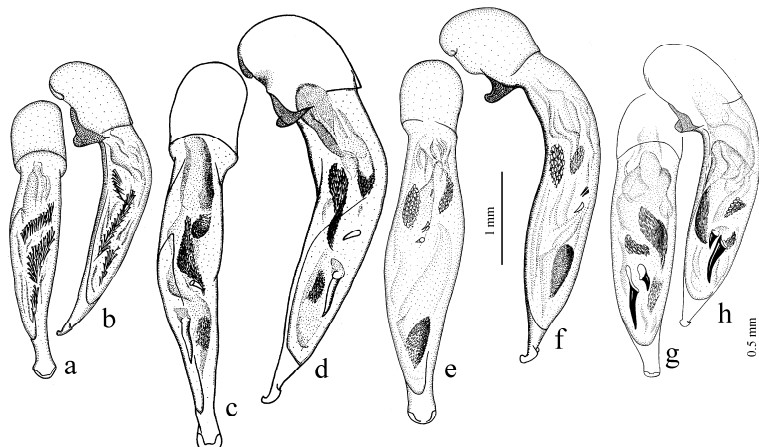

**Figure 40.** Median lobe of aedeagus of *Harpalus* species, dorsal and lateral views: (**a,b**) *H. (Haploharpalus) froelichi* (from [54]); (**c,d**) *H. (H.) brevicornis* (from [54]); (**e,f**) *H. (Acardystus) flavescens*; (**g,h**) *H. (Psammoharpalus) kozlovi* (from [93]).

Ecology. All species of this group occur in rather dry open habitats and are most common in steppe biotopes.

Remarks. The protibiae of the species of this subgenus demonstrate different degrees of development of adaptive changes for burrowing: from less-specialized, weakly widened apically in *H. melaneus* to very specialized, strongly widened apically and with a lobed outer angle in *H. hirtipes* (Figure 9a–f).

The species of this subgenus were revised by Schauberger [64,143,161] and Kataev [54] (as the *hirtipes* species group).

Subgenus *Acardystus* Reitter, 1908

*Acardystus* Reitter, 1908 [39] (pp. 172, 173) (as a subgenus of *Harpalus* Latreille, 1802). Type species *Harpalus rufus* Brüggemann, 1873 (=*Carabus flavescens* Piller et Mitterpacher, 1783), designated by Schauberger [64].

Diagnosis. Comparatively large-sized (length 9.5–12.7 mm). Body slightly elongate, yellow to light brown, without metallic luster on dorsum. Elytral basal border glabrous; interval 3 generally without discal setigerous pore (occasionally with one pore). Protibia with one ventroapical spine, and with several preapical spines on outer margin, forming a single row with spines on ventral surface of tibia (Figure 10b). Internal sac of the aedeagus without large separate spines, only with spiny patches (Figure 40e,f). Gonocoxite in female long, narrow and weakly curved.

Composition and distribution. This subgenus includes only one species, *H. flavescens* (Piller et Mitterpacher, 1783) (Figures 40e,f and 41a), which is widely distributed over Europe, east to the Ural Mountain Range and west Kazakhstan.

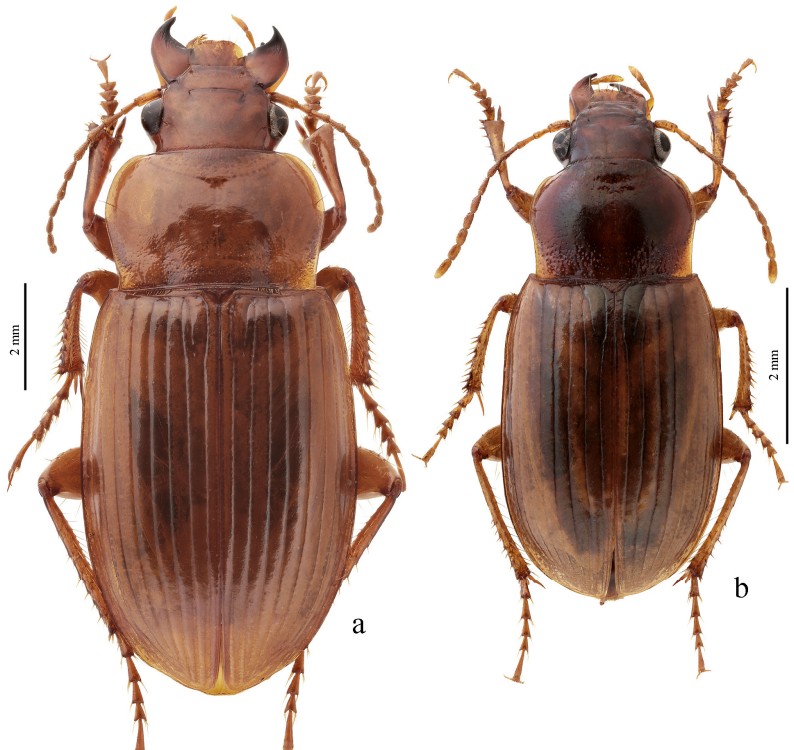

**Figure 41.** Habitus of *Harpalus* species: (**a**) *H. (Acardystus) flavescens*; (**b**) *H. (Psammoharpalus) kozlovi*.

Ecology. The only member of this subgenus is a specialized psammophilous species that occurs on sandy river banks and sea dunes. The species lives in pure sand, in which it burrows deeply.

Remarks. The single species of *Acardystus* differs from the members of two preceding subgenera in a combination of elongate light brown body, glabrous basal elytral border and shape of gonocoxite in female (long, narrow and weakly curved).

This subgenus corresponds to the *flavescens* species group sensu Kryzhanovskij et al. [56] and sensu Kataev [8].

*Psammoharpalus* Subgroup

Diagnosis. Same as for the subgenus.

Composition and distribution. A monobasic subgroup, including only one eastern Palaearctic subgenus.

Subgenus *Psammoharpalus* **subg. n.**

https://zoobank.org/urn:lsid:zoobank.org:act:58D75CA0-89AB-459C-A4B5-43A5E1FA3809
Type species *Harpalus kozlovi* Kataev, 1993.

Diagnosis. Medium-sized (length 5.6–6.9 mm). Body convex, elongate, yellow to light brown, without metallic luster. Head impunctate and glabrous. Pronotum densely punctate basally, with one lateral seta on each side and with setose basal edge. Elytra impunctate and glabrous, with glabrous basal border and one discal setigerous pore on interval 3; intervals 7 and 5 without preapical pores; humeral denticle present, prominent. Metacoxa with one or several additional setae medially. Metafemur with seven to ten setigerous pores along posterior margin. Protibia with one ventroapical spine (longer than that of most other congeneres) and with three to five preapical spines on outer margin, generally isolated from spines on ventral surface of tibia (occasionally spines on outer margin passing on lower surface and forming a single row with spines of lower surface); ventroapical tubercle in male not developed. Male mesotibia without preapical callous thickening on inner margin. Abdominal sternites with additional long setae; last visible abdominal sternite without pronounced sexual dimorphism, its apex in both sexes more or less rounded and not swollen. In male aedeagus with elongate terminal lamella and with disc-shaped apical capitulum; internal sac with two separate spines and three spiny patches (Figure 40g,h). Gonocoxite in female somewhat wide and notably curved.

Etymology. The subgeneric name is a combination of the Greek *psámmos*, meaning "sand", and the name of the carabid taxon *Harpalus*.

Composition and distribution. This subgenus includes only *H. kozlovi* Kataev, 1993 from Qinghai, China (Figures 40g,h and 41b).

Ecology. The single species is found on sandy river banks.

Remarks. Like *H.* (*Acardystus*) *flavescens*, the only species of this subgenus is also adapted for burrowing in river sands, but it is less specialized, with less modified protibiae and shorter and wider gonocoxites. *Harpalus kozlovi* is morphologically separated from other taxa, but its relationship with any of them has not yet been clarified; therefore, it is included in a separate subgenus and, accordingly, in a separate subgroup.

The subgenus *Psammoharpalus* **subg. n.** corresponds to the *kozlovi* species group sensu Kataev [8].

*Ooistus* Subgroup

Diagnosis. Body medium-sized, flattened. Head impunctate and glabrous. Antennae either unicolorous, pale, or bicolorous, dark brown to black, with pale antennomeres 1–2. Pronotum generally impunctate, rarely vaguely punctate latero-basally, with sides rounded to slightly sinuate basally, with one lateral seta on each side and with basal edge glabrous, very rarely (occasionally in some species) with very short, poorly recognizable setae. Elytra impunctate and glabrous, with glabrous basal border; interval 3 generally with one discal pore, rarely without it; intervals 7 and 5 without or (more rarely) with preapical pores; humeral denticle present, more or less prominent. Metepisternum elongate or as wide as long, markedly narrowed basally. Metacoxae with or without additional setae medially.

Metafemur with 3–12 setae along posterior margin. Protibia with one ventroapical spine and with three to four preapical spines on outer margin, not forming a single row with spines on ventral surface of tibia; ventroapical tubercle in male not developed, rarely slightly prominent. Male mesotibia without preapical callous thickening on inner margin. Tarsi glabrous dorsally; length of metatarsomere 1 average for genus. Abdominal sternites usually without additional setae, very rarely with very short, poorly recognizable setae at base of sternites; last visible abdominal sternite without pronounced sexual dimorphism, its apex in both sexes more or less rounded and not swollen. Median lobe of aedeagus with elongate terminal lamella and pronounced horseshoe-shaped apical capitulum; internal sac with one or two separate spines (these spines occasionally absent), with several spiny patches and with or without a group of medium-sized spines.

Composition and distribution. This subgroup comprises one Palaearctic and one Nearctic subgenus.

Ecology. Members of this subgroup occur in open, moderately dry habitats with sparse vegetation, most commonly in steppes and prairies.

Subgenus *Ooistus* Motschulsky, 1864

*Ooistus* Motschulsky, 1864 [61] (p. 209 (as a genus). Type species *Harpalus taciturnus* Dejean, 1829, designated by Noonan [6].

Diagnosis. Length of body 5.8–10.8 mm. Body slightly elongate or wide, dark brown to black, without or with weak metallic luster on dorsum. Pronotum generally impunctate even basally. Internal sac of aedeagus with one or two separate spines (occasionally absent in *H. giacomazzoi*) and with a few spiny patches, including usually two very peculiar, elongate patches in apical half of median lobe; groups of medium-sized spines absent.

Composition and distribution. This subgenus comprises 15 Palaearctic species, with the most species diversity in the steppe and forest-steppe zones: *H. anxius* (Duftschmid, 1812), *H. angustitarsis* Reitter, 1887, *H. anxioides* Kataev, 1991, *H. convexus* Faldermann, 1835, *H. kirgisicus* Motschulsky, 1844, *H. amarellus* Bates, 1891, *H. subcylindricus* Dejean, 1829, *H. servus* (Duftschmid, 1812) (Figures 42a and 43c,d), *H. amplicollis* Ménétriés, 1848 (Figure 43e,f), *H. calathoides* Motschulsky, 1844, *H. taciturnus* Dejean, 1829 (Figure 43a,b), *H. pulchrinulus* Reitter, 1900, *H. amariformis* Motschulsky, 1844, *H. egorovi* Lafer, 1989 and *H. giacomazzoi* Kataev et Wrase, 1996 (with the subspecies *H. g. gracilis* Kataev et Liang, 2007).

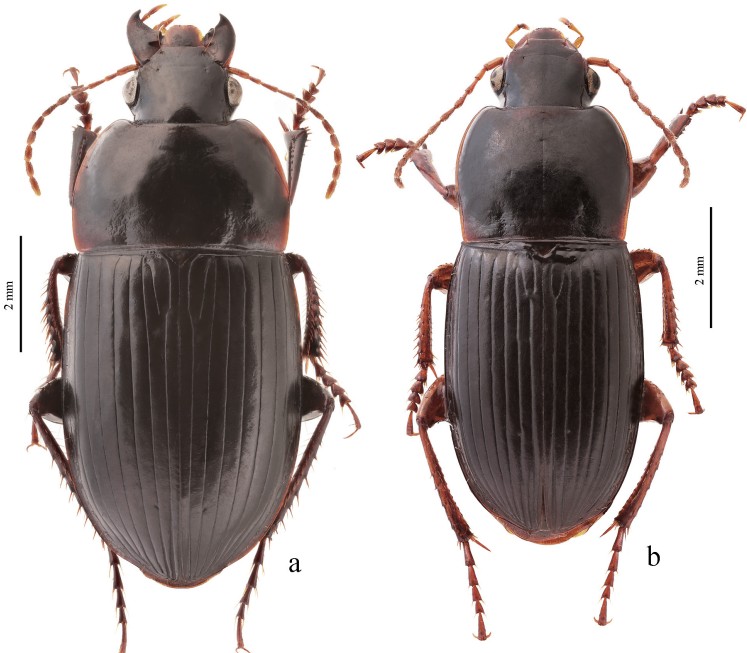

**Figure 42.** Habitus of *Harpalus* species: (**a**) *H.* (*Ooistus*) *servus*; (**b**) *H.* (*Platyharpalus*) *ventralis*.

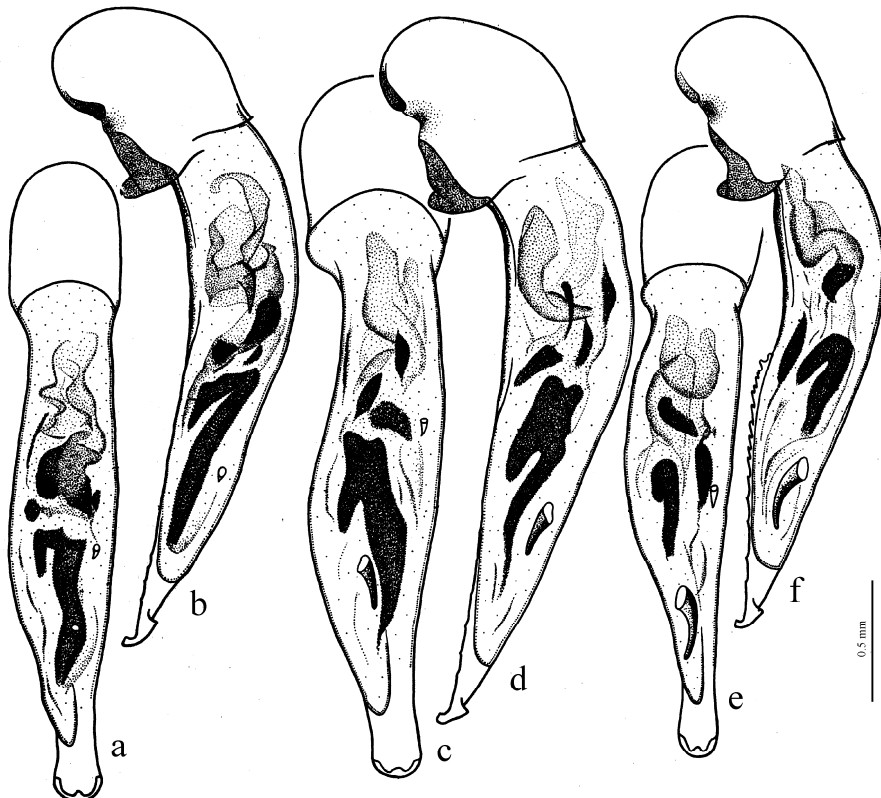

**Figure 43.** Median lobe of aedeagus of *Harpalus* species, dorsal and lateral views (from [54], with modifications): (**a,b**) *H.* (*Ooistus*) *taciturnus*; (**c,d**) *H.* (*O.*) *servus*; (**e,f**) *H.* (*O.*) *amplicollis*.

Ecology. The members of this subgenus occur in open, moderately dry habitats, often on sandy soil with sparse vegetation, and are most common in steppes. *Harpalus egorovi* occurs in dry open biotopes within forests and along river banks.

Remarks. In addition to external morphology, this subgenus is well defined by the structure of the aedeagus with characteristic armament of the internal sac.

The species of this subgenus (as the *anxius* species group) were revised by Kataev [54], with subsequent addition [112]. The taxonomy of some species was heretofore discussed by Mlynář [53].

Subgenus *Platyharpalus* **subg. n.**

https://zoobank.org/urn:lsid:zoobank.org:act:3BECF50A-0ECF-4B1C-ACEE-4138489D46A0
Type species *Harpalus ventralis* LeConte, 1848.

Diagnosis. Length of body [7.2] 7.5–8.2 [10.2] mm. Body moderately wide, dark brown to black, without metallic luster. Pronotum impunctate or vaguely punctate latero-basally. Internal sac of aedeagus with one or two separate spines and with a group of medium-sized spines in middle portion of median lobe in addition to spiny patches in apical portion of median lobe.

Etymology. The subgeneric name is a combination of the Greek *platýs*, meaning "flat", and the name of the carabid taxon *Harpalus*.

Composition and distribution. The subgenus includes two Nearctic species: *H. ventralis* LeConte, 1848 (Figure 42b) and *H. indigens* Casey, 1924.

Ecology. Both species occur in rather dry, open habitats with sandy or silty soil and sparse vegetation: grasslands, prairies, meadows, pastures, cultivated fields, along roadsides, sometimes in open forests [140].

Remarks. This Nearctic subgenus is considered to be related to the Palaearctic subgenus *Ooistus* based on their external similarities, but its true relationship requires further

study as the similarities may be convergent. These subgenera differ in the male genitalia, mainly in the armament of the internal sac of the aedeagus.

*Platyharpalus* **subg. n.** corresponds to the *ventralis* species group sensu Lindroth [15] and sensu Kataev [8] and to the *ventralis* subgroup of the *fraternus* group sensu Noonan [16].

*Asioharpalus* Subgroup

Diagnosis. Same as for the subgenus.

Composition and distribution. A monobasic subgroup, including one eastern Palaearctic subgenus.

Subgenus *Asioharpalus* **subg. n.**

https://zoobank.org/urn:lsid:zoobank.org:act:CB006ACB-5A3E-4503-8A19-F8C3E13A97C1
Type species *Harpalus nigrans* Morawitz, 1862.

Diagnosis. Medium-sized (length 6.2–8.7 mm). Body moderately convex, moderately wide or slightly elongate, brown to black, without metallic luster. Head impunctate and glabrous. Antennae more or less unicolorous, pale or infuscate. Pronotum punctate or impunctate basally, with one lateral seta on each side and with glabrous basal edge. Elytra impunctate and glabrous, with glabrous basal border and one discal setigerous pore on interval 3; intervals 7 and 5 without preapical pores; humeral denticle present, small. Metacoxa with additional setae medially. Metafemur with three to five setae along posterior margin. Protibia with one ventroapical spine and with three to four preapical spines on outer margin, not forming a single row with spines on ventral surface of tibia; ventroapical tubercle in male not developed. Male mesotibia without preapical callous thickening on inner margin. Abdominal sternites glabrous; last visible sternite without pronounced sexual dimorphism, its apex in both sexes more or less rounded and not swollen. Median lobe of aedeagus with elongate terminal lamella and pronounced horseshoe-shaped apical capitulum; internal sac with one separate spine and one or two narrow spiny patches in apical half of median lobe.

Etymology. The subgeneric name is based on a combination of *Asia* and the name of the carabid taxon *Harpalus*.

Composition and distribution. This subgenus comprises three eastern Palaearctic species: *H. nigrans* Morawitz, 1862 (Figures 44a and 45a,b), *H. sinuatus* Tschitschérine, 1893 and *H. parasinuatus* Kataev et Liang, 2007.

Ecology. The species of this group occur in dry, open habitats, mainly within or near forests.

Remarks. *Asioharpalus* **subg. n.** corresponds to the *nigrans* species group sensu Kryzhanovskij et al. [56] and sensu Kataev [8] and to the *sinuatus* species group sensu Kataev and Liang [112]. The taxonomic position of this subgenus is still unclear. Although it is somewhat similar morphologically to the subgenus *Diaharpalus* **subg. n.** of the *Harpalobius* subgroup [112], *Asioharpalus* **subg. n.** is treated here as a member of a separate subgroup. The species of *Asioharpalus* **subg. n.** distinctly differ from those of the *Harpalobius* subgroup in having the antennae reddish brown, only weakly infuscate on antennomeres 2–7, the protibia of male without prominent ventroapical tubercle, and the aedeagus with different pattern of spiny patches in the internal sac. In structure of aedeagus, *Asioharpalus* **subg. n.** is more similar to the subgenera of the *Actephilus*, *Acardystus* and *Ooistus* subgroups.

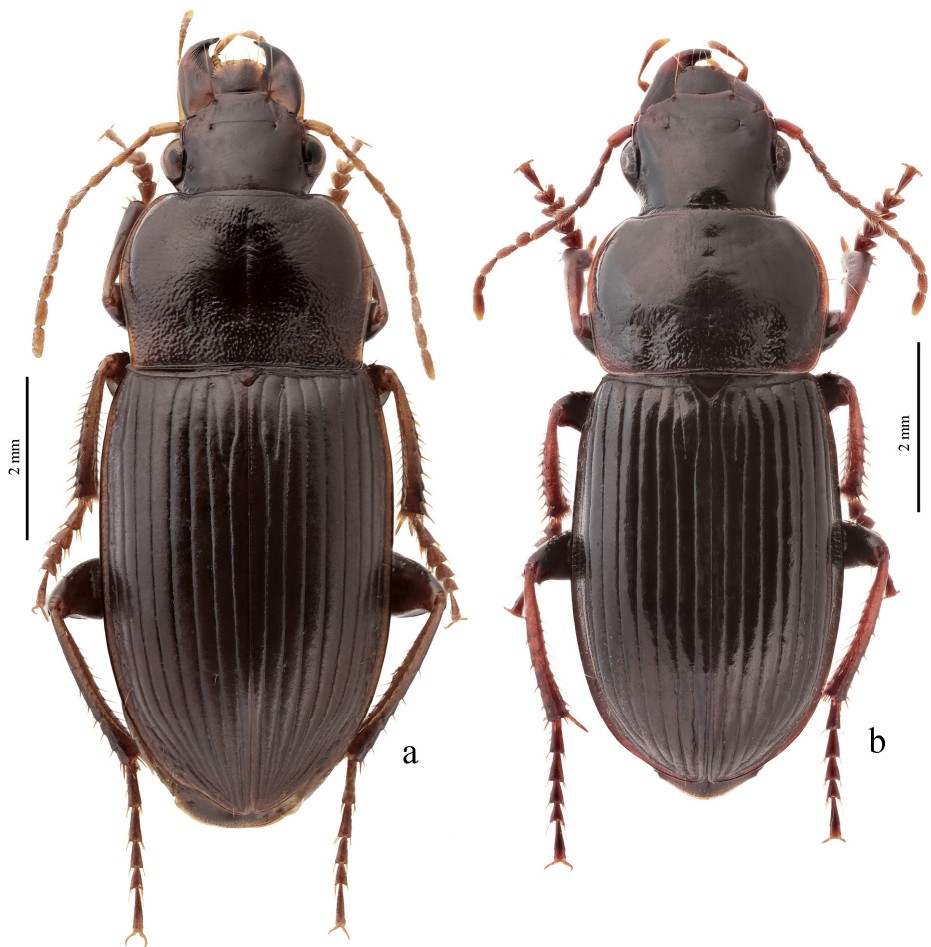

**Figure 44.** Habitus of *Harpalus* species: (**a**) *H.* (*Asioharpalus*) *nigrans*; (**b**) *H.* (*Anamblystus*) *torridoides*.

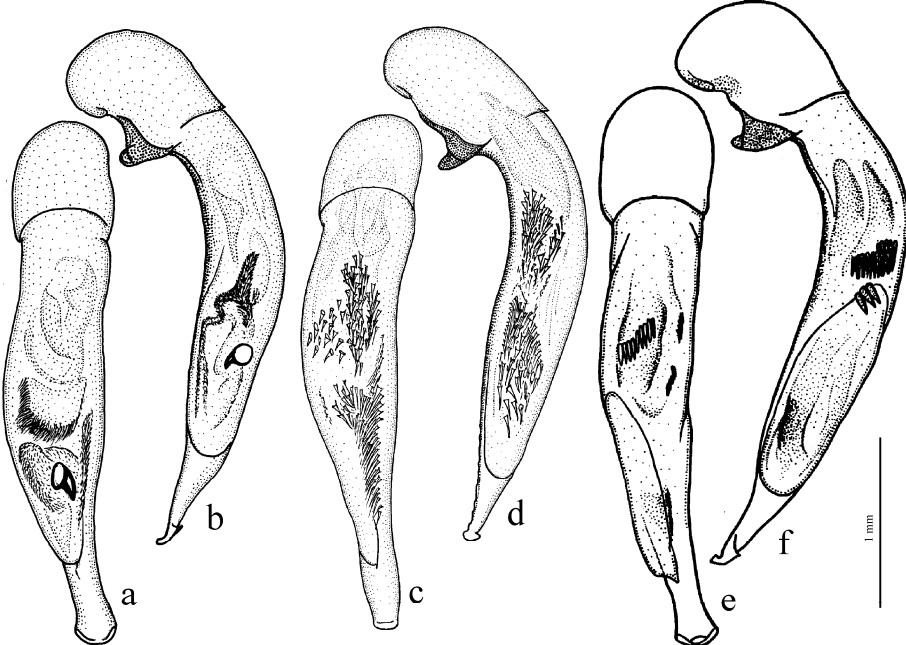

**Figure 45.** Median lobe of aedeagus of *Harpalus* species, dorsal and lateral views: (**a**,**b**) *H.* (*Asioharpalus*) *nigrans*; (**c**,**d**) *H.* (*Anamblystus*) *latus* (from [118]); (**e**,**f**) *H.* (*Homaloharpalus*) *tardus* (from [54]).

*Anamblystus* Subgroup

Diagnosis. Body dark brown to black, without metallic luster. Head impunctate and glabrous. Antennae more or less unicolorous, pale or infuscate. Pronotum punctate or impunctate basally, with one lateral seta on each side; basal edge generally glabrous, rarely with very short, poorly recognizable setae. Elytra in most species impunctate and glabrous, rarely finely punctate and pubescent on lateral intervals, with glabrous basal border (occasionally setose in some species); interval 3 usually with one discal setigerous pore (occasionally without pore); intervals 7 and 5 in most species without preapical pores; basal (parascutellar) pore generally present (absent in *H. albanicus*). Metacoxa generally without additional setae medially, rarely with one or several such setae. Metafemur with three to eight setae along posterior margin. Protibia with one ventroapical spine and with three to six preapical spines on outer margin, not forming a single row with spines on ventral surface of tibia; ventroapical tubercle in male not developed or very small. Male mesotibia without preapical callous thickening on inner margin. Abdominal sternites generally glabrous, in some species with very short, poorly recognizable setae along base of sternites, or with few longer setae; last visible sternite without pronounced sexual dimorphism, its apex in both sexes more or less rounded and not swollen. Median lobe of aedeagus with elongate terminal lamella and pronounced horseshoe-shaped, button-like or almost discoidal apical capitulum; internal sac with one or several spiny patches or groups of spines, without separate spines, rarely without any sclerotic elements.

Composition and distribution. This subgroup includes two subgenera: one Holarctic and one Palaearctic.

Ecology. This subgroup includes both species inhabiting forests and species occurring in dry, open habitats, including steppe ones.

Remarks. The members of this subgroup are characterized by the absence of pronounced sexual dimorphism in the pro- and mesotibia and abdominal sternite, the absence of separate spines in the internal sac of the aedeagus, and the absence of many distinctive features found in members of other subgroups (for example, punctation on head, setae on basal border of elytra and dorsal surface of tarsi, additional setae on pronotum, protibia specialized for burrowing, two and more ventroapical spines on protibia, etc.).

Subgenus *Anamblystus* **subg. n.**

https://zoobank.org/urn:lsid:zoobank.org:act:B6DFCC92-801C-448C-983B-849503B9FC53
Type species *Carabus latus* Linnaeus, 1758.

Diagnosis. Medium-sized (length 5.9–11.5 mm). Body more or less convex, slightly elongate. Pronotal base more or less densely punctate, rarely almost impunctate, in most species with glabrous basal edge (rarely, for example in *H. rufiscapus* and *H. martini*, basal edge setose). Elytra generally impunctate and glabrous (in *H. torridoides* finely punctate and pubescent laterally) and without preapical pores on intervals 7 and 5 (in *H. marginellus* with such pores); basal border generally glabrous (occasionally setose in some Nearctic species). Metacoxa without additional setae medially, rarely (for example in *H. rufiscapus*) with one or several such setae. Metafemur with three, sometimes four (in *H. rufiscapus* generally five, in *H. martini* up to twelve), setigerous pores along posterior margin. Protibia with three or four (in *H. rufiscapus* occasionally five, in *H. martini* up to six) preapical spines on outer margin. Abdominal sternites generally without additional moderately long setae (in *H. martini* with several such setae). Median lobe of aedeagus serrate on ventral side in many species.

Etymology. The subgeneric name is a combination of the Greek *an-*, meaning "not", and the name of the carabid taxon *Amblystus*.

Composition and distribution. This subgenus includes about 22 species from Eurasia and North America. A more precise number of species included in this group cannot now be indicated, since the status of some American taxa requires a revision. Two

species, *H. solitaris* Dejean, 1829 and *H. nigritarsis* C. Sahlberg, 1827 (with the subspecies *H. n. proximus* LeConte, 1848) have a Holarctic distribution.

According to my data, about 13 species are known only from North America: *H. seclusus* Casey, 1914, *H. fanaticus* Casey, 1924, *H. atrichatus* Hatch, 1949, *H. herbivagus* Say, 1823, *H. pleuriticus* Kirby, 1837, *H. somnulentus* Dejean, 1829, *H. celox* Casey, 1914, *H. intactus* Casey, 1914, *H. fallax* LeConte, 1859, *H. carbonatus* LeConte, 1860, *H. uteanus* Casey, 1914, *H. martini* Van Dyke, 1926 and *H. aterrimus* Casey, 1914.

The Palaearctic fauna includes the following species: *H. torridoides* Reitter, 1900 (Figure 43b), *H. latus* (Linnaeus, 1758) (Figure 45c,d), *H. ussuricus* Mlynář, 1979, *H. marginellus* Gyllenhal, 1827, *H. progrediens* Schauberger, 1922, *H. luteicornis* (Duftschmid, 1812), *H. xanthopus* Gemminger et Harold, 1868 (with the subspecies *H. x. winkleri* Schauberger, 1923) and the taxonomically more separated *H. rufiscapus* Gebler, 1833.

Ecology. The species of this subgenus occur mainly in forests or near forests, less often in open landscapes, but, as a rule, in the forest zone; some are in the highlands; *H. rufiscapus* occurs in dry steppe habitats.

Remarks. Although the species of this subgenus show quite a wide variability in some distinctive characters, most of the included taxa form together a morphological continuum separated from species of other subgenera.

The taxonomic position of *H. rufiscapus*, distributed over the Eurasian steppe zone, needs further study since its morphological characteristics are somewhat intermediate between *Anamblystus* **subg. n.** and *Homaloharpalus* **subg. n.** It well differs from the most species of these subgenera also ecologically since it occurs in dry steppe habitats; the internal sac of its aedeagus is without any sclerotic elements.

The Nearctic *H. martini*, which possesses several characters unusual for the subgenus, as noted in the disanosis of the subgenus, is very similar to *H. uteanus* in other characters, including the structure of the aedeagus.

*Anamblystus* **subg. n.** corresponds to the Palaearctic *latus* species group sensu Kryzhanovskij et al. [56], the Nearctic *herbivagus* species group sensu Lindroth [15] and the *latus* species group sensu Kataev [8], including both Palaearctic and Nearctic species. Lindroth [15] referred the Nearctic *herbivagus* species group to the Palaearctic subgenus *Amblystus*, but the type species of the latter subgenus, *H. rubripes*, markedly differs both from the species of the *herbivagus* group and from the Palaearctic species included originally together with *H. rubripes* in the subgenus *Amblystus* by Motschulsky [61] and after him by Reitter [38] and other researchers. In set of characters, *H. rubripes* should be combined with the species related to *H. sulphuripes* (the type species of of the subgenus *Harpaloderus*). Therefore, the new subgenus *Anamblystus* **subg. n.** is erected here for the *latus* species group sensu Kataev [8].

Noonan [16] included most of the Nearctic species of *Anamblystus* **subg. n.** in the *somnulentus* species group, treating *H. somnulentus* very widely, including *H. pleuriticus*, *H. celox*, *H. carbonarius* and *H. uteanus* as its synonyms. He considered *H. atrichatus* as a member of the monobasic *atrichatus* species group, based on the specific armament of the internal sac of the aedeagus, and *H. nigritarsis*, which he also treated very widely, as a member of the *nigritarsis* species group together with *H. megacephalus*.

Subgenus *Homaloharpalus* **subg. n.**

https://zoobank.org/urn:lsid:zoobank.org:act:FE855A57-DBA4-4DCA-B553-0E91ABC6A597
Type species *Carabus tardus* Panzer, 1796.

Diagnosis. Medium-sized (length 5.4–10.5 mm). Body moderately convex, somewhat wide. Pronotal base generally impunctate, more rarely with obliterate punctures; basal edge not setose. Elytra impunctate and glabrous, without preapical pores on intervals 7 and 5; basal border glabrous. Metacoxa generally without additional setae, more rarely (in *H. modestus*) with one or two such setae medially. Metafemur with four (occasionally three) to seven setigerous pores along posterior margin. Protibia with five or six, very rarely three

or four, preapical spines. Abdominal sternites without additional moderately long setae. Median lobe of aedeagus not serrate on ventral side.

Etymology. The subgeneric name is a combination of the Greek *omalós*, meaning "flat", and the name of the carabid taxon *Harpalus*.

Composition and distribution. This subgenus comprises eleven Palaearctic species, with most of them distributed in eastern Asia: *H. tardus* (Panzer, 1796) (Figure 45e,f), *H. tarsalis* Mannerheim, 1825, *H. modestus* Dejean, 1829, *H. bungii* Chaudoir, 1844, *H. chasanensis* Lafer, 1989, *H. tangutorum* Kataev, 1993 (Figure 46a), *H. praecurrens* Schauberger, 1934, *H. stevensi* Kataev, 2011, *H. vernicosus* Kataev et Liang, 2007, *H. albanicus* Reitter, 1900 and the taxonomically more separated *H. dudkoi* Kataev, 2011.

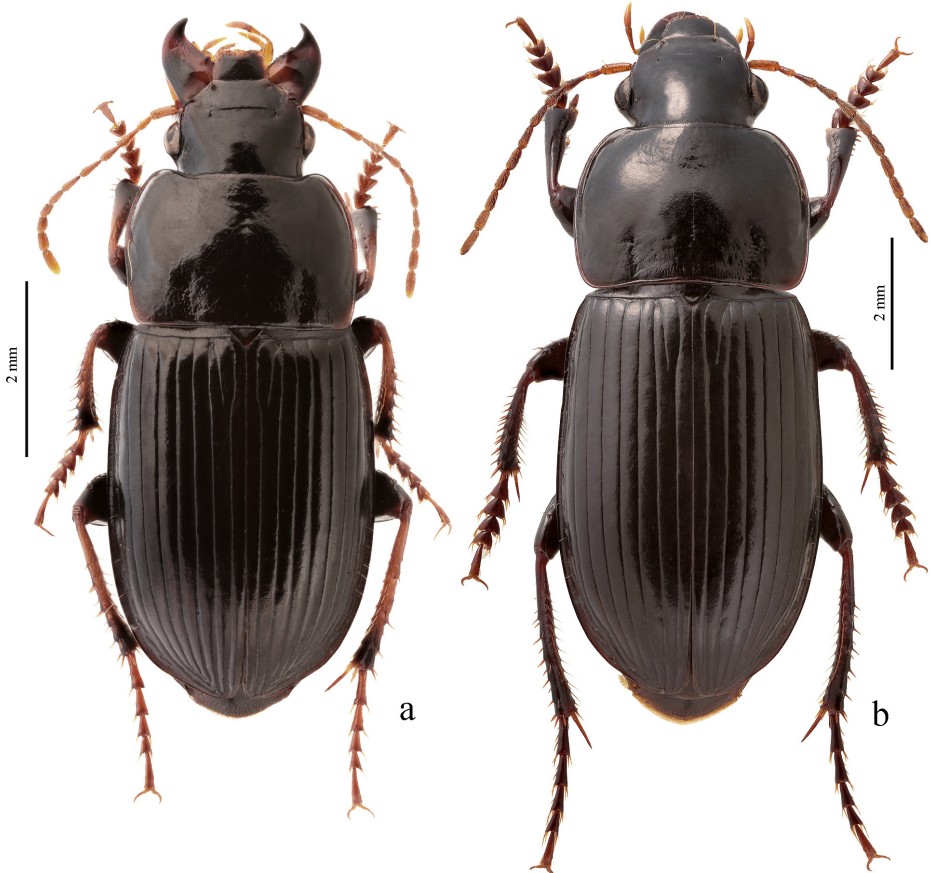

**Figure 46.** Habitus of *Harpalus* species: (**a**) *H.* (*Homaloharpalus*) *tangutorum*; (**b**) *H.* (*Bactroharpalus*) *obnixus*.

Ecology. Species of this subgenus occur in various dry, open habitats, but usually near or within forests; they are most characteristic of the forest-steppe zone.

Remarks. This subgenus is very similar in morphology to the preceding one, differing mainly in the impunctate pronotal base in most species and usually in a more number of preapical spines on the outer margin of the protibia and in a more number of setae along the posterior margin of the metafemur, although the variability in each of these characters taken separately overlaps in the species of these two subgenera.

The subgenus *Homaloharpalus* **subg. n.** corresponds to the *tardus* species group sensu Kryzhanovskij et al. [56], sensu Kataev [8] and sensu Kataev and Liang [112].

*Harpalobius* Subgroup

Diagnosis. Body brown to black, with or without green, blue or copper metallic luster on dorsum; elytra of some members bicolorous, with yellow pattern. Head impunctate and glabrous. Antennae generally bicolorous, with one or two basal antennomeres pale and other antennomeres distinctly infuscate, usually black. Pronotum with sides generally

rounded, rarely slightly sinuate basally, with one lateral seta on each side and with basal edge generally glabrous, more rarely setose; surface generally impunctate, rarely with fine sparse or dense punctures basally. Elytra impunctate and glabrous, with glabrous basal border and with or without one discal setigerous pore on interval 3; intervals 7 and 5 without preapical pores; humeral denticle generally present, more or less prominent or very small, almost indistinct. Metacoxa with or without additional setae medially. Metafemur with four (rarely three) to fifteen setigerous pores along posterior margin. Protibia with one or two (very rarely three) ventroapical spines in transverse row and with three to five preapical spines on outer margin, not forming a single row with spines on ventral surface of tibia; ventroapical tubercle in male more or less prominent. Male mesotibia without preapical callous thickening on inner margin or with small such thickening. Abdominal sternites glabrous or with additional long setae; last visible sternite without pronounced sexual dimorphism, its apex in both sexes more or less rounded and not swollen. Median lobe of aedeagus with elongate terminal lamella and prominent horseshoe-shaped apical capitulum; internal sac with one to three somewhat small spiny patches; separate spines absent.

Composition and distribution. This subgroup includes three Holarctic subgenera, each of which is distributed both in the Palaearctic and in the Nearctic.

Ecology. Most of the species of this subgroup live in open habitats with sandy or gravelly soil, both in the steppes and in the forest zone; many species occur in mountains.

Remarks. This subgroup is similar to the *Anamblystus* subgroup in many distinctive characters listed in their diagnoses but differs from it in having more or less prominent ventroapical tubercle in male protibia.

Subgenus *Bactroharpalus* **subg. n.**

https://zoobank.org/urn:lsid:zoobank.org:act:72DE7CAC-4C63-429B-8761-9695C5050703
Type species *Harpalus cautus* Dejean, 1829.

Diagnosis. Medium-sized (length 6.8–10.6 [11.0] mm). Body elongate or somewhat wide, without metallic luster and with unicolorous elytra. Pronotal basal edge glabrous, not setose. Humeral denticle prominent or almost indistinct. Elytral interval 3 with one discal pore. Metafemur with four (rarely three or five) setigerous pores along posterior margin. Protibia with generally two (very rarely one or three) ventroapical spines, and with four or five preapical spines on outer margin. Male mesotibia generally with a very small preapical callous thickening on inner margin. Abdominal sternites glabrous. Median lobe of aedeagus with rather short terminal lamella; internal sac in most species without sclerotic elements, in some species with one or two indistinct spiny patches (in *H. ochropus* with a distinct small spiny patch apically).

Etymology. The subgeneric name is a combination of the Greek *baktron*, meaning "stick", and the name of the carabid taxon *Harpalus*.

Composition and distribution. This subgenus includes eight species. With the exception of *H. lederi* Tschitschérine, 1899 (Figure 47a,b), which is distributed in the northeast Palaearctic, all other species occur in North America: *H. cautus* Dejean, 1829, *H. obnixus* Casey, 1924 (Figure 46b), *H. opacipennis* (Haldeman, 1843), *H. plenalis* Casey, 1914, *H. ellipsis* LeConte, 1848, *H. balli* Noonan, 1991 and *H. ochropus* Kirby, 1837.

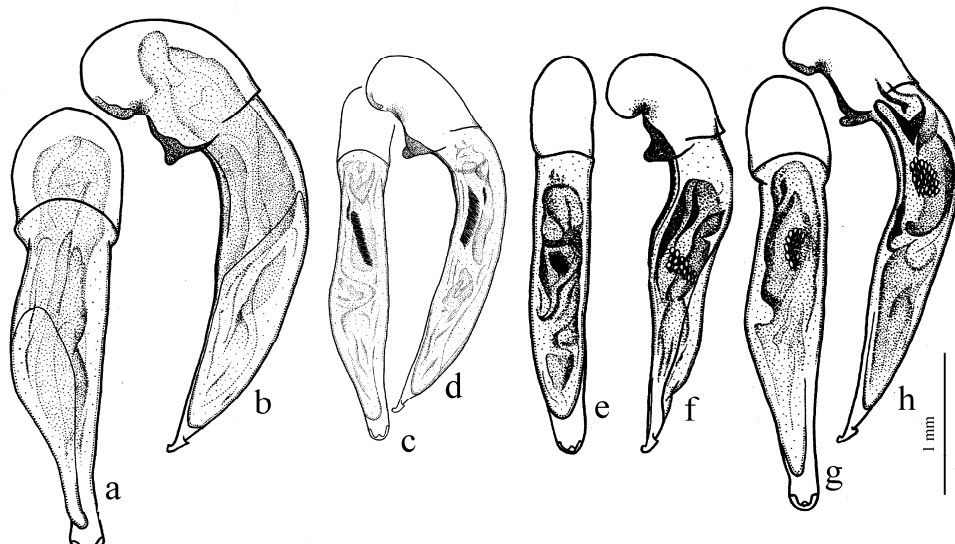

**Figure 47.** Median lobe of aedeagus of *Harpalus* species, dorsal and lateral views: (**a**,**b**) *H.* (*Bactroharpalus*) *lederi* (from [54]); (**c**,**d**) *H.* (*Diaharpalus*) *vittatus* (from [88]); (**e**,**f**) *H.* (*Harpalobius*) *fuscipalpis* (from [54]); (**g**,**h**) *H.* (*H.*) *inexspectatus* (from [54]).

Ecology. Species of this subgenus occur in open habitats with sandy or gravelly soil.

Remarks. Within the *Harpalobius* subgroup, the members of *Bactroharpalus* **subg. n.** are recognizable by the combination of the characters listed in the diagnosis.

This subgenus corresponds to the Nearctic *opacipennis* species group sensu Lindroth [15], the Palaearctic *lederi* species group sensu Kryzhanovskij et al. [56] and the *cautus* species group sensu Kataev [8], including both Nearctic and Palaearctic species. Lindroth [15] referred the Nearctic species of this subgenus (the *opacipennis* species group) to the Palaearctic subgenus *Pheuginus*, but the type species of the latter subgenus, *H. optabilis*, differs markedly both from these Nearctic species and from the Palaearctic species originally included together with *H. optabilis* in *Pheuginus* by Motschulsky [34] and after him by Reitter [38] and other authors. Therefore, the new subgenus *Bactroharpalus* **subg. n.** is erected here for species of the *cautus* species group sensu Kataev [8].

Noonan [16] treated most species of *Bactroharpalus* **subg. n.** (except for *H. obnixus*, *H. plenalis* and *H. opacipennis*) together with the Nearctic species of the *Diaharpalus* **subg. n.** as members of the *cautus* species group. He included *H. obnixus* and *H. plenalis* in the *obnixus* species group based on the absence of sclerotic armament in the internal sac of their aedeagi. *Harpalus opacipennis* was considered by him as a member of the separate monobasic *opacipennis* species group as showing no close cladistic relationships to other Nearctic species.

Subgenus *Diaharpalus* **subg. n.**

https://zoobank.org/urn:lsid:zoobank.org:act:01A17602-6B91-42BE-8D02-A96F17962B7F
Type species *Harpalus vittatus* Gebler, 1833.

Diagnosis. Medium-sized (length 5.2–9.0 [9.7] mm). Body moderately convex, somewhat wide or slightly elongate, with or without metallic luster on dorsum; elytra of some members bicolorous, with yellow pattern. Pronotal basal edge generally glabrous, rarely setose. Humeral denticle generally more or less prominent. Elytral interval 3 with or without discal setigerous pore. Metafemur with four to six setigerous pores along posterior margin. Protibia with one ventroapical spine and with three to four (rarely five) preapical spines on outer margin. Male mesotibia without preapical callous thickening on inner margin. Abdominal sternites generally glabrous, rarely (in *H. metarsius* and occasionally in *H. udege*) with sparse additional somewhat long setae. Median lobe of aedeagus with short

or moderately long terminal lamella and with one to three distinct small spiny patches (or groups of small spines) in internal sac medially.

Etymology. The subgeneric name is a combination of the Greek *dia-*, meaning "through, between", and the name of the carabid taxon *Harpalus*.

Composition and distribution. The subgenus comprises 12 species from the eastern Palaearctic and the Nearctic regions. Among them, *H. vittatus* Gebler, 1833 (Figures 47c,d and 48a) (with the subspecies *H. v. kiselevi* Kataev et Shilenkov, 1990 and *H. v. alaskensis* Lindroth, 1968) are distributed both in Siberia and in North America (Alaska); most other (eight species) are known from Palaearctic Asia: *H. udege* Lafer, 1989, *H. karakorum* Jedlička, 1958, *H. metarsius* Andrewes, 1930, *H. mlynari* Kataev, 1990, *H. kaznakovi* Kataev et Wrase, 1997 (with the subspecies *H. lilliputa* Kataev et Liang, 2007), *H. alexeevi* Kataev, 1990, *H. manas* Kataev, 1990 and *H. plancyi* Tschitschérine, 1897.

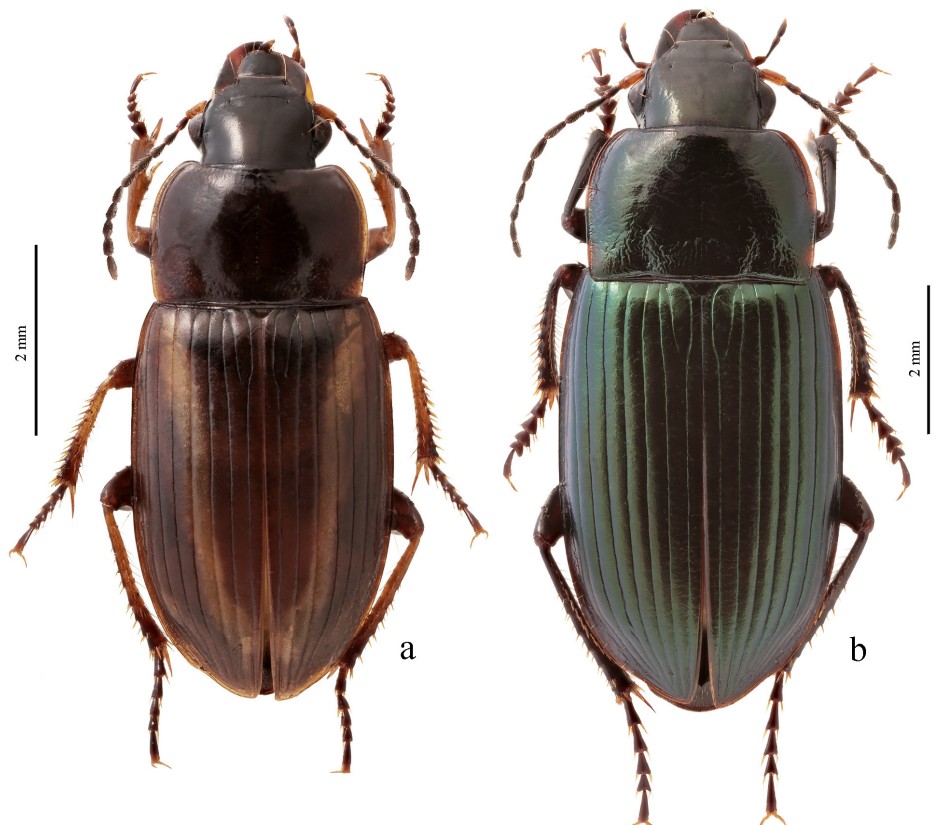

**Figure 48.** Habitus of *Harpalus* species: (**a**) *H.* (*Diaharpalus*) *vittatus*; (**b**) *H.* (*Harpalobius*) *viridanus*.

Three species are known from North America: *H. innocuus* LeConte, 1863, *H. paululus* Casey, 1914 and *H. intactus* Casey, 1924. The status of some American forms needs further study.

Ecology. Species of this subgenus occur in open, dry habitats with sandy or gravelly soil, both in the steppes and at the edge of the forests; many are distributed in mountainous areas.

Remarks. This subgenus is most similar in morphology, including the male genitalia, to the subgenus *Harpalobius* but distinctly distinguished from it by a lesser number of setae along the posterior margin of the protibia and the abdominal sternites without additional dense long setae.

In glabrous abdominal sternites of most species, *Diaharpalus* **subg. n.** is similar to *Bactroharpalus* **subg. n.**, differing from it mainly in having only one ventroapical spine on the protibia, the male mesotibia without preapical callous thickening on the inner

margin and the aedeagus with several distinct spiny patches (groups of small spines) in the internal sac.

The subgenus *Diaharpalus* **subg. n.** corresponds to the *innocuus* species group sensu Lindroth [15] and to the *vittatus* species group sensu Kataev [8,88] and sensu Kryzhanovskij et al. [56].

The Palaearctic species of this subgenus were revised by Kataev [88], with subsequent additions [112,116].

Subgenus *Harpalobius* Reitter, 1900

*Harpalobius* Reitter, 1900 [38] (pp. 76, 103) (as a subgenus of *Harpalus* Latreille, 1802). Type species *Harpalus fuscipalpis* Sturm 1818, designated by Habu [29].

*Harpalellus* Lindroth, 1968 [15] (p. 815) (as a genus). Type species *Harpalus basilaris* Kirby 1837 (=*H. fuscipalpis* Sturm 1818), by original designation.

Diagnosis. Medium-sized to comparatively large (length 6.6–12.0 mm). Body moderately convex, somewhat wide or slightly elongate, with or without metallic luster on dorsum; elytra of some members bicolorous, with yellow pattern. Pronotal basal edge glabrous or setose. Humeral denticle more or less prominent. Elytral interval 3 with or without discal setigerous pore. Metafemur with seven to fifteen setigerous pores along posterior margin. Protibia with one or two ventroapical spines and with five (rarely four) preapical spines on outer margin. Male mesotibia without preapical callous thickening on inner margin. Abdominal sternites with numerous additional long setae. Median lobe of aedeagus with short terminal lamella and with one small spiny patch (group of small spines) in internal sac medially.

Composition and distribution. The subgenus includes four species. One of them, the Holarctic *H. fuscipalpis* Sturm, 1818 (Figure 47e,f), is widely distributed over Eurasia and North America; the other three species occur in the Palaearctic region: *H. fuscicornis* Ménétriés, 1832, *H. inexspectatus* Kataev, 1989 (Figure 47g,h) and *H. viridanus* Motschulsky, 1844 (Figure 48b) (with the subspecies *H. v. angustibasis* Kataev et Liang, 2007 and *H. v. staudingerianus* Schauberger, 1932).

Ecology. All species of this subgenus occur in dry, open, mainly steppe habitats.

Remarks. This subgenus differs from two preceding subgenera in having numerous additional long setae on the abdominal sternites and the metafemur with a more number of setae along the posterior margin. The representatives of *Harpalobius* and *Diaharpalus* **subg. n.** demonstrate a certain parallelism in the variability of some characters, in particular, the appearance of a very similar light pattern on the elytra, the presence or absence of setae on the basal edge of the pronotum and a discal pore on the elytral interval 3 [88].

The subgenus *Harpalobius*, as here treated, corresponds to the *fuscipalpis* species group sensu Kataev [54] and sensu Kryzhanovskij et al. [56]. *Harpalobius* and *Harpalellus* are considered synonyms because the type species of these taxa are conspecific [53,54]. *Harpalus fuscipalpis* has aedeagus with apical orifice in dorsal position which is unusual for *Harpalus*. Perhaps the origin of this species is the result of partial juvenilization of development [118].

The species of this subgenus (as the *fuscipalpis* species group) were revised by Kataev [54], with subsequent addition [112].

*Pheuginus* Subgroup

Diagnosis. Body brown to black, with or without green or blue metallic luster on dorsum. Head impunctate and glabrous, in *Anophonus* **subg. n.** more or less punctate and setose dorsally. Pronotum punctate or impunctate basally, with one lateral seta on each side and with basal edge generally setose, rarely glabrous. Elytra generally impunctate and glabrous, in some taxa more or less widely punctate and pubescent; basal border glabrous or setose; basal (parascutellar) pore present or absent in some taxa; interval 3 in most species with or without one discal setigerous pore, in some species with several discal pores; intervals 7 and 5 with or without preapical pores. Metacoxae with or without

additional setae medially. Protibia generally with one, more rarely two, ventroapical spines and generally with three or four (more rarely five or six) preapical spines on outer margin, usually not forming a single row with spines on lover surface of tibia; ventroapical tubercle in male generally not prominent. Male mesotibia without preapical callous thickening on inner margin. Tarsi glabrous or setose dorsally; metatarsomere 1 often more elongate than average for genus. Abdominal sternites generally with more or less numerous additional setae, in some species glabrous; last visible sternites in most species without pronounced sexual dimorphism, in few species its apex truncate in male and slightly swollen in female. Median lobe of aedeagus with long or short terminal lamella and with pronounced more or less discoidal, more rarely horseshoe-shaped, apical capitulum; internal sac generally with one or two separate spines (more rarely without them), and usually also with one or two spiny patches; in some members without sclerotic elements.

Composition and distribution. This subgroup comprises twelve Palaearctic subgenera, with the largest subgenus and species diversity in Asia.

Ecology. Members of this subgroup occur in various open habitats, mainly in the steppes and deserts, as well as in the mountains. In deserts, the species are usually found near water.

Remarks. The subgenera included in this subgroup form a morphological continuum without distinct apomorphies common to all these subgenera. Many members are characterized by a median lobe of the aedeagus with a pronounced discoidal apical capitulum and by rather slender metatarsus with a comparatively long first tarsomere. The taxonomic position of some subgenera requires further study.

This subgroup corresponds to the *smaragdinus* stock sensu Kataev [104].

Subgenus *Nephoharpalus* Huang, Lei, Yan et Hu, 1996

*Nephoharpalus* Huang, Lei, Yan et Hu, 1996 [168] (pp. 120, 123) (as a subgenus of *Harpalus* Latreille, 1802). Type species *Harpalus jianyangensis* Huang, Lei, Yan et Hu, 1996 (=*H. pallidipennis* Morawitz, 1862), by original designation.

Diagnosis. Medium-sized (length 7.3–11.0 mm). Body moderately convex, slightly elongate, brown to black, with or without greenish or bluish metallic luster on elytra. Pronotum punctate basally, with sides sinuate or not sinuate basally; basal edge setose. Elytra impunctate and glabrous, with glabrous basal border; interval 3 generally with one discal setigerous pore (in *H. pallidipennis* with one or several discal pores); basal (parascutellar) pore present; intervals 7 and 5 without preapical pores. Prosternum with short setae. Metepisternum elongate, longer than wide. Protibia with one ventroapical spine. Tarsi glabrous dorsally or sparsely setose. Abdominal sternites with very short dense pubescence. Median lobe of aedeagus with discoidal apical capitulum and relatively short terminal lamella; internal sac with two parallel elongate groups of small spines and with or without a small sepatate spine medially.

Composition and distribution. This subgenus includes three Palaearctic species: *H. smaragdinus* (Duftschmid, 1812) (Figure 50a,b), *H. pallidipennis* Morawitz, 1862 (Figures 11g,h and 49a) and *H. cyclogonus* Chaudoir, 1844 (with the subspecies *H. c. olenini* Poppius, 1906).

Ecology. Species of this subgenus occur on the plains, in dry, open habitats, both in meadows and steppes.

Remarks. The most distinctive characters of this subgenus are an elongate metepisternum, abdominal sternites with short dense pubescence and a characteristic aedeagus with a short terminal lamella and a large transverse apical capitulum.

The subgenus *Nephoharpalus*, as here treated, corresponds to the *smaragdinus* species group sensu Kryzhanovskij et al. [56] and sensu Kataev [8]. Mlynář [76] first drew attention to the close relationship between *H. smaragdinus* and *H. pallidipennis*.

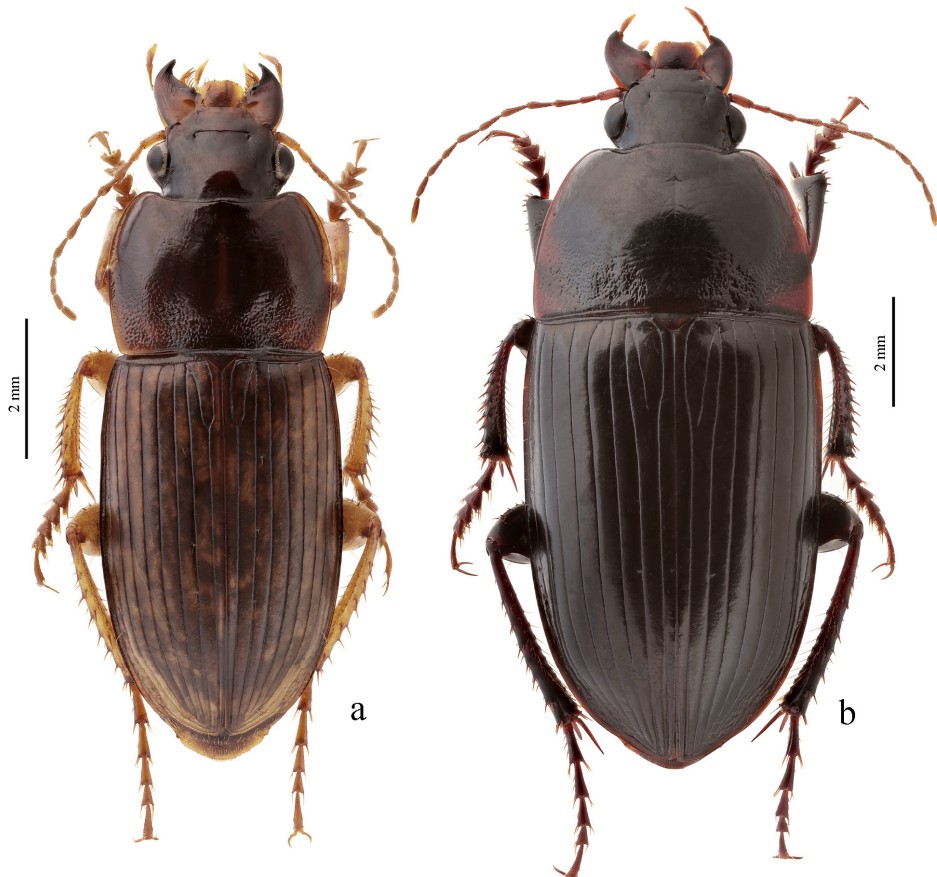

**Figure 49.** Habitus of *Harpalus* species: (**a**) *H.* (*Nephoharpalus*) *pallidipennis*; (**b**) *H.* (*Pheuginus*) *optabilis*.

Subgenus *Pheuginus* Motschulsky, 1844

*Pheuginus* Motschulsky, 1844 [34] (tabl. between pp. 196 and 197, p. 197) (as a subgenus of *Harpalus* Latreille, 1802). Type species *Harpalus optabilis* Dejean, 1829, designated by Noonan [6].

*Conicus* Motschulsky, 1844 [34] (tabl. between pp. 196 and 197, p. 197) (as a taxon within *Harpalus* Latreille, 1802). Type species *Harpalus acuminatus* Motsch. 1844 (=*H. optabilis* Dejean, 1829), designated by Noonan [6].

Diagnosis. Medium-sized to large (length 8.4–13.5 mm). Body more or less convex, moderately wide or elongate, elliptical-shaped, without metallic luster. Pronotum punctate basally, with sides rounded; basal edge generally glabrous, more rarely setose (setae very short and barely noticeable). Elytra impunctate and glabrous, with glabrous basal border and with one discal setigerous pore on interval 3; basal (parascutellar) pore present; interval 7, or intervals 7 and 5, or intervals 7, 5 and 3 with row of preapical pores; these pores in *H. davidianus* almost reaching basal elytral border (Figure 6d). Prosternum with moderately long setae. Metepisternum elongate, longer than wide. Protibia with one ventroapical spine. Abdominal sternites with dense short pubescence. Tarsi glabrous or (in *H. davidianus*) very finely setose dorsally (in the latter species also tibiae finely setose). Median lobe of aedeagus with discoidal apical capitulum and long terminal lamella; internal sac with two or three more or less large groups of medium-sized spines.

Composition and distribution. This subgenus includes three species from the eastern Palaearctic: *H. oodioides* Dejean, 1829, *H. optabilis* Dejean, 1829 (Figures 49b and 50c,d) and *H. davidianus* Tschitschérine, 1903 (with the subspecies *H. d. basharicus* Schauberger, 1933).

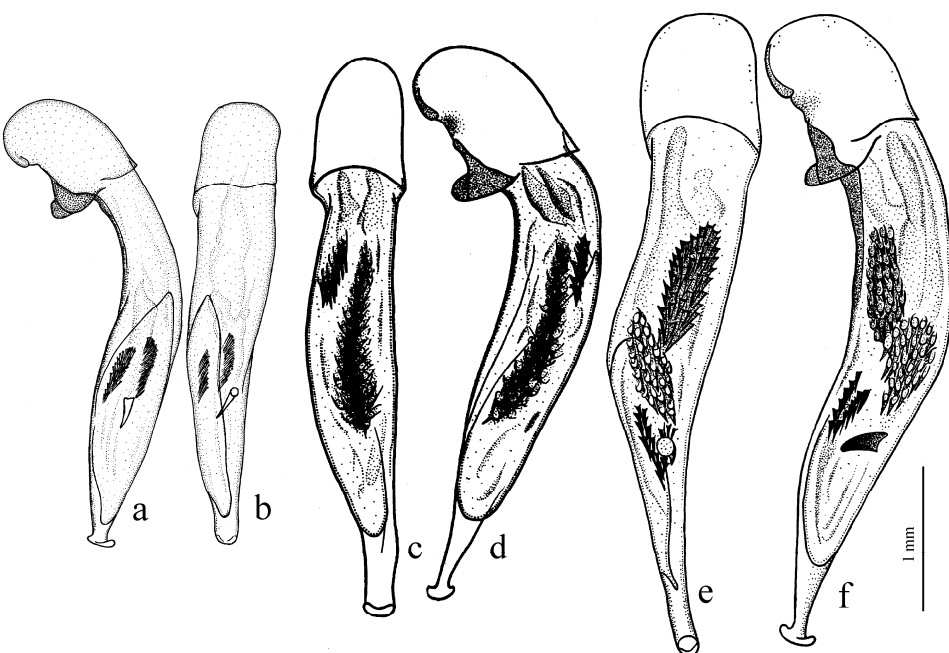

**Figure 50.** Median lobe of aedeagus of *Harpalus* species, dorsal and lateral views: (**a**,**b**) *H.* (*Nephoharpalus*) *smaragdinus*; (**c**,**d**) *H.* (*Pheuginus*) *optabilis* (from [54]); (**e**,**f**) *H.* (*Mesoharpalus*) *gisellae* (from [90], with modifications).

Ecology. All three species occur mainly on the plains, in steppe habitats.

Remarks. The members of *Pheuginus* are similar to those of *Nephoharpalus* in dense short pubescence on abdominal sternites but differ in having preapical pores at least in interval 7 and a more convex body.

The species of this subgenus (as the *optabilis* species group) were revised by Kataev [54]. The taxonomy of some taxa was discussed by Mlynář [53].

Subgenus *Mesoharpalus* **subg. n.**

https://zoobank.org/urn:lsid:zoobank.org:act:333ACC7E-11C3-4742-ADEC-B4DEDF71F825
Type species *Harpalus gisellae* Csiki, 1932.

Diagnosis. Medium-sized to large (length 7.6–12.3 mm). Body moderately convex, elongate, generally with green or violet metallic luster on dorsum. Pronotum impunctate or with more or less distinct punctures basally; sides sinuate basally; basal edge generally setose, more rarely glabrous. Elytra impunctate and glabrous, with glabrous basal border; interval 3 with one discal setigerous pore or without it; intervals 7 and 5 without preapical pores; basal (parascutellar) pore present. Prosternum with short setae. Metepisterna short and wide, at least as wide as long. Protibia with one ventroapical spine. Tarsi glabrous dorsally. Abdominal sternites almost glabrous, with a few short barely noticeable setae. Median lobe of aedeagus with discoidal apical capitulum and comparatively long terminal lamella; internal sac generally with one separate spine and with one to three groups of medium-sized spines (in *H. zhdankoi* without groups of spines, but with an additional separate spine).

Etymology. The subgeneric name is a combination of the Greek *mésos*, meaning "middle, medium", and the name of the carabid taxon *Harpalus*.

Composition and distribution. This subgenus includes five allopatric Palaearctic species with more or less reduced wings: *H. mitridati* Pliginsky, 1915 from the east European and Kazakh steppes, and *H. gisellae* Csiki, 1932 (Figure 50e,f), *H. kiritshenkoi* Kataev, 1990 (Figure 51a), *H. ovtshinnikovi* Kataev, 1990 and *H. zhdankoi* Kataev, 1990 from the Tien Shan.

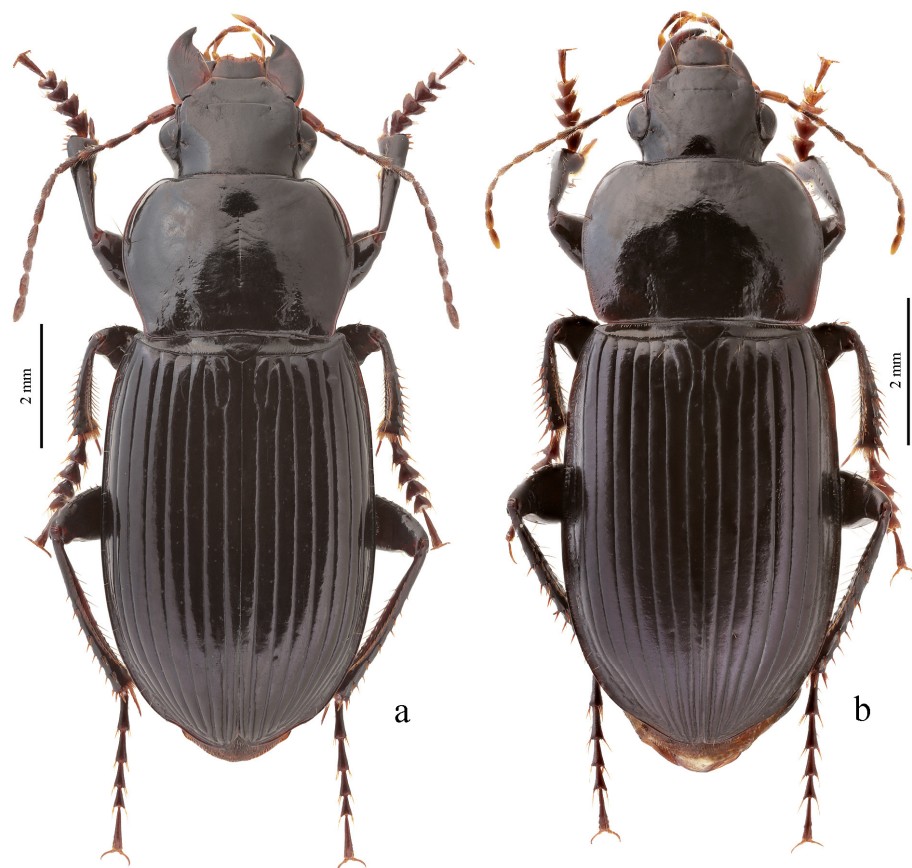

**Figure 51.** Habitus of *Harpalus* species: (**a**) *H.* (*Mesoharpalus*) *kiritshenkoi*; (**b**) *H.* (*Eremoharpalus*) *medvedevi*.

Ecology. Species of this subgenus occur in steppe habitats, *H. mitridati* in lowlands, and other species in mountains.

Remarks. *Mesoharpalus* **subg. n.** differs from the two preceding subgenera in almost glabrous abdominal sternites and a short and wide metepisternum.

This subgenus corresponds to the *gisellae* species group sensu Kataev [8,90] and sensu Kryzhanovskij et al. [56]. A revision of the species of this group was published by Kataev [90].

Subgenus *Eremoharpalus* **subg. n.**

https://zoobank.org/urn:lsid:zoobank.org:act:6C95CFC6-FD16-4D99-95F4-D3765E8920DB
Type species *Harpalus remboides* Solsky, 1874.

Diagnosis. Comparatively large-sized (length 9.3–12.3 mm). Body moderately convex, somewhat wide, without metallic luster; legs long and slender. Pronotum impunctate basally, with rounded sides and setose basal edge. Elytra impunctate and glabrous, with glabrous basal border and with one discal setigerous pore on interval 3; intervals 7 and 5 without preapical pores; basal (parascutellar) pore present. Metepisternum as long as or slightly longer than wide. Protibia with one ventroapical spine. Tarsi glabrous dorsally; metatarsomeres long, weakly widened posteriorly. Abdominal sternites almost glabrous, with a few short barely noticeable setae. Median lobe of aedeagus with discoidal apical capitulum and relatively long terminal lamella; internal sac with two or three spiny patches and without separate spines.

Etymology. The subgeneric name is a combination of the Greek *erēmos*, meaning "desert", and the name of the carabid taxon *Harpalus*.

Composition and distribution. This subgenus includes three allopatric species from Eurasian deserts: *H. remboides* Solsky, 1874 (Figure 52a,b), *H. medvedevi* Kataev, 2006 (Figure 50b) and *H. araraticus* Mlynář, 1979.

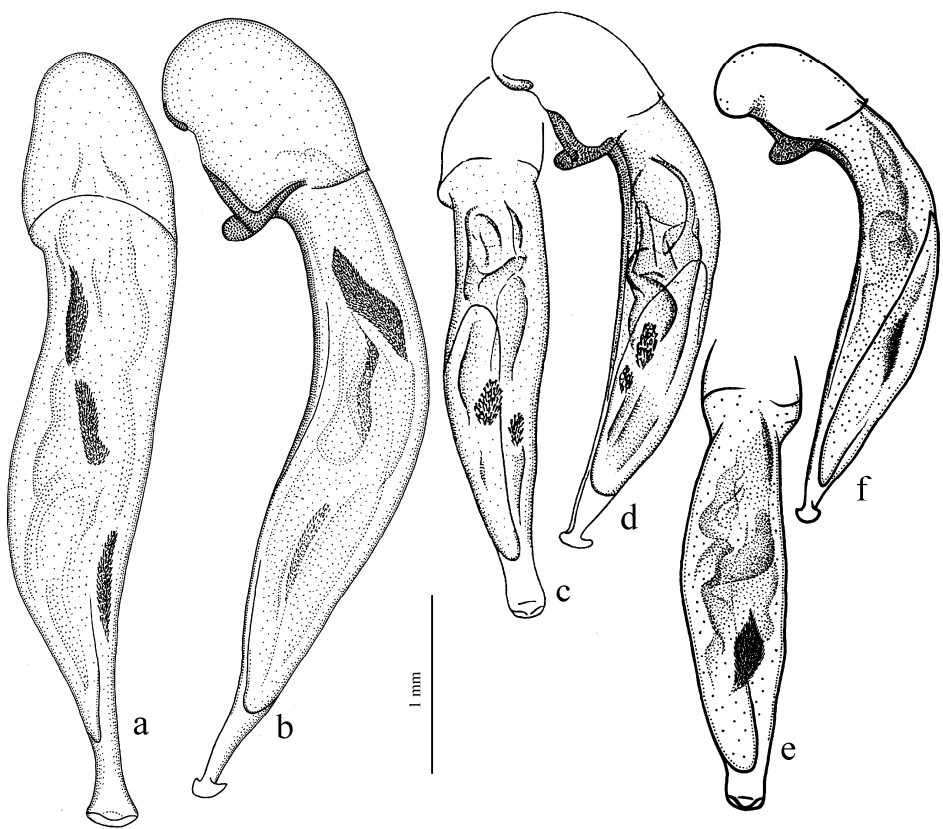

**Figure 52.** Median lobe of aedeagus of *Harpalus* species, dorsal and lateral views: (**a**,**b**) *H. (Eremoharpalus) remboides* (from [104]); (**c**,**d**) *H. (Oreoharpalus) famelicus* (from [114]); (**e**,**f**) *H. (Hypsoharpalus) arnoldii* (from [87]).

Ecology. Species of this subgenus inhabit dry steppes and deserts but occur not far from water, usually in riparian (tugai) forests or in the river coastal zone.

Remarks. This subgenus is most similar to *Nephoharpalus* and particularly to the subgenus *Mesoharpalus* **subg. n.** in most of the structural characters, including the glabrous basal border of elytra and the discoidal apical capitulun of aedeagus. The subgenera *Eremoharpalus* **subg. n.** and *Mesoharpalus* **subg. n.** also share the almost glabrous abdominal sternites and the short metepisterna. The members of *Eremoharpalus* **subg. n.** can be distinguished from those of *Mesoharpalus* **subg. n.** by the rounded sides of the pronotum and the absence of separate spines in the internal sac of the aedeagus.

*Eremoharpalus* **subg. n.** corresponds to the *remboides* species group sensu Kryzhanovskij et al. [56] and Kataev [8,104]. A revision of the species of this group was published by Kataev [104].

Subgenus *Oreoharpalus* **subg. n.**

https://zoobank.org/urn:lsid:zoobank.org:act:3AFCFBF7-8A70-4AD4-8120-B2BC9A709223
Type species *Harpalus famelicus* Tschitschérine, 1898.

Diagnosis. Medium-sized (length 7.7–11.3 mm). Body moderately convex, elongate, without metallic luster. Head impunctate and glabrous. Pronotum impunctate basally, with setose basal edge; sides sinuate or not sinuate basally. Elytra either more or less widely punctate and pubescent or impunctate and glabrous; basal border setose; interval 3 with one discal setigerous pore or without it; intervals 7 and 5 with or without preapical pores;

abbreviate (parascutellar) striole often more or less reduced; basal (parascutellar) pore present or occasionally absent. Metepisternum as long as or slightly longer than wide. Protibia with one ventroapical spine; ventroapical tubercle in male slightly prominent. Tarsi glabrous dorsally. Abdominal sternites almost glabrous, with a few short barely noticeable setae. Median lobe of aedeagus with moderately long terminal lamella and discoidal or horseshoe-shaped apical capitulum; internal sac either without sclerotic armament or with one separate spine, or with two groups of small spines and also spiny patches.

Etymology. The subgeneric name is a combination of the Greek *óros*, meaning "mount", and the name of the carabid taxon *Harpalus*.

Composition and distribution. This subgenus includes three wingless Palaearctic species endemic to the Hissar-Darvaz Mountains: *H. famelicus* Tschitschérine, 1898 (Figures 52c,d and 53a) (with the subspecies *H. f. fanensis* Kataev et Wrase, 1993 and *H. f. loxophonoides* Kataev et Wrase, 1993), *H. diligens* Tschitschérine, 1898 and *H. strenuus* Tschitschérine, 1898.

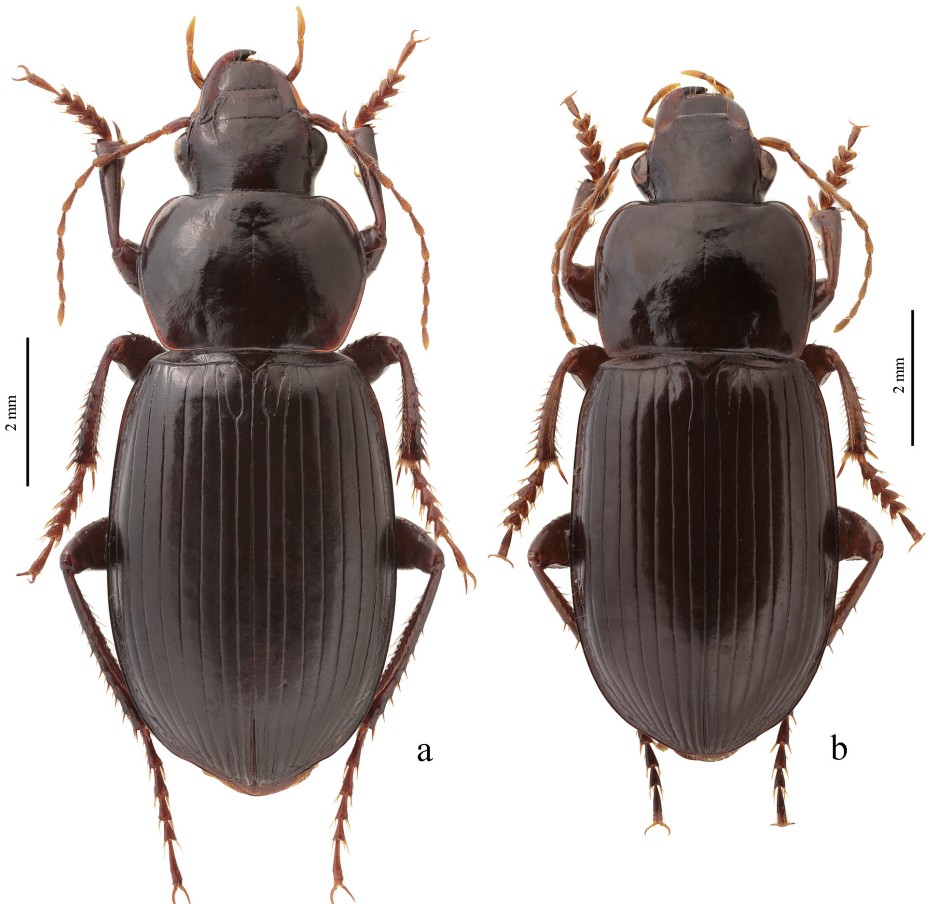

**Figure 53.** Habitus of *Harpalus* species: (**a**) *H.* (*Oreoharpalus*) *famelicus*; (**b**) *H.* (*Hypsoharpalus*) *arnoldii*.

Ecology. Species of this subgenus occur in mountains, in open habitats at altitudes of 1000–4000 m.

Remarks. This subgenus is recognizable by the features listed in the diagnosis, among which the most important are elytra with setose basal border, pronotum with one ventroapical spine, short metepisterna and almost glabrous abdominal sternites.

The subgenus *Oreoharpalus* **subg. n.** corresponds to the *famelicus* species group sensu Kataev and Wrase [114], sensu Kryzhanovskij et al. [56], and sensu Kataev [8]. A revision of the species of this group was published by Kataev and Wrase [114].

Subgenus *Hypsoharpalus* **subg. n.**

https://zoobank.org/urn:lsid:zoobank.org:act:C154293C-9C78-4536-9D57-4AA33C088EE0
Type species *Harpalus arnoldii* Kataev, 1988.

Diagnosis. Medium-sized (length 8.3–9.3 mm). Body moderately convex, somewhat wide or slightly elongate, without metallic luster. Head impunctate and glabrous. Pronotum impunctate basally, with setose basal edge; sides not sinuate basally. Elytra impunctate and glabrous, with glabrous basal border; interval 3 with or without one discal setigerous pore; intervals 7 and 5 without preapical pores; basal (parascutellar) pore present. Metepisternum short and wide, about as long as wide. Protibia with one ventroapical spine and with three or four preapical spines on outer margin, isolated from spines on ventral surface of tibia; ventroapical tubercle in male slightly prominent. Tarsi glabrous dorsally. Abdominal sternites almost glabrous, with a few short barely noticeable setae. Median lobe of aedeagus with a relatively short terminal lamella and with a horseshoe-shaped apical capitulum; internal sac without sclerotic elements or only with small spiny patches.

Etymology. The subgeneric name is a combination of the Greek *húpsos*, meaning "hight, altitude", and the name of the carabid taxon *Harpalus*.

Composition and distribution. This subgenus includes two wingless Palaearctic species endemic to the Tien Shan: *H. kadyrbekovi* Kataev, 1988 and *H. arnoldii* Kataev, 1988 (Figures 52e,f and 53b).

Ecology. Both species of this subgenus inhabit mountains, occurring in the forest and alpine belts at altitudes of 1800–3200 m.

Remarks. The subgenus *Hypsoharpalus* **subg. n.** corresponds to the *kadyrbekovi* species group sensu Kataev [8]. The relationships of this subgenus are still unclear. In some characters (one ventroapical spine on protibia, short metepisternum and almost glabrous abdominal sternites), *Hypsoharpalus* **subg. n.** is similar to *Oreoharpalus* **subg. n.** but distinguished by the elytra with a glabrous basal border. The members of *Hypsoharpalus* **subg. n**. markedly differ from those of *Oreoharpalus* **subg. n.,** which also have a glabrous basal border of the elytra, in the structure of the aedeagus with a short terminal lamella, a horseshoe-shaped apical capitulum and no spines in the internal sac.

Subgenus *Anophonus* **subg. n.**

https://zoobank.org/urn:lsid:zoobank.org:act:C8C61312-23E5-4683-A907-0C008DE7B8CB
Type species *Ophonus cyanopterus* Tschitschérine, 1897.

Diagnosis. Medium-sized (length 7.3–11.0 mm). Body moderately convex, elongate, with or without metallic luster. Head punctate and more or less setose dorsally. Pronotum punctate basally, with setose basal edge; sides sinuate or not sinuate basally. Elytra more or less widely punctate and pubescent, generally with setose basal border; interval 3 with one discal setigerous pore (can be difficult to distinguish against the background of punctation); basal (parascutellar) pore present. Metepisternum short and wide, wider than long. Protibia with one ventroapical spine. Tarsi densely setose dorsally. Abdominal sternites with additional somewhat long setae. Median lobe of aedeagus with discoidal apical capitulum; internal sac with one or two separate spines and one or several groups of small spines.

Etymology. The subgeneric name is a combination of the Greek *an-*, meaning "not", and the name of the carabid taxon *Ophonus*.

Composition and distribution. This subgenus includes two wingless species endemic to the west Tien Shan: *H. cyanopterus* (Tschitschérine, 1897) (Figures 54a and 56a,b) and *H. pterostichus* (Reitter, 1900). Some new taxa of this subgenus are still undescribed.

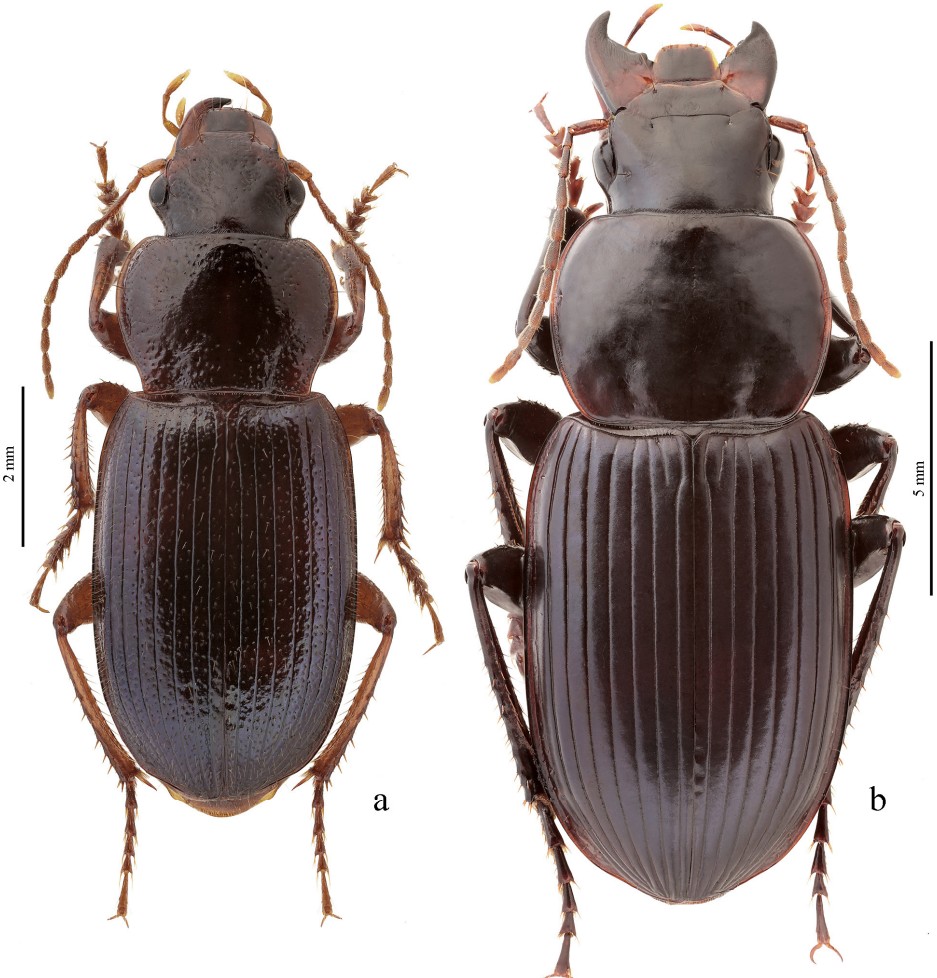

**Figure 54.** Habitus of *Harpalus* species: (**a**) *H.* (*Anophonus*) *cyanopterus*; (**b**) *H.* (*Megaharpalus*) *stoetznerianus*.

Ecology. Both species occur in rather dry open habitats, mainly within the mountain belt of juniper forests.

Remarks. *Harpalus cyanopterus* и *H. pterostichus* have a punctate and pubescent body and densely setose tarsi, and based on these characters, they have long been considered representatives of the genus *Ophonus* [23,38,169]. However, paraglossae in these species are setose at margins, epilobes of mentum are narrow, basal labial palpomere is without an oblique carina, metacoxa is without a posteromedial pore, and, therefore, both species are included in the genus *Harpalus* [170]. Based on the features of their aedeagi and external morphology, they are treated as belonging to a separate subgenus within *Pheuginus* subgroup. A closer relationship of this subgenus, which is distinguished by a punctate and pubescent body, including the head and tarsi, with other taxa of this subgroup requires further study.

This subgenus corresponds to the *cyanopterus* species group sensu Kryzhanovskij et al. [56] and sensu Kataev [8].

Subgenus *Haloharpalus* **subg. n.**

https://zoobank.org/urn:lsid:zoobank.org:act:F0561469-EA8B-4014-8E38-462DB0F6901C
Type species *Harpalus salinulus* Reiter, 1900.

Diagnosis. Medium-sized (length 6.8–7.0 mm). Body moderately convex, slightly elongate, black, without metallic luster. Head impunctate and glabrous. Pronotum impunctate basally, with setose basal edge; sides not sinuate basally. Elytra impunctate and

glabrous, with glabrous basal border and without discal setigerous pore on interval 3; basal (parascutellar) pore present; intervals 7 and 5 without preapical pores. Prosternum with moderately long setae. Metepisternum elongate, markedly longer than wide. Protibia with one ventroapical spine and four preapical spines on outer margin, isolated from spines on ventral surface of tibia. Tarsi glabrous dorsally. Abdominal sternites with additional long setae. Median lobe of aedeagus with a relatively short terminal lamella and with a button-like apical capitulum; internal sac with one or two separate spines and three or four groups of small spines (Figure 55b,c).

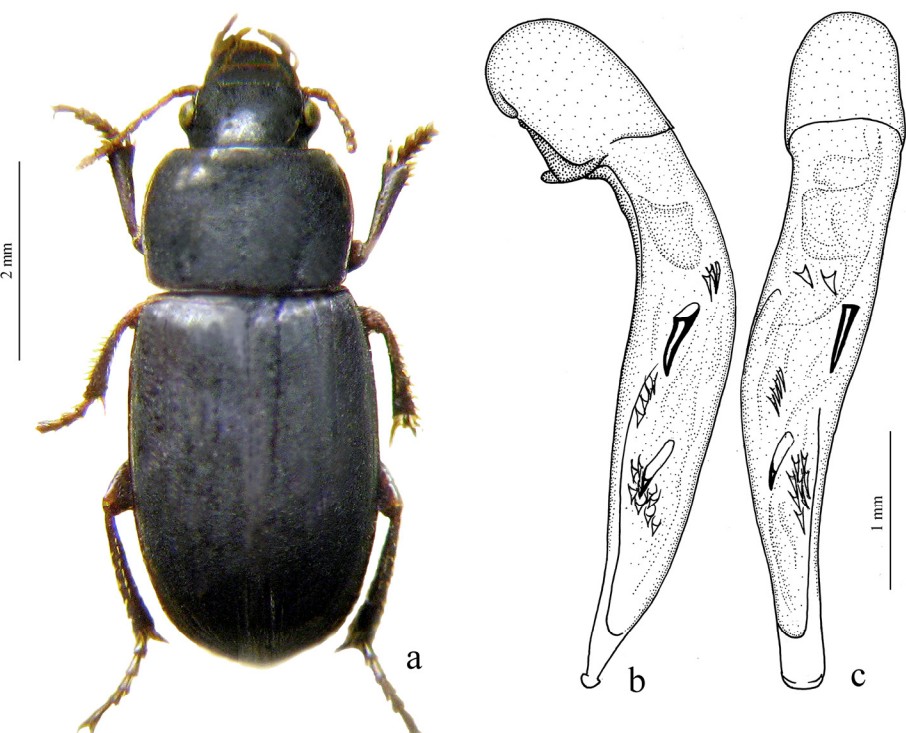

**Figure 55.** *Harpalus* (*Haloharpalus*) *salinulus* (syntype): (**a**) habitus; (**b**,**c**) median lobe of aedeagus in lateral and dorsal views.

Etymology. The subgeneric name is a combination of the Greek *hals*, meaning "salt", and the name of the carabid taxon *Harpalus*.

Composition and distribution. This subgenus includes only one very rare Palaearctic species, *H. salinulus* Reitter 1900 (Figure 55), endemic to central Anatolia.

Ecology. The only species of this subgenus is found on saline lands.

Remarks. This subgenus is well recognizable by unique combination of distinctive characters listed in the diagnosis, but its relationship with other taxa is still unclear.

The subgenus *Haloharpalus* **subg. n.** corresponds to the *salinulus* species group sensu Kataev [8].

## Subgenus *Megaharpalus* **subg. n.**

https://zoobank.org/urn:lsid:zoobank.org:act:F5FF4F1D-AD51-4172-B2F1-3153FBBD17D2
Type species *Harpalus stoetznerianus* Schauberger, 1932.

Diagnosis. Large-sized (length 14.6–15.7 mm). Body moderately convex, elongate, black, with bluish or greenish tinge on dorsum. Head impunctate and glabrous. Pronotum impunctate, with sides not sinuate basally, with one lateral seta on each side and with setose basal edge. Elytra impunctate and glabrous, with glabrous basal border and without discal setigerous pore on interval 3; basal (parascutellar) pore present; intervals 7 and 5 without preapical pores. Prosternum almost glabrous or with very short setae. Metepisternum

short, wider than long. Metacoxa without additional setae medially. Protibia with two (occasionally three) ventroapical spines and with four preapical spines on outer margin, not forming a single row with spines on ventral surface of tibia; ventroapical tubercle in male absent. Tarsi glabrous dorsally. Three last abdominal sternites (V–VII) glabrous; sternite IV densely setose; last visible sternite (VII) without pronounced sexual dimorphism, its apex in both sexes more or less rounded and only slightly swollen in female. Median lobe of aedeagus with an elongate terminal lamella and with a transverse, almost discoidal apical capitulum; internal sac with a wide spiny patch apically and occasionally also with one very small separate spine medially (Figure 56c,d).

Etymology. The subgeneric name is a combination of the Greek *mégas*, meaning "big, large", and the name of the carabid taxon *Harpalus*.

Composition and distribution. This subgenus includes only the wingless eastern Palaearctic *H. stoetznerianus* Schauberger, 1932 (Figures 54b and 56c,d), known from the Chinese province of Sichuan.

Remarks. Like the preceding taxon, the only species of *Megaharpalus* **subg. n.** is also well recognizable by unique combination of distinctive characters listed in the diagnosis, including a large size and a setation of the abdominal sternites, but its relationship with other taxa is still unclear.

This subgenus corresponds to the *stoetznerianus* species group sensu Kataev [8].

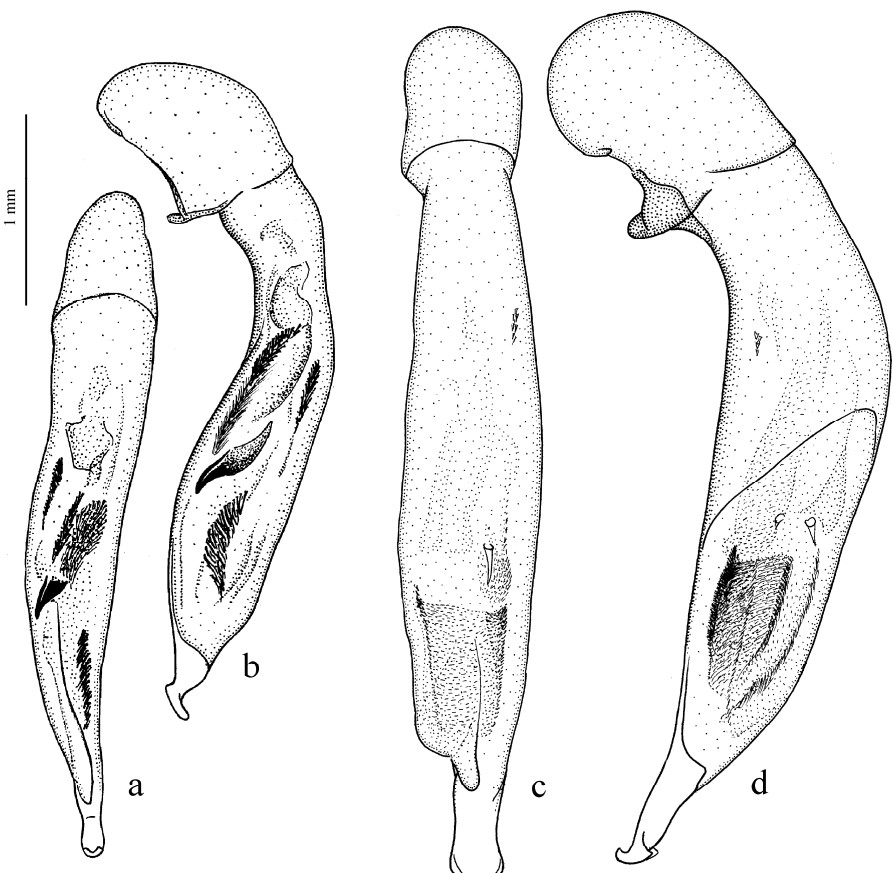

**Figure 56.** Median lobe of aedeagus of *Harpalus* species, dorsal and lateral views: (**a**,**b**) *H.* (*Anophonus*) *cyanopterus*; (**c**,**d**) *H.* (*Megaharpalus*) *stoetznerianus*.

Subgenus *Aristoharpalus* **subg. n.**

https://zoobank.org/urn:lsid:zoobank.org:act:EFEA3413-E5A2-439B-A1D9-A29428FB7BAF
Type species *Harpalus ingenuus* Tschitschérine, 1898.

Diagnosis. Comparatively large-sized (length 9.1–13.0 mm). Body convex, stout and somewhat wide, with or without metallic luster on dorsum. Head impunctate and glabrous. Pronotum impunctate basally, with rounded sides and setose basal edge. Elytra punctate and pubescent basally and often laterally, with setose basal edge, without discal setigerous pore on interval 3 and without basal (parascutellar) pore; intervals 7 and 5 without preapical pores; preapical sinuation rather deep, with a denticle at its base. Metepisternum short and wide, wider than long (Figure 7d,e). Protibia with two ventroapical spines; ventroapical tubercle in male not prominent. Tarsi glabrous dorsally. Abdominal sternites with additional long setae. Median lobe of aedeagus with a long terminal lamella and with a discoidal or horseshoe-shaped apical capitulum; internal sac with two more or less large separate spines, one or two groups of small or medium-sized spines and also spiny patches.

Etymology. The subgeneric name is a combination of the Greek *áristos*, meaning "best, noble", and the name of the carabid taxon *Harpalus*.

Composition and distribution. This subgenus includes two wingless Palaearctic species endemic to the Pamir-Alai region: *H. ingenuus* Tschitschérine, 1898 (Figures 57a and 58a,b) (with the subspecies *H. i. lailakensis* Kataev, 2006 and *H. i. alajanicus* Jedlička, 1957) and *H. arcuatus* Tschitschérine, 1898.

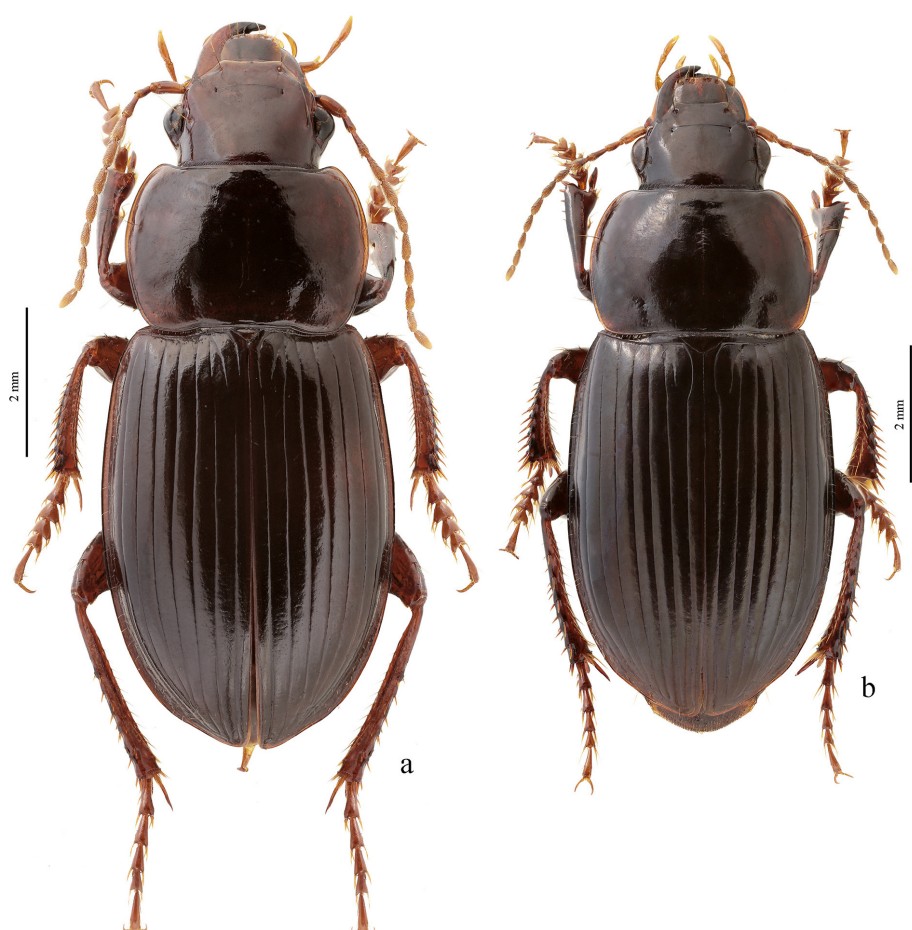

**Figure 57.** Habitus of *Harpalus* species: (**a**) *H.* (*Aristoharpalus*) *ingenuus*; (**b**) *H.* (*Cycloharpalus*) *pulvinatus*.

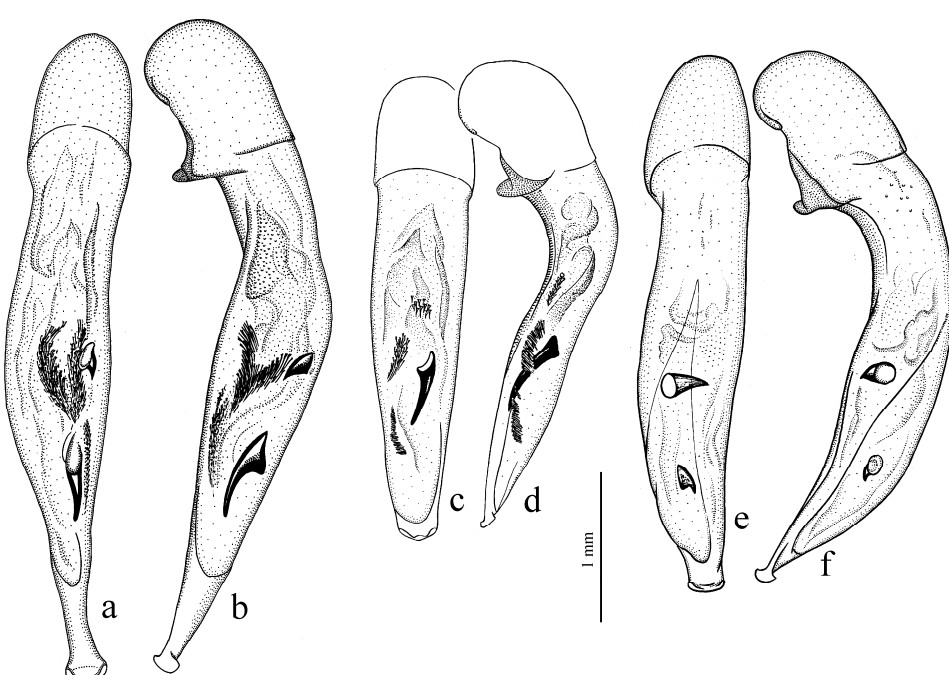

**Figure 58.** Median lobe of aedeagus of *Harpalus* species, dorsal and lateral views: (**a,b**) *H.* (*Aristoharpalus*) *ingenuus* (from [104]); (**c,d**) *H.* (*Cycloharpalus*) *pulvinatus* (from [93]); (**e,f**) *H.* (*Euryharpalus*) *cisteloides* (from [46]).

Ecology. Both species of this subgenus occur in mountains, in open habitats, at altitudes of 2000–3700 m.

Remarks. This subgenus is similar to the subgenera *Oreoharpalus* **subg. n.** and *Cycloharpalus* **subg. n.** in having the basal border of elytra densely setose, but it well differs from both in having a protibia with two ventroapical spines. Species of *Aristoharpalus* **subg. n.** are also recognizable by their elytra, which are finely punctate and setose basally and lack basal (parascutellar) and any discal pores.

The subgenus *Aristoharpalus* **subg. n.** corresponds to the *ingenuus* species group sensu Kryzhanovskij et al. [56] and sensu Kataev [8,104]. A revision of the species of this group was published by Kataev [104].

Subgenus *Cycloharpalus* **subg. n.**

https://zoobank.org/urn:lsid:zoobank.org:act:A8B04353-4C24-4682-9484-F00B09337FCC
Type species *Harpalus pulvinatus* Ménétries, 1848.

Diagnosis. Medium-sized (length 6.7–10.5 mm). Body convex, somewhat wide, without metallic luster. Head impunctate and glabrous. Pronotum impunctate basally, with rounded sides and setose basal edge. Elytra impunctate and glabrous, with setose basal border; basal (parascutellar) pore present; interval 3 generally without discal setigerous pore; interval 7 and 5 with or without preapical pores. Metepisternum as long as or longer than wide. Protibia with one ventroapical spine; preapical spines on outer margin of tibia isolated from spines on ventral surface of tibia or forming a single row with them; ventroapical tubercle in male not developed. Tarsi glabrous dorsally. Abdominal sternites with additional long setae. Median lobe of aedeagus with a relatively short terminal lamella and an oblique horseshoe-shaped apical capitulum; internal sac with one or two separate spines and one or three groups of small spines.

Etymology. The subgeneric name is a combination of the Greek *kýklos*, meaning "circle", and the name of the carabid taxon *Harpalus*.

Composition and distribution. This subgenus includes two Palaearctic species from deserts and semideserts of Transcaucasia, the lower Volga region and middle Asia: *H. pulvinatus* Ménétriés,

1848 (Figures 57b and 58c,d) (with the subspecies *H. p. lubricus* Reitter, 1900) and *H. breviusculus* Chaudoir, 1846.

Ecology. Both species of this subgenus inhabit lowland desert and semi-deserts but are usually found near water, in river floodplains or at lake shores.

Remarks. This subgenus is similar to *Aristoharpalus* **subg. n.** in the presence of setae on the basal border of the elytra and on the abdominal sternites but differs from it in the smaller body size, glabrous elytra without parascutellar pore, longer metepisterna, only one ventroapical spine on the protibia and aedeagus with a shorter terminal lamella.

The subgenus *Cycloharpalus* **subg. n.** corresponds to the *pulvinatus* species group sensu Kryzhanovskij et al. [56] and the *breviusculus* species group sensu Kataev [8]. The taxonomy of *H. pulvinatus* was discussed by Kataev [93].

Subgenus *Euryharpalus* **subg. n.**

https://zoobank.org/urn:lsid:zoobank.org:act:ED5DB44A-1EA6-4870-B68C-E9D02EA172F6
Type species *Harpalus cisteloides* Motschulsky, 1844.

Diagnosis. Medium-sized to large (length 8.7–13.1 mm). Body weakly convex, comparatively wide, without metallic luster. Head impunctate and glabrous. Pronotum impunctate or more or less distinctly punctate basally, with glabrous basal edge; sides not sinuate basally. Elytra impunctate and glabrous, with glabrous basal border and with one discal setigerous pore on interval 3; intervals 7 and 5 with or without preapical pores; basal (parascutellar) pore present. Metepisternum elongate, markedly longer than wide. Protibia with one ventroapical spine and four or five (rarely 6) preapical spines on outer margin, isolated from spines on ventral surface of tibia; ventroapical tubercle in male slightly prominent or indistinct. Tarsi glabrous dorsally. Abdominal sternites either almost glabrous, with a few short barely noticeable setae, or with dense, very short setation. Median lobe of aedeagus with a relatively short terminal lamella and with a horseshoe-shaped apical capitulum; internal sac either with one or two separate spines or with a group of small spines, or only small spiny patches.

Etymology. The subgeneric name is a combination of the Greek *eurys*, meaning "wide", and the name of the carabid taxon *Harpalus*.

Composition and distribution. This subgenus comprises four species from the eastern Palaearctic region: *H. cisteloides* Motschulsky, 1844 (Figure 58e,f) (with the subspecies *H. c. hurkai* Divoky, Pulpan et Rebl, 1990 and *H. c. schouberti* Tschitschérine, 1898), *H. aequicollis* Motschulsky, 1844 (Figure 59a), *H. heyrovskyi* Jedlička, 1928 and the taxonomically more separated *H. compressus* Motschulsky, 1844.

Ecology. Species of this subgenus inhabit the steppes and steppe meadows, both in the lowlands and in the mountains.

Remarks. Within the *Pheuginus* subgroup, the members of this subgenus are recognizable by having a glabrous basal border, an elongate metepisternum and an aedeagus with a horseshoe-shaped apical capitulum. In external characters, *Euryharpalus* **subg. n.** is most similar to the subgenus *Pheuginus* but sharply distinguished from it by the characteristics of the aedeagus. Its taxonopmic position requires further study.

The subgenus *Euryharpalus* **subg. n.** corresponds to the *cisteloides* species group sensu Kryzhanovskij et al. [56] and sensu Kataev [8], but *H. kadyrbekovi*, considered as a member of this group in the former publication, is included in the *Hypsoharpalus* **subg. n.** The taxonomy of *H. aequicollis* and *H. heyrovskyi* was discussed by Kataev [54].

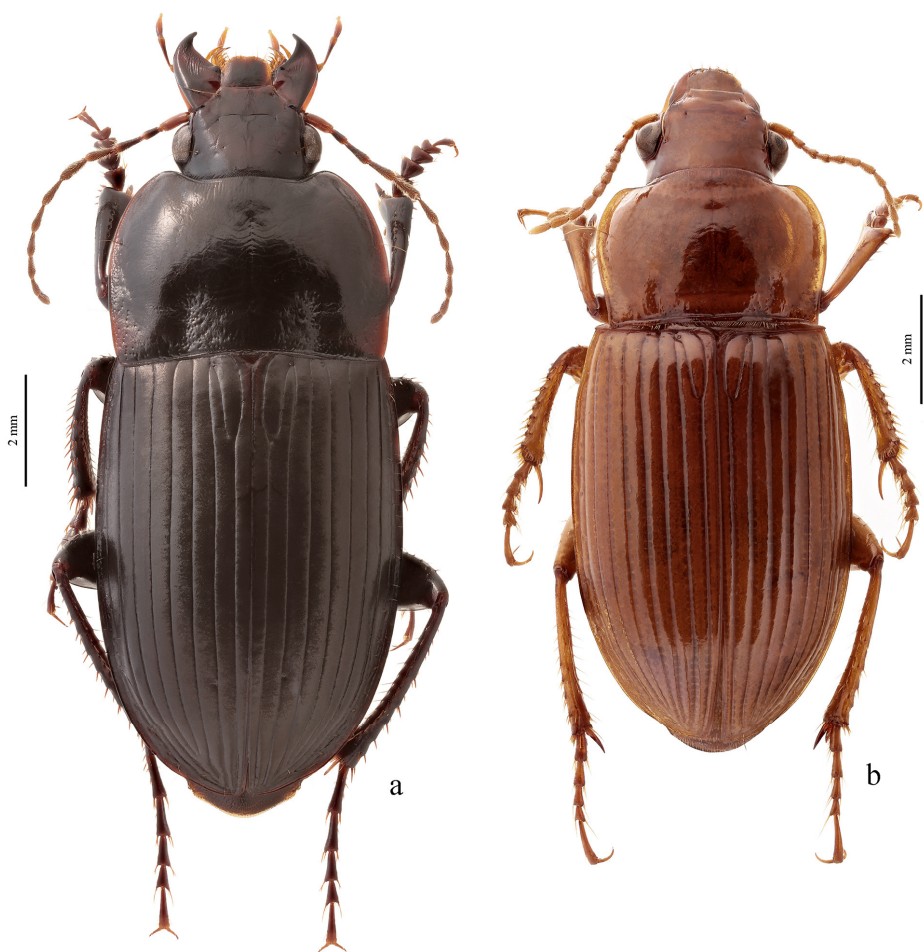

**Figure 59.** Habitus of *Harpalus* species: (**a**) *H.* (*Euryharpalus*) *aequicollis*; (**b**) *H.* (*Pharalus*) *indianus*.

*Pharalus* Subgroup

Diagnosis. Same as for the subgenus.
Composition and distribution. A monobasic subgroup, including only one Nearctic subgenus.

Subgenus *Pharalus* Casey, 1914

*Pharalus* Casey, 1914 [71] (p. 68) (as a genus). Type species *Pangus testaceus* LeConte, 1853 (=*H. indianus* Csiki, 1932), by original designation.

Diagnosis. Medium-sized (length 7.8–9.8 [10.7] mm). Body convex, slightly elongate or somewhat wide, unicolorous testaceous or reddish brown with metallic luster on dorsum. Head impunctate and glabrous. Pronotum finely punctate laterobasally or impunctate, with sides slightly sinuate basally, with one lateral seta on each side and with setose basal edge. Elytra impunctate and glabrous, with glabrous basal border; interval 3 with or without discal pore; intervals 7 and 5 without preapical pores. Prosternum almost glabrous, at most with a few very short setae apically. Metepisternum longer than wide. Metacoxa with one or several setae medially. Protibia with one or two (occasionally three) ventroapical spines and with four to six preapical spines on outer margin, forming a single row with spines on ventral surface of tibia; ventroapical tubercle in male absent. Male mesotibia without preapical callous thickening on inner margin. Tarsi glabrous dorsally; metatarsomere 1 short. Abdominal sternites with additional long setae; last visible sternite without pronounced sexual dimorphism, its apex in both sexes more or less rounded and not swollen. Median lobe of aedeagus with elongate terminal lamella and with large

discoidal apical capitulum; internal sac with spiny patches similar to that of members of *Hypsinephus* subgroup; large separate spines absent.

Composition and distribution. This subgenus includes two Nearctic species from the eastern part of North America: *H. indianus* Csiki, 1932 (Figure 59b) and *H. gravis* LeConte, 1858.

Ecology. Both species of this subgenus occur in various open habitats, but *H. indianus* prefers drier areas than *H. gravis*, and usually areas with a significant proportion of sand in the soil, colonizing sand dunes as well [140].

Remarks. This habitually well-recognized subgenus occupies a somewhat intermediate position in external morphology and male genitalia between the subgenera of *Pheuginus* and *Hypsinephus* subgroups.

Noonan [16] treated *Pharalus* as the *indianus* subgroup of the *desertus* group. This subgenus corresponds to the *indianus* species group sensu Kataev [8].

*Hypsinephus* Subgroup

Diagnosis. Head impunctate and glabrous (occasionally with short setae around basal foveae in *Hemipangus* **subg. n.**), tempora generally glabrous, in *Hemipangus* **subg. n.** and *Loxophonus* usually setose. Pronotum impunctate or punctate basally, with sides generally rounded, more rarely slightly sinuate basally, with one or several lateral setae on each side and with basal edge generally setose. Elytra in most species impunctate and glabrous, more rarely punctate and pubescent; basal border setose; basal (parascutellar) pore generally present (occasionally absent in some species); interval 3 with one or several discal setigerous pores (occasionally pore absent); in most species, interval 7, often also intervals 5 and 3, with short row of preapical setigerous pores (these pores not recognizable in most species with punctate elytra). Metacoxae with additional setae medially, rarely (in one species of subgenus *Hypsinephus*) also with a posteromedial pore. Protibia with one, more rarely two or three, ventroapical spines in a transverse row and with three or four preapical spines on outer margin, isolated from spines on ventral surface of tibia or forming with them a single row; ventroapical tubercle in male not prominent. Male mesotibia without preapical callous thickening on inner margin. Tarsi glabrous dorsally; metatarsomere 1 average for genus or slightly more elongate. Abdominal sternites with more or less numerous additional setae; last visible sternite generally without pronounced sexual dimorphism, in some species its apex slightly truncate in male and slightly swollen in female. Median lobe of aedeagus with more or less long terminal lamella and a horseshoe-shaped apical capitulum; internal sac generally with a characteristic more or less elongate spiny patch in apical half of median lobe or medially, usually in addition to spiny patches or groups of small spines; separate spines absent.

Composition and distribution. This subgroup comprises one Holarctic and four Palaearctic subgenera.

Ecology. Species of this subgroup inhabit open arid and semi-arid landscapes.

Remarks. In addition to the characteristic aedeagus, the main distinctive features of this subgroup are the setae on the pronotal basal edge, elytral basal border and abdominal sternites and a short row of preapical setigerous pores on the elytral interval 7.

Subgenus *Hypsinephus* Bates, 1878

*Hypsinephus* Bates, 1878 [171] (p. 715) (as a genus). Type species *Hypsinephus ellipticus* Bates, 1878 (=*H. salinus* Dejean, 1829), by monotypy.

*Rapahlus* Lutshnik, 1922 [21] (p. 61) (as a subgenus of *Harpalus* Latreille, 1802). Type species *Harpalus salinus* Dejean, 1829, by original designation.

Diagnosis. Medium-sized to large (length 8.2–14.0 mm). Body moderately convex, more or less elongate, brown to almost black, without metallic luster. Basal outer angles of mandibles not angularly prominent. Pronotum with one lateral seta on each side and with basal angles rounded or sharp at tip. Elytra impunctate and glabrous; intervals 3 and

5 with several (up to ten) discal setigerous pores along entire length; occasionally with additional discal pores also on intervals 2, 4 and 6. Metepisternum markedly longer than wide. Protibia with two or three ventroapical spines and with four preapical spines on outer margin, isolated from spines on ventral surface of tibia and not forming a single row with them (Figure 60b,c).

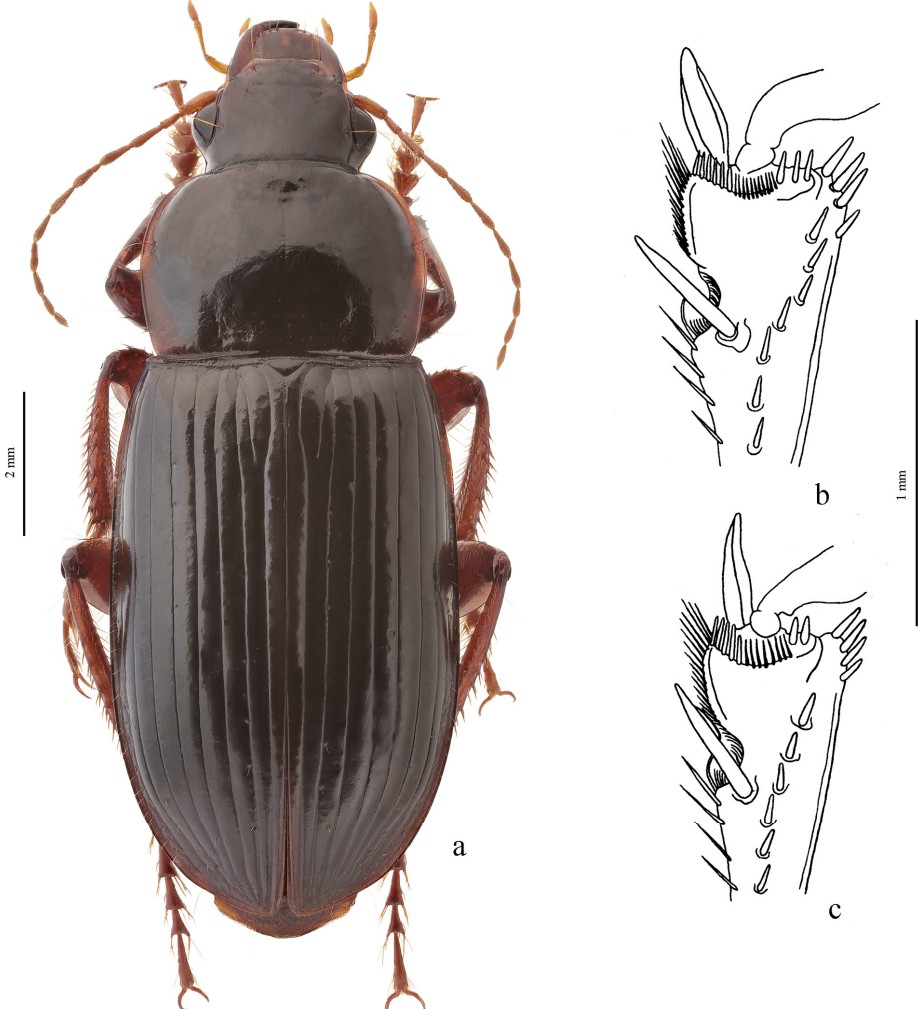

**Figure 60.** Habitus and apical part of left protibia of *Harpalus* species: (**a**,**b**) *H.* (*Hypsinephus*) *salinus* (from [84]); (**c**) *H.* (*H.*) *lumbaris* (from [84]).

Composition and distribution. This subgenus includes two species from the eastern Palaearctic region: *H. salinus* Dejean, 1829 (Figures 60a and 61a,b) (with the subspecies *H. s. agonus* Tschitschérine, 1894 and *H. s. klementzae* Kataev, 1984) and *H. lumbaris* Mannerheim, 1825.

Ecology. Both species occur in dry steppe and semi-desert habitats, both on the plains and in the mountains.

Remarks. Within the subgroup, the members of this subgenus are recognizable by the elytra with numerous discal setigerous pores at least on the intervals 3 and 5. Interestingly, the metacoxa of *H. salinus* has a posteromedial pore, buth this pore is absent in *H. lumbaris*.

This subgenus corresponds to the *salinus* species group sensu Kryzhanovskij et al. [56] and to the *lumbaris* species group sensu Kataev [8]. The revision of *Hypsinephus* was published by Kataev [84].

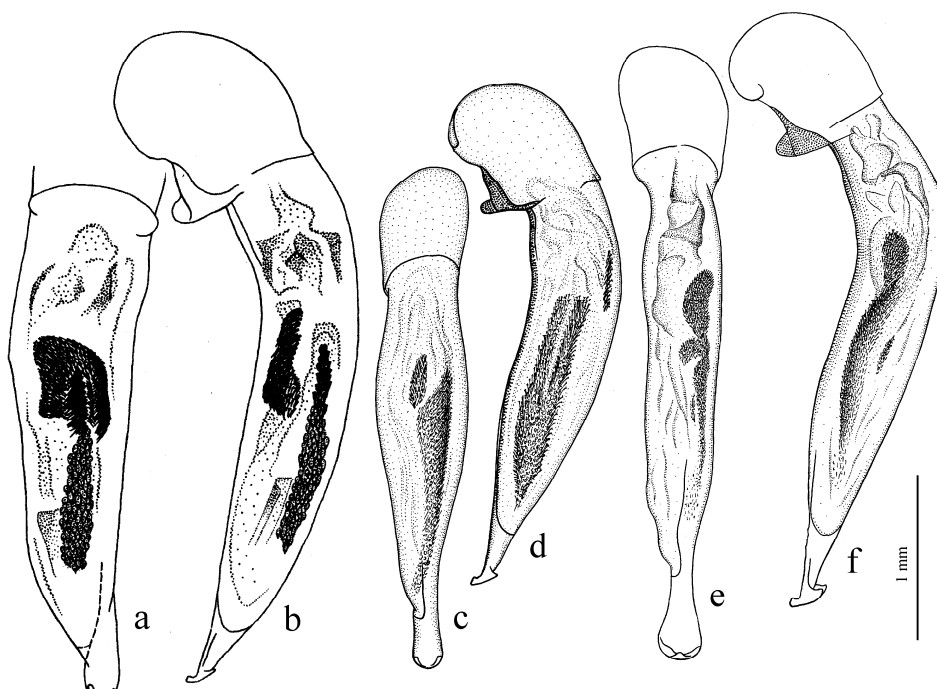

**Figure 61.** Median lobe of aedeagus of *Harpalus* species, dorsal and lateral views: (**a**,**b**) *H.* (*Hypsinephus*) *salinus* (from [84]); (**c**,**d**) *H.* (*Brachyharpalus*) *autumnalis*; (**e**,**f**) *H.* (*B.*) *kunarensis* (from [93]).

Subgenus *Brachyharpalus* **subg. n.**

https://zoobank.org/urn:lsid:zoobank.org:act:45ADC665-7078-4B73-A792-20DB2C1AB3C7
Type species *Carabus autumnalis* Duftschmid, 1812.

Diagnosis. Medium-sized (length 5.9–10.6 mm). Body moderately convex, often stout, brown to black, without metallic luster. Basal outer angles of mandibles not angularly prominent (Figure 63a). Pronotum with one lateral seta on each side and with basal angles rounded or blunted at tip. Elytra impunctate and glabrous; interval 3 with one to five discal setigerous pores (rarely without pores), interval 5 without discal pores (rarely with one or two pores basally). Metepisternum variable, elongate or wider than long (Figure 7f). Protibia generally with one, rarely two, ventroapical spines and with three or four preapical spines on outer margin, usually isolated from spines on ventral surface of tibia and not forming a single row with them.

Etymology. The subgeneric name is a combination of the Greek *brakhús*, meaning "short", and the name of the carabid taxon *Harpalus*.

Composition and distribution. This subgenus comprises about 14 predominantly wingless species from the arid and semiarid regions of the Palaerctic and Nearctic, with four isolated faunal centers.

The largest of them, the European and Middle Eastern center, includes eight winged and wingless species: *H. autumnalis* (Duftschmid, 1812) (Figure 61c,d), *H. danieli* Reitter, 1900, *H. brachypterus* Tschitschérine, 1898, *H. anatolicus* Tschitschérine, 1898 (with the subspecies *H. a. lydius* Kataev et Wrase, 1997, *H. a. caricus* Kataev et Wrase, 1997 and *H. a. lycius* Kataev et Wrase, 1997), *H. triseriatus* Fleischer, 1897 (with the subspecies *H. t. babunensis* Mlynář, 1979), *H. kazanensis* Jedlička, 1958 (Figure 62a), *H. reflexus* Putzeys, 1878 (with the subspecies *H. anadoluensis* Kataev, 1993) and *H. foveiger* Tschitschérine, 1895.

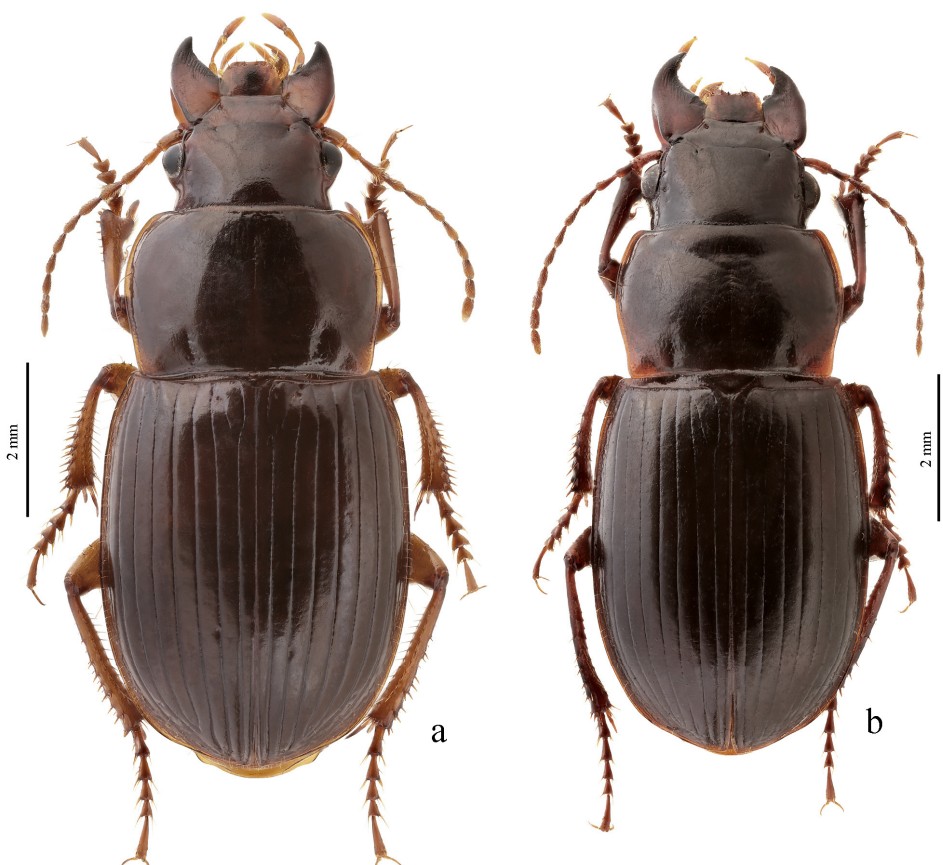

**Figure 62.** Habitus of *Harpalus* species: (**a**) *H.* (*Brachyharpalus*) *kazanensis*; (**b**) *H.* (*Brachypangus*) *antonowi*.

The second faunal center is located in the western part of the Himalayas and is represented there by only one wingless species, *H. kunarensis* Kataev, 1993 (Figure 61e,f).

The third, Tibetan, faunal center includes two sympatric wingless species endemic to Tibet: *H. lama* Kataev et Wrase, 1997 and *H. ascetes* Kataev et Wrase, 1997.

The fourth, Nearctic, faunal center includes three species (wingless or dimorphic) from western North America: *H. desertus* LeConte, 1859, *H. furtivus* LeConte, 1865 and the insufficiently studied *H. durangoensis* Bates, 1891.

Ecology. All species of this subgenus occur in relatively dry open habitats, most in dry steppes and semi-deserts, both in lowlands and in mountains.

Remarks. In combination of characters, this subgenus is very similar to *Hypsinephus* but differs from it in less number of discal setigerous pores (sometimes without pores) on the elytral intervals 3 and 5. *Brachyharpalus* **subg. n.** is distinguished from the monotypical *Brachypangus* by having the basal outer angles of the mandibles not prominent.

This subgenus corresponds to the *desertus* group sensu Lindroth [15], recognized for the Nearctic species, and to the *autumnalis* species group sensu Mlynář [76], sensu Kataev [8,93], sensu Kryzhanovskij et al. [56] and sensu Kataev and Wrase [116], recognized for the Palaearctic species. Noonan [16] considered *H. desertus* (with *H. furtivus* as its synonym, the *furtivus* morph) to be a single member of the *desertus* subgroup of the *desertus* species group, in which he included also the *indianus* subgroup. According to my data, *H. furtivus* distinctly differs from *H. desertus* at least in having a short and wide (wider than long) metepisternum; in *H. desertus*, the metepisternum is slightly longer than wide, markedly narrowed posteriorly. Noonan [16] made no mention of this distinctive character in his revision.

The Palaearctic species of this subgenus were partly revised by Mlynář [76], Kataev [93] and Kataev and Wrase [116]. Several Palaearctic species have not yet been described.

Subgenus *Brachypangus* Tschitschérine, 1898

*Brachypangus* Tschitschérine, 1898 [172] (p. 174) (as a genus). Type species *Brachypangus antonowi* Tschitschérine, 1898, by monotypy.

Diagnosis. Medium-sized (length 8.3–10.0 mm). Body convex and stout, brown to dark brown, without metallic luster. Basal outer angles of mandibles angularly prominent (Figure 63b). Pronotum with one lateral seta on each side and with basal angles sharp, not blunted ar tip. Elytra impunctate and glabrous; interval 3 with one discal setigerous pore or without it; interval 5 without discal pores. Metepisternum short, wider than long. Protibia with one ventroapical spines and with three preapical spines on outer margin, isolated from spines on ventral surface of tibia and not forming a single row with them.

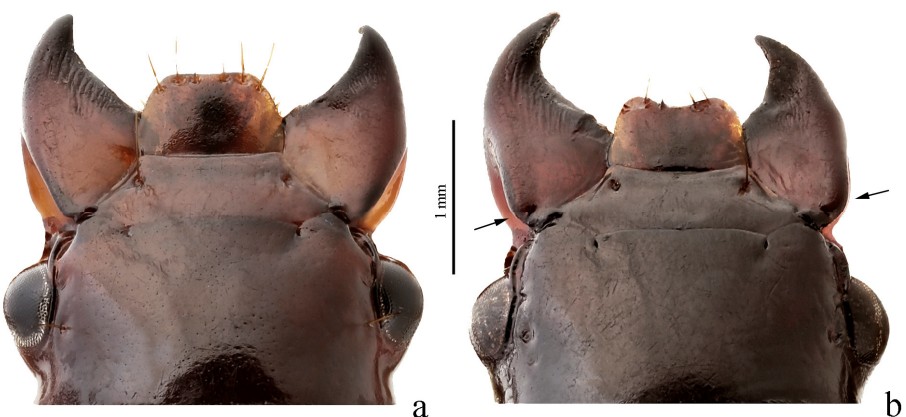

**Figure 63.** Head of *Harpalus* species: (**a**) *H. (Brachyharpalus) kazanensis*; (**b**) *H. (Brachypangus) antonowi*. Arrows point to basal outer angles of mandibles.

Composition and distribution. This subgenus includes only one very rare wingless Palaearctic species, *H. antonowi* (Tschitschérine, 1898) (Figure 62b), endemic to the Kopetdag.

Ecology. The single species of this subgenus occurs in dry, open habitats.

Remarks. In combination of characters, this subgenus coincides with *Brachyharpalus* **subg. n.**, differing from it in the prominent basal outer angles of the mandibles, similar to those in *Ophonus convexicollis* (Ménétriés, 1832).

The subgenus *Brachypangus* corresponds to the *antonowi* species group sensu Kryzhanovskij et al. [56] and sensu Kataev [8].

Subgenus *Hemipangus* **subg. n.**

https://zoobank.org/urn:lsid:zoobank.org:act:CB848051-5D5D-4987-AE3E-0AA164EFD699
Type species *Harpalus klapperichi* Jedlička, 1955.

Diagnosis. Medium-sized (length 8.3–10.6 mm). Body moderately convex, elongate, brown to black, without metallic luster. Tempora generally setose. Ligular sclerite in addition to two long ventroapical setae generally also with several short dorsal setae. Basal outer angles of mandibles not angularly prominent. Pronotum with one lateral seta on each side and with basal angles rounded or somewhat sharp. Elytra coarsely punctate and pubescent on lateral intervals; interval 3 with one to four discal setigerous pores, interval 5 without discal pores or with one or two pores basally. Metepisternum short, wider than long. Protibia with one ventroapical spine and with three or four preapical spines on outer margin, isolated from spines on ventral surface of tibia and not forming a single row with them.

Etymology. The subgeneric name is a combination of the Greek *hēmi-*, meaning "semi-, quasi-, half", and the name of the carabid taxon *Harpalus*.

Composition and distribution. This subgenus includes two wingless Palaearctic species from Afghan and Tadjik Badakhshan: *H. klapperichi* Jedlička, 1955 (Figure 65a,b) and *H. badakschanus* Jedlička, 1955 (Figure 64a).

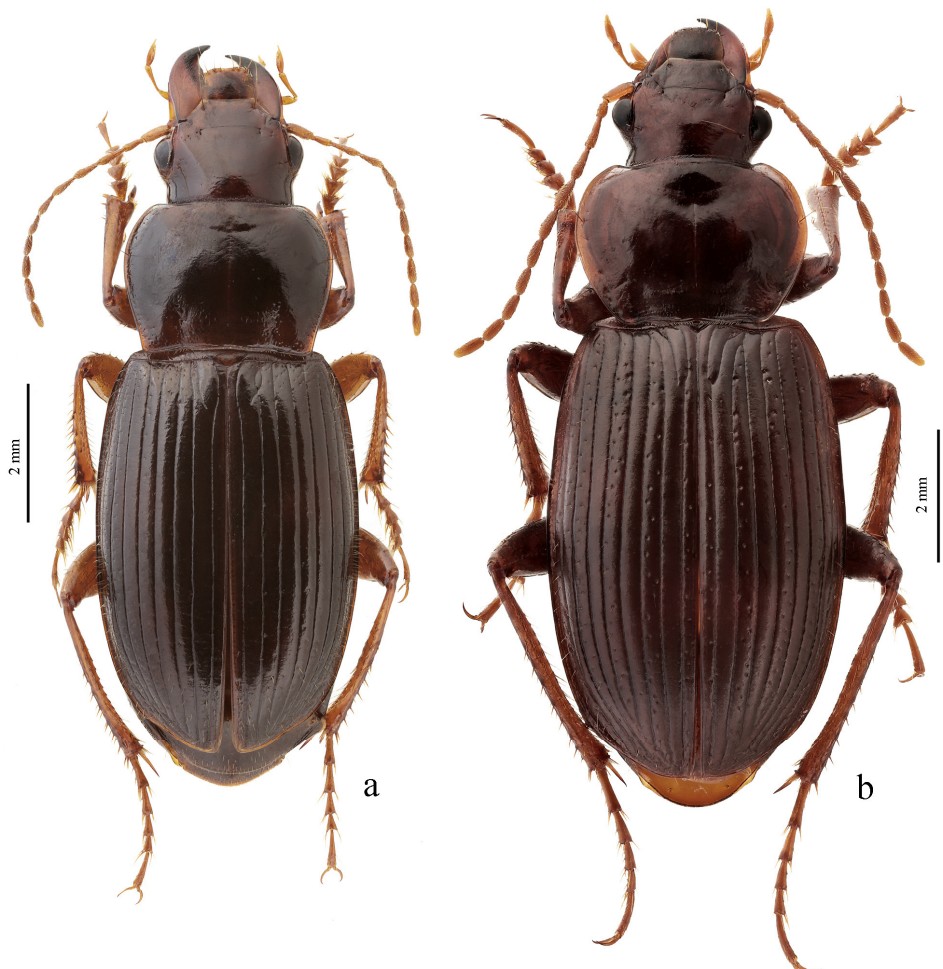

**Figure 64.** Habitus of *Harpalus* species: (**a**) *H.* (*Hemipangus*) *badakschanus*; (**b**) *H.* (*Loxophonus*) *agakhaniantzi*.

Ecology. Both species occur in the highlands (2600–4200 m) in relatively humid habitats.

Remarks. This subgenus differs from *Brachyharpalus* **subg. n.** in the elytra coarsely punctate and setose on the lateral intervals and differs from *Loxophonus* in one lateral seta on each side of the pronotum.

The subgenus *Hemipangus* **subg. n.** corresponds to the *klapperichi* species group sensu Kataev [8].

Subgenus *Loxophonus* Reitter, 1894

*Loxophonus* Reitter, 1894 [173] (p. 124) (as a genus). Type species *Loxophonus setiporus* Reitter, 1894, by monotypy.

Diagnosis. Medium-sized (length 8.5–11.0 mm). Body weakly convex, elongate, brown to almost black, without metallic luster. Tempora setose. Ligular sclerite in addition to two long ventroapical setae also with several short dorsal setae. Basal outer angles of mandibles not angularly prominent. Pronotum with several lateral setae on each side and with basal angles rounded. Elytra coarsely punctate and setose throughout or on odd intervals; interval 3 and 5 without discal setigerous pores (at least they not recognizable against background of punctation). Metepisternum short, as wide as or slightly wider than long. Protibia with one ventroapical spine and with three or four preapical spines on

outer margin, isolated from spines on ventral surface of tibia and not forming a single row with them.

Composition and distribution. This subgenus includes two wingless allopatric Palaearctic species, endemic to the eastern Hissar-Darvaz Mountains: *H. setiporus* (Reitter, 1894) (Figure 65c,d) and *H. agakhaniantzi* (Michailov, 1972) (Figure 64b).

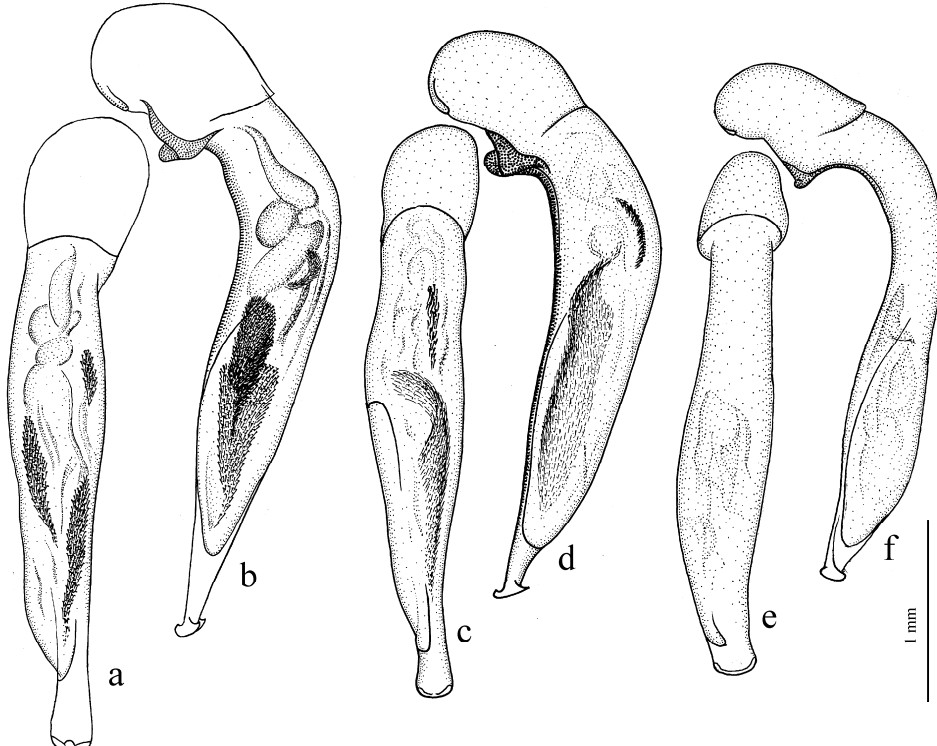

**Figure 65.** Median lobe of aedeagus of *Harpalus* species, dorsal and lateral views: (**a**,**b**) *H. (Hemipangus) klapperichi* (from [93]); (**c**,**d**) *H. (Loxophonus) setiporus*; (**e**,**f**) *H. (Mauriharpalus) cardoni*.

Ecology. Both species occur in relatively humid habitats, both in mountain forests and in the subalpine zone [174].

Remarks. This subgenus is readily distinguished from other taxa of the *Hypsynephus* subgroup by the presence of several lateral setae on the pronotum.

The subgenus *Loxophonus* corresponds to the *setiporus* species group sensu Kryzhanovskij et al. [56] and sensu Kataev [8], but *H. badakschanus*, which was treated as a member of this group in the former publication, is included in the subgenus *Hemipangus* **subg. n.**

*Mauriharpalus* Subgroup

Diagnosis. Same as for the subgenus.
Composition and distribution. This subgenus includes only one western Palaearctic subgenus.

Subgenus *Mauriharpalus* **subg. n.**

https://zoobank.org/urn:lsid:zoobank.org:act:95D29B38-77E7-4C73-A018-C11583331CB7
Type species *Harpalus cardoni* Antoine, 1922.

Diagnosis. Medium-sized (length 7.0–11.0 mm). Body moderately convex, elongate, brown to black, without metallic luster. Head impunctate, glabrous. Pronotum punctate basally, with one lateral seta on each side and with glabrous basal edge. Elytra impunctate and glabrous, with glabrous basal border; interval 3 generally with one, occasionally two, discal setigerous pores. Prosternum with very short setae. Protibia with one ventroapical spine and with three preapical spines on outer margin, not forming a single row with spines

on ventral surface of tibia; ventroapical tubercle in male prominent. Male mesotibia slightly curved, with distinct preapical callous thickening on inner margin. Abdominal sternites almost glabrous, only with very short setae basally; last visible sternite without pronounced sexual dimorphism, its apex in both sexes more or less rounded and not swollen. Median lobe of aedeagus with short terminal lamella and with almost discoidal apical capitulum; internal sac without distinct sclerotic elements (Figure 65e,f).

Etymology. The subgeneric name is a combination of *mauri* (Moor), the name the ancient inhabitants of northwest Africa, and the name of the carabid taxon *Harpalus*.

Composition and distribution. This subgenus includes only *H. cardoni* Antoine, 1922 (Figure 65e,f) from the western Mediterranean region.

Ecology. The only species of this subgenus occur along the edges of small reservoirs on sandy soil, under stones or at the base of herbaceous plants [40].

Remarks. The taxonomic position of *H. cardoni* requires further research since it demonstrates a unique combination of the distinctive characters dissimilar to any other known species. This species is characterized by an anal sternite without sexual differentiation; however, the male has a distinct callous thickening on the inner margin on the inner margin of mesotibia and a markedly prominent ventroapical tubercle on the protibia.

The subgenus *Mauriharpalus* **subg. n.** corresponds to the *cardoni* species group sensu Kataev [8]. Antoine [40] included *H. cardoni* in one species group together with *H. numidicus*, based mainly on coloration of the body, but the latter species differs in many characters, including several discal setigerous pores on the elytral interval 3 and the structure of the aedeagus, and it is apparently unrelated to *H. cardoni*.

*Caloharpalus* Subgroup

Diagnosis. Head glabrous, impunctate or very finely punctate. Pronotum densely punctate basally, with one lateral seta on each side and with glabrous basal edge. Elytra either impunctate and glabrous or punctate and pubescent, with glabrous basal border; interval 3 with one to several discal setigerous pores. Prosternum with short or moderately long setae. Metacoxa with additional setae medially. Protibia with one ventroapical spine and with four to eight preapical spines on outer margin, not forming a single row with spines on ventral surface of tibia; ventroapical tubercle in male indistinct or prominent. Male mesotibia generally with more or less distinct preapical callous thickening on inner margin (in some species this thickening almost not developed). Abdominal sternites generally with additional numerous setae; last visible sternite with pronounced sexual dimorphism: its apex rounded or subtruncate in male, and swollen, slightly expanded posteriorly in female. Median lobe of aedeagus generally with relatively long (at least longer than wide, but usually much longer) terminal lamella and apical capitulum of various shapes; armament of internal sac generally well developed, usually including a characteristic chain of short and wide spines, and with or without one separate spine.

Composition and distribution. This subgroup comprises three Palaearctic subgenera.

Ecology. Species of this subgroup inhabit mainly arid landscapes, but are usually found in fairly humid biotopes.

Remarks. This subgroup is similar to the following three subgroups (*Idioharpalus*, *Artabas* and *Harpalus* subgroups) in having additional setae on the abdominal sternites and the swollen apex of the last visible female sternite, among other common characters. The males of the *Caloharpalus* sugroup are also similar to those of the *Harpalus* subgroup in a more or less distinct preapical callous thickening on the inner margin of the mesotibia. The members of the *Caloharpalus* sugroup differ from members of these three subgroups in the characteristic male genitalia and also differ from the *Idioharpalus* subgroup and *Harpalus* subgroup in having only one ventroapical spine on the protibia, and from the *Artabas* subgroup in the unmodified mesotibia of the male. In addition, the *Caloharpalus* subgroup is distinct from the *Harpalus* subgroup in having a glabrous basal pronotal edge.

Subgenus *Caloharpalus* **subg. n.**

https://zoobank.org/urn:lsid:zoobank.org:act:1FE02295-AC88-4A84-977A-5F6FB74823F4
Type species *Harpalus cupreus* Dejean, 1829.

Diagnosis. Medium-sized to large (length 10.3–16.0 mm). Body moderately convex, somewhat elongate, with metallic luster on dorsum. Elytra impunctate and glabrous, with one discal setigerous pore on interval 3 and without preapical pores on intervals 5 and 7. Male profemur without a dense row of setae along inner anterior margin. Internal sac of aedeagus with a separate spine in addition to a longitudinal chain of spines and groups of small spines (spiny patches).

Etymology. The subgeneric name is a combination of the Greek *kalós*, meaning "beauty", and the name of the carabid taxon *Harpalus*.

Composition and distribution. This subgenus includes three Palaearctic species: *H. cupreus* Dejean, 1829 (Figure 67a,b) (with the subspecies *H. c. asaphus* Antoine, 1940, *H. c. rhodopus* Schauberger, 1930, *H. c. ragusae* Müller, 1924, *H. c. fastuosus* Faldermann, 1836) and *H. reitteri* Wrase et Kataev, 2011 from the western Palaearctic, and *H. chalcentus* Bates, 1873 (Figure 66a) from eastern Asia.

Ecology. At least one of the species of this subgenus (*H. cupreus*) occurs in open, wet habitats, such as along the banks of ponds and swampy lowland areas with dense herbaceous vegetation.

Remarks. The least specialized subgenus within this subgroup, probably basal to the other two subgenera. It is characterized by the absence of the specialized elytral and protibial setations present in *Baryharpalus* **subg. n.** and *Heteroharpalus* **subg. n.**

The subgenus *Caloharpalus* **subg. n.** corresponds to the *cupreus* species group sensu Kataev [8,86] and Kryzhanovskij et al. [56]. The taxonomy of the western Palaerctic species was studied by Schauberger [161] and Puel [175].

Subgenus *Baryharpalus* **subg. n.**

https://zoobank.org/urn:lsid:zoobank.org:act:9E5DCCAF-BE9B-4CDE-AEF4-E340381052DD
Type species *Carabus dimidiatus* Rossi, 1790.

Diagnosis. Medium-sized to large (length 9.7–13.6 mm). Body convex and stout, dark brown to black, with or without metallic luster on dorsum. Elytra either impunctate and glabrous or coarsely punctate and pubescent, with one discal setigerous pore on interval 3 and generally with several preapical pores on intervals 7 and 5, or only 7 (these pores absent in *H. torosensis*). Profemur of male without a dense row of setae along inner anterior margin (in some species with slightly more number of setae in male than in female). Internal sac of aedeagus with a longitudinal chain of spines and often also groups of medium-sized or small spines; a separate spine present or absent.

Etymology. The subgeneric name is a combination of the Greek *barús*, meaning "heavy", and the name of the carabid taxon *Harpalus*.

Composition and distribution. This subgenus comprises six western Palaearctic species, distributed mainly in the eastern Mediterranean: *H. dimidiatus* (Rossi, 1790) (Figure 67c,d), *H. caspius* (Steven, 1806) (Figure 66b), *H. karamani* Apfelbeck, 1902, *H. murzini* Kataev, 2008 and the taxonomically slightly more separated *H. trichophorus* Tschitschérine, 1897 and *H. torosensis* Jedlička, 1961.

Ecology. Species of this subgenus occur in moderately humid habitats, mainly near forests.

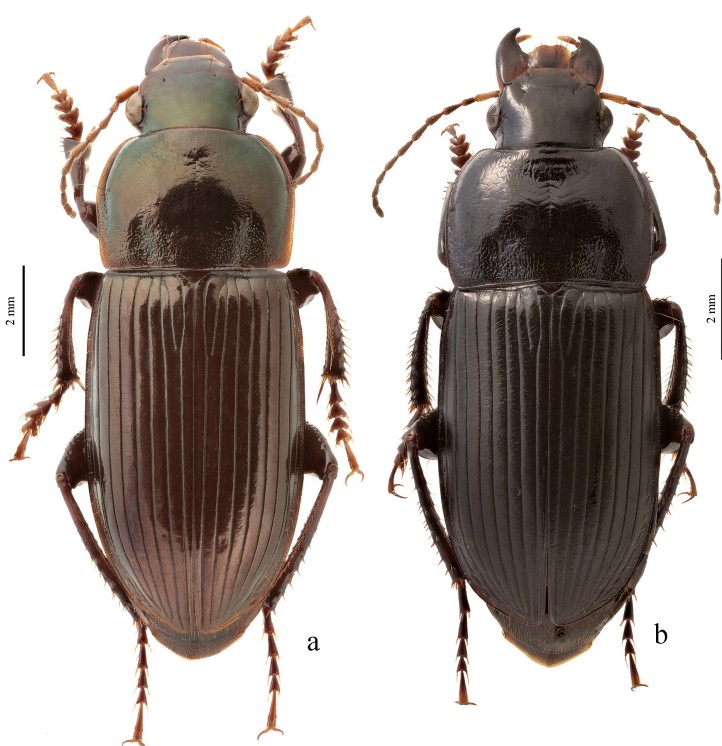

**Figure 66.** Habitus of *Harpalus* species: (**a**) *H.* (*Caloharpalus*) *chalcentus*; (**b**) *H.* (*Baryharpalus*) *caspius*.

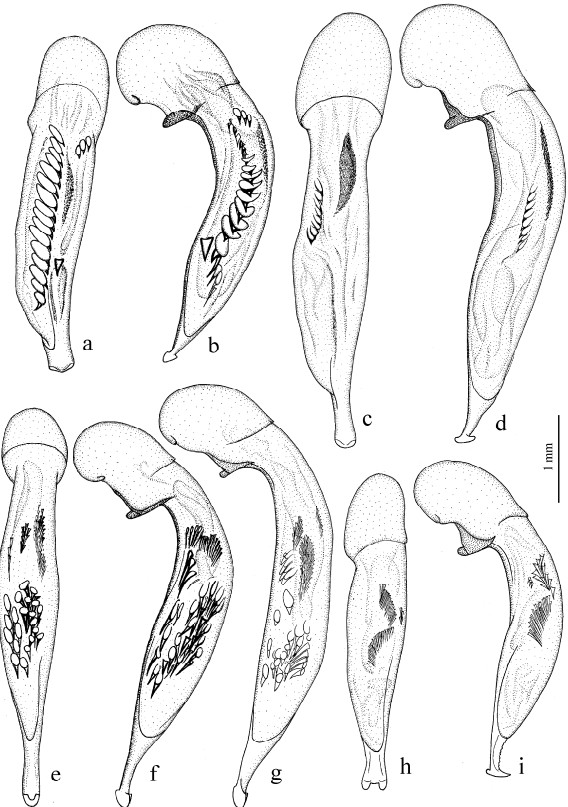

**Figure 67.** Median lobe of aedeagus of *Harpalus* species, dorsal and lateral views: (**a**,**b**) *H.* (*Caloharpalus*) *cupreus*; (**c**,**d**) *H.* (*Baryharpalus*) *dimidiatus* (from [105]); (**e**,**f**) *H.* (*Heteroharpalus*) *metallinus*; (**g**) *H.* (*H.*) *tithonus*; (**h**,**i**), *H.* (*H.*) *tiridates*.

Remarks. This subgenus differs from *Caloharpalus* **subg. n.** mainly in a more convex and stout body, with elytra in most species (except for *H. torosensis* having punctate and pubescent elytra) with several preapical pores on the intervals 7 and 5. In punctate and pubescent elytra, *H. trichophorus* and *H. torosensis* are similar to the species of the subgenus *Heteroharpalus* **subg. n.** but distinguished from them by having protibia in male without a dense row of setae along inner anterior margin and elytral interval 3 with only one discal setigerous pore.

The subgenus *Baryharpalus* **subg. n.** corresponds to the *dimidiatus* species group sensu Kataev [8,86,105] and sensu Kryzhanovskij et al. [56].

The species with impunctate and glabrous elytra were revised by Schauberger [65], Puel [176] and Kataev [105].

Subgenus *Heteroharpalus* **subg. n.**

https://zoobank.org/urn:lsid:zoobank.org:act:9A79D0BF-A7ED-4305-8529-1A44617723A1
Type species *Harpalus metallinus* Ménétries, 1839.

Diagnosis. Medium-sized to somewhat large (length 8.0–12.2 mm). Body moderately convex to rather flat, stout or elongate, brown to black, with or without metallic luster. Elytra generally more or less punctate and pubescent (punctures often arranged in rows along striae), more rarely (in *H. mauritanicus*) impunctate and glabrous; interval 3 generally with several (up to five) discal setigerous pores (occasionally discal pores also present on interval 5 and 7, or discal pores indistinguishable among ordinary punctation); preapical pores on intervals 5 and 7 absent (present in *H. polyglyptus*). Profemur of male with a more or less dense row of setae along inner anterior margin (Figure 15a). Median lobe of aedeagus and its apical capitulum are very variable in shape in different species; internal sac in most species with groups of medium-sized or large spines, without separate spines.

Etymology. The subgeneric name is a combination of the Greek *héteros*, meaning "different, variable", and the name of the carabid taxon *Harpalus*.

Composition and distribution. This subgenus comprises eight western Palaearctic species. Seven of them are distributed mainly in the eastern Mediterranean: *H. metallinus* Ménétriés, 1838 (Figures 67e,f and 68a) (with the subspecies *H. pharisaeus* Reiche et Saulcy, 1855), *H. caiphus* Reiche et Saulcy, 1855), *H. jordanus* Jedlička, 1964, *H. tithonus* Reitter, 1900 (Figure 67g), *H. tiridates* Reitter, 1900 (Figure 67h,i), *H. akbensis* Jedlička, 1958, *H. polyglyptus* Schaum, 1862, and one species, the taxonomically and geographically more separated *H. mauritanicus* Gaubil, 1844, is known from Algeria.

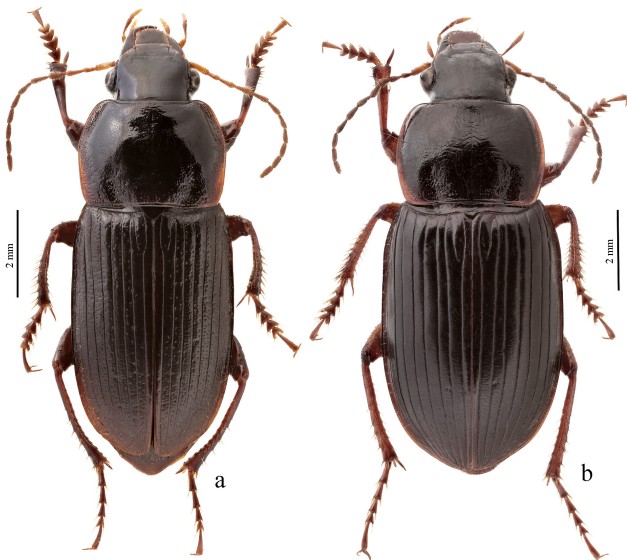

**Figure 68.** Habitus of *Harpalus* species: (**a**) *H.* (*Heteroharpalus*) *metallinus*; (**b**) *H.* (*Idioharpalus*) *numidicus*.

Ecology. Species of this subgenus occur in deserts and semi-deserts, usually not far from water.

Remarks. This subgenus differs from the other two subgenera of the *Caloharpalus* subgroup in having a male profemur with a denser row of setae along the inner anterior margin; many species also have punctate elytra with several discal setigerous pores on the elytral interval 3.

This subgenus corresponds to the *metallinus* species group sensu Jedlička [177], sensu Kataev [8,86] and sensu Kryzhanovskij et al. [56].

### *Idioharpalus* Subgroup

Diagnosis. Same as for the subgenus.

Composition and distribution. A monobasic group, including only one western Palaearctic subgenus.

### Subgenus *Idioharpalus* **subg. n.**

https://zoobank.org/urn:lsid:zoobank.org:act:E3668219-2D5A-4424-9060-704D60965102
Type species *Harpalus numidicus* Bedel, 1893.

Diagnosis. Medium-sized (length 8.2–11.0 mm). Body moderately convex, somewhat elongate, brown to black, without metallic luster. Head impunctate and glabrous. Pronotum more or less widely punctate basally, with one lateral seta on each side and with glabrous basal edge. Elytra impunctate and glabrous, with glabrous basal border and with four to six discal setigerous pores on interval 3; intervals 7, occasionally also 5 and 3, with preapical pores. Prosternum with rather long setae. Profemur of male without a dense row of setae along inner anterior margin. Protibia with three, occasionally two or four, spines in a transverse row and with five or six (occasionally seven or eight) preapical spines on outer margin, not forming a single row with spines on ventral surface of tibia; ventroapical tubercle in male not developed. Male mesotibia without a preapical callous thickening on inner margin. Abdominal sternites with more or less numerous additional setae; last visible sternite with sexual dimorphism: its apex subtruncate in male and slightly swollen (but not expanded posteriorly) in female. Median lobe of aedeagus with a short terminal lamella and a horseshoe-shaped apical capitulum; internal sac with a large group of long and narrow spines.

Etymology. The subgeneric name is a combination of the Greek *idios*, meaning "separate, special, one's own", and the name of the carabid taxon *Harpalus*.

Composition and distribution. This subgenus includes only the western Palaearctic *H. numidicus* Bedel, 1893 (Figure 68b), distributed in northwest Africa and in the south of the Iberian Peninsula.

Ecology. The only species of this subgenus occurs on fairly moist clayey soil near water bodies but at some distance from water [40].

Remarks. This subgenus is somewhat intermediate in its morphology between the *Caloharpalus* subgroup and *Artabas* subgroup. Having generally three ventroapical spines on the protibia, it is similar to *Smirnovia* but differs in the presence of several discal setigerous pores on the elytral interval 3, as in many members of *Heteroharpalus* **subg. n.** and some *Artabas*. Unlike members of the *Artabas* subgroup, the only representative of *Idioharpalus* **subg. n.** has a large group of long and narrow spines in the internal sac of the aedeagus.

The subgenus *Idioharpalus* **subg. n.** corresponds to the *numidicus* species group sensu Kataev [8].

### *Artabas* Subgroup

Diagnosis. Head impunctate and glabrous or punctate and pubescent dorsally. Pronotum more or less punctate at least basally, with one or several lateral setae on each side and with setose or glabrous basal edge. Elytra either punctate and pubescent or impunctate and glabrous, generally with glabrous, very rarely setose, basal border; interval 3 gener-

ally with one discal setigerous pores, more rarely intervals 3, 5 and 7 also with several setigerous pores in middle of intervals. Prosternum generally with more or less long setae. Metatrochanteres with or without additional setae along posterior margin. Profemur of male without a dense row of setae along inner anterior margin. Protibia with one or three ventroapical spines in transverse row and with three to six preapical spines on outer margin, not forming a single row with spines on ventral surface of tibia; ventroapical tubercle in male more or less prominent. Male mesotibia without a preapical callous thickening on inner margin. Abdominal sternites generally with more or less numerous additional setae; last visible sternite with pronounced sexual dimorphism: its apex rounded, truncate or concave in male and swollen and expanded posteriorly in female. Median lobe with somewhat short terminal lamella (at most slightly longer than wide) and a horseshoe-shaped apical capitulum; internal sac generally without sclerotic elements.

Composition and distribution. This subgroup comprises two western Palaearctic subgenera.

Ecology. The species of this subgroup occur on saline soils.

Remarks. In combination of distinctive characters, including male genitalia, this subgroup is most similar and apparently most closely related to the *Harpalus* subgroup, distinguishing from the latter by the presence of one or three ventroapical spines on the protibia, generally longer setae on the prosternum and the absence of a preapical callous thickening on the inner margin of the male mesotibia.

Subgenus *Artabas* Gozis, 1882

*Artabas* Gozis, 1882 [138] (p. 287) (as a genus). Type species *Harpalus punctatostriatus* Dejean, 1829, by monotypy.

Diagnosis. Medium-sized to somewhat large (length 5.2–12.5 mm). Body moderately convex, stout or moderately elongate, brown to black, with or without metallic luster on dorsum. Pronotum generally with several, more rarely one, lateral setae on each side (Figure 5e,f). Elytra either punctate and pubescent or impunctate and glabrous; interval 3 generally with one discal setigerous pores adjacent to stria 2; intervals 3, 5 and 7 in some species also with several setigerous pores apically or along entire length in middle of intervals. Prosternum with comparatively long or moderately long setae. Protibia with one ventroapical spine.

Composition and distribution. This subgenus comprises 14 western Palaearctic species, distributed in the Thetyan region, mainly in the Mediterranean: *H. dispar* Dejean, 1829 (Figures 69a and 70a,b) [with the subspecies *H. d. splendens* (Gebler, 1830) and *H. d. elegantulus* Ménétriés, 1832], *H. kadleci* Kataev et Wrase, 1995, *H. szalliesi* Kataev et Wrase, 1995, *H. punctatostriatus* Dejean, 1829 (Figure 11e,f), *H. suturangulus* Reitter, 1887, *H. basanicus* J. Sahlberg, 1913, *H. wadiensis* Jedlička, 1964, *H. rumelicus* Apfelbeck, 1904, *H. petri* Tschitschérine, 1902, *H. kabakianus* Kataev, 1988, *H. vereschaginae* Kataev, 1988, and the taxonomically more separated *H. pygmaeus* Dejean, 1829 (Figure 70c,d), *H. microthorax* (Motschulsky, 1849) and *H. siculus* Dejean, 1829.

Ecology. Almost all species of this subgenus are pronounced halophiles occurring along the shores of salt lakes or on salt marshes.

Remarks. Although *Artabas* was originally erected for species with several lateral setae on pronotum, it should also include some species with only one such seta (*H. kabakianus*, *H. vereschaginae*, *H. pygmaeus*, *H. microthorax* and *H. siculus*), since they all are very similar to each other in all other characters, including male genitalia. This is true at least for *H. kabakianus* and *H. vereschaginae* [87]; the taxonomic position of *H. pygmaeus*, *H. microthorax* and *H. siculus* needs further study, but now I also prefer to include them in *Artabas*. All species of this subgenus are characterized by one ventroapical spine on the protibia.

The subgenus *Artabas* corresponds to the *dispar* species group sensu Kryzhanovskij et al. [56] and sensu Kataev [8]. The taxonomy of some species of this subgenus was discussed by Kataev and Wrase [115].

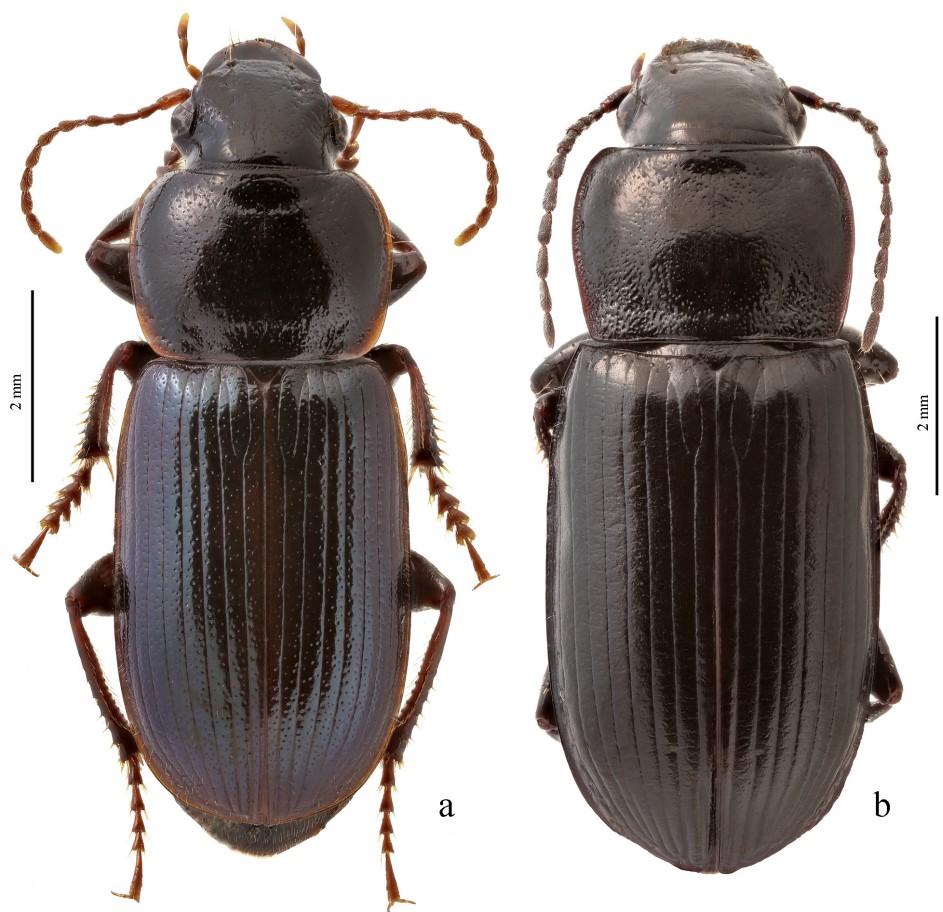

**Figure 69.** Habitus of *Harpalus* species: (**a**) *H. (Artabas) dispar*; (**b**) *H. (Smirnovia) kandaharensis*.

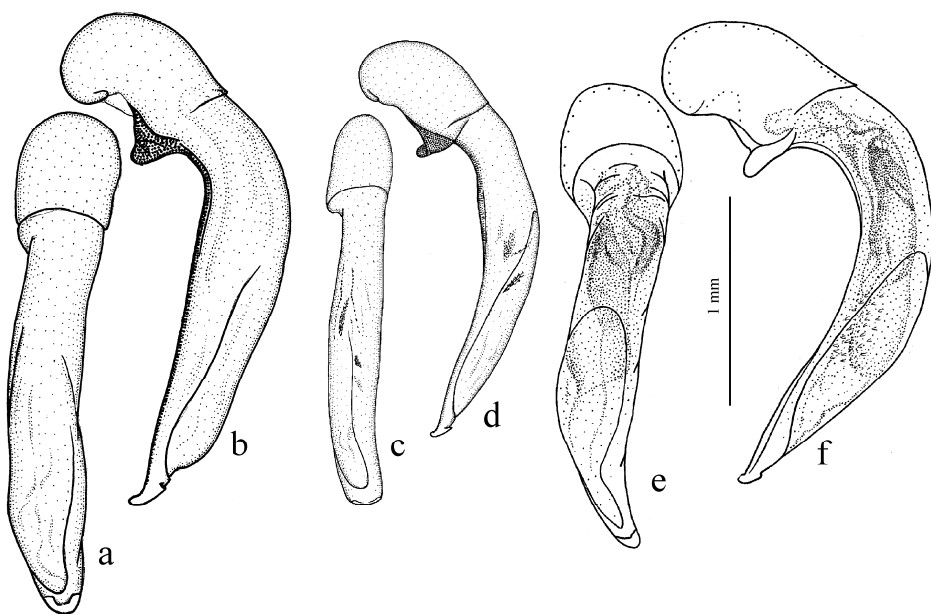

**Figure 70.** Median lobe of aedeagus of *Harpalus* species, dorsal and lateral views: (**a**,**b**) *H. (Artabas) dispar*; (**c**,**d**) *H. (A.) pygmaeus*; (**e**,**f**) *H. (Smirnovia) kandaharensis* (from [85]).

Subgenus *Smirnovia* Lutshnik, 1922

　　　*Smirnovia* Lutshnik, 1922 [21] (p. 62) (as a genus). Type species *Smirnovia tristis* Lutshnik, 1922 (=*H. kandaharensis* Jedlička, 1955), by original designation.

Diagnosis. Medium-sized (length 8.1–10.2 mm). Body moderately convex, somewhat elongate, black, without metallic luster. Pronotum with one lateral seta on each side. Elytra impunctate and glabrous, with one discal setigerous pore on interval 3 adjacent to stria 2. Prosternum with comparatively short setae. Protibia with three ventroapical spines in a transverse row. Median lobe of aedeagus strongly arcuate (Figure 70e,f).

Composition and distribution. This subgenus includes only *H. kandaharensis* Jedlička, 1955 (Figures 69b and 70e,f), known from the deserts of southern Turan.

Ecology. The only species of this subgenus inhabits desert salt marshes.

Remarks. This subgenus is distinguished from *Artabas* by having three ventroapical spines on the protibia and shorter setae on the prosternum [85].

The subgenus *Smirnovia* corresponds to the *lutshnikianus* species group sensu Kryzhanovskij et al. [56] and to the *kandaharensis* species group sensu Kataev [8].

*Harpalus* Subgroup

Diagnosis. Head impunctate and glabrous or punctate and pubescent dorsally. Pronotum punctate or impunctate basally, with one lateral seta on each side and with setose basal edge. Elytra either punctate and pubescent or impunctate and glabrous, generally with glabrous basal border; interval 3 with one discal setigerous pore (rarely pore absent). Metatrochanteres with or without additional setae along posterior margin. Protibia with two ventroapical spines and with four to seven preapical spines on outer margin, not forming a single row with spines on ventral surface of tibia; ventroapical tubercle in male prominent (Figure 15b). Male mesotibia generally with a preapical callous thickening on inner margin (Figure 15d). Tarsi glabrous or setose dorsally. Abdominal sternites generally with more or less numerous additional setae; last visible sternite with pronounced sexual dimorphism: its apex subtruncate or more or less deeply concave in male (Figure 15e,g–i) and swollen and expanded posteriorly in female (Figure 15f). Median lobe with somewhat short terminal lamella and a horseshoe-shaped apical capitulum; internal sac generally without large spines, in most species with small spiny patches and/or groups of small spines, often without any sclerotic elements.

Composition and distribution. This subgroup comprises one Holarctic and five Palaearctic subgenera.

Ecology. Representatives of this subgroup live in open landscapes, with many species within the steppe or desert zone, but they are usually found there in relatively humid places, less often in rather dry habitats. Some species are halophilous.

Remarks. This subgroup is well defined by the characters listed in the diagnosis and certainly represents a monophyletic unit. Its main synapomorphies are: the pronounced dimorphism of the last abdominal sternite, protibia and mesotibia, the setose basal edge of pronotum and two ventroapical spines on the protibia [86].

Subgenus *Pachyharpalus* **subg. n.**

https://zoobank.org/urn:lsid:zoobank.org:act:3245EF50-0E9A-47BC-A64B-59D34F951597
Type species *Harpalus crates* Bates, 1883.

Diagnosis. Large (length 11.5–15.8 mm). Body moderately convex, stout, brown to black, usually with metallic luster on dorsum, glabrous dorsally. Pronotum with rounded sides and widely rounded basal angles. Elytra impunctate, with glabrous basal border and with small humeral denticle; preapical sinuation indistinct; sutural angle of female slightly blunted. Metacoxa without additional setigerous pores medially. Metafemur with five to eight setigerous pores along posterior margin. Metatrochanteres without additional setae along posterior margin. Protibia with six or seven preapical spines on outer margin. Tarsi glabrous dorsally. Male mesotibia occasionally with very small, almost indistinct preapical callous thickening on inner margin. Apex of last visible abdominal sternite of male subtruncate. Median lobe of aedeagus with a group of small spines in internal sac medially (Figure 72a,b).

Etymology. The subgeneric name is a combination of the Greek *pakhús*, meaning "thick", and the name of the carabid taxon *Harpalus*.

Composition and distribution. This subgenus only includes the eastern Palaearctic *H. crates* Bates, 1883 (Figures 71a and 72a,b), distributed in eastern Asia.

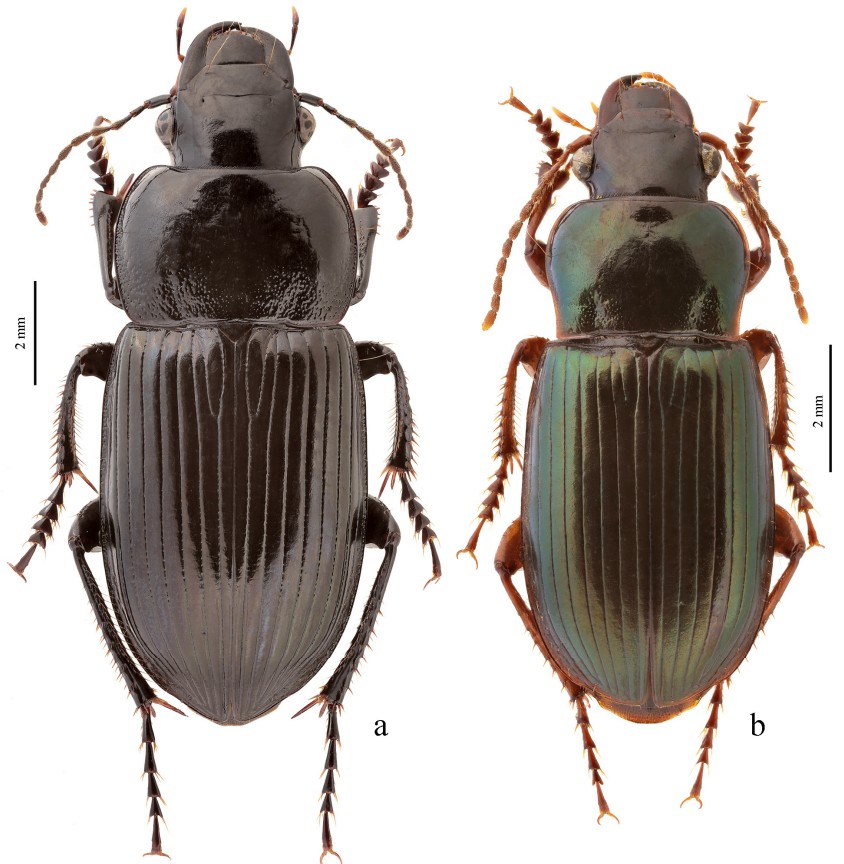

**Figure 71.** Habitus of *Harpalus* species: (**a**) *H.* (*Pachyharpalus*) *crates*; (**b**) *H.* (*Proteonus*) *angularis*.

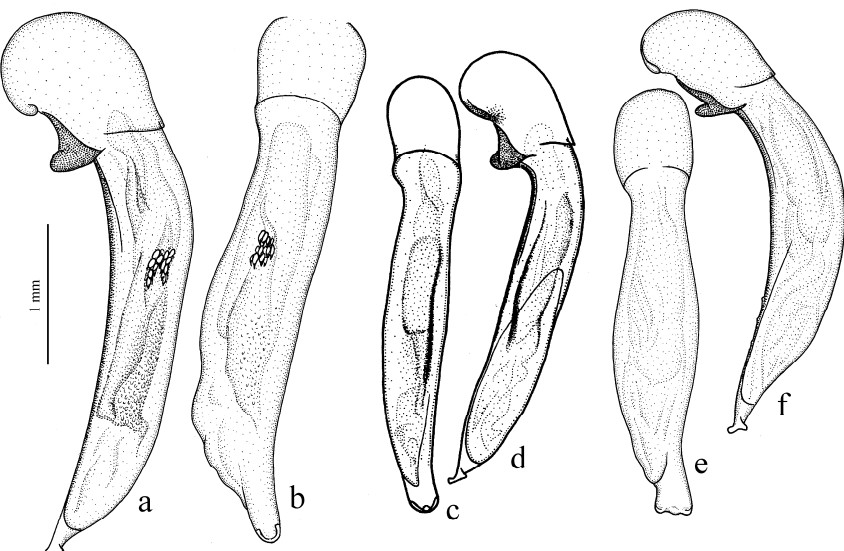

**Figure 72.** Median lobe of aedeagus of *Harpalus* species, dorsal and lateral views: (**a**,**b**) *H.* (*Pachyharpalus*) *crates*; (**c**,**d**) *H.* (*Proteonus*) *distinguendus* (from [54]); (**e**,**f**) *H.* (*P.*) *angulatus*.

Ecology. The only species of this subgenus occurs in open, moderately dry habitats, mainly in meadows but also in various disturbed anthropogenic habitats.

Remarks. The single species of this subgenus is similar to species of the subgenus *Harpalus* s. str. with impunctate and glabrous elytra but differs from them mainly in having protibia with a larger number (six or seven) of preapical spines on the outer margin of the protibia, elytra with a humeral denticle, an indistinct preapical sinuation and a blunted (in female) sutural angle. In the latter character and in metacoxa without additional setigerous pores medially, *Pachyharpalus* **subg. n.** also differs from all other subgenera of the *Harpalus* subgroup.

The subgenus *Pachyharpalus* **subg. n.** corresponds to the *crates* species group sensu Kataev [8,86] and sensu Kryzhanovskij et al. [56].

Subgenus *Proteonus* Fischer von Waldheim, 1829

> *Proteonus* Fischer von Waldheim, 1829 [134] (p. 21) (as a genus). Type species *Carabus distinguendus* Duftschmid, 1812, designated by Bousquet [135].
> *Lasioharpalus* Reitter, 1900 [38] (pp. 75, 86) (as a subgenus of *Harpalus* Latreille, 1802). Type species *Harpalus subangulatus* Reitter, 1900, designated by Antoine [40].
> *Lasiarpalus*: Jakobson, 1907 [63] (p. 378) (print error).

Diagnosis. Medium-sized (length 7.5–10.8 mm). Body moderately convex, somewhat elongate, brown to black, often with metallic luster on dorsum, glabrous dorsally. Pronotum with distinct basal angles (sharp to narrowly rounded at apex) and sides usually straight or sinuate basally. Elytra impunctate, with glabrous basal margin; humeri more or less rounded, with a small denticle, or angulate, without denticle; preapical sinuation very shallow or indistinct; sutural angle of female generally acute and prominent posteriorly. Metacoxa with additional setigerous pores medially. Metafemur with five to seven setigerous pores along posterior margin. Metatrochanteres without additional setae along posterior margin. Protibia with four to five preapical spines on outer margin. Male mesotibia often with almost indistinct preapical callous thickening on inner margin. Tarsi glabrous dorsally. Apex of last visible abdominal sternite of male generally not markedly concave (at most very shallowly emarginate). Internal sac of aedeagus without sclerotic elements or with small spiny patches or a group of few small spines.

Composition and distribution. This subgenus comprises four western Palaearctic species, distributed mainly in the western part of the region: *H. distinguendus* (Duftschmid, 1812) (Figure 72c,d) (with the subspecies *H. kidanicus* Kataev, 1989), *H. contemptus* Dejean, 1829, *H. saxicola* Dejean, 1829 and *H. angulatus* Putzeys, 1878 (Figures 71b and 72e,f) (with the subspecies *H. a. subangulatus* Reitter, 1900 and *H. a. scytha* Tschitschérine, 1899).

Ecology. Species of this subgenus occur in meadows and in steppe habitats, *H. distinguendus* is often also in destroyed antropogenic biotopes and agricultural fields.

Remarks. The species of this subgenus are very similar in their distinctive characters to those of *Harpalus* s. str. with impunctate and glabrous elytra and of *Paraharpalus* **subg. n.** but can be distinguished from them by the elytral humer either angulate, without denticle, or rounded, with a denticle at the tip; in addition, the pronotum is generally with more distinct basal angles and usually with straight or sinuate sides basally; *Proteonus* also differs from *Harpalus* s. str. in having the last abdominal sternite of male not markedly concave at the apex.

The subgenus *Proteonus* corresponds to the *distinguendus* species group sensu Kataev [8,86] and sensu Kryzhanovskij et al. [56]. Some species of this subgenus were revised by Puel [178] and Mlynář [76]. The *distinguendus* species group sensu Puel [178] also includes the species of *Paraharpalus* **subg. n.**

Subgenus *Paraharpalus* **subg. n.**

> https://zoobank.org/urn:lsid:zoobank.org:act:7D9C8D41-DDF9-4181-8AB7-30AAC578302C
> Type species *Harpalus oblitus* Dejean, 1829.

Diagnosis. Medium-sized to large (length 7.1–12.4 mm). Body moderately convex, moderately robust to somewhat elongate, brown to black, often with metallic luster on dorsum, glabrous dorsally. Pronotum generally with rounded sides and widely rounded basal angles; more rarely basal angles more or less angulate, blunted at apex. Elytra impunctate, with glabrous basal border and generally with rounded humeral angle lacking denticle; preapical sinuation rather deep, often with a denticle at its base; sutural angle of female generally acute and prominent posteriorly. Metacoxa with additional setigerous pores medially. Metafemur with five to seven setigerous pores along posterior margin. Metatrochanteres without additional setae along posterior margin. Protibia with five to seven preapical spines on outer margin. Male mesotibia often with almost indistinct preapical callous thickening on inner margin. Tarsi glabrous dorsally. Apex of last visible abdominal sternite of male not concave (at most slightly truncate). Internal sac of aedeagus with characteristic armament consisting of one or two groups of small spines medially.

Etymology. The subgeneric name is a combination of the Greek *pará*, meaning "near", and the name of the carabid taxon *Harpalus*.

Composition and distribution. This subgenus comprises six western Palaearctic species, distributed in the Tethyan region, mainly in the Mediterranean: *H. oblitus* Dejean, 1829 (Figure 74a,b) (with the subspecies *H. o. patruelis* Dejean, 1829), *H. akinini* Tschitschérine, 1895, *H. lateralis* Dejean, 1829, *H. smyrnensis* Heyden, 1888 (Figure 73a) (with the subspecies *H. s. raddei* Tschitschérine, 1897 and *H. s. medicus* Kataev, 1993), *H. quadratus* Chaudoir, 1846 and *H. subtruncatus* Chaudoir, 1846.

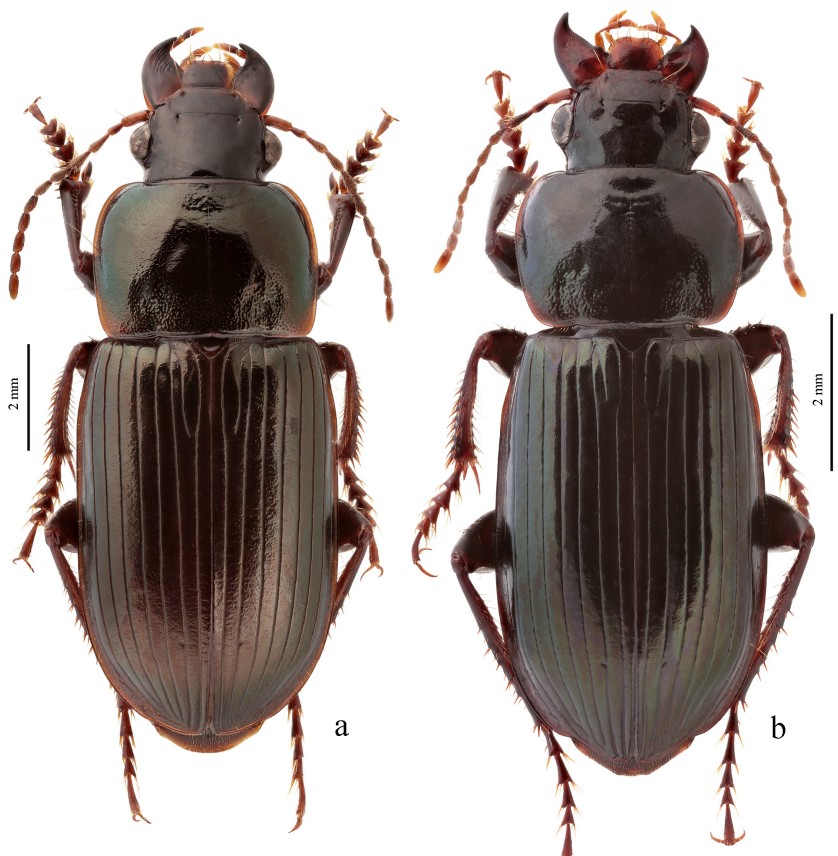

**Figure 73.** Habitus of *Harpalus* species: (**a**) *H. (Paraharpalus) smyrnensis*; (**b**) *H. (Harpalus) kabakovi*.

Ecology. The species of this subgenus inhabit arid landscapes, steppes and semi-deserts, but they usually occur in humid places; some species are found on saline soils.

Remarks. The species of this subgenus are very similar in their distinctive characters to those of *Harpalus* s. str. with impunctate and glabrous elytra differing from them in having

the last abdominal sternite of male not concave and the aedeagus with a characteristic armament consisting of one or two groups of small spines.

The subgenus *Paraharpalus* **subg. n.** corresponds to the *oblitus* species group sensu Kataev [8,86,92] and sensu Kryzhanovskij et al. [56]. A revision of the species of this group was published by Kataev [92]. Some species were previously revised by Puel [178].

Subgenus *Harpalus* Latreille, 1802

*Harpalus* Latreille, 1802 [10] (p. 92) (as a genus). Type species *Carabus proteus* Paykull, 1790 (=*C. affinis* Schrank, 1781), designated by Andrewes [12], tentatively accepted as type species.

*Harpaleus* Billberg, 1820 [121] (p. 23) (unjustified emendation).

*Bioderus* Motschulsky, 1848 [59] (p. 487) (as a genus). Type species *Microderus petreus* Motschulsky, 1844 (=*Anisodactylus obtusus*, Gebler, 1833), by monotypy.

*Harpalomerus* Casey, 1914 [71] (p. 76) (as a subgenus of *Harpalus* Latreille, 1802). Type species *Harpalus amputatus* Say, 1830, designated by Lindroth [15].

Diagnosis. Medium-sized to large (length 7.0–13.0 mm). Body moderately convex, somewhat elongate, brown to black, often with metallic luster on dorsum, glabrous dorsally or pubescent on pronotum and elytra. Pronotum with more or less widely rounded sides and basal angles. Elytra either impunctate or punctate throughout or laterally; basal border glabrous (very rarely setose); humeral angle rounded, generally lacking denticle; preapical sinuation rather deep, often with a denticle at its base, rarely very shallow; sutural angle of female generally acute and prominent posteriorly. Metacoxa with additional setigerous pores medially, rarely without them. Metafemur with five to eight setigerous pores along posterior margin. Metatrochanteres without additional setae along posterior margin. Protibia with four to five preapical spines on outer margin. Tarsi generally glabrous or very sparsely setose dorsally. Apex of last visible abdominal sternite of male distinctly concave (Figure 15g–i). Internal sac of aedeagus with groups of small spines and/or distinct spiny patches medially, in some species also with a separate spine.

Composition and distribution. This subgenus has a Holarctic distribution; it includes one Holarctic and 13 Palaearctic species, with two centers of diversity—in the western Mediterranean and in the eastern part of the Tethyan region; among them, one species, *H. affinis* (Schrank, 1781) (Figure 74c,d), has a trans-Palaearctic distribution.

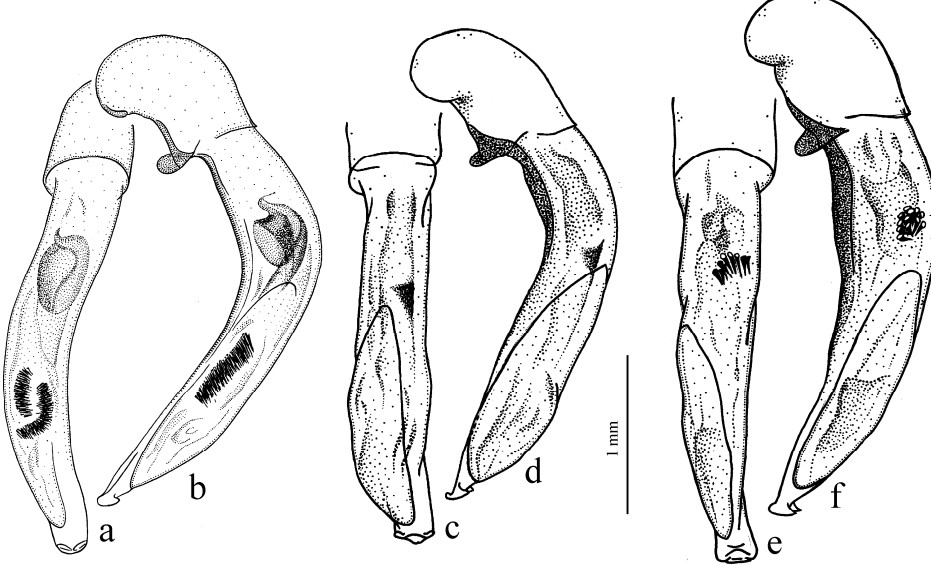

**Figure 74.** Median lobe of aedeagus of *Harpalus* species, dorsal and lateral views: (**a**,**b**) *H.* (*Paraharpalus*) *oblitus* (from [92]); (**c**,**d**) *H.* (*Harpalus*) *affinis* (from [86]); (**e**,**f**) *H.* (*H.*) *ganssuensis* (from [86]).

The fauna of the west Mediterranean region includes two endemic species: *H. semipunctatus* Dejean, 1829 and *H. lethierryi* Reiche, 1859 (with the subspecies *H. l. azrouanus* Emden et Schauberger, 1932).

The fauna of the eastern part of the Tethyan region comprises eleven species, ten (except for *H. affinis*) Palaearctic: *H. tjanschanicus* Semenov, 1889, *H. glasunovi* Kataev, 1987 (with the subspecies *H. g. spinulifer* Kataev, 2006 and *H. g. opaculus* Kataev, 2006), *H. anisodactyliformis* Solsky, 1874, *H. bucharicus* Tschitschérine, 1898, *H. michailovi* Kataev, 1987, *H. kabakovi* Kataev, 1987 (Figure 73b), *H. caeruleatus* Bates, 1878, *H. uniformis* Motschulsky, 1844 (with the subspecies *H. u. staudingeri* Jedlička, 1953), *H. ganssuensis* Semenov, 1889 (Figure 74e,f) and *H. erosus* Mannerheim, 1825, and one Holarctic species: *H. amputatus* Say, 1830 [with the Palaearctic subspecies *H. a. amputatoides* Mlynář, 1979, *H. a. obtusus* (Gebler, 1833), and *H. a. inschanicus* Breit, 1914]; the nominotypical subspecies of the latter species is widely distributed in North America.

Ecology. The species of this subgenus, like the previous one, usually inhabit arid and semi-arid landscapes, steppes and semi-deserts, both on the plain and in the mountains, but they occur usually in humid places; some species occur on saline soils.

Remarks. The most important features of this subgenus, distinguishing it from other representatives of this subgroup, are as follows: the basal pronotal angles rounded, the elytra with a rounded humeral angle without denticle and with a deep preapical sinuation, the tarsi glabrous dorsally and the apex of the last abdominal sternite of male distinctly concave.

The nominotypical subgenus, as it is treated here, corresponds to the *affinis* species group sensu Kataev [8,86] and sensu Kryzhanovskij et al. [56]. A revision of this group was published by Kataev [86], with a later addition [104]. Lindroth [15] and Noonan [16] treated the single Nearctic species, *H. amputatus*, as a member of the monobasic *amputatus* species group corresponding to *Harpalomerus*.

Subgenus *Harpalophonus* Ganglbauer, 1892

*Harpalophonus* Ganglbauer, 1892: 341, 346 [36] (as a subgenus of *Ophonus* Dejean, 1821). Type species *Harpalus hospes* Sturm, 1818, by monotypy.

Diagnosis. Medium-sized to large (length 8.3–14.0 mm). Body moderately convex, somewhat elongate, brown to black, often with metallic luster on dorsum. Head impunctate and glabrous dorsally. Pronotum with rounded sides and widely rounded basal angles. Elytra punctate and pubescent throughout or laterally, with setose basal border and rounded humeral angle without denticle; preapical sinuation more or less deep, often with a denticle at its base; sutural angle of female generally acute and prominent posteriorly. Metacoxa with additional setigerous pores medially. Metafemur with five to seven setigerous pores along posterior margin. Metatrochanteres without additional setae along posterior margin. Protibia with four to five preapical spines on outer margin. Tarsi densely setose dorsally. Apex of last visible abdominal sternite of male distinctly concave. Internal sac of aedeagus with one or two groups of small spines medially, in some species also with a spiny patch.

Composition and distribution. This subgenus comprises seven western Palaearctic species, distributed in the Tethyan region: *H. hospes* Sturm, 1818 (Figure 76a,b) [with the subspecies *H. h. ciscaucasicus* Lutshnik, 1921 and *H. h. armenus* (Daniel, 1904)], *H. kagyzmanicus* Kataev, 1984, *H. mairei* Peyerimhoff, 1928, *H. italus* Schaum, 1860, *H. circumpunctatus* Chaudoir, 1846, *H. terrestris* (Motschulsky, 1844) and *H. stevenii* Dejean, 1829 (Figure 75a).

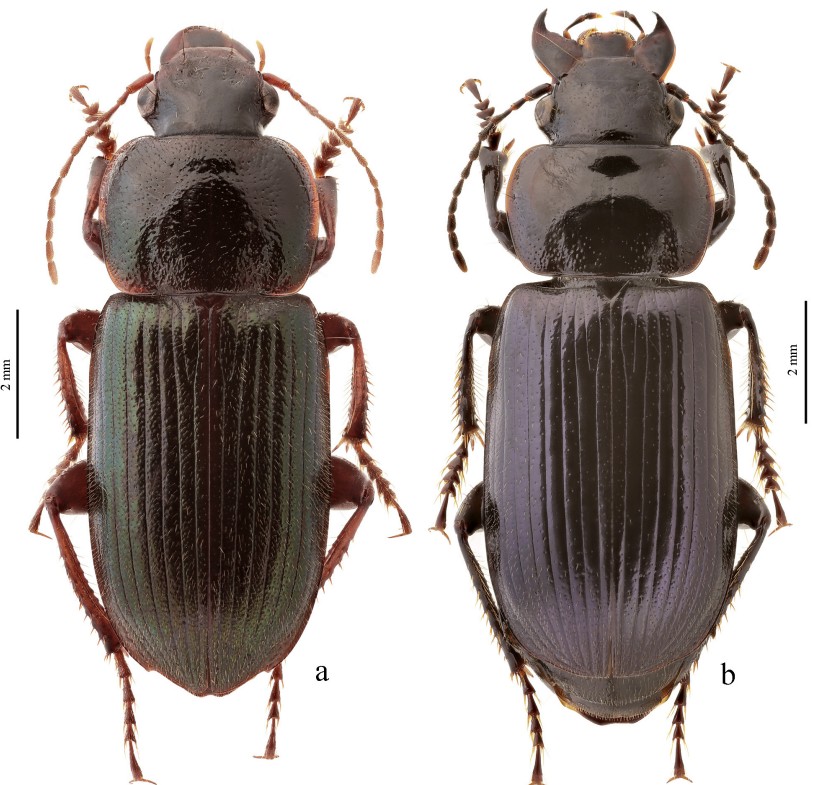

**Figure 75.** Habitus of *Harpalus* species: (**a**) *H.* (*Harpalophonus*) *stevenii*; (**b**) *H.* (*Cephalotypsis*) *kryzhanovskii*.

Ecology. Species of this subgenus inhabit arid and semi-arid territories, often found near water bodies, on more or less saline soils.

Remarks. This taxon was previously either included in the genus *Ophonus* ([14,36,37,40], etc.) or considered as a subgenus or even a synonym of *Pseudoophonus* [6,28–30,38]. My research [83,86] fully confirmed the opinion of Mlynář [53,75] that *Harpalophonus* is very similar and undoubtedly very closely related to the subgenus *Harpalus* s. str. since the only character distinguishing it from the latter subgenus is densely punctate and pubescent tarsomeres [86].

The subgenus *Harpalophonus* corresponds to the *hospes* species group sensu Kryzhanovskij et al. [56] and sensu Kataev [8]. The revision of the subgenus was published by Kataev [83], with a subsequent addition [110]. *Hapalus altaicus* Jedlička, 1968, which was regarded as a distinct species of *Harpalophonus* in Kataev [83], was later treated as a synonym of the nominotypical subspecies of *H. uniformis* (Motschulsky, 1844) of the subgenus *Harpalus* s. str. [86, 100].

Subgenus *Cephalotypsis* Tschitschérine, 1901

*Cephalotypsis* Tschitschérine, 1901 [20] (p. 238) (as a subgenus of *Harpalus* Latreille, 1802). Type species *Harpalus semenowi* Tschitschérine, 1901, by monotypy.

Diagnosis. Large (length 11.2–14.5 mm). Body moderately convex, elongate, brown to dark brown, with metallic luster on dorsum. Head, pronotum and elytra punctate and pubescent dorsally. Pronotum with rounded sides and widely rounded basal angles. Elytra with setose basal border and rounded humeral angle lacking denticle; preapical sinuation absent; sutural angle of female generally acute and prominent posteriorly. Metacoxa with additional setigerous pores medially. Metafemur with numerous (more than ten) setigerous pores along posterior margin. Metatrochanteres with several additional setae along posterior margin. Protibia with five to six preapical spines on outer margin. Tarsi glabrous dorsally. Apex of last visible abdominal sternite of male distinctly concave.

Internal sac with two groups of medium-sized spines, occasionally also with a small separate spine.

Composition and distribution. This subgenus includes two species from the Tethyan region of Asia: *H. semenowi* Tschitschérine, 1901 and *H. kryzhanovskii* Kataev, 1988 (Figures 75b and 76c,d).

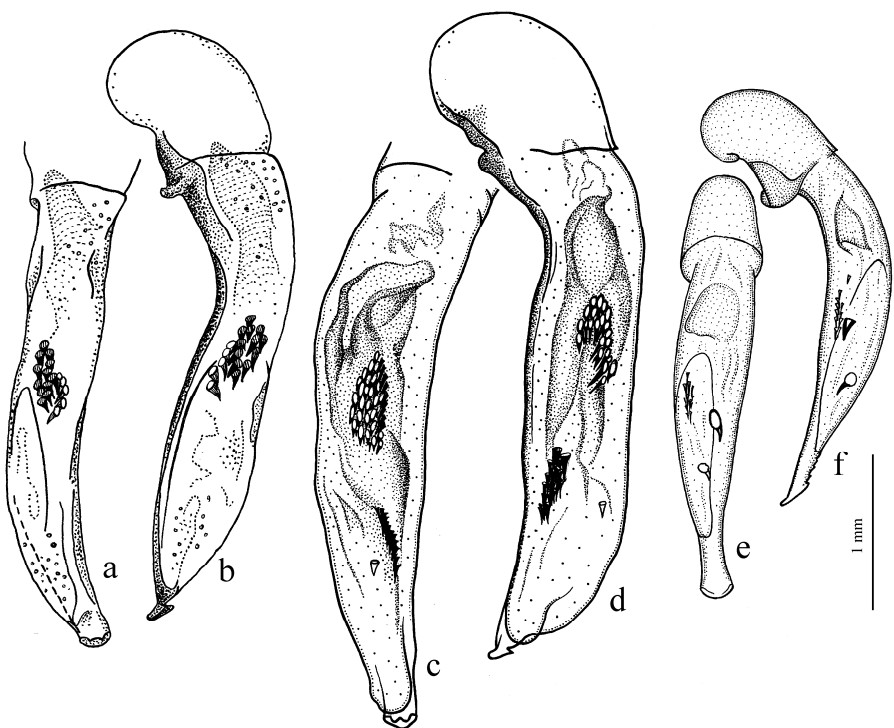

**Figure 76.** Median lobe of aedeagus of *Harpalus* species, dorsal and lateral views: (**a**,**b**) *H.* (*Harpalophonus*) *hospes* (from [83]); (**c**,**d**) *H.* (*Cephalotypsis*) *kryzhanovskii* (from [87]); (**e**,**f**) *H.* (*Opadius*) *cordatus* (from [106]).

Ecology. Both species of this subgenus inhabit deserts.

Remarks. This subgenus is similar to the subgenera *Harpalus* s. str. and *Harpalophonus* in many characters including the general habitus, elytra punctate and pubescent, humeral angle rounded, without denticle and last abdominal sternite of male deeply concave. *Cephalotypsis* distinctly differs from them in having head punctate and pubescent, elytra without preapical sinuation, metafemur with numerous (more than ten) setigerous pores along posterior margin and metatrochanter with several additional setae along posterior margin. In addition, *Cephalotypsis* differs from *Harpalophonus* in tarsi glabrous dorsally.

The subgenus *Cephalotypsis* corresponds to the *semenowi* species group sensu Kataev [8,86] and sensu Kryzhanovskij et al. [56].

### 3.3.10. *Glanodes* Group

Description. Body glabrous on dorsal side. Head dorsally, at least on frons and vertex, more or less distinctly punctate; tempora not setose. Mentum without median tooth, separated from submentum by complete transverse suture; ligular sclerite in addition to two long ventroapical setae also with several short dorsal setae; labial basal palpomere more or less cylindrical, without oblique carina on ventral side (as in Figure 3p). Antennae pubescent from antennomere 3. Pronotum with one lateral seta on each side, bordered along basal margin. Elytra impunctate, with one discal setigerous pore on interval 3 and with a short row of preapical pores at least on intervals 7 and 5 (sometimes also on interval 3); elytral preapical sinuation absent or shallow. Metepisternum short, as wide as or wider than long. Metacoxa without posteromedial setigerous pore. Protibia generally with four to six (sometimes three) ventroapical spines arranged in a transverse row along apical margin

of tibia; its outer distal margin with at least three spines forming a single row with spines on ventral side of tibia (as in Figure 10a). Tarsi glabrous on dorsal side; mesotarsomere 1 of male ventrally either without adhesive vestiture or with a pair of small scales at apex. Two penultimate abdominal sternites with numerous long setae in addition to two obligatory fixed setae; last visible sternite and tergite without pronounced sexual dimorphism. Median lobe of aedeagus with an oblique or transverse apical capitulum (in *H. cordatus* capitulum only suggested); internal sac with two separate spines and usually one or several groups of small spines.

Composition and distribution. This group comprises two Nearctic subgenera.

Remarks. The species of this subgeneric group are easily distinguished from other members of the genus *Harpalus* by the combination of the following characters: body and tarsi glabrous on dorsal side, head dorsally more or less distinctly punctate, protibia with four to six (sometimes three) ventroapical spines arranged in a transverse row, elytra with a short row of preapical pores on intervals 5 and 7 and two penultimate abdominal sternites with numerous long setae.

Some of features of the two included subgenera (punctation on the head, structure of the protibia and aedeagus) are similar to those of some genera of *Harpali*, such as the Palaearctic *Microderes* Faldermann, 1935 and *Neophygas* Noonan, 1976, and perhaps further research will show that *Glanodes* warrant generic status.

This group corresponds to the subgenus *Glanodes* sensu Kataev [8,106].

Subgenus *Opadius* Casey, 1914

*Opadius* Casey, 1914 [71] (p. 66) (as a genus). Type species *Cratognathus cordatus* LeConte, 1853, by original designation.

Diagnosis. Body length 8.4–9.2 mm. Pronotal sides sinuate in basal half; basal angles more or less right, with sharp apex. Outer distal margin of protibia with three to five spines.

Composition and distribution. This subgenus includes two wingless species from the southwestern United States: *H. cordatus* (LeConte, 1853) (Figures 76e,f and 77a) and *H. apache* Kataev, 2010.

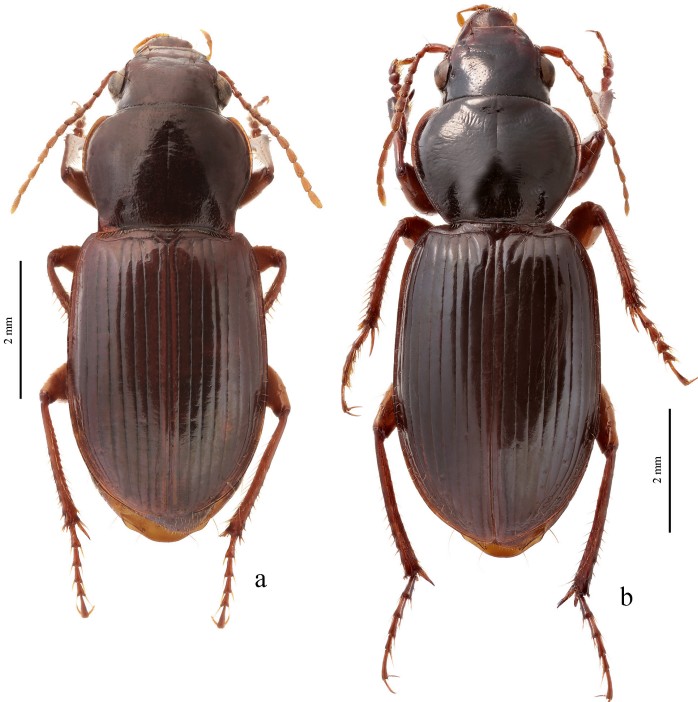

**Figure 77.** Habitus of *Harpalus* species: (**a**) *H.* (*Opadius*) *cordatus*; (**b**) *H.* (*Glanodes*) *obliquus*.

Ecology. Species of this subgenus inhabit uplands, occurring in grasslands, pastures, roadsides, the borders of ponds and streams, on dry soil composed of sandy silt or loam (sometimes gravel-mixed) and areas covered with sparse vegetation [140].

Remarks. Lindroth [15] and Ball [80] considered *Opadius* as a subgenus of *Harpalus*, very similar to the subgenus *Glanodes*. Noonan [16], without mentioning of the name of *Opadius* in his revision, united *H tadorcus* (=*H. cordatus*) and *H. cordifer* into one species group *cordifer* (=*Cordoharpalus*), which was considered by him separately from the species of the subgenus *Glanodes*, based mainly on the similar shape of the pronotum in both these species. This treatment was accepted by Ball and Bousquet [7] and Bousquet [82], who also regarded the names *Opadius* and *Cordoharpalus* as synonyms. In my opinion, *Opadius* and *Cordoharpalus* are two unrelated taxa [106].

The subgenus *Opadius* corresponds to the *tadorcus* species group sensu Kataev [8,106].

Subgenus *Glanodes* Casey, 1914

*Glanodes* Casey, 1914 [71] (p. 60) (as a genus). Type species *Harpalus obliquus* Horn, 1880, by original designation.

Diagnosis. Body length 9.0–10.5 mm. Pronotal sides in basal half roundly or nearly rectilinearly converging posteriorly, not sinuate; basal angles obtuse, often more or less widely rounded at apex. Outer distal margin of protibia with four to six spines.

Composition and distribution. This subgenus comprises six wingless species from the southwestern United States: *H. corpulentus* (Casey, 1914), *H. puncticeps* (Casey, 1914), *H. huachuca* Ball, 1972, *H. stephani* Ball, 1972, *H. obliquus* Horn, 1880 (Figure 77b) and *H. cohni* Ball, 1972.

Ecology. Species of this subgenus inhabit uplands, occurring in grasslands, in open dry or damp habitats, on clay and on muddy or sandy soils [140].

Remarks. The subgenus *Glanodes* was revised by Ball [80]. This subgenus corresponds to the *obliquus* species group sensu Kataev [8,106].

*3.4. Uncertain Nominal Taxa of Species Group*

The following nominal taxa, described within the genus *Harpalus*, are known to me only from the original descriptions, and their interpretation is impossible without an examination of the type specimens:

*Harpalus atripes* Casey, 1914 (Mexico); *H. agonoderus* Putzeys, 1878 ("Gouv. Baku", Azerbaijan); *H. bicolor* Fischer von Waldheim, 1829 (non Fabricius, 1775; non Marsham, 1802) ("Caucasus"); *H. tridentatus* Fischer von Waldheim, 1821 ("Orenburg—Buchara"); *H. brunnipes* Fischer von Waldheim, 1829 [non Dejean, 1829] ("Caucasus"); *H. erythrocerus* Fischer von Waldheim, 1829 ("Tauria, Caucasus"); *H. foliarius* Gistlel, 1857 ("Turcia europaea"); *H. frontalis* Fischer von Waldheim, 1829 ("Caucasus"); *H. interruptus* Fischer von Waldheim, 1829 ("Caucasus"); *H. lutarius* Gistlel, 1857 ("Lombardiae Alpina"); *H. minorita* Gistlel, 1857 (Italy: "Aosta"); *H. nigripalpis* Stephens, 1835 ("near Windsor and near London"); *H. opacus* Gravenhorst, 1807 (*Carabus*) (locality not recorded); *H. platypterus* Fischer von Waldheim, 1829 ("Tauria"); *H. polychrous* Gistlel, 1857 ("Tirol"); *H. praedatorius* Gistlel, 1857 ("Helvetia"); *H. pumilus* Gravenhorst, 1807 (*Carabus*) (locality not recorded); *H. punctus* Gravenhorst, 1807 (*Carabus*) (locality not recorded); *H. serbicus* Matits, 1910 ("in der Naehe der Stadt Nisch (Serbien)"; *H. solaris* Gistlel, 1857 ("Helvetia et Lombardia"); *H. typhonius* Gistlel, 1857 ("Jura"); *H. silipes* Dejean, 1831 (Argentina); *H. egenus* Dejean, 1829 (Brazil); *H. aulicus* Dejean, 1829 (Brazil); *H. octopunctatus* Dejean, 1829 (Argentina); *H. pallipes* Chaudoir, 1837 (Brazil); *H. laetus* Reiche, 1843 (Colombia); *H. viridicupreus* Reiche, 1843 (Colombia); *H. turmalinus* Erichson, 1847 (Peru); *H. gory* Gory, 1933 (Australia); *H. goryi* Boisduval, 1835 (Australia) and *H. ignobilis* Boheman, 1858 ["Sydney (N. S. W.)"].

The taxa of this list originally described from South America and Australia almost certainly belong to different genera, not to *Harpalus*. Most of the names of the Palaearctic taxa are probably synonyms of the known species.

*3.5. List of the Valid SupraSpecific Taxa of Harpalus*

Genus *Harpalus* Latreille, 1802
*Cephalophonus* group
  Subgenus *Cephalophonus* Ganglbauer, 1892
*Pseudoophonus* group
  Subgenus *Cephalomorphus* Tschitschérine, 1897
  Subgenus *Pseudoophonus* Motschulsky, 1844
  Subgenus *Platus* Motschulsky, 1844
*Megapangus* group
  Subgenus *Megapangus* Casey, 1914
*Plectralidus* group
  Subgenus *Plectralidus* Casey, 1914
*Loboharpalus* group
  Subgenus *Loboharpalus* Schauberger, 1932
*Semiophonus* group
  Subgenus *Semiophonus* Schauberger, 1933
*Zangoharpalus* group
  Subgenus *Zangoharpalus* Huang, 1998
*Cryptophonus* group
  Subgenus *Cryptophonus* Brandmayr et Zetto Brandmayr, 1982
*Harpalus* group
*Afroharpalus* subgroup
  Subgenus *Afroharpalus* **subg. n.**
*Hyloharpalus* subgroup
  Subgenus *Hyloharpalus* **subg. n.**
  Subgenus *Sinoharpalus* **subg. n.**
  Subgenus *Macroharpalus* **subg. n.**
  Subgenus *Meroharpalus* **subg. n.**
  Subgenus *Ameroharpalus* **subg. n.**
*Amblystus* subgroup
  Subgenus *Drymoharpalus* **subg. n.**
  Subgenus *Epiharpalus* Reitter, 1900
  Subgenus *Caucasoharpalus* **subg. n.**
  Subgenus *Calathoderus* **subg. n.**
  Subgenus *Licinoderus* Sainte-Claire Deville, 1905
  Subgenus *Amblystus* Motchulsky, 1864
*Cordoharpalus* subgroup
  Subgenus *Cordoharpalus* Hatch, 1949
*Actephilus* subgroup
  Subgenus *Actephilus* Stephens, 1833
  Subgenus *Isoharpalus* **subg. n.**
*Acardystus* subgroup
  Subgenus *Euharpalops* Casey, 1924
  Subgenus *Haploharpalus* Schauberger, 1926
  Subgenus *Acardystus* Reitter, 1908
*Psammoharpalus* subgroup
  Subgenus *Psammoharpalus* **subg. n.**
*Ooistus* subgroup
  Subgenus *Ooistus* Motschulsky, 1864
  Subgenus *Platyharpalus* **subg. n.**
*Asioharpalus* subgroup
  Subgenus *Asioharpalus* **subg. n.**
*Anamblystus* subgroup
  Subgenus *Anamblystus* **subg. n.**
  Subgenus *Homaloharpalus* **subg. n.**
*Harpalobius* subgroup
  Subgenus *Bactroharpalus* **subg. n.**
  Subgenus *Diaharpalus* **subg. n.**

Subgenus *Harpalobius* Reitter, 1900
*Pheuginus* subgroup
Subgenus *Nephoharpalus* Huang, Lei, Yan et Hu, 1996
Subgenus *Pheuginus* Motschulsky, 1844
Subgenus *Mesoharpalus* **subg. n.**
Subgenus *Eremoharpalus* **subg. n.**
Subgenus *Oreoharpalus* **subg. n.**
Subgenus *Hypsoharpalus* **subg. n.**
Subgenus *Anophonus* **subg. n.**
Subgenus *Haloharpalus* **subg. n.**
Subgenus *Megaharpalus* **subg. n.**
Subgenus *Aristoharpalus* **subg. n.**
Subgenus *Cycloharpalus* **subg. n.**
Subgenus *Euryharpalus* **subg. n.**
*Pharalus* subgroup
Subgenus *Pharalus* Casey, 1914
*Hypsinephus* subgroup
Subgenus *Hypsinephus* Bates, 1878
Subgenus *Brachyharpalus* **subg. n.**
Subgenus *Brachypangus* Tschitschérine, 1898
Subgenus *Hemipangus* **subg. n.**
Subgenus *Loxophonus* Reitter, 1894
*Mauriharpalus* subgroup
Subgenus *Mauriharpalus* **subg. n.**
*Caloharpalus* subgroup
Subgenus *Caloharpalus* **subg. n.**
Subgenus *Baryharpalus* **subg. n.**
Subgenus *Heteroharpalus* **subg. n.**
*Idioharpalus* subgroup
Subgenus *Idioharpalus* **subg. n.**
*Artabas* subgroup
Subgenus *Artabas* Gozis, 1882
Subgenus *Smirnovia* Lutshnik, 1922
*Harpalus* subgroup
Subgenus *Pachyharpalus* **subg. n.**
Subgenus *Proteonus* Fischer von Waldheim, 1829
Subgenus *Paraharpalus* **subg. n.**
Subgenus *Harpalus* Latreille, 1802
Subgenus *Harpalophonus* Ganglbauer, 1892
Subgenus *Cephalotypsis* Tschitschérine, 1901
*Glanodes* group
Subgenus *Opadius* Casey, 1914
Subgenus *Glanodes* Casey, 1914

**Funding:** This research received no external funding.

**Institutional Review Board Statement:** Not applicable.

**Data Availability Statement:** The data are available from the author on reasonable request.

**Acknowledgments:** I am very grateful to all the staff of the Laboratory of Insect Systematics of the Zoological Institute, Russian Academy of Sciences, Saint Petersburg, Russia, many of whom, unfortunately, are no longer with us, for their many years of comprehensive support and assistance, including help in working with the collections of this institute, which was the main basis of this study. First of all, I fondly remember my teacher Oleg L. Kryzhanovskij (1918–1997), who recommended this group to me as an object of study. I express my special gratitude to my friend and colleague David W. Wrase (Gusow-Platkow, Germany), an excellent specialist on ground beetles and co-author of many of my publications, for many years of cooperation; many problems of Harpalini taxonomy were discussed with him. I am also very appreciative to all my many friends and colleagues, both professional and amateur entomologists, for the opportunity to study material from the collections under their care. Special thanks to museum staff and curators of museum collections for their hospitality and

**Conflicts of Interest:** The author declares no conflict of interest.

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
