# Peer review of "Classification of the Genus Harpalus (Coleoptera, Carabidae) of the World Based on Imaginal Morphology"

_diversity, doi:10.3390/d15090971_

Round 1

Reviewer 1 Report

This is an important and comprehensive study devoted entirely on the supra-speces classification of the huge carabid genus Harplaus. The numerous subgenera (among them 36 new for the science subgenera) arranged into 19 subgroups and 10 groups are treated in details. For each group, subgroup and subgenus are given diagnoses, composition and distribution of the taxa included, short ecological and other notes.

The research design is appropriate. The methods of research used include the well-known (for the taxonomitsts) traditional methods for morphological examination of the imagoes and assestment of the charact states of the important morphological features, which is the main basis for such kind exhaustive taxonomic investigation. The structure of the protibia is bassically used together with some other important characteristics for species diagnosis and for elucidating the relationship between taxa of the genus.

As the author is the top specialist on complex genus Harplaus and is well-recognized to the ground-beetle professional and amateur community, his effort and knowledge is a guarantee for a quality work.

There is some faults and lapses on the manuscript which is not surprasing in view of its size. In order to improve the manuscript, I have made numerous recommndations and comments on an corrected PDF file attached to this message. I recommend the author to carefully examine these recommendations and to estimate if these need acception or not.

Single characters used to delimit few new subgenera (for instance,  Macroharpalus subg. n.) may seem "weak" because of the relative nature of these characteristics in particular.

The illustrations are of good quality and clear, and thus do not need further improvement. Check if the "posteromedial" pore of metacoxa is not actually a "anteromedial" pore in view of its location toward the body.

As a whole, I recommend the publication of this important and original work in the journal Diversity and congratulate the author for his great effort to end it as well as for his contunuously strive to improve the knowledge on the complex genus Harpalus.

Some sentences are too long and sound unclear for the non-English reader. Probably the best choice is to re-write and separate them.

Author Response

I express my deep gratitude to the first reviewer for the great work done and such a thorough review of the article. All the comments made, undoubtedly, contributed to its significant improvement.

Point 1. There is some faults and lapses on the manuscript which is not surprasing in view of its size. In order to improve the manuscript, I have made numerous recommndations and comments on an corrected PDF file attached to this message. I recommend the author to carefully examine these recommendations and to estimate if these need acception or not.

Response 1: I have tried to follow all the recommendations of the reviewer and have made appropriate notes in the corrected PDF file attached to this message. Almost all comments were accepted. The entire text of the article was carefully checked, and many additional minor corrections were made.

Point 2. Single characters used to delimit few new subgenera (for instance,  Macroharpalus subg. n.) may seem "weak" because of the relative nature of these characteristics in particular.

Responce 2. I think that the vast majority of taxa recognized, including Macroharpalus subg. n., are monophyletic, and I really struggled to find their distinctive features. Of course, I would be glad if all the recognized monophyletic units were characterized by several clear synapomorphies, but, unfortunately, nature provides material that is more complex. Larger combinations will cause even more trouble finding clear synapomorphies. The status of supraspecific taxa is always subjective; there are no objective criteria for them.

Point 3. Check if the "posteromedial" pore of metacoxa is not actually a "anteromedial" pore in view of its location toward the body.

Responce 3. I have made the relevant comments in the PDF file.

Point 4. Some sentences are too long and sound unclear for the non-English reader. Probably the best choice is to re-write and separate them.

Response 4. I have made some changes to the text.

Reviewer 2 Report

This is an impressive and very important piece of work. The author has a long, stellar history of quality publications on the tribe Harpalini and this treatment of Harpalus of the world will be another. As pointed out in the manuscript, this is a genus in desperate need of a global review. The author provides this. While I think that the manuscript can be further improved (as noted below and in my comments in the PDF) it absolutely is worthy of publication upon the revisions.  

The first 30 or so pages, that cover the most general and introductory aspects, I read carefully and made numerous comments. Many of these comments and suggestions are language related. I tried to mark places that seemed to me to be odd grammar and unusual word choice. I fully understand the challenge one faces writing in a second language and so I hope my suggestions will be well received. More checking and critical review of the text is needed, but this is all I have time for at the moment.

I did not have a chance to properly test the key to groups, but I look forward to using it. Superficially, it looks excellent. However, it does not leverage any figures! I think some figures associated with the key would be a big improvement. Using existing illustrations (which are excellent) and adding a few more, key-specific ones would be a big upgrade.

The bulk of the text covering the subgenus and group diagnoses and descriptions I spot read, but did not have time to read line by line. I noted a few errors therein. Another round of editing is advised. The content and level of description is wonderful, generally very clear and well organized. This is probably the most important part of the contribution. I heartily thank the author for the work that went into these.

Figures are great, high quality and very useful. I didn’t check all the in-text references to figures but noted some errors with those.

Regarding the classification and the many, many new subgenera- Chances are that this is over-split. Time and evidence will test this. Given the extremely poor and confused state of Harpalus taxonomy at this time, this work is a huge step forward. I’m willing to accept the likely extra names for now. Let’s see how these hold.

One general caution to the author- Frequently propositions of behaviors, evolutionary transitions, and processes are portrayed as “known” or “obvious”. However, in many of those cases no specific studies, example species, or testable evidence is given. I strongly encourage phrasing these as “hypothesized” and “apparent” and to be clear if speculations are based on the author’s direct observations or published (and cited) previous works. Just one example would be the implication of processes or evolutionary transition, when there is no phylogeny to justify a statement. Take elytral microsculpture for example. It is either present and evident (i.e., deeply impressed microlines) or present but hardly visible or absent. To say that the elytral microsculpture is “present” or “reduced” implies a process and a directionality for transition, but without a phylogeny this is not possible. Look throughout the text for language that implies such over extension of assumed process.

See my previous comments and marks on the pdf.

Author Response

Response to reviewer 2 Comments

Point 1: The first 30 or so pages, that cover the most general and introductory aspects, I read carefully and made numerous comments. Many of these comments and suggestions are language related. I tried to mark places that seemed to me to be odd grammar and unusual word choice. I fully understand the challenge one faces writing in a second language and so I hope my suggestions will be well received. More checking and critical review of the text is needed, but this is all I have time for at the moment.

Response 1: I have tried to follow all the recommendations of the reviewer and have made appropriate notes in the corrected PDF file attached to this message. Almost all suggestions and all English corrections were accepted. The entire text of the article was carefully checked, and many additional orrections were made.

 Point 2: I did not have a chance to properly test the key to groups, but I look forward to using it. Superficially, it looks excellent. However, it does not leverage any figures! I think some figures associated with the key would be a big improvement. Using existing illustrations (which are excellent) and adding a few more, key-specific ones would be a big upgrade.

 Response 2: The references to the figures are included in the key.

Point 3: One general caution to the author- Frequently propositions of behaviors, evolutionary transitions, and processes are portrayed as “known” or “obvious”. However, in many of those cases no specific studies, example species, or testable evidence is given. I strongly encourage phrasing these as “hypothesized” and “apparent” and to be clear if speculations are based on the author’s direct observations or published (and cited) previous works. Just one example would be the implication of processes or evolutionary transition, when there is no phylogeny to justify a statement. Take elytral microsculpture for example. It is either present and evident (i.e., deeply impressed microlines) or present but hardly visible or absent. To say that the elytral microsculpture is “present” or “reduced” implies a process and a directionality for transition, but without a phylogeny this is not possible. Look throughout the text for language that implies such over extension of assumed process.

Response 3: I tried to correct all these places in the text.

 I express my deep gratitude to the reviewer for the great work done and such a thorough review of the article. All the comments made, undoubtedly, contributed to its significant improvement.

Reviewer 3 Report

The article made a successful attempt to solve an old problem - to create a natural classification of the genus Harpalus. This genus, belonging to the family Carabidae, consists of about 400 species, many of which are difficult to distinguish due to the weak expression and transitional nature of the characters. The author was able to subdivide this genus into a limited number of subgenera. I hope this system will be convenient for carabidologists in their practical scientific work.

Quality of English in article is fine.

Author Response

I am very grateful to the second reviewer for the work done and for the review.

No points.